

# A General Lake Model (GLM 3.0) for linking with high-frequency sensor data from the Global Lake Ecological Observatory Network (GLEON)

Matthew R. Hipsey[1,*], Louise C. Bruce[1], Casper Boon[1], Brendan Busch[1], Cayelan C. Carey[2], David P. Hamilton[3], Paul C. Hanson[4], Jordan S. Read[5], Eduardo de Sousa[1], Michael Weber[6], Luke A. Winslow[7]

[1] UWA School of Agriculture & Environment, The University of Western Australia, Crawley WA, 6009, Australia
[2] Department of Biological Sciences, Virginia Tech, Blacksburg VA, USA
[3] Australian Rivers Institute, Griffith University, Brisbane QLD, Australia
[4] Center for Limnology, University of Wisconsin - Madison, Madison WI, USA
[5] U.S. Geological Survey, Water Mission Area, Middleton WI, USA
[6] Department of Lake Research, Helmholtz Centre for Environmental Research - UFZ, Magdeburg, Germany
[7] Department of Biological Sciences, Rensselaer Polytechnic Institute, Troy NY, USA

*Correspondence to*: Matthew R. Hipsey (matt.hipsey@uwa.edu.au)

*Keywords:*     climate change, lake, mixing, stratification, observatory network, wetland, water balance, water quality

**Abstract.** The General Lake Model (GLM) is a one-dimensional open-source code designed to simulate the hydrodynamics of lakes, reservoirs and wetlands. GLM was developed to support the science needs of the Global Lake Ecological Observatory Network (GLEON), a network of researchers using lake sensors to understand lake functioning and address questions about how lakes around the world respond to climate and land-use change. The scale and diversity of lake types, locations and sizes, as well as the expanding observational datasets, created the need for a robust community model of lake dynamics with sufficient flexibility to accommodate a range of scientific and management questions relevant to the GLEON community. This paper summarises the scientific basis and numerical implementation of the model algorithms, including details of sub-models that simulate surface heat exchange and ice-cover dynamics, vertical mixing and inflow/outflow dynamics. We demonstrate the suitability of the model for different lake types, that vary substantially in their morphology, hydrology and climatic conditions. GLM supports a dynamic coupling with biogeochemical and ecological modelling libraries for integrated simulations of water quality and ecosystem health, and options for integration with other environmental models are outlined. Finally, we discuss utilities for the analysis of model outputs and uncertainty assessments, model operation within a distributed cloud-computing environment, and as a tool to support learning of network participants.



**Revision 26 Oct 2018**

## 1 Introduction

Lakes and other standing waters support extensive ecosystem services such as water supply, flood mitigation, hydropower, aesthetic and cultural benefits, as well as fisheries and biodiversity (Mueller et al., 2016). Lakes are often considered to be "*sentinels of change*", providing a window into the sustainability of activities in their catchments (Williamson et al., 2009).

They are also particularly susceptible to impacts from invasive species and land use development, which often lead to water quality deterioration and loss of ecosystem integrity. Recent estimates have demonstrated their significance in the earth system, contributing to heterogeneity in land surface properties and feedbacks to regional and global climate through energy, water and biogeochemical transfers (Martynov et al., 2012; Cole et al., 2007). For example, Tranvik et al. (2009) suggest carbon burial in lakes and reservoirs is substantial on a global scale, on the order of 0.6 Pg yr$^{-1}$, or four times the oceanic burial rate.

Given the diversity of lakes among continents, region-specific pressures and local management approaches, the Global Lake Ecological Observatory Network (GLEON: gleon.org) was initiated in 2004 as a grass-roots science community with a vision to observe, understand and predict freshwater systems at a global scale (Hanson et al., 2016). In doing so, GLEON has been a successful example of collaborative research within the hydrological and ecological science disciplines. GLEON aims to bring

together environmental sensor networks, numerical models, and information technology to explore ecosystem dynamics across a vast range of scales - from individual lakes or reservoirs (Hamilton et al., 2015) to regional (Read et al., 2014; Klug et al., 2012), and global extents (Rigosi et al., 2015; O´Reilly et al., 2015). Ultimately, it is the aim of the network to facilitate discovery and synthesis, and to provide an improved scientific basis for sustainable freshwater resource management.

Environmental modelling forms a critical component of observing systems, as a way to make sense of the "data deluge" (Porter et al., 2012), allowing users to build virtual domains to support knowledge discovery at the system scale (Ticehurst et al., 2007; Hipsey et al., 2015). In lake ecosystems, the tight coupling between physical processes and water quality and ecological dynamics has long been recognised. Modellers have capitalized on a comprehensive understanding of physical processes (e.g., Imberger and Patterson, 1990; Imboden and Wüest, 1995) to use hydrodynamic models as an underpinning basis for coupling

to ecological models. Such models have contributed to our understanding of lake dynamics, including applications associated with climate change (Winslow et al., 2017), eutrophication dynamics (Matzinger et al., 2007), harmful algal bloom dynamics (Chung et al., 2014), and fisheries (Makler-Pick et al., 2011).

In recent decades, a range of 1, 2, and 3-dimensional hydrodynamic models has emerged for lake simulation. Depending on

the dimensionality, the horizontal resolution of these models may vary from metres to tens of kilometres with vertical resolutions from sub-metre to several metres. As in all modelling disciplines, identifying the most parsimonious model structure and degree of complexity and resolution is challenging, and users in the lake modelling community often tend to rely on heuristic rules or practical reasons for model choice (Mooij et al., 2010). High-resolution models are suited to studying events that occur at the time scale of flow dynamics, but are not always desirable for ecological studies over longer time scales

due to their computational demands and level of over-parameterisation. On the other hand, simple models may be more agile for a particular application, and more suited to parameter identification and scenario testing workflows. However, it has been

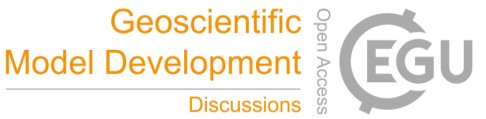

the case within GLEON that simple models are often less applicable across a wide variety of domains, making them less generalizable, which is a key requirement of synthesis studies across many waterbodies. Despite the fact that there is a relatively large diversity of models and approaches for aquatic ecosystem simulation (Janssen et al., 2015), it is generally agreed that to improve scientific collaboration within the limnological modelling community, there is an increasing need for

flexible, open-source community models (Trolle et al., 2012). Whilst acknowledging that there is no single model suitable for all applications, a range of open-source community models and tools can enhance scientific capabilities, and foster scientific collaboration and combined efforts (Read et al., 2016). There are examples of such initiatives being successful in the oceanography, hydrology and climate modelling communities.

With this in mind, the General Lake Model (GLM), a one-dimensional (1D) hydrodynamic model for enclosed aquatic ecosystems, was developed. The lake modelling community has often relied on 1D models, which originated to capture lake water balance and thermal stratification dynamics (e.g., Imberger and Patterson, 1981; Saloranta and Andersen, 2007; Perroud et al., 2009; Stepanenko et al., 2013). The use of 1D structure is justified across a diverse range of lake sizes given the dominant role of seasonal changes in vertical stratification on lake dynamics, including oxygen, nutrient and metal cycling and plankton

dynamics (Hamilton and Schladow, 1997; Gal et al., 2009). Despite advances in computing power and more readily available 3D hydrodynamic drivers, 1D models continue to remain attractive as they are easily linked with biogeochemical and ecological modelling libraries for complex ecosystem simulations. This allows 1D models to be used to capture the long-term trajectory and resilience of lakes and reservoirs to climate change, hydrologic change and land use change. For example, such models have been used to study long-term changes to oxygen, nutrient cycles, and the changing risk of algal blooms (e.g.,

Peeters et al., 2007; Hu et al., 2016; Snortheim et al., 2017). Furthermore, the low computational requirements of this approach relative to 3D models is more suited to parameter identification and uncertainty analysis, making it an attractive balance between process complexity and computational intensity.

   GLM emerged as a new open-source code in 2012, with the design goal of balancing the complexity of dimensional

representation, applicability to a wide range of standing waters, and availability to a broad community (e.g., GLEON has >700 members from around 50 countries). The scope and capability of the model has developed rapidly with application to numerous lakes and lake-types within the GLEON network and beyond (e.g., Read et al., 2014; Bueche et al., 2017; Snortheim et al., 2017; Weber et al., 2017; Menció et al., 2017; Bruce et al., 2018). It is unique in that its suitability now ranges from ephemeral wetlands and ponds to deep lakes, from natural systems to heavily managed man-made reservoirs, and across climatic regions.

Given that individual applications rarely engage the full array of features or describe the full details of the model structure, the aim of this paper is to present a complete description of GLM, including the scientific background (Section 2), and model code organization (Section 3). The approach to coupling with biogeochemical models is also discussed (Section 4), since a main objective of GLM's development is to link its hydrodynamic simulation with water quality models to explore the effects of stratification and vertical mixing on biogeochemical cycles and lake ecology. Finally, an overview of the use of the model

within the context of GLEON specific requirements for model analysis, integration and education (Section 5-6) is described.



**Revision 26 Oct 2018**

In order to better define the typical level of model performance across these diverse lake types, a companion paper by Bruce et al. (2018) has undertaken a systematic assessment of the model's error structure against 31 lakes.

## 2 Model Overview

### 2.1 Background and layer structure

The 1D approach adopted by GLM resolves a vertical series of layers that capture the variation in water column properties. Users may configure any number of inflows and outflows, and more advanced options exist for simulating aspects of the water and heat balance (Figure 1). Depending on the context of the simulation, either daily or hourly meteorological time series data for surface forcing is required, and daily time series of volumetric inflow and outflow rates can also be supplied. The model is suitable for operation in a wide range of climate conditions and is able to simulate ice formation, as well as accommodating

a range of atmospheric forcing conditions.

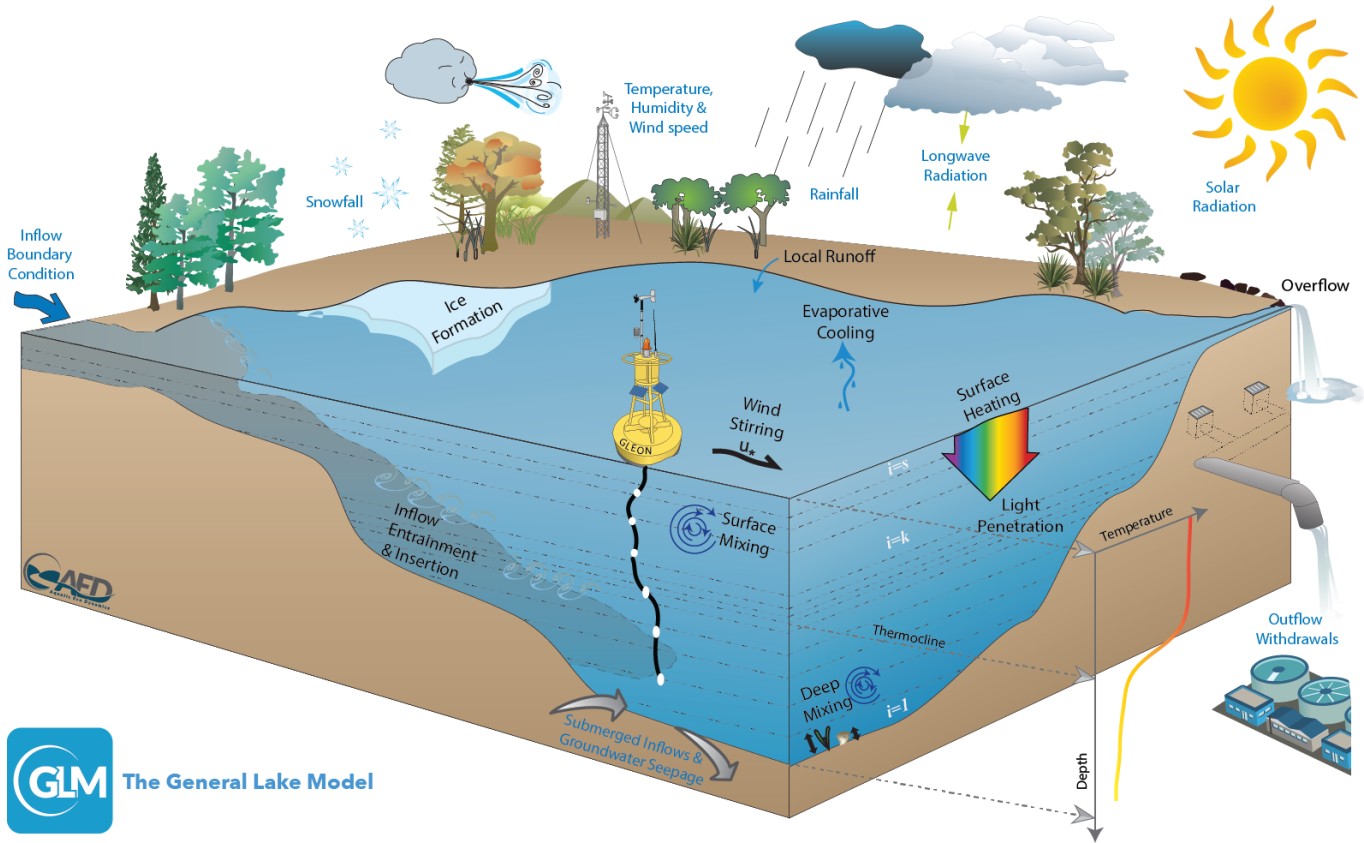

**Figure 1: Schematic of a GLM simulation domain, input information (blue text) and key simulated processes (black text).**

Although GLM is a new model code written in the C programming language, the core layer structure and mixing algorithms are founded on principles and experience from model platforms including the DYnamic REservoir Simulation Model (DYRESM; Imberger and Patterson, 1981; Hamilton and Schladow, 1997) and the Dynamic Lake Model (DLM; Chung et al., 2008). Other variations have been introduced to extend this underlying approach through applications to a variety of lake and



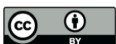

**Revision 26 Oct 2018**

reservoir environments (e.g., Hocking & Patterson, 1991; McCord and Schladow, 1998; Gal et al., 2003; Yeates and Imberger, 2003). The layer structure is numbered from the lake bottom to the surface, and adopts the flexible Lagrangian layer scheme first introduced by Imberger et al. (1978) and Imberger and Patterson (1981). The approach defines each layer, $i$, as a 'control volume' (Figure 1) that can change thickness by contracting and expanding in response to inflows, outflows, mixing with

adjacent layers, and surface mass fluxes. As the model simulation progresses, density changes due to surface heating, vertical mixing, and inflows and outflows lead to dynamic changes in the layer structure, associated with layers amalgamating, expanding, contracting or splitting. Notation used throughout the model description is provided in Table 1.

As layers change, their volumes change based on the site-specific hypsographic curve, whereby the overall lake volume, $V_{max}$,

is defined as $\int_{H_0}^{H_{max}} A[H]\,dH$, with the elevation ($H$), and area ($A$) relationship provided as a series of points based on bathymetric data. This computation requires the user to provide a number, $N_{BSN}$, of elevations with corresponding areas. The cumulative volume at any lake elevation is first estimated as:

$$V_b = V_{b-1} + [0.5(A_b + A_{b-1})](H_b - H_{b-1}) \qquad (1)$$

where $2 \leq b \leq N_{BSN}$. Using this raw hypsographic data, a refined height-area-volume relationship is then internally computed using finer height increments (e.g., $\Delta H_{mi} \sim 0.1$ m), giving $N_{MORPH}$ levels that are used for subsequent calculations. The area

and volume at the height of each increment, $H_{mi}$, are interpolated from the supplied information as:

$$V_{mi} = V_b \left(\frac{H_{mi}}{H_b}\right)^{\alpha_b} \quad \text{and} \quad A_{mi} = A_b \left(\frac{H_{mi}}{H_b}\right)^{\beta_b} \qquad (2)$$

where $V_{mi}$ and $A_{mi}$ are the volume and area at each of the elevations of the interpolated depth vector, and $V_b$ and $A_b$ refer to the nearest $b$ level below $H_{mi}$ such that $H_b < H_{mi}$. The interpolation coefficients are computed as:

$$\alpha_b = \left(\frac{\log_{10}\left[\frac{V_{b+1}}{V_b}\right]}{\log_{10}\left[\frac{H_{b+1}}{H_b}\right]}\right) \quad \text{and} \quad \beta_b = \left(\frac{\log_{10}\left[\frac{A_{b+1}}{A_b}\right]}{\log_{10}\left[\frac{H_{b+1}}{H_b}\right]}\right). \qquad (3)$$

Within this lake domain, the model solves the water balance by including several user configurable water fluxes that change the layer structure. Initially, the layers are assumed to be of equal thickness, and the initial number of layers, $N_{LEV}(t = 0)$ is

computed based on the initial water depth. Water fluxes include surface mass fluxes (evaporation, rainfall and snowfall), inflows (surface inflows, submerged inflows and local runoff from the surrounding exposed lake bed area) and outflows (withdrawals, overflow and seepage). Surface mass fluxes operate on a sub-daily time step, $\Delta t$, by impacting the surface layer thickness (described in Section 2.2), whereby the dynamics of inflows and outflows modify the overall lake water balance and layer structure on a daily time step, $\Delta t_d$, by adding, merging or removing layers (described in Section 2.7). Depending on

whether a surface (areal) mass flux or volumetric mass flux is being applied, the layer volumes are updated by interpolating changes in layer heights, whereby $V_i = f[h_i]$, and $i$ is the layer number, or layer heights are updated by interpolating changes in layer volumes, whereby $h_i = f[V_i]$.

Each layer also contains heat, salt ($S$) and other constituents ($C$), which are generically referred to as scalars. These are subject

to mass conservation as layers change thickness or are merged or split. The specific number of other constituents depends on the configuration of the associated water quality model, but typically includes attributes such as oxygen, nutrients and

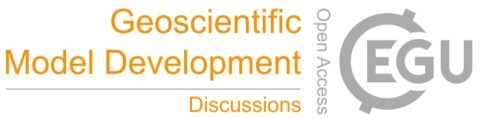



phytoplankton. Layer density is computed from the local salinity and temperature according to TEOS-10 (http://teos-10.org), whereby $\rho_i = \rho[T_i, S_i]$. When density instabilities occur between adjacent layers, or when sufficient turbulent kinetic energy becomes available to overcome stable density gradients, then layers merge, thereby accounting for the process of mixing (Section 2.6). For deeper systems, a stable vertical density gradient forms seasonally in response to periods of high solar

radiation creating warm, buoyant water overlying cooler, denser water, separated by a metalimnion region which includes the thermocline. Layer volumes change due to depth-specific changes in mixing, inflows and outflows. Thickness limits, $\Delta z_{min}$ and $\Delta z_{max}$, are enforced to adequately resolve the vertical density gradient, generally with fine resolution occurring in the metalimnion and thicker cells where gradients are weak. The number of layers, $N_{LEV}[t]$, is adjusted throughout the simulation to maintain homogenous properties within a layer. It has been reported that numerical diffusion at the thermocline can be

restricted using this layer structure and mixing algorithm (depending on the minimum and maximum layer thickness limits set by the user), making it particularly suited to long-term investigations, and ideally requiring limited site-specific calibration (Patterson et al., 1984; Hamilton and Schladow, 1997; Bruce et al., 2018).

Because this approach assumes layer properties are laterally averaged, the model is suitable for investigations where resolving

the horizontal variability is not a requirement of the study. This is often the case for ecologists and biogeochemists studying central basins of natural lakes (e.g., Gal et al., 2009), managers simulating drinking water reservoirs (e.g., Weber et al., 2017), or mining pit lakes (e.g., Salmon et al., 2017), or for analyses exploring the coupling between lakes and regional climate (e.g., Stepanenko et al., 2013). Further, whilst the model is able to resolve vertical stratification, the approach is also able to be used to simulate shallow lakes, wetlands, wastewater ponds and other small waterbodies that experience well mixed conditions. In

this case, the layer resolution, with upper and lower layer bounds specified by the user, will automatically be reduced, and the mass of water and constituents, and energy will continue to be conserved. The remainder of this section outlines the model components and provides example outputs for five water bodies that experience a diverse hydrology.

**2.2 Water balance**

The general nature of the model to accommodate a wide diversity of lake types has necessitated flexibility in configuration of

water inputs and outputs (schematically depicted in Figure 1). The net water flux over the entire lake is summarised as:

$$\frac{dV}{dt} = A_S \frac{dh_S}{dt} + \sum_{I}^{N_{INF}} Q_{inf_{0_I}} - \sum_{O}^{N_{OUTF}} Q_{outf_O} - Q_{seepage} - Q_{ovfl} \qquad (4)$$

where $V$ is the total lake volume, $t$ is time, $A_S$ is the lake surface area, the changes due to fluxes at the water surface, $h_S$, are expanded upon below, and the remaining inflow and outflow terms are described in detail in Section 7. For practical reasons

the equation is numerically solved in two stages with different times steps for the surface flux change and all other fluxes. Furthermore, in any given application, not all the inputs and outputs are relevant and users may customise the water balance components accordingly; examples demonstrating lake hydrology from wetlands to reservoirs to deep lakes are presented in

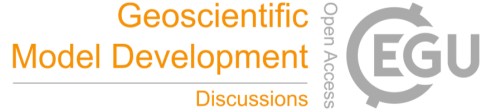

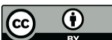

Figure 2. Note that Eq. 4 accounts for the liquid water balance, and in cold climates the model will also track the amount of water allocated into an overlying ice layer (Section 2.4), which interacts with the surface water balance as indicated next.

The mass balance of the surface layer is computed at each model time step ($\Delta t$; usually hourly), by modifying the surface layer height, $h_S$, according to:

$$\frac{dh_S}{dt} = R_F + S_F + \frac{Q_R}{A_S} - E - \frac{d\Delta z_{ice}}{dt} \qquad (5)$$

**Figure 2: A two-year times series of the simulated daily water balance for five example lakes, a-e, that range in size and hydrology. The water balance components summarised are depicted schematically in the inset, and partitioned into inputs and outputs. The daily net water flux is computed from Eq. 4. For more information about each lake, the simulation configuration and input data, refer to the Data availability section.**

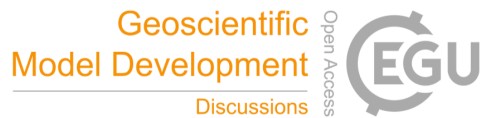

**Revision 26 Oct 2018**

where $E$ is the evaporation mass flux computed from the latent heat flux $\phi_E$, described below ($E = \phi_E/\lambda_v\rho_s$ ; m s$^{-1}$), $R_F$ is rainfall and $S_F$ is snowfall (m s$^{-1}$). Depending on the meteorological conditions, precipitation will either be added to the water volume, or to the surface of the ice cover (see section 2.4), and $R_F$ and $S_F$ therefore influence the water surface height depending on the presence of ice cover according to:

$$R_F = \begin{cases} f_R R_x/c_{secday} \,, & \text{if } \Delta z_{ice} = 0 \\ f_R R_x/c_{secday} \,, & \text{if } \Delta z_{ice} > 0 \text{ and } T_a > 0 \\ 0 \,, & \text{if } \Delta z_{ice} > 0 \text{ and } T_a \leq 0 \end{cases} \qquad (6)$$

and

$$S_F = \begin{cases} f_S\, f_{SWE}\, S_x/c_{secday}\,, & \text{if } \Delta z_{ice} = 0 \\ 0, & \text{if } \Delta z_{ice} > 0 \end{cases} \qquad (7)$$

Here $f_R$ and $f_S$ are user defined scaling factors that may be applied to adjust the input data values, $R_x$ and $S_x$ respectively. The surface height of the water column is also impacted by ice formation or melting of the ice layer sitting on the lake surface, according to $d\Delta z_{ice}/dt$, as described in Section 2.4.

$Q_R$ is an optional term to account for runoff to the lake from the exposed riparian banks, which may be important in reservoirs with a large drawdown range, or wetlands where periodic drying of the lake may occur. The runoff volume generated is averaged across the area that the active lake surface area ($A_s$) is not occupying, and the amount is calculated using a simple model based on exceedance of a rainfall intensity threshold, $R_L$ (m day$^{-1}$), and runoff coefficient:

$$Q_R = max[0, f_{ro}(R_F - R_L/c_{secday})](A_{max} - A_S) \qquad (8)$$

where $f_{ro}$ is the runoff coefficient, defined as the fraction of rainfall that is converted to runoff at the lake's edge, and $A_{max}$ is
the maximum possible area of inundation of the lake (the area provided by the user as the $N_{BSN}$ value).

Note that mixing dynamics (i.e., the merging or splitting of layers to enforce the layer thickness limits), will impact the thickness of the surface mixed layer, $z_{sml}$, but not change the overall lake height. However, in addition to the terms in Eq. 5, $h_S$ is modified due to volume changes associated with river inflows, withdrawals, seepage or overflows, which are described
in subsequent sections.

**2.3 Surface energy balance**

A balance of shortwave and longwave radiation fluxes, and sensible and evaporative heat fluxes (all W m$^{-2}$) determines the net cooling and heating across the surface.  The general heat budget equation for the uppermost layer is described as:

$$c_w\rho_s z_s \frac{dT_s}{dt} = \phi_{SW_S} - \phi_E + \phi_H + \phi_{LWin} - \phi_{LWout} \qquad (9)$$

where $c_w$ is the specific heat capacity of water, $T_s$ is the surface temperature, and $z_s$ and $\rho_s$ are the depth and density of the
surface layer ($i = N_{LEV}$), respectively. The right-hand side (RHS) heat flux terms are numerically computed at each time step, and include several options for customizing the individual surface heat flux components, which are expanded upon below.

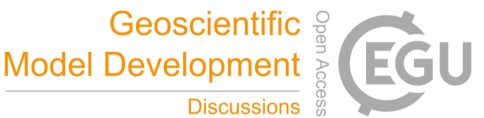

**Revision 26 Oct 2018**

**2.3.1 Solar heating and light penetration**

Solar radiation is the key driver of the lake thermodynamics and may be input based on daily or hourly measurements from a nearby pyranometer. If data are not available then users may choose to have GLM compute surface irradiance from a theoretical approximation based on the Bird Clear Sky insolation model (BCSM) (Bird, 1984), modified for cloud cover and latitude. The options for input are summarised as:

$$\phi_{SW_0} = \begin{cases} (1 - \alpha_{SW}) \, f_{SW} \, \phi_{SW_x} \, f[d, t - \lfloor t \rfloor], & \text{Option 1: daily insolation data provided} \\ (1 - \alpha_{SW}) \, f_{SW} \, \phi_{SW_x}, & \text{Option 2: sub-daily insolation data provided} \\ (1 - \alpha_{SW}) \, f_{SW} \, \hat{\phi}_{SW}, & \text{Option 3: insolation computed from the BCSM} \end{cases}$$

(10a-c)

where $\phi_{SW_0}$ is the solar radiation flux entering the surface layer, $\phi_{SW_x}$ is the incoming shortwave radiation flux supplied by the user, $f_{SW}$ is a scaling factor that may be applied and adjusted as part of the calibration process (for example, to capture the effects of shading), and $\alpha_{SW}$ is the albedo for shortwave radiation. If daily data are supplied (Option 1), the model continues to run at a sub-daily time step, but applies the algorithm outlined in Hamilton and Schladow (1997) to distribute the daily solar energy flux over a diurnal cycle, based on the day of the year, $d$, and time of day, $t - \lfloor t \rfloor$. For Option 3 the BCSM is used (Bird, 1984; Luo et al., 2010):

$$\hat{\phi}_{SW} = \frac{\hat{\phi}_{DB} + \hat{\phi}_{AS}}{1 - (\alpha_{SW} \, \alpha_{SKY})} \, f[C_x]$$

(11)

where the total irradiance, $\hat{\phi}_{SW}$, is computed from direct beam $\hat{\phi}_{DB}$, and atmospheric scattering $\hat{\phi}_{AS}$ components (refer to Appendix A for a detailed outline of the BCSM equations and parameters). In GLM, the clear sky value is then reduced according to the cloud cover data provided by the user, $C_x$, according to:

$$f[C_x] = 0.66182 \, C_x^2 - 1.5236 \, C_x + 0.98475$$

(12)

which is based on a polynomial regression of cloud data from Perth Airport, Australia, compared against nearby sensor data ($R^2 = 0.952$; see also a similar relationship by Luo et al., 2010).

The albedo, $\alpha_{SW}$, is the reflected fraction of the incoming radiation and depends on surface conditions including the presence of ice, waves and the angle of incident radiation. For open water conditions, users may configure:

Option 1 : Daily approximation from Hamilton and Schladow (1997)

$$\alpha_{SW} = \begin{cases} \alpha_{SW_{mean}} - \delta\alpha_{SW} \, sin\left[\frac{2\pi}{365}d - \frac{\pi}{2}\right] & \text{:northern hemisphere, Lat>0} \\ \alpha_{SW_{mean}} & \text{:equator} \\ \alpha_{SW_{mean}} - \delta\alpha_{SW} \, sin\left[\frac{2\pi}{365}d + \frac{\pi}{2}\right] & \text{:southern hemisphere, Lat<0} \end{cases}$$

(13a)

Option 2 : Sub-daily approximation from Briegleb et al. (1986)

$$\alpha_{SW} = \frac{1}{100}\left(\frac{2.6}{cos[\Phi_{zen}]^{1.7}+0.065} + 15(cos[\Phi_{zen}] - 0.1)(cos[\Phi_{zen}] - 0.5)(cos[\Phi_{zen}] - 1)\right)$$

(13b)

Option 3 : Sub-daily approximation from Yajima and Yamamoto (2015)

$$\alpha_{SW} = max[\, 0.02, \; 0.001 \, f_{RH} \, RH_x \, (1 - cos[\Phi_{zen}])^{0.33} - 0.001 \, U_{10} \, (1 - cos[\Phi_{zen}])^{-0.57} - 0.001 \, \varsigma \, (1 - cos[\Phi_{zen}])^{0.829}]$$

(13c)

Option 4 : Daily approximation from Grischenko look-up table in Cogley (1979)

$$\alpha_{SW} = \alpha_{SW_G}[Lat, d\,]$$

(13d)

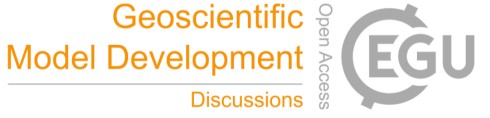

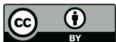

**Revision 26 Oct 2018**

where $\Phi_{zen}$ is the solar zenith angle (radians) as outlined in Appendix A, $RH_x$ is the relative humidity, $\varsigma$ is the percentage of atmospheric diffuse radiation, $d$ is the day of year, and $U_{10}$ is wind speed. The second (oceanic) and third (lacustrine) options are included to allow for diel and seasonal variation of albedo from approximately 0.01 to 0.4 depending on the sun-angle (Figure 3). Albedo is calculated separately during ice cover conditions using a customised algorithm, outlined below in Section

2.4.

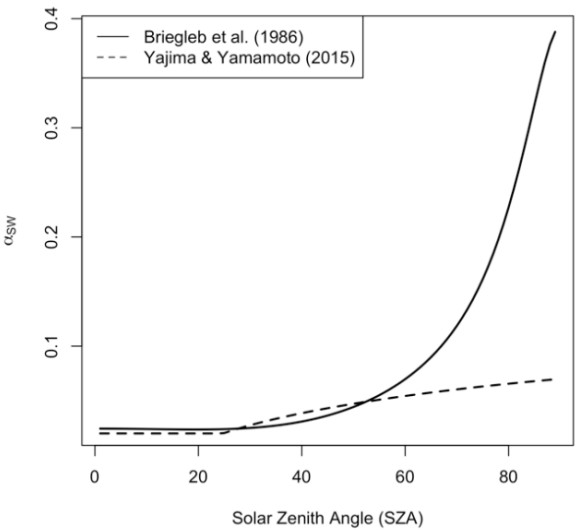

**Figure 3: Variation of albedo ($\alpha_{SW}$) with solar zenith angle (SZA = $\Phi_{zen}180/\pi$, degrees) for Options 2 and 3 (Eq. 13). For Option 3, settings of $RH_x$ = 80% and $U_{10}$ = 6 m s$^{-1}$ were assumed.**

The depth of penetration of shortwave radiation into the lake is wavelength specific, and depends on the water clarity via the light extinction coefficient, $K_w$ (m$^{-1}$). Two approaches are supported in GLM. The first option assumes the Photosynthetically Active Radiation (PAR) fraction of the incoming light is the most penetrative, and follows the Beer-Lambert Law:

$$\phi_{PAR}[z] = f_{PAR}\ \phi_{SW_0}\ exp[-K_w z] \tag{14}$$

where $z$ is the depth of any layer from the surface. $K_w$ may be set by the user as constant, read in from a time-series file, or linked with the water quality model (e.g., FABM or AED2, see Section 4); in the latter case the extinction coefficient will

change as a function of depth and time according to the concentration of dissolved and particulate constituents. For this option Beer's Law is only applied for the photosynthetically active fraction, $f_{PAR}$, which is set as 45% of the incident light. The amount of radiation heating the surface layer, $\phi_{SW_S}$, is therefore the photosynthetically active fraction that is attenuated across $z_s$, plus the entire $(1 - f_{PAR})$ fraction, $\phi_{SW_S} = \phi_{SW_0} - \phi_{PAR}[z_s],$ which implicitly assumes the near infra-red and ultraviolet bandwidths of the incident shortwave radiation have significantly higher attenuation coefficients (Kirk, 1994). The second

option adopts a more complete light absorption algorithm that integrates the attenuated light intensity across the bandwidth spectrum:


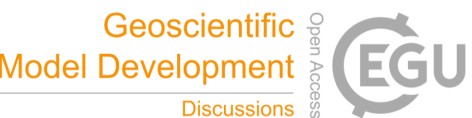
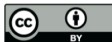

**Revision 26 Oct 2018**

$$c_w \rho_i \Delta z_i \frac{dT_i}{dt} = \sum_{l=1}^{N_{SW}} \phi_{SW_{i_l}}[z_i] - \sum_{l=1}^{N_{SW}} \phi_{SW_{i-1_l}}[z_{i-1}]$$

(15)

where $l$ is the bandwidth index and $\phi_{SW_{i_l}}[z_i]$ is the radiation flux at the top of the $i$ th layer for the $l$ th bandwidth fraction. For this option, the model by Cengel and Ozisk (1984) is adopted to compute the penetration of individual bandwidth fractions, which more comprehensively resolves the incident and diffuse radiation components of the light climate, taking into account the angle of incident light and transmission across the light surface (based on the Fresnel equations), and reflection off the bottom. These processes are wavelength specific and the user specifies the number of simulated bandwidths, $N_{SW}$, their respective absorption coefficients, $K_{w_l}$, and reflectivity of light at the sediment, $\alpha_{sed}$.

The light reaching the benthos is relevant in some applications as an indicator of benthic productivity, or as a proxy for the type of benthic habitat that might emerge. In addition to the light profiles, GLM therefore predicts the benthic area of the lake where light intensity exceeds a user defined fraction of the surface irradiance, $f_{BEN_{crit}}$, (Figure 4):

$$A_{BEN} = A_S - A[h_{BEN}]$$

(16)

where $h_{BEN} = h_S - z_{BEN}$, and $z_{BEN}$ is calculated from Beer's law:

$$z_{BEN} = - \frac{ln[f_{BEN_{crit}}]}{K_w}$$

(17)

and the daily average benthic area above the threshold is then reported as a percentage ($100 \times A_{BEN}/A_s$).

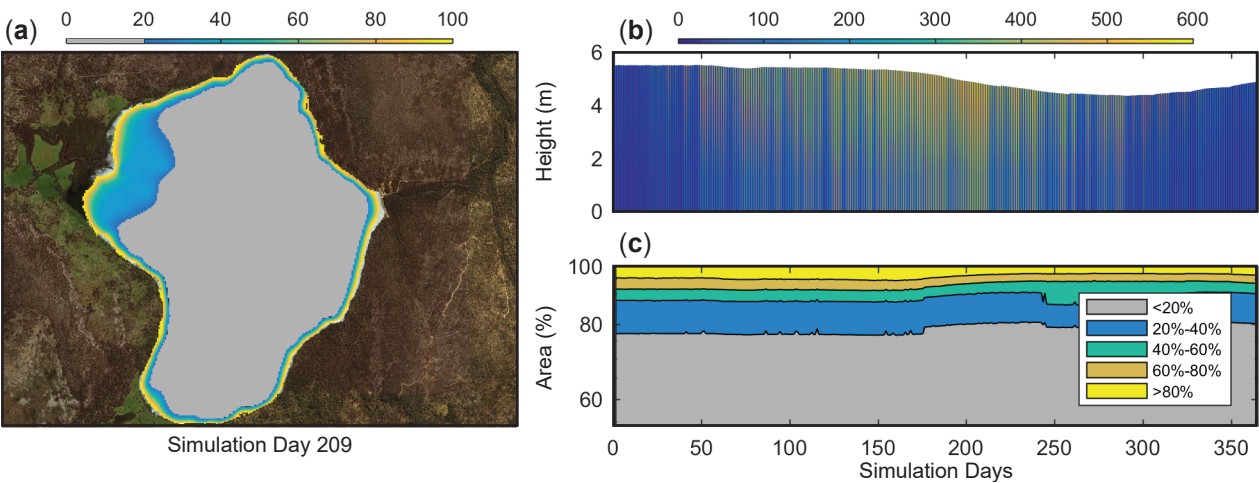

**Figure 4: Example light data outputs from a GLM application to Woods Lake, Australia, showing a) the ratio of benthic to surface light, $100\, \phi_{PAR_{BEN}}/\phi_{SW_0}$ (%), overlain on the lake map based on the bathymetry, with the area where $f_{BEN_{crit}} < 0.2$ (i.e. less than 20% of surface irradiance) depicted in grey, b) a time series of the depth variation in light (W m$^{-2}$), and c) a time series of $A_{BEN}/A_s$ (as %) for various $f_{BEN_{crit}}$. Note the 2D projection of the 1D lake model in (a) can assist in managing lake condition but assumes uniformity of K$_w$.**



**Revision 26 Oct 2018**

## 2.3.2 Longwave radiation

Longwave radiation can be provided as a net flux, an incoming flux or, if there is no radiation data from which longwave radiation can be computed, then it may be calculated by the model internally based on the cloud cover fraction and air temperature. Net longwave radiation is described as:

$$\phi_{LW_{net}} = \phi_{LW_{in}} - \phi_{LW_{out}} \tag{18}$$

where

$$\phi_{LW_{out}} = \varepsilon_w \sigma (\theta_s)^4 \tag{19}$$

and $\sigma$ is the Stefan-Boltzman constant and $\varepsilon_w$ the emissivity of the water surface, assumed to be 0.985. If the net or incoming longwave flux is not provided, the model will compute the incoming flux from:

$$\phi_{LW_{in}} = (1 - \alpha_{LW}) \, \varepsilon_a^* \, \sigma \, (\theta_a)^4 \tag{20}$$

where $\alpha_{LW}$ is the longwave albedo (0.03). The emissivity of the atmosphere can be computed considering emissivity for cloud-free conditions ($\varepsilon_a$), based on air temperature ($T_a$) and vapour pressure, and extended to account for reflection from clouds, such that $\varepsilon_a^* = f[T_a, e_a, C_x]$ (see Henderson-Sellers, 1986; Flerchinger, 2009). Options adapted from a range of authors include:

$$\varepsilon_a^* = \begin{cases} (1 + 0.275 \, C_x)(1 - 0.261 \exp[-0.000777 \, T_a{}^2]), & \text{Option 1: Idso and Jackson (1969)} \\ (1 + 0.17 \, C_x^2) \, (9.365 \times 10^{-6} (\theta_a)^2), & \text{Option 2: Swinbank (1963)} \\ (1 + 0.275 \, C_x) \, 1.24 \, (e_a/\theta_a)^{1/7}, & \text{Option 3: Brutsaert (1975)} \\ (1 - C_x^{2.796}) \, 1.24 \, (e_a/\theta_a)^{1/7} + 0.955 \, C_x^{2.796}, & \text{Option 4: Yajima and Yamamoto (2015)} \end{cases} \tag{21a-d}$$

where $C_x$ is the cloud cover fraction (0-1) and $e_a$ the air vapour pressure calculated from relative humidity. Note that cloud cover is typically reported in octals (0-8), thus a value of 1 would correspond to a fraction of 0.125. Some data may also include cloud type and their respective heights. If this is the case, good correspondence has been reported by averaging the octal values for all cloud types to get an average cloud cover.

If longwave radiation data does not exist and cloud data are also not available, but solar irradiance is measured, then GLM `rad_mode` setting 3 will instruct the model to compare the measured and theoretical clear-sky solar irradiance (estimated by the BCSM; Eq. 11) to approximate the cloud cover fraction by assuming that $\phi_{SW_x}/\hat{\phi}_{SW} = f[C_x]$. Note that if neither shortwave or longwave radiation is provided, then the model will use the BCSM to compute incoming solar irradiance, and cloud cover will be assumed to be 0 (noting that this is likely to overestimate of downwelling shortwave radiation).



**Revision 26 Oct 2018**

### 2.3.3 Sensible and latent heat transfer

The model accounts for the surface fluxes of sensible heat and latent heat using commonly adopted bulk aerodynamic formulae. For sensible heat:

$$\phi_H = -\rho_a c_a C_H U_{10} (T_s - T_a) \tag{22}$$

where $c_a$ is the specific heat capacity of air, $C_H$ is the bulk aerodynamic coefficient for sensible heat transfer, $T_a$ the air temperature and $T_s$ the temperature of the water surface layer. The air density (kg m$^{-3}$) is computed from $\rho_a = 0.348\,(1 + r)/(1 + 1.61r)\,p/T_a$, where $p$ is air pressure (hPa) and $r$ is the water vapour mixing ratio, which is used to compute the gas constant.

For latent heat:

$$\phi_E = -\rho_a C_E\, \lambda_v\, U_{10}\frac{\omega}{p}\,(e_s[T_s] - e_a[T_a]) \tag{23}$$

where $C_E$ is the bulk aerodynamic coefficient for latent heat transfer, $e_a$ the air vapour pressure, $e_s$ the saturation vapour pressure (hPa) at the surface layer temperature (°C), $\omega$ the ratio of molecular mass of water to molecular mass of dry air ($= 0.622$) and $\lambda_v$ the latent heat of vaporisation. The vapour pressure is calculated by the linear formula from Tabata (1973):

$$e_s[T_s] = 10^{\left(9.28603523\, -\frac{2322.37885}{T_s+273.15}\right)} \tag{24}$$

and

$$e_a[T_a] = (f_{RH}\, RH_x/100)\, e_s[T_a] \tag{25}$$

The net heat fluxes for the example lakes are shown in Figure 5.

Correction for non-neutral atmospheric stability: For long time integrations (e.g., seasonal), the bulk-transfer coefficients for momentum, $C_D$, sensible heat, $C_H$, and latent heat, $C_E$, can be assumed approximately constant because of the negative feedback between surface forcing and the temperature response of the water body (e.g., Strub and Powell, 1987). At finer timescales (hours to weeks), the thermal inertia of the water body is too great and so the transfer coefficients should be specified as a function of the degree of atmospheric stratification experienced in the internal boundary layer that develops over the water (Woolway et al., 2017). Monin and Obukhov (1954) parameterised the stratification in the air column using the now well-known stability parameter, $z/L$, which is used to define corrections to the bulk aerodynamic coefficients $C_H$ and $C_E$, using the numerical scheme presented in Appendix B. The corrections may be optionally applied within a simulation, and if enabled, the transfer coefficients used above are automatically updated. To ensure data provided are from within the internal boundary layer over the lake surface, users should preferably provide wind speed, air temperature and relative humidity data that has been collected over the lake surface (at a height of 2-10 m, depending on lake size), supplied at approximately hourly resolution.

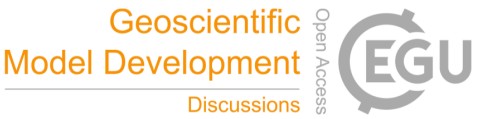

**Revision 26 Oct 2018**

Wind sheltering: Wind sheltering may be important depending on the lake size and shoreline complexity, and is parameterised according to several methods based on the context of the simulation and data available. For example, Hipsey and Sivapalan (2003) presented a simple adjustment to the bulk transfer equation to account for the effect of wind sheltering in small reservoirs using a shelter index to account for the length scale associated with the vertical obstacle relative to the horizontal length scale associated with the water body itself. Markfort et al. (2010) estimate the effect of a similar sheltering length scale on the overall lake area calculated based on surrounding topography and canopy heights relative to the water surface. Therefore, within GLM, users may specify the degree of sheltering or fetch limitation using either constant or direction-specific options for computing an "effective" area:

$$
A_E = \begin{cases} A_S, & \text{Option 0: no sheltering (default)} \\ A_S \tanh\left(\dfrac{A_S}{A_{WS}}\right), & \text{Option 1: Yeates \& Imberger (2003)} \\ \dfrac{L_D{}^2}{2} arccos\left(\dfrac{x_{WS}{}^{\Phi}}{L_D}\right) - \dfrac{x_{WS}{}^{\Phi}}{2}\sqrt{L_D{}^2 - (x_{WS}{}^{\Phi})^2}, & \text{Option 2: Markfort et al. (2010)} \\ f_{WS}[\Phi_{wind}] \, A_S, & \text{Option 3: user} - \text{defined shelter index} \end{cases}
$$

(26a-d)

where $A_{WS}$ is a user defined critical lake area for wind sheltering to dominate, $x_{WS}$ is a user defined sheltering distance, and $L_D$ the lake diameter ($L_D = 0.5(L_{crest} + W_{crest})$). For Option 1, the sheltering factor is held constant for the simulation based on the size of the lake, whereas options 2 and 3 require users to additionally input wind direction data, and a direction function, $f_{WS}[\Phi_{wind}]$, to allow for a variable sheltering effect over time. In the case of option 2, this function scales the sheltering distance, $x_{WS}$, as a function of wind direction, $x_{WS}{}^{\Phi} = x_{WS}(1 - min(f_{WS}[\Phi_{wind}], 1))$, whereas in the case of option 3 the function reads in an effective area scaling fraction directly based on a pre-calculated shelter index.

The ratio of the effective area to the total area of the lake, $A_E/A_S$, is then used to scale the wind speed data input by the user, $U_x$, as a means of capturing the average wind speed over the entire lake surface, such that $U_{10} = f_U U_x A_E/A_S$, where $f_U$ is a wind speed adjustment factor that can be used to assist calibration, or to correct the raw wind speed data to the reference height of 10 m.

Still-air limit: The above formulations apply when sufficient wind exists to create a defined boundary layer over the surface of the water. As the wind tends to zero (the 'still-air limit'), Eqs. 22-23 become less appropriate as they do not account for free convection directly from the water surface. This is a relatively important phenomenon for small lakes, cooling ponds and wetlands since they tend to have small fetches that limit the energy input from wind. These water bodies may also have large areas sheltered from the wind and will develop surface temperatures warmer than the atmosphere for considerable periods. Therefore, users can optionally augment Eqs. 22-23 with calculations for low wind speed conditions by calculating the evaporative and sensible heat flux values for both the given $U_{10}$ and for an assumed $U_{10} = 0$. The chosen value for the surface energy balance (as applied in Eq. 9) is found by taking the maximum value of the two calculations:

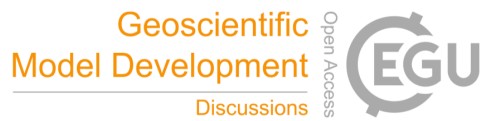

**Revision 26 Oct 2018**

$$\phi_X^* = \begin{cases} \max\left[\phi_X, \phi_{X_0}\right], & \text{Option 1: no} - \text{sheltering area} \\ \max\left[\phi_X, \phi_{X_0}\right] A_E/A_S + \phi_{X_0}(A_S - A_E)/A_S, & \text{Option 2: still} - \text{air sheltered area} \end{cases} \tag{27}$$

where $\phi_{X_0}$ is the zero-wind flux for either the evaporative or sensible heat flux ($\phi_{E_0}$ and $\phi_{H_0}$, respectively) and $\phi_X$ is calculated from Eqs. 22-23. The two zero-wind speed heat flux equations are from TVA (1972), but modified to return energy flux in SI units (W m$^{-2}$):

$$\phi_{E_0} = \rho_s \lambda_v \alpha_e (\vartheta_s - \vartheta_a) \tag{28a-b}$$
$$\phi_{H_0} = \alpha_h (T_s - T_a)$$

$$\alpha_e = 0.137 \ f_0 \ \frac{K_{air}}{c_a \rho_s} \left( g \frac{|\rho_a - \rho_o|}{\rho_a \ \nu_a \ D_a} \right)^{1/3} \tag{29a-b}$$

$$\alpha_h = 0.137 \ f_0 \ K_{air} \left( g \frac{|\rho_a - \rho_o|}{\rho_a \ \nu_a \ D_a} \right)^{1/3}$$

where $\vartheta = \kappa \ e/p$, with the appropriate vapour pressure values, *e*, for both surface and ambient atmospheric values. Here, $K_{air}$ is the molecular heat conductivity of air (J m$^{-1}$ s$^{-1}$ C$^{-1}$), $\nu_a$ is the kinematic viscosity of the air (m$^2$ s$^{-1}$), $\rho_o$ is the density of the saturated air at the water surface temperature, $\rho_s$ is the density of the surface water, $f_0$ is a dimensionless roughness correction coefficient for the lake surface, and $D_a$ is the molecular heat diffusivity of air (m$^2$ s$^{-1}$). Note that the impact of low wind speeds

on the drag coefficient is captured by the modified Charnock relation (Eq. A24-A25), which includes an additional term for the smooth flow transition (see also Figure A1).

### 2.4 Snow and ice dynamics

The extent of ice and snow cover can significantly impact the lake water balance and mixing regime, depending on the

prevailing environmental conditions. The algorithms for GLM ice and snow dynamics are based on previous ice modelling studies that adopt a three-layer scheme for resolving ice and snow, split into blue ice (or black ice), white ice (or snow ice) and snow layers (Patterson and Hamblin, 1988; Gu and Stefan, 1993; Rogers et al., 1995; Vavrus et al., 1996; Launiainen and Cheng, 1998; Magee et al., 2016). Blue ice is formed through direct freezing of lake water into ice, whereas white ice is generated in response to seeping of lake water onto the ice surface, when the mass of snow that can be supported by the

buoyancy of the ice cover is exceeded (see below; Rogers et al., 1995). The snow layer is subject to compaction and melting based on surface meteorological conditions and the ice layers are affected by the lake water temperature at the lower boundary.

Blue ice initially forms when the water at the lake surface goes below 0 $^{\circ}$C. Once fresh snow deposits on the surface it is subject to densification, which depends on the air temperature and amount of rainfall (Figure 6); the density of fresh snowfall

is determined as the ratio of measured snowfall height to water-equivalent height, with any values exceeding the assigned

**Revision 26 Oct 2018**

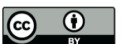

**Figure 5: A two-year times series of the simulated daily heat fluxes for the five example lakes, a-e, that were depicted in Figure 2. The heat balance components summarised are depicted schematically in the inset, as described in Section 2.3 and the "Heat Balance" line refers to the LHS of Eq. 9.**

maximum or minimum snow density (defaults: $\rho_{s,max}$ = 300 kg m$^{-3}$, $\rho_{s,min}$ = 50 kg m$^{-3}$) truncated to the appropriate limit. The snow compaction equation is based on the exponential decay formula of McKay (1968), with selection of snow compaction parameters based on air temperature and depending on whether rainfall or snowfall is being added. When the weight of snow

10   exceeds the buoyancy of the ice layer:

$$\Delta z_{snow}\, \rho_{snow} \;>\; \{\Delta z_{blue}\, (\rho_w - \rho_{blue}) + \Delta z_{white}\, (\rho_w - \rho_{white})\} \tag{30}$$



the ice will be forced downward, and lake water will seep into the snow layer leading to formation of white ice. This downward movement and white ice formation is limited to the snow amount matching the buoyancy deficit of the ice layer, and the lake height is reduced accordingly.

To capture the changing thickness of the ice and snow layers due to melting or freezing, the model employs a quasi steady state assumption to solve the heat transfer equation through the layers by assuming that the time scale for heat conduction is short relative to the time scale of changes in meteorological forcing (Patterson and Hamblin, 1988; Rogers et al., 1995). By assigning appropriate boundary conditions at the ice-atmosphere and the ice-water interfaces, the model computes the upward conductive heat flux through the ice and snow cover to the atmosphere, termed $\phi_0$.

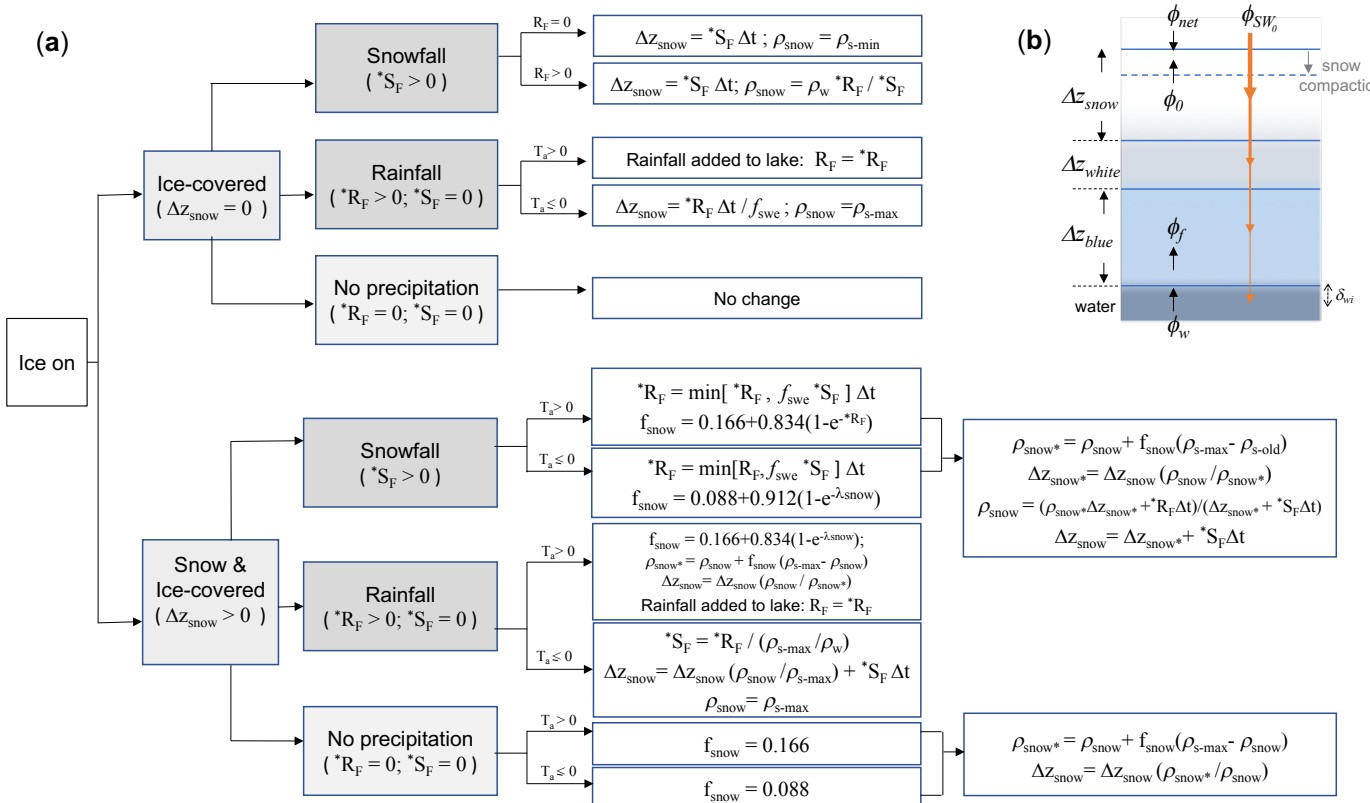

Figure 6: a) Decision tree describing updates to the snow cover each time step according to the amount of incident rainfall ($^*R_F$) and snowfall ($^*S_F$), air temperature ($T_a$) and snow compaction rules. b) Schematic of ice and snow layers and heat fluxes. Refer to text and Table 1 for definitions of other variables. Here $^*R_F = f_R R_x/c_{secday}$ and $^*S_F = f_S S_x/c_{secday}$ if ice cover is present, otherwise they are set to 0 and the model reverts to Eqs. 6-7.

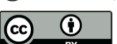

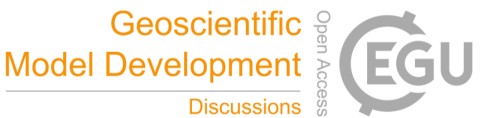

**Revision 26 Oct 2018**

At the upper surface (which could be ice or snow), a heat flux balance is employed to provide the condition for surface melting:

$$\phi_0[T_0] + \phi_{net}[T_0] = 0 \qquad\qquad T_0 < T_m \qquad\qquad (31)$$

$$\phi_{net}[T_0] = -\rho_{ice,snow} \lambda_f \frac{d\Delta z_{ice,snow}}{dt} \qquad\qquad T_0 = T_m \qquad\qquad (32)$$

where $\lambda_f$ is the latent heat of fusion, $\Delta z_{ice,snow}$ is the height of either the upper snow or ice layer, $\rho_{ice,snow}$ is the density of the relevant snow or ice layer, determined from the surface medium properties, $T_0$ is the temperature at the solid surface, $T_m$ is the melt-water temperature (0 $^\circ$C). $\phi_{net}[T_0]$ is the net incoming heat flux for non-penetrative radiation at the solid surface:

$$\phi_{net}[T_0] = \phi_{LWin} - \phi_{LWout}[T_0] + \phi_H[T_0] + \phi_E[T_0] + \phi_R[T_0] \qquad\qquad (33)$$

where the heat fluxes between the solid boundary and the atmosphere are calculated as outlined previously, but with modification for the determination of vapour pressure over ice or snow ( $e_{s_{ice}}[T_0] = e_s[T_0]\big(1 + 9.72{\times}10^{-3}T_0 + 4.2{\times}10^{-5}\,T_0{}^2\big)$; Jeong, 2009), and the addition of the rainfall heat flux, $\phi_R$, ($\phi_R = {}^*R_F\rho_w\lambda_f$ to capture the freezing effect if

$T_0 < T_m$, or simply as ${}^*R_F c_w\rho_w(T_a - T_0)$ if $T_0 = T_m$; Rogers et al., 1995). To determine the flow of heat through the layers, Rogers et al. (1995) derived:

$$\Lambda\big(\phi_0 - \phi_{SW_0}\big) = T_m - T_0 \qquad\qquad (34)$$

$$- \left\{ f_{VIS}\, \phi_{SW_0} \left( \frac{(1 - e^{-K_{s1}\Delta z_{snow}})}{K_{snow}K_{s1}} + e^{-K_{s1}\Delta z_{snow}} \frac{(1 - e^{-K_{w1}\Delta z_{white}})}{K_{w1}\Delta z_{white}} \right.\right.$$

$$\left.\left. + e^{-K_{s1}\Delta z_{snow}-K_{w1}\Delta z_{white}} \frac{(1 - e^{-K_{b1}\Delta z_{blue}})}{K_{b1}\Delta z_{blue}} \right) \right\}$$

$$- \left\{ \big(1 - f_{VIS}\big) \phi_{SW_0} \left( \frac{(1 - e^{-K_{s2}\Delta z_{snow}})}{K_{snow}K_{s2}} + e^{-K_{s2}\Delta z_{snow}} \frac{(1 - e^{-K_{w2}\Delta z_{white}})}{K_{w2}\Delta z_{white}} \right.\right.$$

$$\left.\left. + e^{-K_{s2}\Delta z_{snow}-K_{w2}\Delta z_{white}} \frac{(1 - e^{-K_{b2}\Delta z_{blue}})}{K_{b2}\Delta z_{blue}} \right) \right\}$$

$$+ \phi_{si}\Delta z_{snow}\Lambda - \frac{\phi_{si}\Delta z_{snow}{}^2}{2K_{snow}}$$

where $\Lambda = \left(\frac{\Delta z_{snow}}{K_{snow}} + \frac{\Delta z_{white}}{K_{white}} + \frac{\Delta z_{blue}}{K_{blue}}\right)$, $\phi_{SW_0}$ is the shortwave radiation penetrating the ice/snow surface, $K$ refers to the light

attenuation coefficient of the ice and snow components designated with subscripts $s$, $w$ and $b$ for snow, white ice and blue ice respectively, and the $\Delta z$ terms refers to the thickness of snow, white ice and blue ice. This is rearranged and solved for $T_0$ and $\phi_0$ by using a bilinear iteration until surface heat fluxes are balanced (i.e., $\phi_0[T_0] = -\phi_{net}[T_0]$) and $T_0$ is stable ($\pm$ 0.001 $^o$C). In the presence of ice (or snow) cover, a surface temperature $T_0 > T_m$ indicates that energy is available for melting. The amount of energy for melting is calculated by setting $T_0 = T_m$ to determine the reduced thickness of snow or ice (as shown in

Eq. 32). The estimation of $\phi_0$ applies an empirical equation to estimate snow conductivity, $K_{snow}$, from its density (Ashton, 1986):



**Revision 26 Oct 2018**

$$K_{snow} = 0.021 + 0.0042\, \rho_{snow} + (2.2{\times}10^{-9}\, \rho_{snow}{}^3) \tag{35}$$

The heat flux in the ice at the ice-water interface is:

$$\phi_f = \phi_0 \;- f_{VIS}\, \phi_{SW_0}\, (1 - \exp[-K_{s1}\Delta z_{snow} - K_{w1}\Delta z_{white} - K_{b1}\Delta z_{blue}]) \tag{36}$$
$$- (1 - f_{VIS})\, \phi_{SW_0}\, (1 - \exp[-K_{s2}\Delta z_{snow} - K_{w2}\Delta z_{white} - K_{b2}\Delta z_{blue}])$$
$$- \phi_{si}\, \Delta z_{snow}$$

where $\phi_{si}$ is a volumetric heat flux for the formation of white ice, which is given in Eq. 14 of Rogers et al. (1995) and ice and

snow light attenuation coefficients in GLM are also fixed to the same values as those given by Rogers et al. (1995). Shortwave albedo for the ice or snow surface (required for Eq. 10) is a function of surface medium (see Table 1 of Vavrus et al., 1996) with values varying from 0.08 to 0.6 for ice and from 0.08 to 0.7 for snow, depending on the surface temperature and the layer thicknesses; an additional scaling factor for the snow albedo, $f_\alpha$, is also implemented to aid calibration.

Accretion or ablation of blue ice occurs at the ice-water boundary based on the conductive heat flux from water into the ice, $\phi_w$, as given by the finite difference approximation:

$$\phi_w = -K_{water}\, \frac{\Delta T}{\delta_{wi}}, \tag{37}$$

where $K_{water}$ is the molecular conductivity of water (assuming the water is stagnant under the ice), and $\Delta T$ is the temperature difference between the surface water of the lake and the bottom of the blue ice layer, $T_m - T_s$. This occurs across an assigned length scale, $\delta_{wi}$, for which a value of 0.1–0.5 m is usual, based on the reasoning given in Rogers et al. (1995) and the typical

vertical water layer resolution of a model simulation (0.125–1.5 m). Note that a wide variation in techniques and values is used to determine the basal heat flux immediately beneath the ice pack (e.g., Harvey, 1990), which suggests that this may need careful consideration during calibration.

The imbalance between $\phi_f$ moving through the blue ice layer and the heat flux from the water into the ice, $\phi_w$, gives the rate

of change of ice thickness at the interface with water:

$$\frac{d\Delta z_{blue}}{dt} = \frac{\phi_f - \phi_w}{\rho_{blue}\, \lambda_f} \tag{38}$$

The ice thickness is set to its minimum value of 0.05 m, which is suggested by Patterson and Hamblin (1988) and Vavrus et al. (1996). The need for a minimum ice thickness relates primarily to horizontal variability of ice cover during the formation

and closure (ice-on) periods. The ice cover equations are discontinued and open water conditions are restored in the model when the thermodynamic balance first produces ice thickness < 0.05 m. Example outputs are shown in Figure 7; see also Yao et al. (2014) for a previous application.



**Revision 26 Oct 2018**

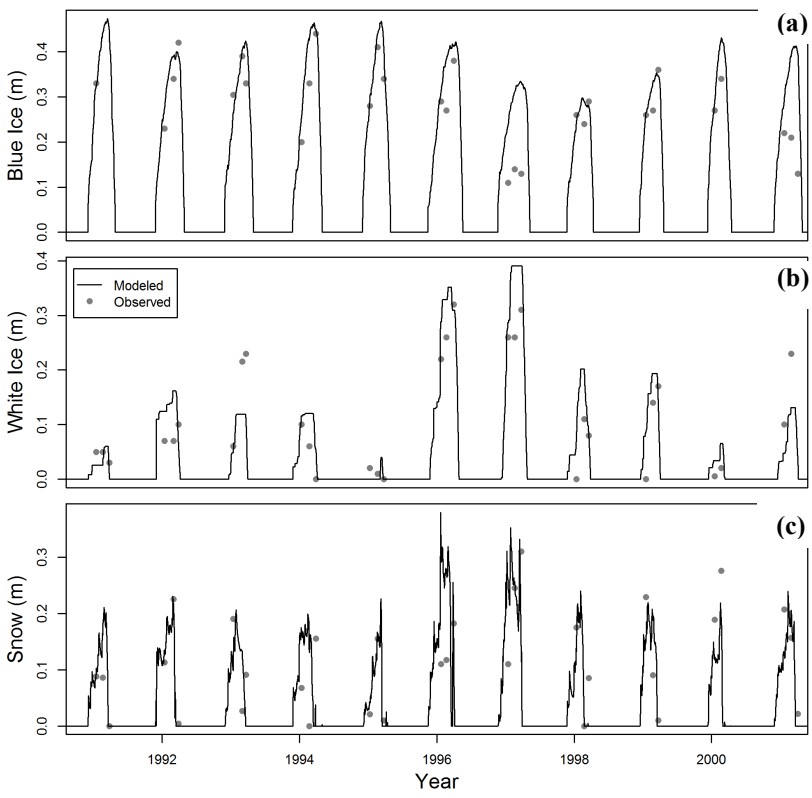

**Figure 7: Example of modelled and observed thickness of (a) blue ice, $\Delta z_{blue}$, (b) white ice, $\Delta z_{white}$, and (c) snow, $\Delta z_{snow}$, for Sparkling Lake, Wisconsin. Points are the average observed thicknesses.**

## 2.5 Sediment heating

The water column thermal budget may also be affected by heating or cooling from the soil/sediment below. For each layer, the rate of temperature change depends on the temperature gradient and the relative area of the layer volume in contact with bottom sediment:

$$c_w \rho_i \, \Delta V_i \frac{dT_i}{dt} = K_{soil} \frac{(T_{z_i} - T_i)}{\delta z_{soil}} \; (A_i - A_{i-1}) \tag{39}$$

where $K_{soil}$ is the soil/sediment thermal conductivity and $\delta z_{soil}$ is the length scale associated with the heat flux. The temperature of the bottom sediment varies seasonally, and also depending on its depth below the water surface, such that:

$$T_{z_i} = T_{z_{mean}} + \delta T_z \cos\left[\frac{2\pi}{365}(d - d_{T_z})\right] \tag{40}$$

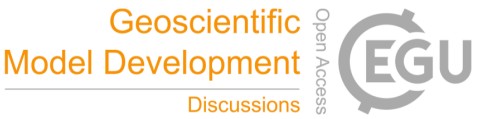

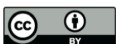

**Revision 26 Oct 2018**

where $z$ is the soil/sediment zone that the $i$ [th] layer overlays (see Section 4 for details), $T_z$ , is the temperature of this zone, $T_{z_{mean}}$ is the annual mean sediment zone temperature, $\delta T_z$ is the seasonal amplitude of the sediment temperature variation, and $d_{T_z}$ is the day of the year when the sediment temperature peaks. By defining different sediment zones, the model can therefore allow for a different mean and amplitude of littoral waters compared to deeper waters. A dynamic sediment

temperature diffusion model is also under development, which will be suitable when empirical data for the above parameters in Eq. 40 are not available.

## 2.6 Stratification and vertical mixing

Mixing processes in lakes are varied and depend upon the degree of meteorological and hydrological forcing, the lake
morphometry, and nature of thermal stratification experienced by the lake at the time of forcing. Numerous models adopt an eddy-diffusivity approach whereby mixing is captured using the advection-dispersion equation (e.g., Riley and Stefan, 1988). GLM adopts an energy balance approach as used in DYRESM whereby the mixing dynamics are based on estimating the amount of turbulent energy available, which is separately computed for the surface mixed layer (surface mixing), and for mixing below the thermocline (deep mixing).

### 2.6.1 Surface mixed layer

To compute mixing of layers, GLM works on the premise that the balance between the available energy, $E_{TKE}$, and the energy required for mixing to occur, $E_{PE}$, provides for the surface mixed layer (sml) deepening rate $dz_{sml}/dt$, where $z_{sml}$ is the depth from the surface to the bottom of the surface mixed layer. For an overview of the dynamics, readers are referred to early works on bulk mixed layer depth models by Kraus and Turner (1967) and Kim (1976), which were subsequently extended by
Imberger and Patterson (1981) and Spigel et al. (1986) as a basis for hydrodynamic model design. Using this approach, the available kinetic energy is calculated due to contributions from wind stirring, convective overturn, shear production between layers, and Kelvin-Helmholtz (K-H) billowing. Overall, the turbulent energy generated for mixing is summarised as (Hamilton and Schladow, 1997):

$$E_{TKE} = \underbrace{0.5 C_K (w_*^3)\,\Delta t}_{convective\ overturn} + \underbrace{0.5 C_K (C_W\,u_*^3)\,\Delta t}_{wind\ stirring} + 0.5\,C_S \underbrace{\left[ u_b^2 + \frac{u_b^2}{6}\frac{d\delta_{KH}}{dz_{sml}} + \frac{u_b\delta_{KH}}{3}\frac{du_b}{dz_{sml}} \right]}_{\substack{shear\ production \\ K-H\ production}} \Delta z_{k-1} \tag{41}$$

where $\delta_{KH}$ is the K-H billow length scale (described below), $u_b$ is the shear velocity at the interface of the mixed layer, and
$C_K$, $C_W$, and $C_S$ are mixing efficiency constants. For mixing to occur, the energy must be sufficient to lift up water in the layer below the bottom of the mixed layer, denoted here as the layer $k-1$, with thickness $\Delta z_{k-1}$, and accelerate it to the mixed layer velocity, $u_*$. This must also account for energy consumption associated with K-H billowing. In total, the energy required to entrain a layer into the mixed layer is expressed as $E_{PE}$:

$$E_{PE} = \left[ \underbrace{0.5 C_T (w_*^3 + C_W\,u_*^3)^{2/3}}_{acceleration} + \underbrace{\frac{\Delta\rho}{\rho_o} g\,z_{sml}}_{lifting} + \underbrace{\frac{g\delta_{KH}^2}{24\rho_o}\frac{d(\Delta\rho)}{dz_{sml}} + \frac{g\delta_{KH}\Delta\rho}{12\rho_o}\frac{d\delta_{KH}}{dz_{sml}}}_{K-H\ consumption} \right] \Delta z_{k-1} \tag{42}$$

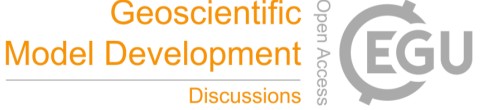

where $C_T$ is a mixing efficiency constant to account for unsteady turbulence. To numerically resolve Eq. 41 and 42 the model sequentially computes the different components of the above expressions with respect to the layer structure, checking the available energy relative to the required amount (depicted schematically in Figure 8). GLM follows the sequence of the algorithm presented in detail in Imberger and Patterson (1981), whereby layers are combined due to convection and wind stirring first, and then the resultant mixed layer properties are used when subsequently computing the extent of shear mixing and the effect of K-H instabilities. Plots indicating the role of mixing in shaping the thermal structure of the example lakes are shown in Figure 9.

To compute the mixing energy available due to convection, in the first step, the value for $w_*$ is calculated, which is the turbulent velocity scale associated with convection brought about by cooling at the air-water interface. The model adopts the algorithm used in Imberger and Patterson (1981), whereby the potential energy that would be released by mixed layer deepening is computed from the first moment of layer masses in the epilimnion (surface mixed layer) about the lake bottom, relative to the well-mixed condition. This is numerically computed by summing from the bottom-most layer of the epilimnion, $k$, up to $N_{LEV}$:

$$w_*^3 = \frac{g}{\rho_{sml}\, \Delta t} \sum_{i=k}^{N_{LEV}} \left[ (\rho_i\, \Delta z_i)\left(\widetilde{h_\iota} - \widetilde{h_{sml}}\right) \right] \tag{43}$$

where $\rho_{sml}$ is the mean density of the mixed layer including the combined layer, $\rho_i$ is the density of the $i$th layer, $\Delta z_i$ is the height difference between two consecutive layers within the loop ($\Delta z_i = h_i - h_{i-1}$), $\widetilde{h_\iota}$ is the mean height of layers to be mixed ($\widetilde{h_\iota} = 0.5[\, h_i + h_{i-1}]$), and $\widetilde{h_{sml}}$ is the epilimnion mid height, calculated as: $\widetilde{h_{sml}} = 0.5(h_S + h_{k-1})$ where $h_S$ is the height of the surface water level.

The velocity scale $u_*$ of the surface layer is associated with wind stress and calculated according to the wind strength:

$$u_*^2 = \frac{\rho_a}{\rho_{sml}} C_D U_{10}^2 \tag{44}$$

where $C_D$ is the drag coefficient for momentum. The model first checks to see if the energy available (Eqs. 43 and 44) can overcome the energy required to mix the $k-1$ layer into the surface mixed layer (Figure 8e); i.e., mixing of $k-1$ occurs if:

$$C_K(w_*^3 + C_W\, u_*^3)\, \Delta t \geq \left( g'_k\, z_{sml} + C_T(w_*^3 + C_W\, u_*^3)^{2/3} \right) \Delta z_{k-1} \tag{45}$$

where $g'_k = g\frac{\Delta\rho}{\rho_o}$ is the reduced gravity between the mixed layer and the $k-1$ layer, calculated as $g\,(\rho_{sml} - \rho_{k-1})/(0.5(\rho_{sml} + \rho_{k-1}))$. If the mixing condition is met, the layers are combined, the energy required to combine the layer is removed from the available energy, $k$ is adjusted, and the loop continues to the next layer. Where the mixing energy is substantial and the mixing reaches the bottom layer, then the mixing routine ends. If the condition in Eq. 45 is not met, then any residual energy is stored for the next time step, and the mixing algorithm continues as outlined below.

Once stirring is completed, mixing generated due to velocity shear is then accounted for. Parameterising the shear velocity, denoted $u_b$, in a one-dimensional model can be problematic, however the approximation used in Imberger and Patterson (1981) is applied, whereby:

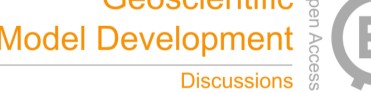

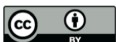

**Revision 26 Oct 2018**

$$u_b = \begin{cases} \dfrac{u_*^2 \Delta t}{z_{sml}} + u_{b_{old}}, & t \le t_b + \delta t_{shear} \\ 0, & t > t_b + \delta t_{shear} \end{cases} \tag{46}$$

where $u_{b_{old}}$, is from the previous time step, and zeroed between shear (wind) events. Therefore, this model yields a simple linear increase in the shear velocity over time for a constant wind stress. This is considered relative to $\delta t_{shear}$, which is the cut-off time, beyond which it is assumed no further shear-induced mixing occurs for that event. This cut-off time assumes use

of only the energy produced by shear at the interface during a period equivalent to half the basin-scale seiche duration, $\delta t_{iw}$, which can be modified to account for damping (Spigel, 1978):

$$\delta t_{shear} = \begin{cases} 1.59 \, \delta t_{iw} & \dfrac{\delta t_{damp}}{\delta t_{iw}} \ge 10 \\ \left(1 + 0.59 \left[1 - cosh\left(\dfrac{\delta t_{damp}}{\delta t_{iw}} - 1\right)^{-1}\right]\right) \delta t_{iw} & \dfrac{\delta t_{damp}}{\delta t_{iw}} < 10 \end{cases} \tag{47}$$

where $\delta t_{damp}$ is the time scale of damping. The wave period is approximated based on the stratification as $\delta t_{iw} = L_{META}/2c$, where $L_{META}$ is the length of the basin at the thermocline, calculated from $\sqrt{A_{k-1}(4/\pi)(L_{crest}/W_{crest})}$, whereby an ellipse

shape is assumed and $c$ is the internal wave speed:

$$c = \sqrt{|g'_{EH}| \frac{\delta_{epi}\delta_{hyp}}{(\delta_{epi} + \delta_{hyp})}} \tag{48}$$

where $\delta_{epi}$ and $\delta_{hyp}$ are characteristic vertical length scales associated with the epilimnion and hypolimnion:

$$\delta_{epi} = \frac{\Delta V_{epi}}{0.5(A_s + A_{k-1})} \; ; \; \delta_{hyp} = \frac{V_{k-1}}{0.5A_{k-1}} \tag{49}$$

where $\Delta V_{epi}$ and $V_{k-1}$ are the associated volumes.

The time for damping of internal waves in a two-layer system can be parameterised by estimating the length scale of the

15 oscillating boundary layer, through which the wave energy dissipates, and the period of the internal standing wave (see Spigel and Imberger, 1980):

$$\delta t_{damp} = \frac{\sqrt{\nu_w}}{c_{damp}\,\delta_{ss}} \frac{2(\delta_{epi} + \delta_{hyp})}{u_*^2} \sqrt{\frac{c}{2\,L_{META}} \frac{\delta_{hyp}}{\delta_{epi}}} \, (\delta_{epi} + \delta_{hyp}) \tag{50}$$

Once the velocity is computed from Eq. 46, the energy for mixing from velocity shear is compared to that required for lifting and accelerating the next layer down, and layers are combined if there is sufficient energy (Figure 6f), i.e. when:

$$0.5 \, C_S \left[\frac{u_b^2(\widetilde{z_{sml}} + \Delta\delta_{KH})}{6} + \frac{u_b\delta_{KH}\Delta u_b}{3}\right] + \left[g'_k\delta_{KH}\left(\frac{\delta_{KH}\Delta z_{k-1}}{24z_{sml}} - \frac{\Delta\delta_{KH}}{12}\right)\right] \tag{51}$$

$$\ge \left(g'_k \, z_{sml} + C_T(w_*^3 + C_W \, u_*^3)^{2/3}\right)\Delta z_{k-1}$$

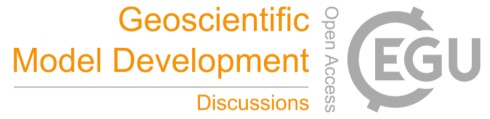

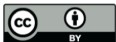

**Revision 26 Oct 2018**

where the billow length scale is $\delta_{KH} = C_{KH} u_b^2 / g'_{EH}$ and $\Delta\delta_{KH} = 2\,C_{KH}\,u_b\,\Delta u_b / g'_{EH}$; in this case the reduced gravity is computed from the difference between the bulk epilimnion and hypolimnion waters (Eq. 49), and $C_{KH}$ is a measure of the billow mixing efficiency.

5    Once energy from shear mixing is exhausted, the model checks the resultant density interface to see if it remains unstable to shear, such that K-H billows would be expected to form, i.e., if the metalimnion thickness is less than the K-H length scale, $\delta_{KH}$. If this condition is met, a six-layer set is created about the thermocline to relax the stratification, set to have a linear density profile over $\delta_{KH}$ (Figure 8g), and the surface layer properties are updated.

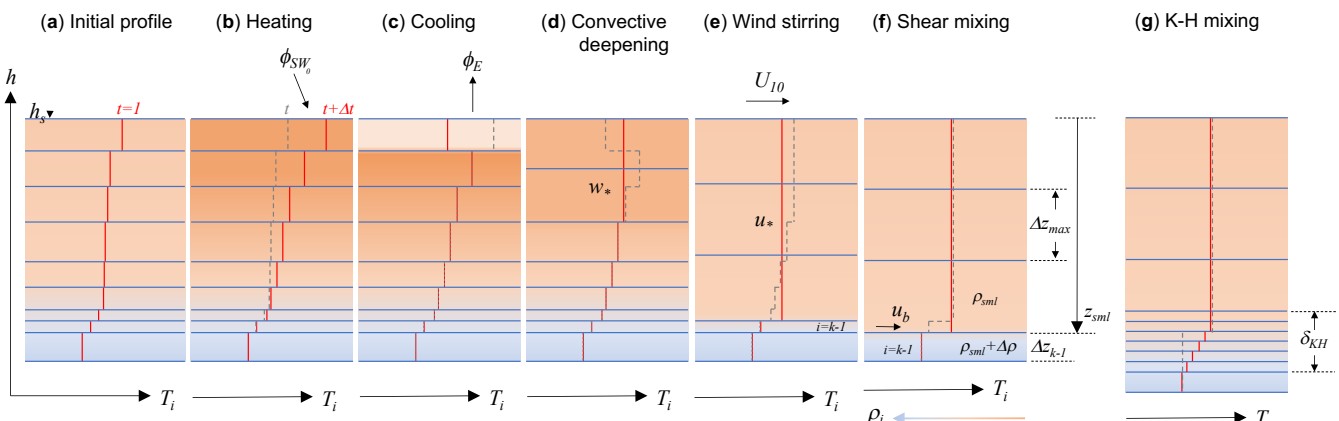

**Figure 8: Schematic depiction of layer changes during stratification and mixing. Consecutive panels show changes from a) the initial layer and thermal profile, to b) heating due to solar radiation, to c) evaporative cooling, which creates d) convective mixing, which is followed by e) a wind event causing stirring, and f) shear mixing across the thermocline. If the metalimnion remains unstable to shear it may be subjected to mixing from K-H billowing, which opens up the thermocline as depicted in panel (g).**

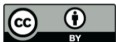

**Figure 9: A two-year time-series of the simulated temperature profiles for five example lakes, a-e, that range in size and hydrology. For more information about each lake and the simulation configuration refer to the Data availability section (refer also to Fig. 2 and 5). Sparkling Lake (b) also indicates the simulated depth of ice on the RHS scale.**

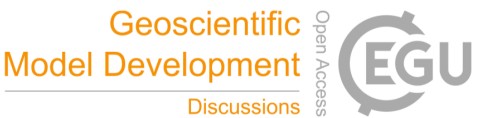

### 2.6.2 Deep mixing

Mixing below the thermocline in lakes, in the deeper hypolimnion, is modelled using a characteristic vertical diffusivity, $D_Z = D_\varepsilon + D_m$, where $D_m$ is a constant molecular diffusivity for scalars and $D_\varepsilon$ is the turbulent diffusivity. Three hypolimnetic mixing options are possible in GLM including: (1) no diffusivity, $D_Z = 0$, (2) a constant vertical diffusivity $D_Z$ over the water

5   depth below the surface mixed layer or (3) a derivation by Weinstock (1981) used in DYRESM, which is described as being suitable for regions displaying weak or strong stratification, whereby diffusivity increases with dissipation and decreases with heightened stratification. For the constant vertical diffusivity option, the coefficient $C_{HYP}$ is interpreted as the vertical diffusivity (m² s⁻¹), i.e., $D_z = C_{HYP}$, and applied uniformly below the surface mixed layer. For the Weinstock (1981) model, the diffusivity varies depending on the strength of stratification and the rate of turbulent dissipation, according to:

$$D_z = \frac{C_{HYP}\, \varepsilon_{TKE}}{N^2 + 0.6\, k_{TKE}{}^2\, u_*{}^2} \tag{52}$$

where $C_{HYP}$ in this case is the mixing efficiency of hypolimnetic TKE (~0.8 in Weinstock, 1981), $k_{TKE}$ is the turbulence wavenumber defined below, and $u_*$ is defined as above. The stratification strength is computed using the Brunt–Väisälä (buoyancy) frequency, $N^2$, defined for a given layer i as:

$$N^2{}_i = \frac{g\Delta\rho}{\rho\Delta z} \approx \frac{g(\rho_{i-2} - \rho_{i+2})}{\rho_{ref}(h_{i+2} - h_{i-2})} \tag{53}$$

where $\rho_{ref}$ is the average of the layer densities. This is computed from layer 3 upwards, averaging over the span of 5 layers, until the vertical density gradient exceeds a set tolerance. $N^2$ varies following an approximate normal distribution with height, centred at the height where the centre of buoyancy is located, computed each time-step from the 1ˢᵗ moment of the vertical $N^2$ distribution. Additionally, GLM estimates the vertical length scale associated with one standard deviation about the centre of

20   the $N^2$ distribution, denoted $\delta z_\sigma$.

The diffusivity increases in line with the turbulent dissipation rate. This can be complex to estimate in stratified lakes, however, GLM adopts a simple approach as described in Fischer et al. (1979) where a "net dissipation" is approximated by assuming dissipation is in equilibrium with energy inputs from external forcing:

$$\varepsilon_{TKE} \approx \overline{\varepsilon_{TKE}} = \varepsilon_{WIND} + \varepsilon_{INFLOW} \tag{54}$$

which is expanded and calculated per unit mass as:

$$\overline{\varepsilon_{TKE}} = \underbrace{\frac{1}{\tilde{V}_{N^2}\, \bar{\rho}}\, m\, C_D\, \rho_a\, U_{10}{}^3\, A_S}_{\text{rate of working by wind}} + \underbrace{\frac{1}{(\tilde{V}_{N^2} - \Delta V_S)\, \bar{\rho}} \sum_I^{N_{INF}} g\, \left(\rho_{ins_I} - \rho_{i_{ins_I}}\right)\, Q_{inf_{ins_I}} \left(\left(h_S - z_{inf_{ins_I}}\right) - h_{i_{ins_I}-1}\right)}_{\text{rate of work done by inflows}} \tag{55}$$

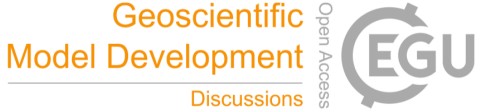

**Revision 26 Oct 2018**

where $\bar{\rho} = 0.5\left(\rho_1 + \rho_{N_{LEV}}\right)$ is the mean density of the water column. The work done by inflows is computed based on the flow rate, and considers the depth the inflow plunges to and the difference in density between the inflow water and layer into which it inserts, summed over all configured inflows (refer Section 2.7). These sources are normalised over the mass of water contained above the area of mixing. This is estimated as $\tilde{V}_{N^2}$, the fractional volume of the lake that is contained above the

height that corresponds to being one standard deviation below the centre of buoyancy, and is therefore the volume of the lake over which 85% of the $N^2$ variance is captured. The turbulence wavenumber, $k_{TKE}$, is then estimated from:

$$k_{TKE}^2 = \frac{c_{wn} A_s}{\tilde{V}_{N^2} \Delta z_{sml}} \tag{56}$$

where $c_{wn}$ is a coefficient. Since the dissipation is assumed to concentrate close to the level of strongest stratification, the

"mean" diffusivity suggested by Eq. 52 is modified to decay exponentially within the layers as they increase their distance from the thermocline:

$$D_{Z_i} = \begin{cases} 0 & h_i \geq (h_S - z_{sml}) \\ \dfrac{C_{HYP}\,\overline{\varepsilon_{TKE}}}{N^2_i + 0.6\,k_{TKE}^2\,u_*^2}\,exp\left[-\dfrac{(h_S - z_{sml} - h_i)^2}{\delta z_\sigma^2}\right] & h_i < (h_S - z_{sml}) \end{cases} \tag{57}$$

where $\delta z_\sigma$ is used to scale the depth over which the mixing is assumed to decay below the bottom of the mixed layer, $h_S -$

$z_{sml}$.

Once the diffusivity is approximated (either using a constant value or Eq. 57), the diffusion of any scalar, $C$ (including temperature, salinity and any water quality attributes), between two layers is numerically accounted for by the following mass transfer expressions:

$$C_{i+1} = \bar{C} - e^{-f_{dif}}\frac{\Delta z_i \Delta C}{(\Delta z_{i+1} + \Delta z_i)} \tag{58a,b}$$

$$C_i = \bar{C} + e^{-f_{dif}}\frac{\Delta z_{i+1}\Delta C}{(\Delta z_{i+1} + \Delta z_i)}$$

where $\bar{C}$ is the weighted mean concentration of $C$ for the two layers, and $\Delta C$ is the concentration difference between them. The smoothing function, $f_{dif}$, is related to the diffusivity according to:

$$f_{dif} = \frac{D_{Z_{i+1}} + D_{Z_i}}{(\Delta z_{i+1} + \Delta z_i)^2}\Delta t \tag{59}$$

and the above diffusion algorithm is run once up the water column and once down the water column as a simple explicit method for capturing diffusion of mass to both the upper and lower layers. An example of the effect of hypolimnetic mixing

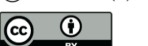

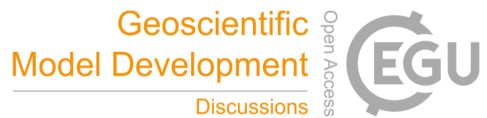

**Revision 26 Oct 2018**

on a hypothetical scalar concentration released from the sediment to the water column layers and accumulating in the hypolimnion is shown in Figure 10.

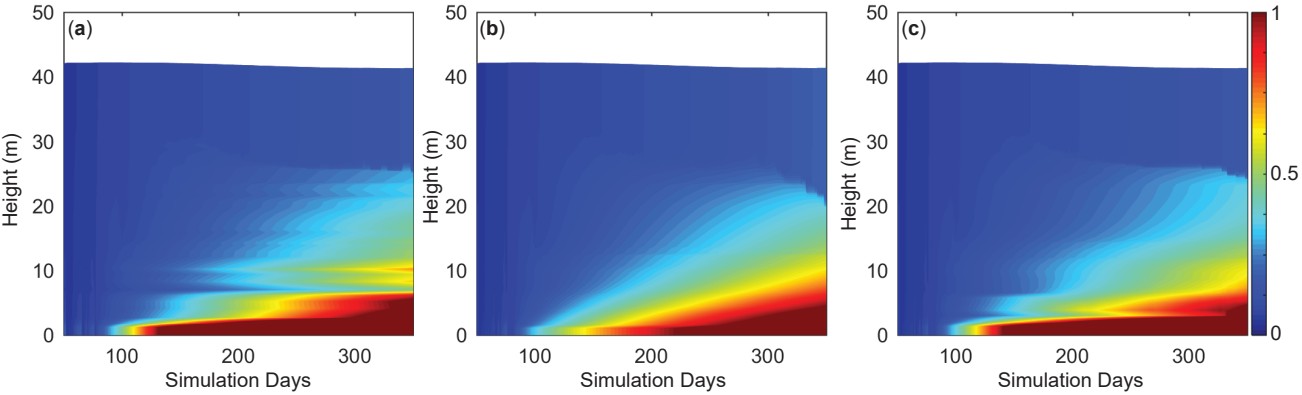

**Figure 10: Example simulations for Lake Kinneret showing the hypolimnetic concentration of a passive tracer (normalised units)**
**released from the sediment into the water layers at a constant rate for the case: a) without deep mixing, b) with a constant vertical diffusivity, $D_z = 2 \times 10^{-6}$ m² s⁻¹, and c) with the depth-dependent vertical diffusivity formulation (Eq. 57). The thermal structure for this case is in Figure 9e.**

**2.7 Inflows and outflows**

Aside from the surface fluxes of water described above, the water balance of a lake is controlled by inflows and outflows. Inflows can be specified as local runoff from the surrounding (dry) lake domain ($Q_R$ described separately above, Eq. 8), rivers entering at the surface of the lake that will be buoyant or plunge depending on their momentum and density (Section 2.7.1), or submerged inflows (including groundwater) that enter at depth (Section 2.7.2). Four options for outflows are included in GLM. These include withdrawals from a specified depth (Section 2.7.3), adaptive offtake (Section 2.7.4), vertical groundwater 15 seepage (Section 2.7.5), and river outflow/overflow from the surface of the lake (Section 2.7.6). Any number of lake inflows and outflows can be specified, and, except for the local runoff term, all are applied at a daily time step. Depending on the specific settings of each, these water fluxes can impact the volume of the individual layers, $\Delta V_i$, as well as the overall lake volume (Eq. 4). Inflows have a prescribed composition (temperature, salinity, and scalars), except local runoff which is assumed to be at air temperature, with zero salinity.

**2.7.1 River inflows**

As water from an inflowing river connects with a lake or reservoir environment, it will form a positively or negatively buoyant intrusion depending on the density of the incoming river water in the context of the water column stratification. As the inflow progresses towards insertion, it will entrain water at a rate depending on the turbulence created by the inflowing water mass (Fischer et al., 1979). For each configured inflow the entrainment coefficient, $E_{inf}$, is computed based on the bottom drag 25 being experienced by the inflowing water, $C_{D_{inf}}$, and the water stability using the approximation given in Imberger and Patterson (1981) as written in Ayala et al. (2014):



**Revision 26 Oct 2018**

$$E_{inf} = 1.6 \frac{C_{D_{inf}}^{3/2}}{Ri_{inf}} \tag{60}$$

where the inflow Richardson number, $Ri_{inf}$, characterises the stability of the water in the context of the inflow. Imberger and Patterson (1981) derived a simple estimate of $Ri_{inf}$ based on the drag coefficient by assuming the velocity (and Froude number) is typically small and considering the channel geometry, which is adapted in GLM as:

$$Ri_{inf} = \frac{C_{D_{inf}} \left( 1 + 0.21 \sqrt{C_{D_{inf}}} \sin \alpha_{inf} \right)}{\sin \alpha_{inf} \tan \Phi_{inf}} \tag{61}$$

where $\alpha_{inf}$ is the stream half angle assuming an approximate triangular cross-section, and $\phi_{inf}$ is the angle of the slope of the inflow thalweg relative to horizontal in the region where it meets the water body (Figure 11). Therefore, using Eq. 60 and 61, a simple approximation of stream geometry and bottom roughness can be used to parameterise the characteristic rate of entrainment as it enters the waterbody.

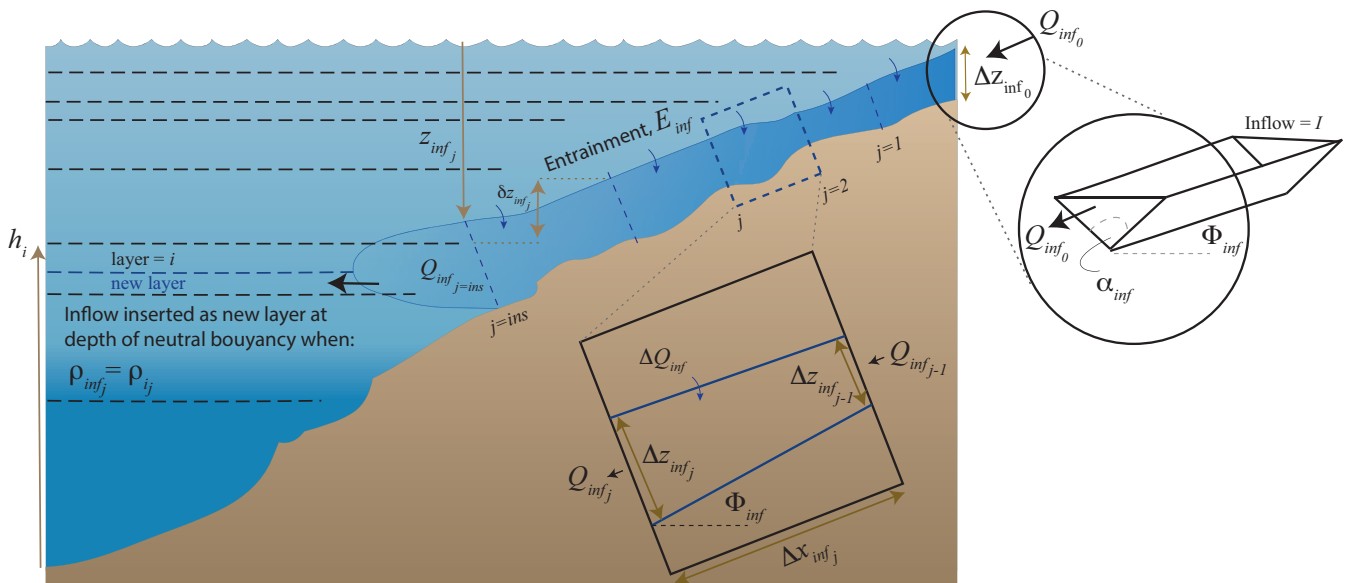

**Figure 11: Schematic showing inflow insertion depth, entrainment, $E_{inf}$, slope, $\Phi_{inf}$ and bottom slope angle, $\alpha_{inf}$, of an inflowing river, $I$, entering with at a flow rate of $Q_{inf_0}$, and estimated starting thickness of $\Delta z_{inf_0}$.**

On entry, the inflow algorithm captures two phases: first, the inflowing water crosses the layers of the lake until it reaches a level of neutral buoyancy, and second, it then undergoes insertion. In the first part of the algorithm, the daily inflow parcel is tracked down the lake-bed and its mixing with layers is updated until it is deemed ready for insertion. The initial estimate of the intrusion thickness, $\Delta z_{inf_0}$, is computed as in Antenucci et al. (2005) and Ayala et al. (2014):

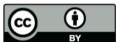


**Revision 26 Oct 2018**

$$\Delta z_{inf_0} = \left( 2 \, \frac{Ri_{inf}}{g'_{inf}} \left( \frac{Q_{inf_0}}{tan \, \alpha_{inf}} \right)^2 \right)^{1/5} \tag{62}$$

where $Q_{inf_0} = f_{inf} \, Q_{inf_x}/c_{secday}$ is the inflow discharge entering the domain, based on the data provided as a boundary condition, $Q_{inf_x}$, and $g'_{inf}$ is the reduced gravity of the inflow as it enters:

$$g'_{inf} = g \, \frac{(\rho_{inf} - \rho_s)}{\rho_s} \tag{63}$$

where $\rho_{inf}$ is the density of the inflow, computed from the supplied inflow properties of temperature and salinity ($T_{inf_x}, S_{inf_x}$), and $\rho_s$ is the density of the surface layer. If the inflowing water is deemed to be positively buoyant $(\rho_{inf} < \rho_s)$, or the model

only has one layer ($N_{LEV} = 1$), then the inflow water over the daily time step is added to the surface layer volume ($\Delta V_{N_{LEV}} = Q_{inf_0} \, \Delta t_d$), and $h_s$ is updated accordingly. Otherwise, this inflow volume is treated as a parcel which travels down through the lake layers, and its properties are subsequently incremented over each time step, $j$, (currently daily) until it inserts. The thickness of an inflow parcel increases over each increment due to entrainment, assuming:

$$\Delta z_{inf_j} = 1.2 \, E_{inf} \, \Delta x_{inf_j} + \Delta z_{inf_{j-1}} \tag{64}$$

where $\Delta z_{inf_j}$ is the inflow thickness and $\Delta x_{inf_j}$ is the distance travelled by the inflowing water parcel over the $j$ th time step.

The distance travelled is estimated based on the change in the vertical height of the inflow, $\delta z_{inf}$, and the angle of the inflow river, $\phi_{inf}$, as given by:

$$\Delta x_{inf_j} = \frac{\delta z_{inf_j}}{\sin \Phi_{inf}}. \tag{65}$$

The vertical excursion for the step is approximated as the difference between its starting height and the bottom of the nearest layer that it sits above, $h_{i_{j-1}}$, such that $\delta z_{inf_j} = \left( h_s - z_{inf_{j-1}} \right) - h_{i_{j-1}}$, where $z_{inf_{j-1}}$ is the depth of the inflow from the surface at the start of the time step, and this is subsequently updated from $z_{inf_j} = z_{inf_{j-1}} + \Delta x_{inf_j} \sin \Phi_{inf}$. The average

velocity of the inflow parcel is updated based on the incoming flow rate from:

$$u_{inf_j} = \frac{Q_{inf_{j-1}}}{\left( \Delta z_{inf_j} \right)^2 \tan \alpha_{inf}} \tag{66}$$

where the numerator links the relationship between inflow height and channel width in order to define the cross-sectional area of the flow. This velocity is used to estimate the time scale of transport of the parcel $(\delta t_d = \Delta x_{inf_j}/u_{inf_j})$. Following conservation of mass, the flow is estimated to increase according to Fischer et al. (1979) (see also Antenucci et al., 2005):

$$\Delta Q_{inf_j} = Q_{inf_{j-1}} \left[ \left( \frac{\Delta z_{inf_j}}{\Delta z_{inf_{j-1}}} \right)^{5/3} - 1 \right] \tag{67}$$

whereby $\Delta Q_{inf_j}$ is removed from the volume of the corresponding layer, $i_j$, and added to the previous time-step inflow $Q_{inf_{j-1}}$

to capture the entrainment effect on the inflow for the next increment. The properties associated with $\Delta Q_{inf_j}$ are assumed to match those of the $i_j$ layer, and mixed into the inflow parcel, to update temperature, salinity and density, $\rho_{inf_j}$. The inflow travel algorithm (Eqs. 62-67) increments through $j$ until the density of the inflow first reaches its depth of neutral buoyancy:

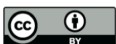

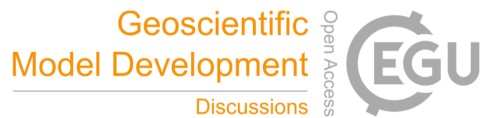

**Revision 26 Oct 2018**

$\rho_{inf_j} \le \rho_{i_j}$. Once this condition is met, the insertion depth is defined as $z_{inf_{insI}}$, its density as $\rho_{inf_{insI}}$, and the second part of the algorithm then creates a new layer of thickness dependent on the inflow's volume at that time, $Q_{inf_{insI}} c_{secday}$, which includes the successive layer additions from entrainment; Eq. 67.

5 Since a new inflow parcel is created each day, and the user may configure multiple inflows, $N_{INF}$, a complex set of parcels being tracked via Eqs. 60-67, and a queue of new layers to be inserted, is created. Following creation of a new layer for an inflow parcel, $N_{LEV}$ is incremented and all layer heights above the new layer are updated, paying attention to the lake hypsography. The new inflow layer is then subject to the thickness limits criteria within the layer limit checking routine and may amalgamate with adjacent layers or be divided into thinner layers.

Aside from importing mass into the lake, river inflows also contribute turbulent kinetic energy that may dissipate in the hypolimnion, as discussed in Sect 2.6.2 (e.g., see Eq. 55), and they contribute to the scalar transport in the water column by adding mass contained within the inserted inflow parcels, and contributing to mixing of properties via entrainment as described above (Figure 12a); see also Fenocchi et al. (2017).

15 **2.7.2 Submerged inflows**

Submerged inflows are inserted at the user-specified depth, $h_{inf}$, with zero entrainment by utilising the second part of the algorithm described in Section 2.7.1. Once the submerged inflow volume is added as a new layer it may then be mixed with adjacent layers (above or below) depending on the density difference and layer thickness criteria (Figure 12b). This option can be used across one or more inflow elevations to account for groundwater input to a lake, or for capturing a piped inflow, for 20 example.

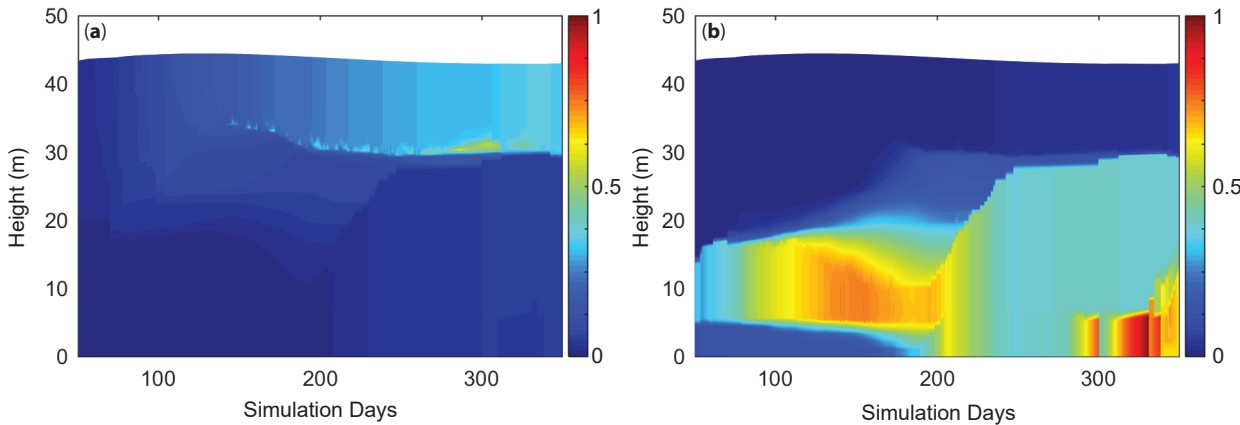

**Figure 12: Example simulations with tracers demonstrating inflow insertion example for the case where a) the inflow was set as a surface river inflow and subject to the insertion algorithm (Eqs. 60-67) prior to insertion, and b) the inflow was set as a submerged**
25 **inflow at a specified height ($h_{inf}$ = 5 m). Once entering the water column, the tracer, $C$, is subject to mixing during inflow entrainment in case (a), and by surface and/or deep mixing once inserted, for both cases (a) and (b). The colour scale represents an arbitrary inflow concentration which entered with a value of 1.**



**Revision 26 Oct 2018**

### 2.7.3 Withdrawals

Outflows from a specific depth can include outlets from a dam wall offtake or other piped withdrawal, or removal of water that may be lost due to groundwater recharge. For a stratified water column, the water will be removed from the layer corresponding to the specified withdrawal height, $h_{outf}$, as well as layers above or below, depending on the strength of

discharge and stability of the water column. Accordingly, the model assumes an algorithm where the thickness of the withdrawal layer is dependent on the internal Froude ($Fr$) and Grashof ($Gr$) numbers, and the parameter, $R$ (see Fischer et al., 1979; Imberger and Patterson, 1981):

$$Fr = \frac{f_{outf}\ Q_{outf_x}/c_{secday}}{N_{outf}\ W_{outf}\ L_{outf}^2} \tag{68}$$

$$Gr = \frac{N_{outf}^2 A_{outf}^2}{D_{outf}^2} \tag{69}$$

$$R = Fr Gr^{1/3} \tag{70}$$

where $W_{outf}$, $L_{outf}$ and $A_{outf}$ are the width, length and area of the lake at the outlet elevation, and $D_{outf}^2$ is the vertical diffusivity averaged over the layers corresponding to the withdrawal thickness, $\delta_{outf}$ (described below). To calculate the width

and length of the lake at the height of the outflow, it is assumed, firstly, that the lake shape can be approximated as an ellipse, and secondly, that the ratio of length to width at the height of the outflow is the same as that at the lake crest. The length of the lake at the outflow height, $L_{outf}$ and the lake width, $W_{outf}$ are given by:

$$L_{outf} = \sqrt{A_{outf}\ \frac{4}{\pi}\frac{L_{crest}}{W_{crest}}} \tag{71}$$

$$W_{outf} = L_{outf}\ \frac{W_{crest}}{L_{crest}} \tag{72}$$

where $A_{outf}$ is the area of the lake at the outflow height, $L_{crest}$ is the length and $W_{crest}$ the width of the lake at the crest height.

The thickness of the withdrawal layer is calculated depending on the value of $R$ (Fischer et al., 1979), such that:

$$\delta_{outf} = \begin{cases} 2L_{outf}\ Gr^{-1/6} & R \le 1 \\ 2L_{outf}\ Fr^{1/2} & R > 1 \end{cases} \tag{73}$$

If stratification is apparent near $h_{outf}$, either above or below this elevation, then the thickness computed in Eq. 73 may not be symmetric about the offtake level (Imberger and Patterson, 1981); therefore the algorithm separately computes the thickness of the withdrawal layer above and below, denoted $\delta_{outf_{top}}$ and $\delta_{outf_{bot}}$, respectively. The Brunt-Väisälä frequency is averaged over the relevant thickness, $N_{outf}^2$, and calculated as:

$$N_{outf}^2 = \frac{g}{\delta_{outf}}\frac{\rho_{outf} - \rho_i}{\rho_{outf}} \tag{74}$$

where $\rho_{outf}$ is the density of the layer corresponding to the height of the withdrawal, $i_{outf}$, and $\rho_i$ is the density of the water column at the edge of the withdrawal layer, as determined below. The flow of water taken from each layer influenced by the withdrawal, $Q_{outf_i}$, either above or below the layer of the outlet elevation, requires identification of the upper and lower-most

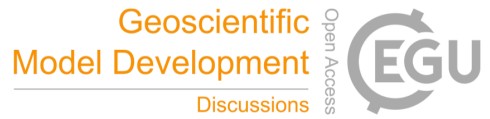

layer indices influenced by the outflow, denoted $i_{top}$ and $i_{bot}$. Once the layer range is defined, $Q_{outf_i}$ is computed for the layers between $i_{outf}$ and $i_{top}$, and $i_{outf}$ and $i_{bot}$, by partitioning the total outflow using a function to calculate the proportion of water withdrawn from any layer that fits the region of water drawn in a given time ($Q_{outf_i} = f\left[f_{outf}\, Q_{outf_x}/c_{secday}\,, h_i, h_{i-1}, h_{outf}, \delta_{outf_{bot}}, \delta_{outf_{top}}\right]$; see Imberger and Patterson, 1981, Eq. 65). Given that

users configure any height for a withdrawal outlet and flow rates of variable strength, the upper ($h_{outf} + \delta_{outf_{top}}$) and lower ($h_{outf} - \delta_{outf_{bot}}$) elevation limits computed by the algorithm are limited to the lake surface layer or bottom layer. Once computed, the volumes are removed from the identified layer set, and their height and volumes updated accordingly. $Q_{outf_i}$ is constrained within the model to ensure no more than 90% of a layer can be removed in a single time step. Depending on the fractional contribution from each of the layers the water is withdrawn from, the water taken will have the associated weighted

average of the relevant scalar concentrations ($T_{outf}, S_{outf}, C_{outf}$ ) which are reported in the outlet file for the particular withdrawal. This routine is repeated for each withdrawal considered, denoted $O$, and the model optionally produces a summary file of the combined outflow water and its properties.

### 2.7.4 Adaptive offtake dynamics

For reservoir applications, a special outflow option has been implemented that extends the dynamics in Section 2.7.3 to

simulate an adaptive offtake or selective withdrawal. This approach is used for accommodating flexible reservoir withdrawal regimes and their effects on the thermal structure within a reservoir. For this option, a target temperature is specified by the user and GLM identifies the corresponding withdrawal height within a predefined (facility) range to meet this target temperature during the runtime of the simulation, i.e., the withdrawal height adaptively follows the thermal stratification in the reservoir. The target temperature can be defined as a constant temperature or a time series (via a *.csv file), such as a

measured water temperature from an upstream river that could be used to plan environmental releases from the reservoir to the downstream river. The selected height of the adaptive offtake is printed out in a *.txt file for assisting reservoir operation. In addition to the basic adaptive offtake function, GLM can also simulate withdrawal mixing, i.e., water from the adaptive offtake is mixed with water from another predefined height (e.g., the bottom outlet). For this option, the discharges at both locations need to be predefined by the user (via the standard outflow *.csv files) and GLM chooses the adaptive withdrawal

from a height, where the water temperature is such that the resulting mixing temperature meets the target temperature. This withdrawal mixing is a common strategy in reservoir operation where deep water withdrawal and temperature control are required simultaneously to prevent deleterious downstream impacts.

An example of the adaptive offtake function with and without withdrawal mixing, assuming a constant water temperature of

14 °C for the outflow water, shows that GLM is able to deliver a constant outflow temperature of 14 °C during the stratified period (Figure 13). In winter, when the water column is cooler than 14 °C, the model withdraws surface water. The adaptive offtake functionality can be used in a stand-alone mode or also linked to the dissolved oxygen concentration (when operated with the coupled water quality model AED2, see Section 4). In the latter case, the effect of the withdrawal regime on the oxygen dynamics in the hypolimnion can be simulated (see Weber et al., 2017). In this setting, the simulated hypolimnetic



dissolved oxygen concentration at a specified height is checked against a user-defined critical threshold. If the hypolimnetic oxygen falls below the critical threshold, the height of the adaptive offtake will be automatically switched to a defined height (usually deep outlets in order to remove the oxygen-depleted water) to withdraw water from this layer, until the oxygen concentrations have recovered.

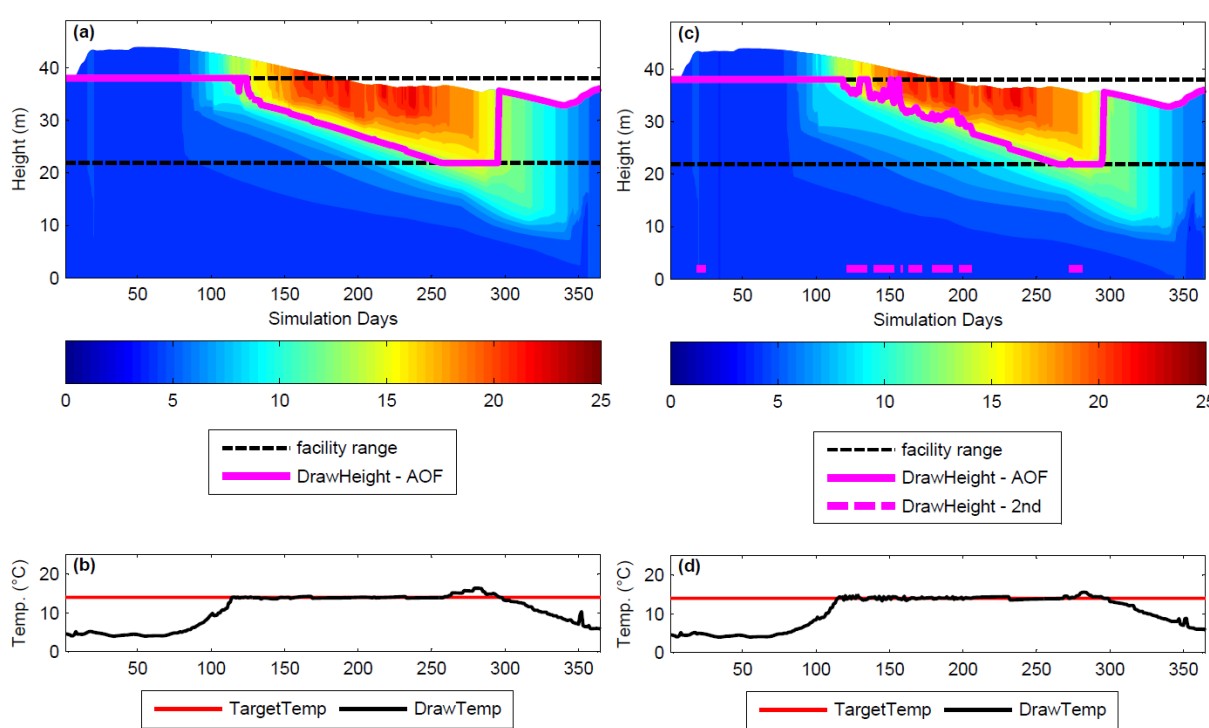

**Figure 13: Adaptive offtake reservoir simulation; water temperatures of the adaptive offtake model assuming a constant target temperature of 14 °C (a,b) without and (c,d) with mixing with the bottom outlet withdrawal. The black dashed line (a,c) represents**
10 **the height range of the variable withdrawal facility (AOF) and the magenta lines the adaptive offtake and second withdrawal height (here: bottom outlet). In the scenario with the second withdrawal activated (c), the bottom outlet was periodically opened during flooding conditions. Panels (b) and (d) indicate where the actual withdrawal temperature (DrawTemp, $T_{outf}$) was able to meet the target (TargetTemp).**

15 **2.7.5 Seepage**

Seepage of water from the lake can also be configured within the model, for example, as might be required in a wetland simulation or for small reservoirs perched above the water table that experience leakage to the soil below. The seepage rate, $Q_{seepage}$, can be assumed constant or dependent on the overlying lake head:

$$Q_{seepage} = \begin{cases} -GA_S/c_{secday}, & \text{Option 1: constant rate} \\ -\left(\dfrac{K_{seep}}{\delta z_{soil}}\right)\dfrac{A_S\, h_S}{c_{secday}}, & \text{Option 2: Darcy flux based on water height} \end{cases}$$

(75)





**Revision 26 Oct 2018**

where $G$ is the seepage rate (m day$^{-1}$), $K_{seep}$ is the sediment hydraulic conductivity (m day$^{-1}$) and $\delta z_{soil}$ is an assumed sediment thickness over which the seepage is assumed to occur. The water leaving the lake is treated as a "vertical withdrawal" whereby the water exits via the bottom-most layer(s), and the amount $\Delta V_G = Q_{seepage}\Delta t_d$, is generally all taken from the bottom-most layer ($i = 1$), however, it is constrained within the model to ensure no more than 90% of the layer can be reduced in any one time step; where $\Delta V_G > 0.9 V_{i=1}$ then the routine sequentially loops up through the above layers until enough lake volume has been identified to cover the seepage demand. Once the individual layer volumes are incremented due to the seepage flux, $\Delta V_{G_i}$, the heights of all layers ($h_1 : h_s$) are re-computed based on the hypsographic curve using $h_i = f[V_i]$. Where seepage reduces the lake below 0.05 m, the lake becomes dry and will continue to have zero volume until new inputs from rain or inflows (e.g., Figure 9a).

**2.7.6 Overflows**

Once the lake volume exceeds the maximum volume, the excess water is assumed to leave the domain as an overflow. The flow rate, $Q_{ovfl}$, is computed based on the interim volume, $V_S^*$, prior to the end of the daily time-step, where $V_S^* = V_S^t + \Delta h_S A_S + \Delta t \left( \sum_I^{NINF} Q_{inf_{0_I}} - \sum_O^{NOUT} Q_{outf_O} - Q_{seepage} \right)$, and $\Delta h_s$ is the cumulative change in the daily water level over the day. Users can optionally also specify a crest elevation which lies below the elevation of maximum lake volume, and support a rating curve linking the height of water above the crest level with the overflow volume:

$$Q_{weir} = \begin{cases} 0, & V_S^* \leq V_{crest} \\ \dfrac{2}{3} C_{D_{weir}} \sqrt{2g} \, W_{weir} \, (h_S^* - h_{crest})^{3/2}, & V_S^* > V_{crest} \end{cases} \qquad (76)$$

where $h_S^*$ is the interim update to the water surface height prior to the overflow computation, $C_{D_{weir}}$ is a coefficient related to the drag of the weir, $W_{weir}$ is the width of the weir crest and $h_{crest}$ is the height of the crest level. The overflow rate is then computed as the sum of the flow over the weir crest and the volume of water exceeding the volume of the domain:

$$Q_{ovfl} = \begin{cases} Q_{weir}, & V_S^* \leq V_{max} \\ Q_{weir} + (V_S^* - V_{max})/\Delta t_d, & V_S^* > V_{max} \end{cases} \qquad (77)$$

If no crest is configured below the maximum lake height, then Eq. 77 assumes $Q_{weir} = 0$.

**2.8 Wave height and bottom stress**

Resuspension of sediment from the bed of lakes depends on the stresses created by water movement across the lake bottom. Wind induced resuspension in particular is sporadic and occurs as the waves at the water surface create oscillatory currents that propagate down to the lake-bed and exceed a critical threshold. The wave climate that exists on a lake can be complex and depend on the fetch over which the wind has blown, the time period over which the wind has blown, and complicating factors such as wind-sheltering and variations in bottom topography. The horizontally averaged nature of GLM means that only a single set of wave characteristics across the entire lake surface can be computed for a given time-step and these are assumed to be at steady-state. Note that GLM does not predict resuspension and sediment concentration directly, but computes the bottom shear stress for later use by coupled sediment and water quality modules. Since each layer has a component that is

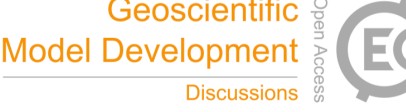

considered to overlay sediment (Section 4), the stress experienced at the sediment-water interface is able to approximated as a function of depth in relation to the surface wave climate. The model can therefore identify the depth-range and areal extent where there is potential for bed-sediment resuspension to occur, i.e., by computing the area of the lake over which the bed shear stress exceeds some critical value required for resuspension.

The model estimates the surface wave conditions using a simple, fetch-based, steady state wave model (Laenen and LeTourneau, 1996; Ji 2008). The average wave geometry (wave period, significant wave height and wave length), is predicted based on the wind speed and fetch over which the waves develop (Figure 14), whereby the average fetch is approximated in the one-dimensional model formulation from the lake area:

$$F = 2\sqrt{A_s/\pi} \tag{78}$$

and the wave period, $\delta t_{wave}$, is calculated from fetch based on:

$$\delta t_{wave} = 7.54 \left(\frac{U_{10}}{g}\right) tanh[\xi] \, tanh\left[\frac{0.0379\left(\frac{gF}{U_{10}^2}\right)^{0.333}}{tanh[\xi]}\right] \tag{79}$$

where:

$$\xi = 0.833\left(\frac{gz_{avg}}{U_{10}^2}\right)^{0.375} \tag{80}$$

and $z_{avg} = h_S/2$ is the average lake depth. The typical wave length is then estimated from:

$$\delta x_{wave} = \frac{g(\delta t_{wave})^2}{2\pi} \, tanh\left[\frac{2\pi \, z_{avg}}{\frac{g(\delta t_{wave})^2}{2\pi}}\right] \tag{81}$$

and the significant wave height from:

$$\delta z_{wave} = 0.283\left(\frac{U_{10}^2}{g}\right) tanh[\zeta] tanh\left[\frac{0.00565\left(\frac{gF}{U_{10}^2}\right)^{0.5}}{tanh[\zeta]}\right] \tag{82}$$

15 where

$$\zeta = 0.53\left(\frac{gz_{avg}}{U_{10}^2}\right)^{0.75} \tag{83}$$

Based on these properties the orbital wave velocity at the surface can be translated down the depth of the water column, such that in the $i$<sup>th</sup> layer it is calculated as (Sheng and Lick, 1979):

$$U_{orb_i} = \frac{\pi \, \delta z_{wave}}{\delta t_{wave} \, sinh\left[\frac{2\pi \, z_{i-1}}{\delta x_{wave}}\right]} \tag{84}$$

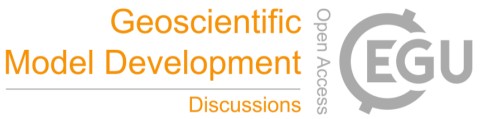

**Revision 26 Oct 2018**

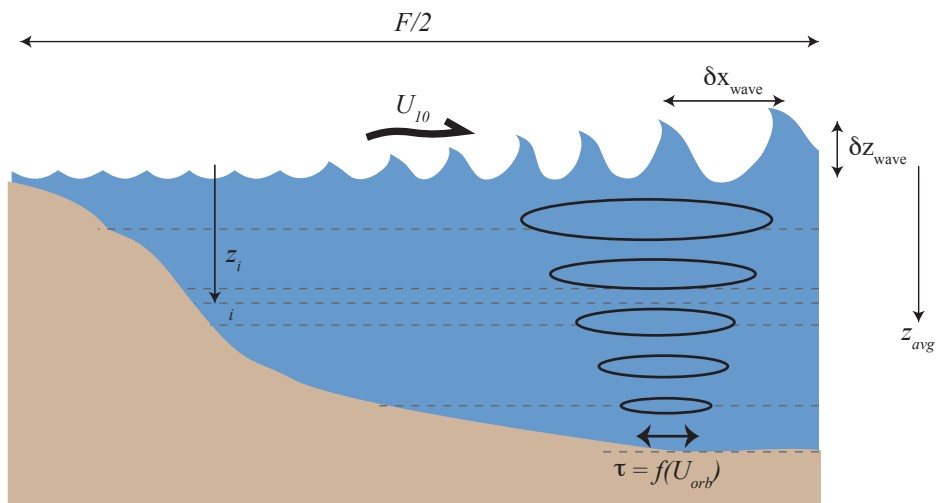

**Figure 14: Schematic of the wave estimation approach depicting the lake fetch, surface wind speed, wave height, wavelength and bottom stress created by the orbital velocity.**

For each layer, the total shear stress experienced at the lake bed portion of that layer (equivalent in area to $A_i - A_{i-1}$) is calculated from:

$$\tau_i = \frac{1}{2} \rho_i \left( f_{w_i} U_{orb_i}{}^2 + f_{c_i} U_{m_i}{}^2 \right) \tag{85}$$

where $U_m$ is the mean layer velocity, which for simplicity is assumed based on the velocity estimate made during the mixing

10   calculations (Eq. 44) in the surface mixed layer, such that:

$$U_{m_i} = \begin{cases} u_*, & i \geq k \\ 0, & i < k \end{cases} \tag{86}$$

The friction factors depend upon the characteristic particle diameter of the lake bottom sediments, $\delta_{ss}$ and the fluid velocity. For the current induced stress, we compute $f_{c_i} = 0.24/\log\left[12 z_{avg}/2.5 \delta_{ss_{z_i}}\right]$, and for waves (Kleinhans and Grasmeijer,

15   2006):

$$f_{w_i} = \exp\left[-5.977 + 5.213 \left(\frac{U_{orb_i} \, \delta t_{wave}}{5\pi \, \delta_{ss_{z_i}}}\right)^{-0.194}\right] \tag{87}$$

where $\delta_{ss_{z_i}}$ is specific for each layer $i$, depending on which sediment zone it overlays (see Section 4). The current and wave induced stresses at the lake bottom manifest differently within the lake, as demonstrated in Figure 15 for a shallow lake.

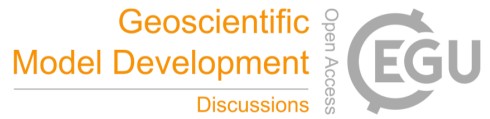

**Revision 26 Oct 2018**

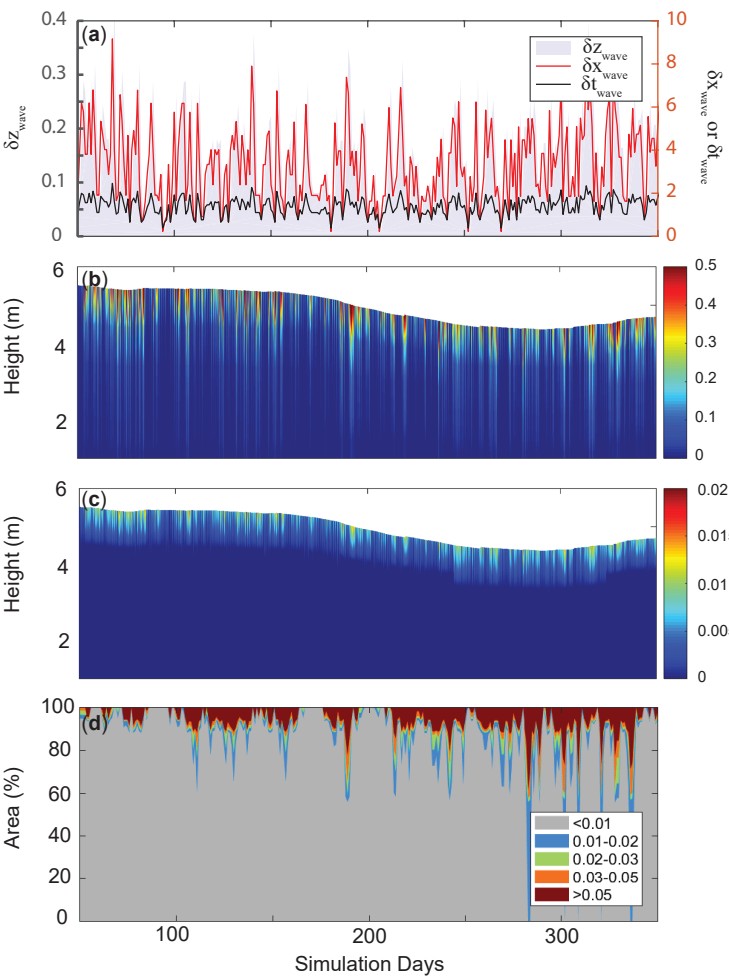

**Figure 15: Simulation from Woods Lake, Australia, showing a) time series of surface wave properties, b) orbital velocity, $U_{orb}$ (m s$^{-1}$), c) layer mean velocities, $U_m$ (m s$^{-1}$), and d) lake-bed area fraction within each of five depicted shear stress categories (based on $\tau_i$, N m$^{-2}$).**

## 3 Code organization and model operation

Aside from the core water balance and mixing functionality, the model features numerous options and extensions in order to make it a fast and easy-to-use package suitable for a wide range of contemporary applications. Accommodating these

10    requirements has led to the modular code structure outlined in Figure 16. The model is written in C, with a Fortran-based interface module to link with Fortran-based water quality modelling libraries described in Section 4. The model compiles with gcc, and gfortran, and commercial compilers, with support for Windows, OS X and Linux.

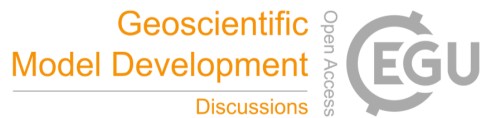

**Revision 26 Oct 2018**

**Figure 16: Overview of GLM code structure and program flow. Modules are depicted as a box with the main routines and functions summarised. Three entry points to the main model routines are possible depending on the desired treatment of the inflow and outflow boundary condition data: do_model uses the flow boundary condition data over the present and previous day in order to get the midday value, do_model_nonavg uses that from the present day only, and do_model_coupled passes in the present day flows from the host.**

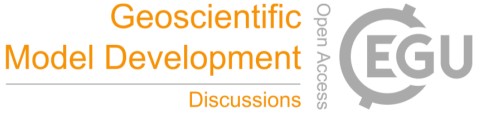

Revision 26 Oct 2018

The model may also be compiled as a library, termed libGLM, that can be called as a plugin into other models (e.g., see Section 5.4). Whilst the model is not object-oriented, users may easily customise specific modules described in Section 2 by adding or extending options for alternate schemes or functions.

To facilitate the use of the model in teaching environments and for users with limited technical support, it may be operated without any third-party software, as the input files consist of "namelist" (`nml`) text files for configuration and csv files for meteorological and flow time series data (Figure 17). The outputs from predictions are stored into a structured netCDF file, which can be visualised in real-time through the simple inbuilt plotting library (libplot) or may be opened for post-processing in MATLAB, R, or any other tool supporting the open netCDF format (see Section 5.1). Parameters and configuration details

are input through the main `glm.nml` text file (Figure 17) and default parameters and their associated descriptions are outlined in Table 1.

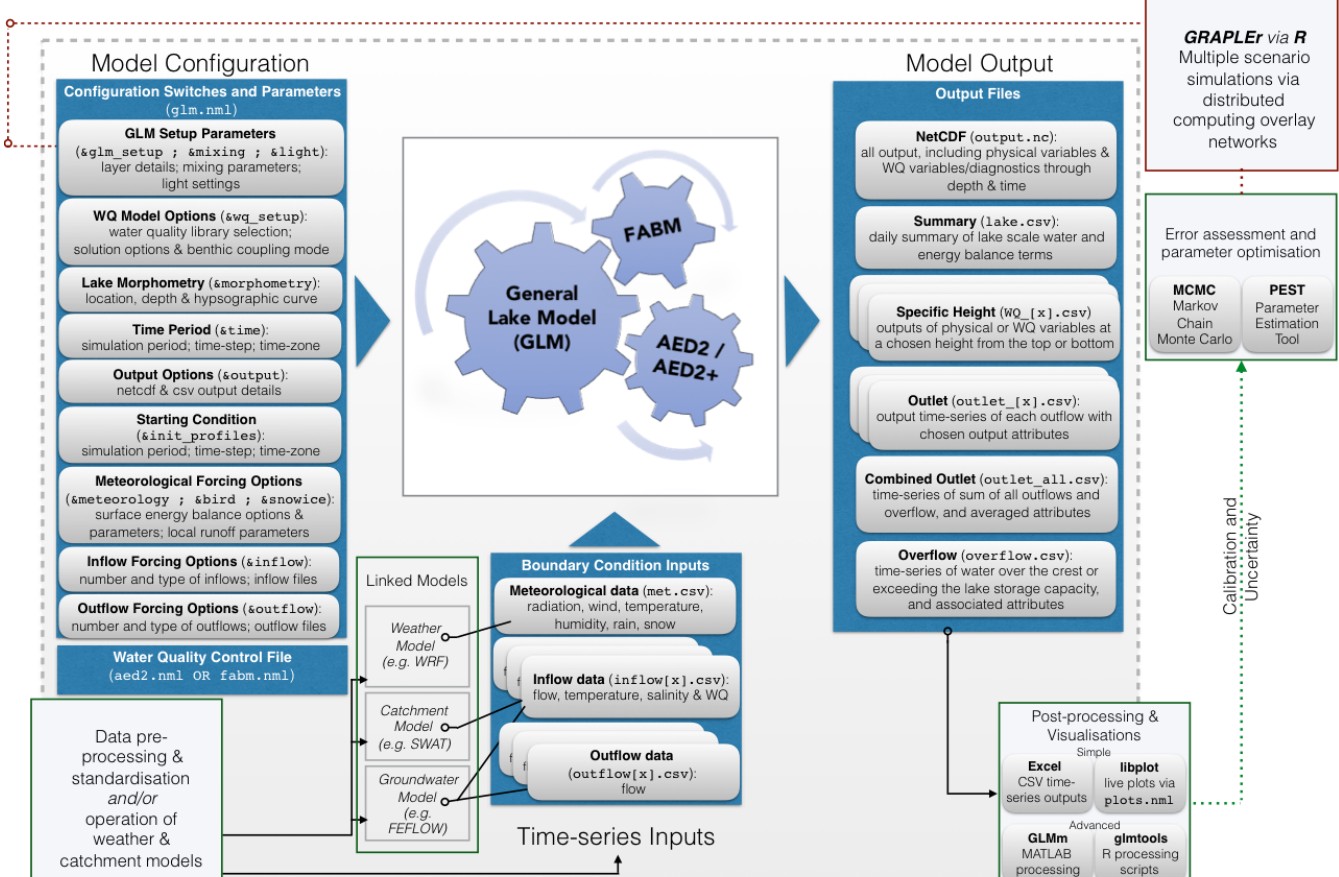

**Figure 17: Flow diagram showing the input information required for operation of the model, the outputs, and analysis pathways.**

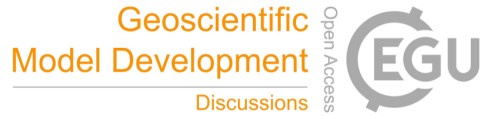

**Revision 26 Oct 2018**

## 4 Dynamic coupling with biogeochemical and ecological model libraries

Beyond modelling the vertical temperature distribution, the water, ice and heat balance, as well as the transport and mixing in a lake, the model has been designed to couple with biogeochemical and ecological model libraries. Currently the model is distributed pre-linked with the AED2 simulation library (Hipsey et al., 2013) and the Framework for Aquatic Biogeochemical

Models (FABM; Bruggeman and Bolding, 2014). Through connection with these libraries, GLM creates a set $\mathbb{C}$ of scalar variables, where by $C \in \mathbb{C}$, which resolve the vertical profiles and mass balance of turbidity, oxygen, nutrients, phytoplankton, zooplankton, pathogens and other water quality variables of interest. Documentation of these models is beyond the scope of the present paper, however, two features associated with the coupling are highlighted here as relevant to managing physical-ecological interactions.

Firstly, the model is designed to allow a user defined number of sediment zones that span the depth of the lake. Using this approach, the current setup allows for depth-dependent sediment properties, both for physical properties such as roughness or sediment heat flux (as outlined in previous sections), and also biogeochemical properties such as sediment nutrient fluxes and benthic ecological interactions. Since the GLM layer structure is flexible over time (i.e., layer heights are not fixed), any

interactions between the water and sediment/benthos must be managed at each time step. The model supports disaggregation and/or aggregation of layer properties, for mapping individual water layers to one or more sediment zones (Figure 18). The weightings provided by each layer to the sediment are based on the relative depth overlap of a layer with the depth range of the sediment zone, with the heights of zone boundaries denoted $h_z$. This approach makes the model suitable for long-term assessments of wetland, lake and reservoir biogeochemical budgets, as is required for C, N and other attribute balances

(Stepanenko et al., 2016).

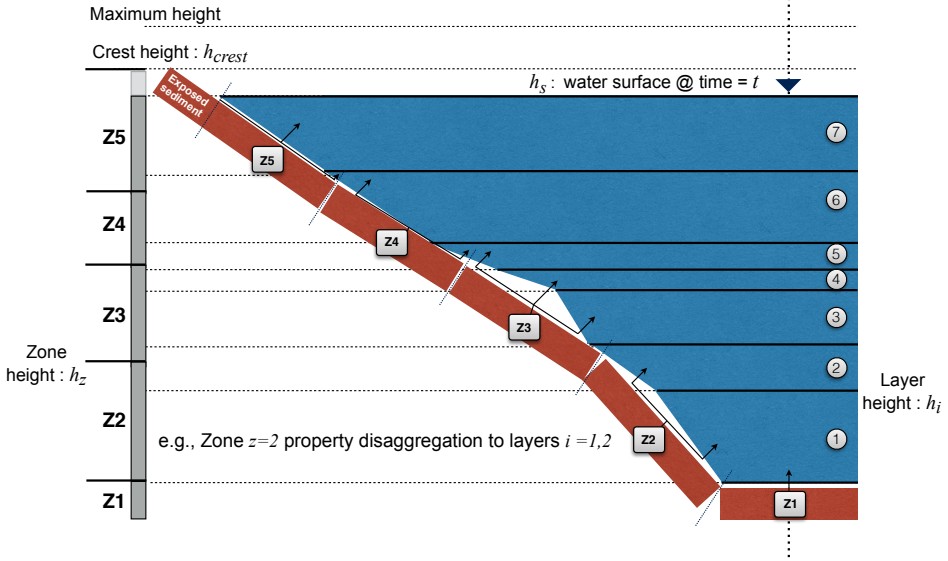

**Figure 18: Schematic of a lake model layer structure (indicated by layers $i$ =1:7), in conjunction with five sediment "zones" (Z1-Z5) activated when `benthic_mode` = 2. The dynamically varying layer structure is re-mapped to the fixed sediment zone locations at**
**each time step in order for the sediment zone to receive the average overlying water properties, and for the water to receive the appropriate information from benthic/sediment variables.**

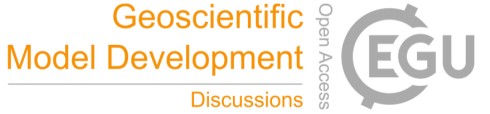

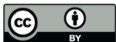

**Revision 26 Oct 2018**

Secondly, the water quality modules feedback to GLM properties related to the water and/or heat balance. Feedback options are included for external libraries to provide water density additions, and to modify bottom friction, $f_w$, the light attenuation coefficient, $K_w$, solar shading, $f_{SW}$, and rainfall interception, $f_R$.

## 5 Workflow tools for integrating GLM with sensor data and supporting models

The GLM model has been designed to support integration of large volumes of data coming from instrumented lakes, including many GLEON sites. These data consist of high-frequency and discrete time series observations of hydrologic fluxes, meteorology, temperature, and water quality (e.g., Hamilton et al., 2015). To facilitate research that requires running the model using these data sources, we have created GLM interfaces in the R and MATLAB analysis environments. These tools support user-friendly access to the model and include routines that streamline the process of calibrating models or running various

scenarios. In addition, for assessment of lake dynamics in response to catchment or climatic forcing, it is desirable to be able to connect GLM with other model platforms associated with surface and groundwater simulation, and weather prediction (Read et al., 2016).

### 5.1 R and MATLAB libraries for model setup and post-processing

The R and MATLAB scientific languages are commonly used in aquatic research, often as part of automated modelling and

analysis workflows. GLM has a client library for both, and these tools are shared freely online. The R package is called "`glmtools`" and the MATLAB library is called "`GLMm`"(available via the GLM website). Both tools have utilities for model output pre- and post-processing. The pre-processing components can be used to format and modify data inputs and configuration files, and define options for how GLM executes. Post-processing tools include visualizations of simulation results (as shown in the results figures above), comparisons to field observations, and various evaluations of model

performance.

### 5.2 Utilities for assessing model performance, parameter identification and uncertainty analysis

In order to compare the performance of the model for various types of lakes, numerous metrics of model performance are relevant. These include simple measures like surface or bottom temperature, or ice thickness. It is also possible to assess the model's performance in capturing higher-order metrics relevant to lake dynamics, including Schmidt Stability, thermocline

depth, ice on/off dates (see also Bruce et al., 2018, for a detailed assessment of the model's accuracy across a wide diversity of lakes across the globe).  With particular interest in the model's ability to interface with high frequency sensor data for calculation of key lake stability metrics (Read et al., 2011), continuous wavelet transform comparisons are also possible (Kara et al., 2012), allowing assessment of the time scales over which the model is able to capture the observed variability within the data.

As part of the modelling process, it is common to adjust parameters to get the best fit with available field data and, as such, the use of a Bayesian Hierarchical Framework in the aquatic ecosystem modelling community has become increasingly useful (e.g., Zhang and Arhonditsis, 2009; Romarheim et al., 2015). Many parameters described throughout Section 2 are physically

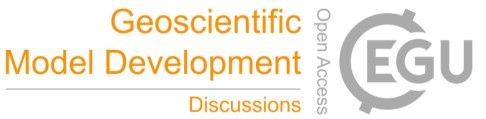

**Revision 26 Oct 2018**

based descriptions where there is relatively little variation (Bruce et al., 2018), thereby reducing the number of parameters that remain uncertain. For others, however, their variation reflects imperfect formulation of some processes that are not completely described numerically. Therefore, within MATLAB, support scripts for GLM to work with the Markov Chain Monte Carlo (MCMC) code outlined in Haario et al. (2006) can be used to provide improved parameter estimates and uncertainty assessment

(Figure 19; see also Huang et al., 2017). Example setups for use of GLM within the PEST (Parameter Estimation Tool) have also been developed, giving users access to a wide range of assessment methodologies. The PEST framework allows for calibration of complex models using highly-parameterised regularisation with pilot-points (Doherty, 2015). Sensitivity matrices derived from the calibration process can also be utilised in linear and non-linear uncertainty analysis.

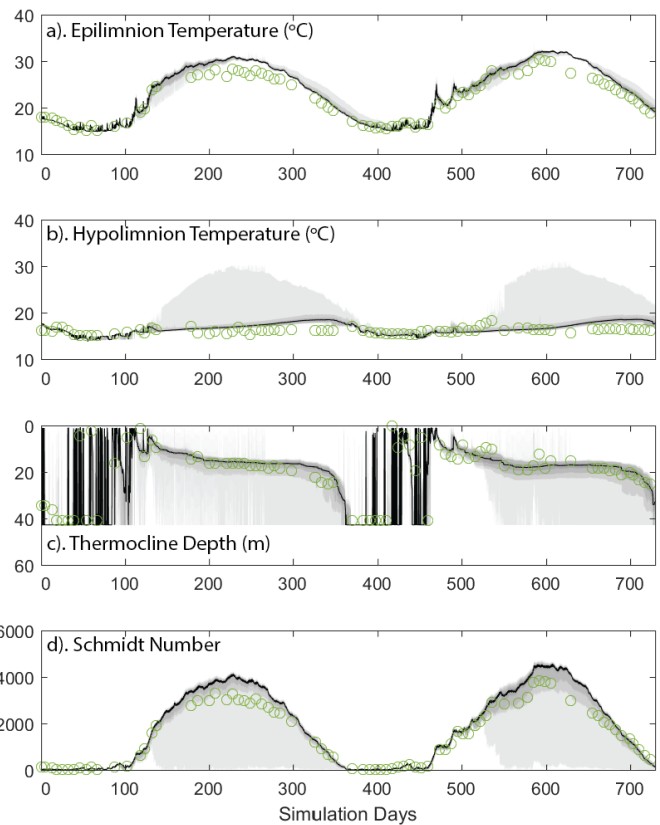

**Figure 19: Depiction of parameter uncertainty for a GLM simulation of Lake Kinneret, Israel, following calibration against observations (green circles) via MCMC for a) epilimnion temperature, b) hypolimnion temperature, c) thermocline depth, and d) Schmidt number. The black line indicates the 50th-percentile likelihood of the prediction, and the grey bands depict the 40th, 60th and 80th percentile.**

**5.3 Operation in the cloud: GRAPLEr**

Questions relevant to land use and climate changes are driving scientists to develop numerous scenarios for how lake ecosystems might respond to changing exogenous drivers. An important approach to addressing these questions is to simulate lake or reservoir physical-biological interactions in response to changing hydrology, nutrient loads or meteorology, and then

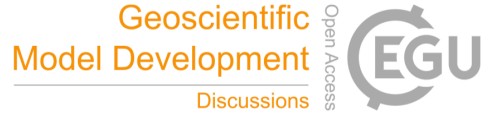

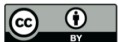

infer consequences from the emergent properties of the simulation, such as changes in water clarity, extent of anoxia, mixing regime, or habitability to fishes (Hipsey et al., 2015). Often, it takes years or even decades for lakes to respond fully to changes in exogenous drivers, requiring simulations to recreate lake behavior over extended periods. While most desktop computers can run a decade-long, low-resolution simulation in less than one minute, high-resolution simulations of the same extent may

require minutes to hours of processor time. When questions demand hundreds, thousands or even millions of simulations, the desktop approach is no longer suitable.

Through access to distributed computing resources, modellers can run thousands of GLM simulations in the time it takes to run a few simulations on a desktop computer. Collaborations between computer scientists in the Pacific Rim Applications and

Grid Middleware Assembly (PRAGMA) and GLEON have led to the development of GRAPLEr (GLEON Research and PRAGMA Lake Expedition in R), software, written in R, that enables modellers to distribute batches of GLM simulations to pools of computers (Subratie et al., 2017). Modellers use GRAPLEr in two ways: by submitting a single simulation to the GRAPLEr Web service, along with instructions for running that simulation under different climate scenarios, or by configuring many simulations on the user's desktop computer, and then submitting them as a batch to the Web service. The first approach

provides a high degree of automation that is well suited to training and instruction, and the second approach has the full flexibility often needed for research projects. In all approaches, GRAPLEr converts the submitted job to a script that is used by the scheduling program HTCondor (Thain et al., 2005) to distribute and manage jobs among the computer pool and ensure that all simulations run and return results. An iPOP overlay network (Ganguly et al., 2006) allows the compute services to include resources from multiple institutions, as well as cloud computing services.

GRAPLEr's Web service front-end shields the modeller from the compute environment, greatly reducing the need for modellers to understand distributed computing; they therefore only need to install the R package, know the URL of the GRAPLEr Web service, and decide how the simulations should be set up.

**5.4 Integration with catchment and climate models**

GLM simulations may be coupled with catchment models, such as the Soil Water Assessment Tool (SWAT) or similar catchment models, simply by converting the catchment model output into the inflow file format via conversion scripts (e.g., Bucak et al., 2018). Similarly, scripts exist for coupling GLM with the Weather Research Forecasting (WRF) model, or similar climate models, for specification of the meteorological input file from weather prediction simulations (e.g., Hansen et al.,

30    2018).

The above coupling approaches require the models to be run in sequence. For the simulation of lake-wetland-groundwater systems, however, two-way coupling is required to account for the flow of water into and out of the lake throughout the simulation. For these applications, the interaction has been simulated using GLM coupled with the 3D groundwater flow

model, FEFLOW (https://www.mikepoweredbydhi.com/products/feflow). For this case, the GLM code is compiled as a

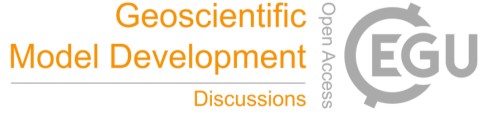

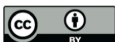

**Revision 26 Oct 2018**

Dynamic Link Library (DLL), termed libGLM, and loaded into FEFLOW as a plug-in module. The coupling between GLM and FEFLOW is implemented using a one-step lag between the respective solutions of the groundwater and lake models. This approach, in most simulations, does not introduce a significant error, however, error can be assessed and reduced using smaller time step lengths. The GLM module was designed to accommodate situations of variable lake geometry, by using a dry-lake/wet-lake approach, whereby dry-lake areas are defined as those above the current lake level and wet-lake areas as below the current lake level. Different boundary types in FEFLOW are assigned to dry-lake and wet-lake areas (Figure 20). The calibration of such coupled models is often complex, given the large number of parameters and sensitivities when different sources of information are utilised (for example flow and water level measurements). The FEFLOW-GLM coupling structure allows for a relatively straightforward integration with PEST, based on existing FEFLOW workflows.

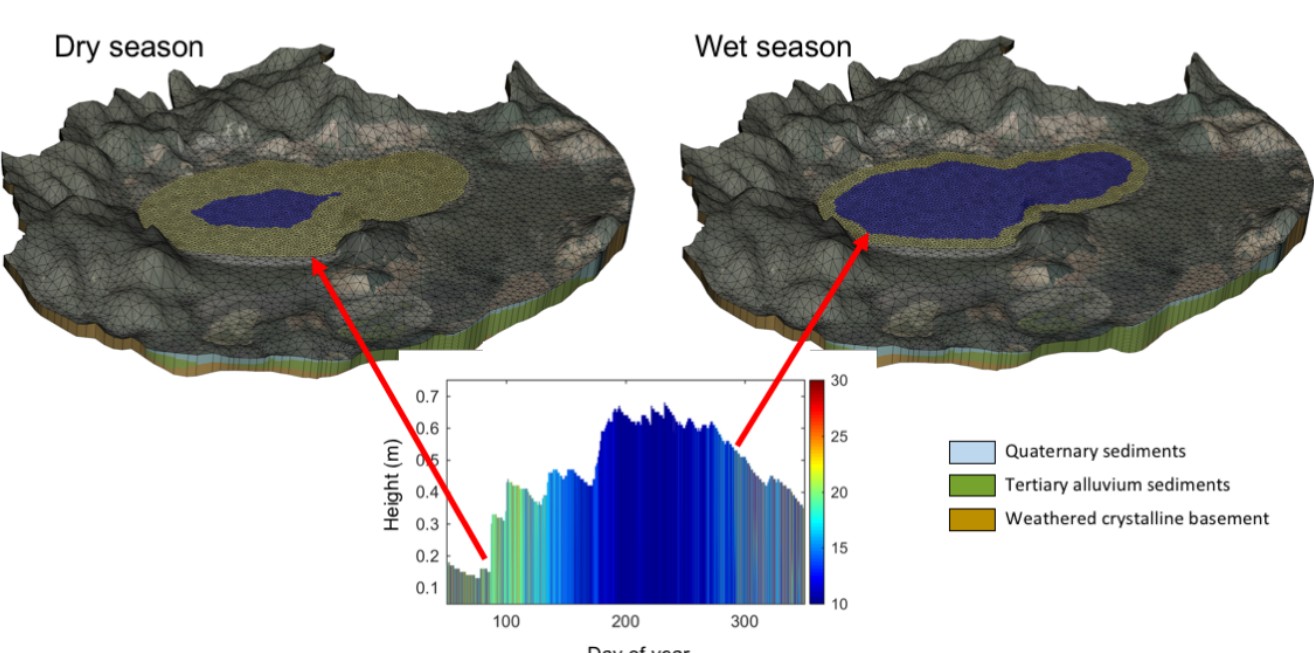

**Figure 20: Example of water level changes during a seasonal cycle from Lake Muir, Australia. GLM water level is periodically communicated to the 3D FEFLOW groundwater model via a plugin calling libGLM, and used as a constant head boundary condition for all wet cells within the FEFLOW mesh.**

## 6 GLM as a tool for teaching environmental science and ecology

Environmental modelling is integral for understanding complex ecosystem responses to anthropogenic and natural drivers, and also provides a valuable tool for engaging students learning environmental science (Carey and Gougis, 2017). Previous pedagogical studies have demonstrated that engaging students in modelling provides cognitive benefits, enabling them to build new scientific knowledge and conceptual understanding (Stewart et al., 2005; Schwarz et al., 2009). For example, modelling forces students to analyze patterns in data, create evidence-based hypotheses for those patterns and make their hypotheses

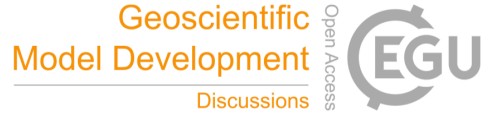

**Revision 26 Oct 2018**

explicit, and develop predictions of future conditions (Stewart et al., 2005). As a result, the U.S. National Research Council has recently integrated modelling into the *Next Generation Science Standards*, which provide recommendations for primary and secondary school science pedagogy in the United States (NRC, 2013). However, it remains rare for undergraduate and graduate science courses to include the computer-based modelling that environmental scientists need to manage natural

ecosystems.

A teaching module for the use of GLM within undergraduate and graduate classrooms has been developed to explore lake responses to climate change (Carey and Gougis, 2017). The GLM module, called the "Climate Change Effects on Lake Temperatures", teaches students how to set up a simulation for a model lake within R. After they are able to successfully run

their lake simulations, they force the simulation with climate scenarios of their own design to examine how lakes may change in the future. To improve computational efficiency, students also learn how to submit, retrieve, and analyze hundreds of model simulations through distributed computing overlay networks embedded via the GRAPLEr interface (Section 5.3). Hence, students participating in the module learn computing and quantitative skills in addition to improving their understanding of how climate change affects lake ecosystems.

Initial experiences teaching GLM as well as pre- and post-assessments indicate that participation in the module improves students' understanding of lake responses to climate change (Carey and Gougis, 2017). By modifying GLM boundary condition data and exploring model output, students are able to better understand the processes that control lake responses to altered climate, and improve their predictions of future lake change. Moreover, the module exposes students to computing and

modelling tools not commonly experienced in most university classrooms, building competence with manipulating data files, scripting, creating figures and other visualizations, and statistical and time series analysis; all skills that are transferrable for many other applications.

## 7 Conclusions

As part of GLEON activities, the emergence of complex questions about how different lake types across the world are responding to climate change and land-use change has created the need for a robust, accessible community code suitable for a diverse range of lake types and simulation contexts. Here, GLM is presented as a tool that meets many of the needs of network participants with suitability to a wide array of lake types and sizes, whilst also meeting the need for a distributed simulation across tens to thousands of lakes as is required for regional and global scale assessments (e.g., Kirillin et al., 2011). Recent

examples have included an application of the model for assessing how the diversity of >2000 lakes in a lake-rich landscape in Wisconsin respond to climate, including projected warming (Read et al., 2014; Winslow et al., 2017). Given its computationally efficient nature, it is envisioned that GLM can be made available as a library for use within in land-surface models (e.g., the Community Land Model, CLM), allowing improved representation of lake dynamics in regional hydrological or climate assessments. With further advances in the degree of resolution and scope of earth system models, we further

envisage GLM as an option suitable to be embedded within these models to better allow the simulation of lake stratification,

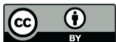

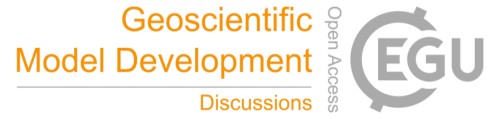

**Revision 26 Oct 2018**

air-water interaction of momentum and heat, and also biogeochemically relevant variables associated with contemporary questions about greenhouse gases emissions such as $CO_2$, $CH_4$, and $N_2O$.

Since the model is one-dimensional, it assumes no horizontal variability in the simulated water layers and users must therefore ensure their application of the model is suited to this simplifying assumption. For stratified systems, the parameterization of mixing due to internal wave and inflow intrusion dynamics is relatively simple, making the model ideally suited to longer-term investigations ranging from weeks to decades (depending on the domain size), and for coupling with biogeochemical models to explore the role that stratification and vertical mixing play on lake ecosystem dynamics. However, the model can also be used for shallow lakes, ponds and wetland environments where the water column is relatively well mixed. In cases where the assumption of one-dimensionality is not met for a particular lake application, a two or three dimensional model may be preferred.

This paper has focused on description of the hydrodynamic model, but we highlight that the model is a platform for coupling with advanced biogeochemical and ecological simulation libraries for water quality prediction and integrated ecosystem assessments. As with most coupled hydrodynamic-ecological modelling platforms, GLM handles the boundary conditions and transport of variables simulated within these libraries, including the effects of inflows, vertical mixing, and evapo-concentration. Whilst the interface to these libraries is straightforward, the Lagrangian approach adopted within GLM for simulation of the water column necessitates the adoption of sediment zones on a static grid that is independent from the water column numerical grid.

More advanced workflows for operation of the model within distributed computing environments and with data assimilation algorithms is an important application when used within GLEON capabilities related to high frequency data and its interpretation. The 1D nature of the model makes the run-times modest and therefore the model suitable for application within more intensive parameter identification and uncertainty assessment procedures. This is particularly relevant as the needs for network participants to expand model configurations to further include biogeochemical and ecological state variables. It is envisioned that continued application of the model will allow us to improve parameter estimates and ranges, and this will ultimately support other users of the model in identifying parameter values, and assigning parameter prior distributions. Since many of the users the model is intended for may not have access to the necessary cyberinfrastructure, the use of GLM with the open-source GRAPLEr software in the R environment provides access to otherwise unavailable distributed computing resources. This has the potential to allow non-expert modellers within the science community to apply good modelling practices by automating boundary condition and parameter sensitivity assessments, with technical aspects of simulation management abstracted from the user.

Finally, the role of models in informing and educating members of the network and the next generation of hydrologic and ecosystem modellers has been identified as a critical element of synthesis activities and supporting cross-disciplinary

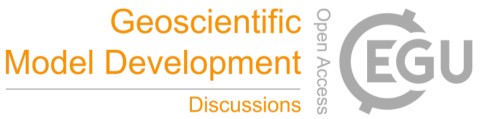

**Revision 26 Oct 2018**

collaboration (Weathers et al., 2017). Initial use of GLM within the classroom has found that teaching modules integrating GLM into classes improves students' understanding of lake ecosystems.

### Code availability

The GLM code is provided as open-source under the GNU GPL3 license, and version controlled via the GitHub repository:

https://github.com/AquaticEcoDynamics/GLM; *DOI: 10.5281/zenodo.1297331*

### Data availability

The five example lakes used to demonstrate the model operation are described along with model input files (and associated hydrologic and meteorological forcing data) within the GitHub repository:
https://github.com/AquaticEcoDynamics/GLM_Examples; *DOI: 10.5281/zenodo.1297415*

**Acknowledgments**

The primary code of GLM has been developed by MH, LB, CB, BB and DH at The University of Western Australia in collaboration with researchers participating in GLEON, with support provided by a National Science Foundation (NSF) (USA) Research Coordination Network Award. Whilst GLM is a new code, it is based on the large body of historical research and publications produced by the Centre for Water Research at the University of Western Australia, which we acknowledge for
the inspiration, development and testing of several of the model approaches that have been adopted. Funding for initial development of the GLM code was from the U.S. NSF Cyber-enabled Discovery and Innovation grant awarded to PH (lead investigator) and colleagues from 2009-2014 (NSF CDI-0941510), and subsequent development was supported by the Australian Research Council projects awarded to MH and colleagues (ARC projects LP0990428, LP130100756 and DP130104087). Funding for the optimization and improvement of the snow and ice model was provided by NSF MSB-
1638704. Funding for development of the GLM teaching module and GRAPLEr was supported from NSF ACI-1234983 and NSF EF-1702506 awarded to CC. Funding for glmtools was provided by the Department of the Interior Northeast Climate Science Center. Any use of trade, firm, or product names is for descriptive purposes only and does not imply endorsement by the U.S. Government. Provision of the environmental symbols used for the GLM scientific diagrams are courtesy of the Integration and Application Network, University of Maryland Center for Environmental Science. Joanne Moo and Aditya
Singh also provided support in model setup and testing. We gratefully acknowledge the anonymous reviewers whose contribution and editing have significantly improved the paper and model.



**Revision 26 Oct 2018**

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



**Revision 26 Oct 2018**

**Table 1. Summary of GLM notation with values for constants, suggested (default) values for parameters, and supporting information and references, where relevant.**

| Symbol | Description | Units | Value * | Comments | Reference |
|---|---|---|---|---|---|
| | | | Indices | | |
| $N_{BSN}$ | user provided number of basin height points | | configurable | set in `&morphometry` | |
| $N_{MORPH}$ | internally computed number of vertical elevation/height increments for the hypsographic curve | | computed | $H_{b=N_{BSN}}\, \Delta H_{mi} + 10$ | |
| $N_{LEV}$ | number of layers | | variable | active layers varies over time | |
| $N_{INF}$ | number of inflows configured | | configurable | set in `&inflows` | |
| $N_{OUTF}$ | number of outlets configured | | configurable | set in `&outflows` | |
| $N_{SW}$ | number of shortwave radiation bands configured | | configurable | set in `&light` | |
| $N_{SZ}$ | number of sediment zones configured | | configurable | set in `&sediment` | |
| $b$ | hypsographic data point index | | index | | |
| $mi$ | internal hypsographic curve increment | | index | | |
| $i$ | index of computational layer | | index | | |
| $i_j$ | index of the lake layer at an equivalent depth to inflow parcel $j$ | | index | | |
| $i_{bot}$ | index of lower most layer impacted by a given withdrawal/outflow | | index | | |
| $i_{top}$ | index of the upper-most layer impacted by a given withdrawal/outflow | | index | | |
| $i_{outf}$ | index of the lake layer aligning with a withdrawal/outflow extraction point | | index | | |
| $s$ | layer index of the layer at the surface of the lake | | index | | |
| $k$ | layer index of the layer at the bottom of the surface mixed layer (sml; epilimnion) | | index | | |
| $j$ | index of inflow parcel transport step, prior to insertion | | index | | |
| $z$ | index of sediment zone | | index | | |
| $l$ | index of light bandwidth fraction | | index | | |
| $I$ | inflow index | | index | | |
| $O$ | outflow index | | index | | |
| | | | Time | | |
| $t$ | time | s | - | | |
| $t_b$ | time when a shear event begins | s | - | | |
| $\lfloor t \rfloor$ | time marker | s | - | used to compute the time within a day | |
| $\Delta t$ | time step used by the model | s | 3600 | numerical time increment the model uses | |
| $TZ$ | time zone indicated by number of hours from GMT | hr | configurable | set in `&time` | |
| $N_{\Delta t}$ | number of time-steps to simulate | - | configurable | set in `&time` | |
| $c_{secday}$ | number of seconds per day | s day$^{-1}$ | 86400 | | |

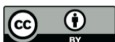

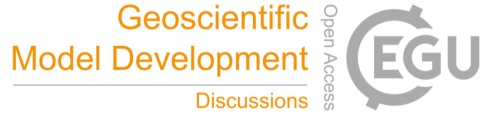

| Symbol | Description | Units | Value * | Comments | Reference |
|---|---|---|---|---|---|
| $d$ | day of the year | - | variable | | |
| $\delta t_d$ | time-scale of inflow parcel transport | s | computed | | |
| $\delta t_{wave}$ | period of surface waves | s | computed | Eq. 79 | |
| $\delta t_{iw}$ | period for internal waves | s | computed | $\delta t_{iw} = L_{META}/2c$ | Spigel and Imberger (1980) |
| $\delta t_{shear}$ | cut-off time for internal wave induced velocity shear | s | computed | Eq. 47 | |
| $\delta t_{damp}$ | time-scale of internal wave damping | s | computed | Eq. 50 | Spigel and Imberger (1980) |
| Lake setting (volumes, areas and lengths) | | | | | |
| $V_{max}$ | maximum volume of the lake | m³ | computed | once exceeded, excess water is passed to overflow | |
| $V_{crest}$ | volume of the lake at the crest height | m³ | computed | volume corresponds to height, $h_{crest}$ | |
| $V_b$ | lake volume at the hypsographic data point $b$ | m³ | configurable | Eq. 1 | |
| $V_{mi}$ | interpolated volume at internal morphometry table increment $mi$ | m³ | computed | Eq. 2 | |
| $V_i$ | volume of the lake at the top of the $i$ th layer | m³ | variable | $V_i = \sum_{j=1}^{i} \Delta V_j$ | |
| $V_s$ | volume of the lake at the top of the surface layer ($i = N_{LEV}$) | m³ | variable | $V[h_{i=N_{LEV}}]$ | |
| $V_S^*$ | interim calculation of the volume of the lake at the top of the surface layer | m³ | variable | used to estimate lake volume prior to overflow calculation | |
| $\tilde{V}_{N^2}$ | a fractional volume of the lake that contains 85% of $N^2$ variance | m³ | variable | computed as the volume of the lake above the height which is one standard deviation ($\delta z_\sigma$) below the height at the centre of buoyancy | |
| $\Delta V_i$ | volume of the $i$ th layer | m³ | variable | $V[h_i] - V[h_{i-1}]$ | |
| $\Delta V_{epi}$ | volume of the epilimnion | m³ | variable | $\Delta V_{epi} = V_S - V_{k-1}$ | |
| $\Delta V_{k-1}$ | volume of the layer below the surface mixed layer/epilimnion | m³ | variable | $V[h_{i=k-1}]$ | |
| $\Delta V_G$ | volume of water lost to seepage | m³ | variable | $\Delta V_G = Q_{seepage}\Delta t_d$ | |
| $A_{max}$ | maximum possible area of the lake | m² | configurable | $A_{max} = A_{b=N_{BSN}}$ | |
| $A_b$ | lake area above datum at the hypsographic data point b | m² | configurable | set in &morphometry | |
| $A_{mi}$ | lake area at internal morphometry table increment mi | m² | computed | | |
| $A_i$ | lake area of the $i$ th layer | m² | variable | | |
| $A[H]$ | lake area at a given height / elevation | m² | configurable | area-height relationship set in &morphometry | |
| $A_S$ | area of the lake surface | m² | variable | | |
| $A_{BEN}$ | lake bottom (benthic) area exceeding the critical light threshold $\phi_{BEN_{crit}}$ | m² | variable | Eq. 16 | |
| $A_E$ | effective area of the lake surface exposed to wind stress | m² | computed | Eq. 26 | |
| $A_{WS}$ | critical area below which wind sheltering may occur | m² | $10^7$ | set in &fetch | Xenopoulos and Schindler (2001) |
| $A_{outf}$ | area of the lake at the height of the relevant outflow | m² | computed | | |
| $A_{k-1}$ | lake area at the top of the metalimnion | m² | variable | | |

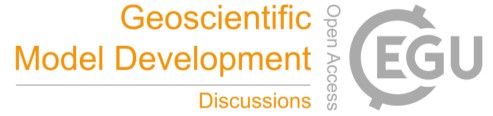

**Revision 26 Oct 2018**

| Symbol | Description | Units | Value * | Comments | Reference |
|--------|-------------|-------|---------|----------|-----------|
| $H$ | variable referring to elevation; height above datum | m above datum | | $H = H_0 + h$ | |
| $H_{max}$ | maximum elevation of the lake, above which water will overflow | m above datum | configurable | | |
| $H_{crest}$ | elevation at the lake crest | m above datum | configurable | set in `&morphometry` | |
| $H_0$ | bottom elevation of the lake | m above datum | configurable | | |
| $H_b$ | elevation above datum at the hypsographic data point $b$ | m above datum | configurable | | |
| $H_{mi}$ | elevation above datum at internal morphometry table increment mi | m above datum | computed | | |
| $\Delta H_{mi}$ | elevation increment used for the model's internal hypsograhic curve interpolation function | m | 0.1 | | |
| $h$ | height | m above lake bottom | | | |
| $h_i$ | height at the top of layer $i$ | m above lake bottom | variable | | |
| $h_S$ | height of the top surface of the uppermost (surface) layer | m above lake bottom | variable | Eq. 5 | |
| $h_S^*$ | interim surface height computed prior to overflow calculation | m above lake bottom | computed | | |
| $h_B$ | height of the top surface of the bottom-most layer | m above lake bottom | variable | | |
| $h_{BEN}$ | height at which the $\phi_{BEN_{crit}}$ is reached | m above lake bottom | variable | | |
| $h_z$ | height of the uppermost limit of the sediment zone $z$ | m above lake bottom | configurable | set in `&sediment` if `benthic_mode=2` | |
| $\widetilde{h_i}$ | height of the middle of the $i$ th layer | m above lake bottom | variable | | |
| $\widetilde{h_{sml}}$ | height of the middle of the epilimnion | m above lake bottom | variable | | |
| $h_{outf}$ | height of a configured outflow | m above lake bottom | configurable | set in `&outflow` | |
| $h_{inf}$ | height of a submerged inflow | m above lake bottom | configurable | set in `&inflow` | |
| $h_{i_{j-1}}$ | bottom of the nearest layer that inflow parcel $j$ sits above | m above lake bottom | variable | | |
| $h_{i_{ins_I}-1}$ | height of the bottom of the layer where an inflow parcel associated with the $I$ th inflow inserted | m above lake bottom | variable | | |
| $h_{crest}$ | height of the lake crest where water begins to overflow | m above lake bottom | computed | $h_{crest} = H_{crest} - H_0$ | |
| $z$ | depth from the lake surface, or height above the lake surface | m from water surface | | | |
| $z_{avg}$ | average depth of the lake | m | variable | | |
| $z_{BEN}$ | depth to the lake where critical light threshold is exceeded | m from water surface | variable | Eq. 17 | |
| $z_{sml}$ | depth to the metalimnion from the surface | m from water surface | variable | equivalent to vertical thickness of the surface mixed layer (sml). | |
| $z/L$ | Monin-Obukhov stability parameter | - | computed | Eq. A26 | |
| $z_o$ | water surface roughness length | m | computed | Eq. A24 | |
| $z_\theta$ | water surface heat roughness length | m | computed | | |


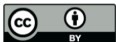

**Revision 26 Oct 2018**

| Symbol | Description | Units | Value * | Comments | Reference |
|---|---|---|---|---|---|
| $z_q$ | water surface moisture roughness length | m | computed | | |
| $z_{inf_j}$ | depth of an inflow parcel during its transit | m | computed | $z_{inf_j} = z_{inf_{j-1}} + \Delta x_{inf_j} \sin \Phi_{inf}$ | |
| $z_{inf_{insI}}$ | depth that an inflow parcel associated with inflow I inserts | m from water surface | variable | depth from the surface where an inflow reaches its level of neutral buoyancy | |
| $\Delta z_i$ | thickness of the $i$ th layer | m | variable | | |
| $\Delta z_{k-1}$ | thickness of the layer below the epilimnion | m | variable | | |
| $\Delta z_{min}$ | minimum layer thickness | m | 0.5 | should be estimated relative to lake depth; set in `&glm_setup` | Bruce et al. (2018) Bueche et al. (2017) |
| $\Delta z_{max}$ | maximum layer thickness | m | 1.5 | | |
| $\Delta z_{ice}$ | combined thickness of the white ice and blue ice | m | computed | $\Delta z_{white} + \Delta z_{blue}$ | |
| $\Delta z_{ice,snow}$ | thickness of top layer of ice cover, depending on ice or snow presence | m | computed | Eq. 32 | |
| $\Delta z_{snow}$ | thickness of snow | m | variable | Eq. 32; Figure 6 | |
| $\Delta z_{white}$ | thickness of white ice | m | variable | Eq. 32 | |
| $\Delta z_{blue}$ | thickness of blue ice | m | variable | Eq. 38 | |
| $\Delta z_{inf_0}$ | thickness of an inflow parcel before transport into the lake | m | computed | Eq. 62 | |
| $\Delta z_{inf_j}$ | thickness of inflow parcel $j$ | m | variable | Eq. 64 | |
| $\delta z_\sigma$ | vertical length scale of one standard deviation about the centre of the $N^2$ distribution | m | computed | | |
| $\delta z_{inf_j}$ | vertical transport length of inflow parcel $j$ | m | variable | $\delta z_{inf_j} = \left( h_s - z_{inf_j} \right) - h_{i_{j-1}}$ | |
| $\delta z_{wave}$ | significant wave height of surface waves | m | computed | Eq. 82 | |
| $\delta z_{soil}$ | depth of soil layer | m | 0.5 | relevant soil depth for computing sediment heat diffusion or water seepage | |
| Other notation (sorted alphabetically) | | | | | |
| $a$ | Charnock constant | - | 0.012 | relates roughness length to wind speed | |
| $a_W$ | constant relating precipitable water vapour, $W$, to $T_d$ | - | 0.07-0.09 | | Luo et al. (2010) |
| $AM$ | air mass factor | - | computed | Eq. A7 | |
| $AM_p$ | air mass factor at pressure $p$ | - | computed | Eq. A7 | |
| $AOD_{500}$ | aerosol optical depth at 500 nm | - | 0.033 – 0.10 | set in `&bird` | Luo et al. (2010) |
| $AOD_{380}$ | aerosol optical depth at 380 nm | - | 0.038 – 0.15 | | |
| $W_{weir}$ | width of the weir crest | m | configurable | used if $H_{crest} < H_{max}$; set in `&outflow` | |
| $b_W$ | constant relating precipitable water vapour, $W$, to $T_d$ | - | 1.88-2.12 | | Luo et al. (2010) |
| $c$ | internal wave speed | m s⁻¹ | computed | Eq. 48 | |
| $c_a$ | specific heat capacity of air | J kg⁻¹ °C⁻¹ | 1005 | | |

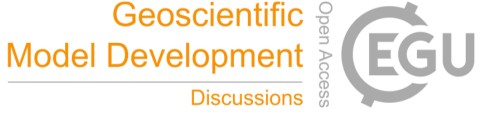

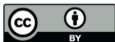

**Revision 26 Oct 2018**

| Symbol | Description | Units | Value * | Comments | Reference |
|---|---|---|---|---|---|
| $c_i$ | specific heat capacity of ice | J kg$^{-1}$ $^o$C$^{-1}$ | 2050 | | |
| $c_w$ | specific heat capacity of liquid water | J kg$^{-1}$ $^o$C$^{-1}$ | 4185.5 | | |
| $c_{damp}$ | coefficient related to damping rate of internal waves | - | 104.1 | | Spigel (1978) |
| $c_{wn}$ | coefficient related to wavenumber calculation | - | 12.4 | | |
| $\mathbb{C}$ | set of scalars being simulated | various | variable | variable number of scalars managed by GLM which are subject to mixing and mass conservation | |
| $C_i$ | concentration of relevant scalar in the $i$$^{th}$ layer | various | variable | | |
| $\bar{C}$ | mean concentration of two or more layers | various | computed | | |
| $\Delta C$ | difference in concentration of two layers | various | computed | | |
| $C_{inf}$ | concentration of relevant scalar in the outflowing water | various | time-series input | | |
| $C_{outf}$ | concentration of relevant scalar in the outflowing water | various | variable | | |
| $C_{KH}$ | mixing efficiency - Kelvin-Helmholtz billows | - | 0.3 | set in &mixing | Sherman et al. (1978) |
| $C_{HYP}$ | mixing efficiency - hypolimnetic turbulence | - | 0.5 | applied differently based on deep_mixing model (see text), set in &mixing | Weinstock (1981); general diffusivities in Jellison and Melack (1993) |
| $C_T$ | mixing efficiency - unsteady turbulence (acceleration) | - | 0.51 | set in &mixing | Sherman et al. (1978) Spigel et al. (1986) Yeates & Imberger (2003) Wu (1973) |
| $C_S$ | mixing efficiency - shear production | - | 0.3 | | |
| $C_W$ | mixing efficiency - wind stirring | - | 0.23 | | |
| $C_K$ | mixing efficiency - convective overturn | - | 0.2 | | |
| $C_{Dinf}$ | stream-bed drag of an inflowing river | - | 0.016 | set based on inflow bed roughness in &inflow | |
| $C_{D_{weir}}$ | drag associated with weir crest | - | 0.62 | set in &outflow | |
| $C_D$ | bulk aerodynamic coefficient for momentum | - | 0.0013 | see also Appendix B; Eq. A23 | Fischer et al. (1979) Bruce et al. (2018) Bueche et al. (2017) |
| $C_E$ | bulk aerodynamic coefficient for latent heat transfer | - | 0.0013 | from Hicks' (1972) collation of ocean and lake data; many studies since use similar values; internally calculated if atmospheric stability correction is on. | |
| $C_H$ | bulk aerodynamic coefficient for sensible heat transfer | - | 0.0013 | | |
| $C_{XN}$ | generic notation for neutral value of bulk transfer coefficient | - | selected | X = H or E | |
| $C_{DN-10}$ | value of bulk transfer coefficient for momentum under neutral atmospheric conditions, referenced to 10m height. | - | computed | see also Appendix B | |
| $C_{HEN}$ | value of bulk transfer coefficient for heat/moisture under neutral atmospheric conditions, referenced to 10m height. | - | 0.0013 | | |
| $C_x$ | cloud cover fraction | - | time-series input | | |
| $d_{T_z}$ | day of the year when the soil temperature peaks, for the $z$$^{th}$ zone | - | 1-365 | | |
| $D_Z$ | effective vertical diffusivity of scalars in water | m$^2$ s$^{-1}$ | computed | | |
| $D_\varepsilon$ | diffusivity of scalars in water due to turbulent mixing | m$^2$ s$^{-1}$ | computed | | |

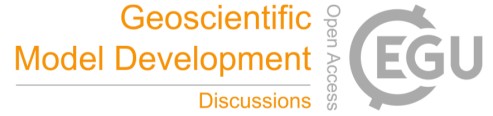

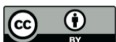

**Revision 26 Oct 2018**

| Symbol | Description | Units | Value * | Comments | Reference |
|---|---|---|---|---|---|
| $D_m$ | molecular diffusivity for scalars in water | m$^2$ s$^{-1}$ | 1.25×10$^{-9}$ | | |
| $D_a$ | molecular heat diffusivity of air | m$^2$ s$^{-1}$ | 2.14×10$^{-5}$ | reported as 0.077 m$^2$ hr$^{-1}$ | TVA (1972) |
| $D_{outf}$ | average vertical diffusivity of scalars in layers spanning the withdrawal thickness | m$^2$ s$^{-1}$ | computed | values from Eq. 57 averaged over the relevant layers | Imberger and Patterson (1981) |
| $e_s$ | saturation vapour pressure | hPa | computed | Eq. 24 | Tabata (1973) |
| $e_a$ | atmospheric vapour pressure | hPa | computed | Eq. 25 | |
| $e_*$ | humidity scale | hPa | computed | | |
| $E_{TKE}$ | turbulent kinetic energy per unit mass available for mixing, integrated over layer depth | m$^3$ s$^{-2}$ | computed | Eq. 41 | Imberger and Patterson (1981) |
| $E_{PE}$ | potential energy within the stratified water column integrated over layer depth | m$^3$ s$^{-2}$ | computed | Eq. 42 | Hamilton and Schladow (1997) |
| $E$ | evaporation mass flux | m s$^{-1}$ | variable | $E = \phi_E / \lambda_v \rho_s$ | |
| $E_{inf}$ | inflow entrainment | - | computed | Eq. 60 | |
| $EQT$ | equation of time | - | computed | Eq. A4 | |
| $F$ | fetch | m | computed | approximated from the square root of the lake area, Eq. 78 | |
| $Fr$ | internal Froude number of the lake subject to a water withdrawal | - | computed | Eq. 68 | |
| $f_R, f_S$ | rainfall, snowfall scaling factor | - | 1 | | |
| $f_{SW}$ | solar radiation scaling factor | - | 1 | used to adjust/calibrate model to meteorological data | |
| $f_U$ | wind-speed scaling factor | - | 1 | | |
| $f_{AT}$ | air temperature scaling factor | - | 1 | | |
| $f_{RH}$ | relative humidity scaling factor | - | 1 | | |
| $f_{inf}$ | inflow rate scaling factor | - | 1 | used to adjust flow boundary condition data | |
| $f_{outf}$ | outflow rate scaling factor | - | 1 | | |
| $f_{SWE}$ | snow water equivalent fraction | m rain/m snow | 0.1 | | |
| $f_{snow}$ | snow compaction constant | - | computed | Figure 6 | |
| $f_{WS}$ | wind-sheltering scaling factor | - | 1 | function used to scale the wind-sheltering length scale or lake surface area, based on the direction of the wind | |
| $f_{ro}$ | runoff coefficient | m runoff/m rain | 0.2 | depends on land slope and soil type | |
| $f_{PAR}$ | fraction of global incoming radiation flux which is photosynthetically active | - | 0.45 | | Jellison and Melack (1993) |
| $f_{VIS}$ | visible bandwidth fraction | - | 0.3 | | Rogers et al. (1995) |
| $f_{BEN_{crit}}$ | fraction of surface irradiance at the benthos, which is considered critical for productivity | - | 0.2 | set in `&light` | |
| $f_w$ | friction factor used for current stress calculation | - | computed | Eq. 87 | Kleinhans and Grasmeijer (2006) |
| $f_c$ | friction factor used for wave stress calculation | - | computed | | |
| $f_0$ | roughness correction coefficient for the lake surface | - | 0.5 | | TVA (1972) |

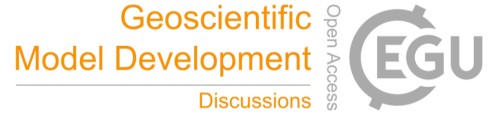

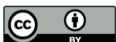

**Revision 26 Oct 2018**

| Symbol | Description | Units | Value * | Comments | Reference |
|---|---|---|---|---|---|
| $f_{dif}$ | smoothing factor used for diffusion | - | computed | Eq. 59 | |
| $f_\alpha$ | snow albedo scaling factor | - | 1.0 | set in &snowice | |
| $g$ | acceleration due to gravity | m s$^{-2}$ | 9.81 | | |
| $g'_k$ | reduced gravity between the mixed layer and the $k-1$ layer | m s$^{-2}$ | computed | | |
| $g'_{EH}$ | reduced gravity between the epilimnion and the hypolimnion | m s$^{-2}$ | computed | | |
| $g'_{inf}$ | reduced gravity between the inflowing water and adjacent lake water | m s$^{-2}$ | computed | | |
| $G$ | seepage rate | m day$^{-1}$ | 0 | | |
| $Gr$ | Grashof number related to an outflow extraction | - | computed | Eq. 69 | Imberger and Patterson (1981) |
| $k_{TKE}$ | turbulence wavenumber | m$^{-1}$ | computed | Eq. 56 | |
| $K_w$ | light extinction coefficient | m$^{-1}$ | 0.5 | set in &light, or updated via the linked water quality model; can be estimated from Secchi depth data | |
| $K_{w_l}$ | light extinction coefficient of the $l$ th bandwidth fraction | m$^{-1}$ | configurable | set in &light, and used if light_model = 2 | Cengel and Ozisk (1984) |
| $K_{w1}$ | waveband 1, white ice light extinction | m$^{-1}$ | 48.0 | | |
| $K_{w2}$ | waveband 2, white ice light extinction | m$^{-1}$ | 20.0 | | |
| $K_{b1}$ | waveband 1, blue ice light extinction | m$^{-1}$ | 1.5 | | |
| $K_{b2}$ | waveband 2, blue ice light extinction | m$^{-1}$ | 20.0 | | Rogers et al., (1995), |
| $K_{s1}$ | waveband 1, snow light extinction | m$^{-1}$ | 6 | | Patterson and Hamblin (1988) |
| $K_{s2}$ | waveband 2, snow light extinction | m$^{-1}$ | 20 | | Ashton (1986) |
| $K_{snow}$ | molecular heat conductivity of snow | J m$^{-1}$ s$^{-1}$ °C $^{-1}$ | computed | dependent on snow density according to Eq. 35 | Yao et al., (2014) |
| $K_{ice_{white}}$ | molecular heat conductivity of white ice | J m$^{-1}$ s$^{-1}$ °C $^{-1}$ | 2.3 | | |
| $K_{ice_{blue}}$ | molecular heat conductivity of blue ice | J m$^{-1}$ s$^{-1}$ °C $^{-1}$ | 2.0 | | |
| $K_{water}$ | molecular heat conductivity of water | J m$^{-1}$ s$^{-1}$ °C $^{-1}$ | 0.57 | | |
| $K_{air}$ | molecular heat conductivity of air | J m$^{-1}$ s$^{-1}$ °C $^{-1}$ | 2.8×10$^{-3}$ | reported as 0.1 kJ m$^{-1}$ hr$^{-1}$ K$^{-1}$ | TVA (1972) |
| $K_{soil}$ | heat conductivity of soil/sediment | J m$^{-1}$ s$^{-1}$ °C $^{-1}$ | 1.2 | varies from 0.25 for organic soil to 2.9 for inorganic particles | |
| $K_{seep}$ | hydraulic conductivity of soil below the lake | m day$^{-1}$ | configurable | set in &outflows | |
| $Lat$ | Latitude | degrees, + for N | configurable | set in &morphometry | |
| $Long$ | Longitude | degrees + for E | configurable | | |
| $L_D$ | equivalent circular diameter of the lake | m | computed | | |
| $L_{META}$ | length of the lake at the depth of the thermocline region (metalimnion) | m | computed | | |
| $L_{outf}$ | length of the lake at the height of the relevant outflow | m | computed | | |
| $L_{crest}$ | length of the lake at the upper most height of the domain | m | configurable | | |
| $m$ | constant used to compute the rate at which work from the wind is converted | - | 4.6×10$^{-7}$ | | |





| Symbol | Description | Units | Value * | Comments | Reference |
|---|---|---|---|---|---|
| $N^2$ | the buoyancy frequency, a measure of water column stratification | s⁻² | computed | Eq. 53 | |
| $N^2_{outf}$ | the buoyancy frequency, a measure of water column stratification, about the layers impacted by the water outflow | s⁻² | computed | Eq. 74 | |
| $Oz$ | ozone concentration | atm-cm | 0.279 - 0.324 | set in &bird | Luo et al. (2010) |
| $p$ | air pressure | hPa | 1013 | assumed constant, or set in &bird | |
| $Q_{inf_x}$ | rate of a single water inflow provided by the user as input to the model | m³ day⁻¹ | time-series input | based on the non_avg flag in &glm_setup, the supplied value at the current time step or the average of the current and past time step is used, depending on whether the daily data are referenced from midday or midnight | |
| $Q_{outf_x}$ | rate of a single water outflow provided by the user as input to the model | m³ day⁻¹ | time-series input | | |
| $Q_{inf_0}$ | rate of a single water inflow prior to the inflow entering the lake | m³ s⁻¹ | computed | $Q_{inf_0} = f_{inf}\, Q_{inf_x}/c_{secday}$ | |
| $Q_{inf_j}$ | flow rate of inflow water parcel during transit, at the $j$ th increment | m³ s⁻¹ | variable | Eq. 67 used to increment between $j$ steps | |
| $\Delta Q_{inf_j}$ | rate of entrained water for an inflow at the $j$ th increment | m³ s⁻¹ | computed | | |
| $Q_{inf_{ins_I}}$ | flow rate of inflowing water at the point of insertion, for inflow $I$ | m³ s⁻¹ | variable | Figure 11 | |
| $Q_{outf}$ | rate of a single water outflow exiting the lake | m³ s⁻¹ | computed | $Q_{outf} = f_{outf}\, Q_{outf_x} / c_{secday}$ | |
| $Q_{outf_i}$ | flow rate of water being extracted from the $i$ th layer | m³ s⁻¹ | computed | $Q_{outf} = \sum_{i}^{N_{LEV}} Q_{outf_i}$ | |
| $Q_{ovfl}$ | rate of over flowing water leaving the lake | m³ s⁻¹ | computed | Eq. 77 | |
| $Q_{weir}$ | flow rate of water discharging over the crest, before flooding | m³ s⁻¹ | computed | Eq. 76 | |
| $Q_{seepage}$ | flow rate of water discharging from the lake bottom via seepage | m³ s⁻¹ | computed | Eq. 75 | |
| $Q_R$ | boundary run-off into the lake surface layer | m³ s⁻¹ | computed | Eq. 8 | |
| $R$ | dimensionless parameter describing a water withdrawal flow regime | - | computed | Eq. 70 | |
| $R_x$ | rainfall rate supplied in the input file | m day⁻¹ | time-series input | user supplied rainfall rate | |
| $R_L$ | rainfall intensity threshold before run-in occurs | m day⁻¹ | 0.04 | depends on land slope and soil type | |
| $R_{snow}$ | critical rainfall rate incident on snow that controls densification | m day⁻¹ | configurable | set in &snowice | |
| $R_F$ | rainfall rate entering the water column | m s⁻¹ | computed | Eq. 6 | |
| $^*R_F$ | rainfall rate incident on the ice/snow layer | m s⁻¹ | computed | $^*R_F = f_R R_x/c_{secday}$ | |
| $r$ | water vapour mixing ratio | - | computed | ratio of water mass to total air mass | |
| $Ri_{inf}$ | Richardson number of the inflow water | - | computed | Eq. 61 | |
| $Ri_B$ | bulk Richardson number of the atmosphere over the lake | - | computed | Eq. A34 | |
| $S_x$ | snowfall rate supplied in the input file | m day⁻¹ | time-series input | user supplied snowfall rate | |
| $S_F$ | snowfall rate entering the water column | m s⁻¹ | computed | Eq. 7 | |
| $^*S_F$ | snowfall rate incident on the ice/snow layer | m s⁻¹ | computed | $^*S_F = f_S S_x/c_{secday}$ | |
| $S_i$ | salinity of the $i$ th layer | ‰ | variable | | |
| $S_{inf_x}$ | salinity of water entering in an inflow | g m⁻³ | time-series input | | |

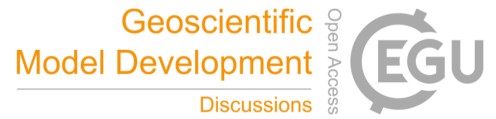

**Revision 26 Oct 2018**

| Symbol | Description | Units | Value * | Comments | Reference |
|---|---|---|---|---|---|
| $S_{outf}$ | salinity of outflowing water | g m⁻³ | variable | | |
| SZA | solar zenith angle | degrees | variable | $SZA = \Phi_{zen} 180/\pi$ | |
| $T_s$ | temperature of the surface layer | °C | variable | Eq. 9 | |
| $T_x$ | air temperature supplied by the user | °C | time-series input | user supplied air temperature | |
| $T_a$ | air temperature | °C | computed | $T_a = f_{AT} T_x$ | |
| $T_d$ | dew point temperature | °C | computed | | |
| $T_i$ | temperature of the $i$ th layer | °C | variable | | |
| $T_m$ | melt-water temperature | °C | 0 | | |
| $T_0$ | temperature at the solid (ice/snow) surface | °C | variable | | |
| $T_{inf_x}$ | temperature of water entering in an inflow | °C | time-series input | | |
| $T_{outf}$ | salinity of outflowing water | °C | variable | | |
| $T_{z_i}$ | temperature of the sediment zone $z$ which underlies layer $i$ | °C | computed | Eq. 40 | |
| $T_{z_{mean}}$ | annual mean temperature of the $z$ th sediment zone | °C | configurable | set in &sediment; corresponds to the sediment temperature at a depth of $\delta z_{soil}$ | |
| $\delta T_z$ | seasonal amplitude of the soil temperature variation | °C | configurable | set in &sediment | |
| $T_{aa}$ | aerosol absorptance of incoming light in the atmosphere | - | computed | Eq. A17 | |
| $T_{aerosol}$ | light scattering due to aerosols in the atmosphere | - | computed | Eq. A16 | |
| $T_{mix}$ | light absorptance due to mixed gases in the atmosphere | - | computed | Eq. A11 | |
| $T_{ozone}$ | ozone absorptance of incident light in the atmosphere | - | computed | Eq. A10 | |
| $T_{rayleigh}$ | Rayleigh scattering of incident light in the atmosphere | - | computed | Eq. A9 | |
| $T_{watvap}$ | absorptance of incident light in the atmosphere due to water vapour | - | computed | Eq. A14 | |
| $U_{10}$ | wind speed above the lake referenced to 10m height | m s⁻¹ | computed | wind speed corrected to reference height | |
| $U_x$ | wind speed above the lake surface provided by the user | m s⁻¹ | time-series input | user supplied snowfall rate | |
| $U_{orb_i}$ | orbital wave velocity experienced at the bottom of the $i$ th layer | m s⁻¹ | variable | Eq. 84 | |
| $U_{m_i}$ | mean layer velocity of the $i$ th layer | m s⁻¹ | variable | Eq. 86 | |
| $u_{inf_j}$ | average velocity of an inflow parcel being tracked, prior to insertion | m s⁻¹ | variable | Eq. 66 | |
| $u_*$ | friction velocity | m s⁻¹ | computed | Eq. 44 | |
| $u_b$ | velocity shear at the base of the thermocline | m s⁻¹ | variable | Eq. 46 | |
| $u_{b_{old}}$ | velocity shear at the thermocline at the previous time-step | m s⁻¹ | variable | reset between shear events | |
| $w_*^3$ | turbulent velocity scale within the surfaced mixed layer, due to convective cooling | m³ s⁻³ | computed | Eq. 43 | Imberger and Patterson (1981) |
| $W$ | total precipitable water vapour | atm-cm | 1.1 - 2.2 | Eq. A13 | Luo et al. (2010) |
| $Wm$ | atmospheric water mass factor, computed for calculating water scattering | - | computed | Eq. A12 | |

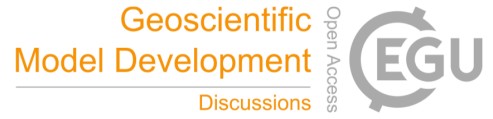

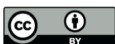

**Revision 26 Oct 2018**

| Symbol | Description | Units | Value * | Comments | Reference |
|---|---|---|---|---|---|
| $W_{crest}$ | width of the lake at the upper most point | m | configurable | set in `&morphometry` | |
| $W_{outf}$ | width of the lake at the height of an outflow | m | computed | Eq. 72 | |
| $W_{weir}$ | width of the weir crest | m | configurable | used if $H_{crest} < H_{max}$ ; set in `&outflow` | |
| $x_{WS}$ | default sheltering distance defined as the distance from the shoreline at which wind stress is no longer affected by sheltering | m | configurable | set in `&fetch`; approximated as 50× the vertical height of the sheltering obstacle/landform | Markfort et al. (2010) |
| $x_{WS}{}^\Phi$ | sheltering distance adjusted for changes in wind direction | m | computed | $x_{WS}{}^\Phi = x_{WS}\,(1 - min(f_{WS}[\Phi_{wind}], 1))$ | |
| $\delta x_{wave}$ | wave length of surface waves | m | computed | Eq. 81 | |
| $\Delta x_{inf_j}$ | lateral distance travelled by an inflow parcel per $j$ increment, prior to insertion | m | computed | Eq. 65 | |
| $\alpha_{inf}$ | angle describing the width of an inflow river channel | degrees | configurable | user supplied in `&inflow` based on width and depth of the relevant river | |
| $\alpha_h$ | coefficient for sensible heat flux into still air | J m⁻² s⁻¹ °C⁻¹ | computed | Eq. 29b | TVA (1972) |
| $\alpha_e$ | coefficient for evaporative flux into still air | m s⁻¹ | computed | Eq. 29a | TVA (1972) |
| $\alpha_{LW}$ | longwave albedo | - | 0.03 | | |
| $\alpha_{SW}$ | albedo of shortwave radiation at the water surface | - | computed | Eq. 13 for water; uses empirical algorithm for ice cover | |
| $\alpha_{SW_{mean}}$ | annual mean albedo of shortwave radiation | | 0.08 | active if `albedo_mode=1` in `&light` | |
| $\alpha_{SW_G}$ | function to compute albedo of shortwave radiation as a function of latitude and day number | | computed | function $\alpha_{SW_G}[d, Lat]$ is called if `albedo_mode=4` in `&light` | Table 5 in Cogley (1979) |
| $\alpha_{SKY}$ | scattered radiation within the sky | - | computed | Eq. A22 | Bird (1984) |
| $\alpha_{sed}$ | reflected fraction of radiation reaching the sediment | - | configurable | set in `&sediment`, and can vary with depth if sediment zones are configured | |
| $\alpha_b$ | interpolation coefficient for volume | - | computed | Eq. 3 | |
| $\beta_b$ | interpolation coefficient for area | - | computed | Eq. 3 | |
| $\delta_{wi}$ | length-scale associated with conduction of heat at the ice-water interface | m | 0.039 | | Rogers et al. (1995) |
| $\delta_{KH}$ | length-scale associated with formation of Kelvin-Helmholtz billows at the interface of two-layer stratification | m | computed | $\delta_{KH} = C_{KH} u_b^2 / g'_{EH}$ | Imberger and Patterson (1981) |
| $\delta_{epi}$ | length-scale associated with the epilimnion | m | computed | Eq. 49 | |
| $\delta_{hyp}$ | length-scale associated with the hypolimnion | m | computed | Eq. 49 | |
| $\delta_{outf}$ | length-scale associated with the vertical thickness of the zone of influence of a withdrawal | m | computed | Eq. 73 | Imberger and Patterson (1981) |
| $\delta_{outf_{top}}$ | thickness of withdrawal layer above the withdrawal height | m | computed | | |
| $\delta_{outf_{bot}}$ | thickness of withdrawal layer below the withdrawal height | m | computed | | |
| $\delta\alpha_{SW}$ | seasonal amplitude of albedo change | - | 0.01-0.08 | active if `albedo_mode=1` in `&light` | Cogley (1979) |
| $\delta_{ss}$ | particle diameter of bottom sediment | m | 80×10⁻⁶ | | |
| $\varepsilon_{TKE}$ | TKE dissipation flux per unit mass | m² s⁻³ | - | Eq. 54 | |
| $\overline{\varepsilon_{TKE}}$ | steady-state/equilibrium TKE dissipation flux per unit mass | m² s⁻³ | computed | Eq. 55 | |

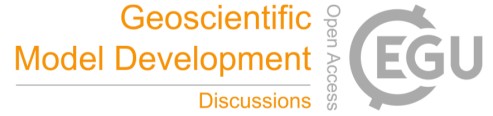



| Symbol | Description | Units | Value * | Comments | Reference |
|---|---|---|---|---|---|
| $\varepsilon_{WIND}$ | TKE dissipation flux created by power introduced by the wind | $m^2 \, s^{-3}$ | computed | Eq. 55 | |
| $\varepsilon_{INFLOW}$ | TKE dissipation flux caused by inflow plunging creating seiching | $m^2 \, s^{-3}$ | computed | Eq. 55 | |
| $\varepsilon_w$ | emissivity of the water surface | - | 0.985 | | |
| $\varepsilon_a$ | emissivity of the atmosphere under cloud-free conditions | - | computed | $\varepsilon_a^*[T_a, e_a, 0]$ | |
| $\varepsilon_a^*$ | emissivity of the atmosphere including cloud effects | - | computed | Eq. 21; options 1-4 are chosen via the cloud_mode variable in &meteorology | Henderson-Sellers (1986) |
| $\epsilon$ | factor for incident radiation from atmospheric calculation | - | computed | Eq. A18 | Bird (1984) |
| $\phi_{SW_x}$ | shortwave radiation flux provided in the input file | $W \, m^{-2}$ | time-series input | user supplied solar radiation data | |
| $\phi_{SW_0}$ | shortwave radiation flux crossing the water surface | $W \, m^{-2}$ | computed | Eq. 10 | |
| $\phi_{SW_S}$ | shortwave radiation flux heating the surface mixed layer | $W \, m^{-2}$ | computed | | |
| $\phi_{SW_{il}}$ | radiation flux at the top of the $i^{th}$ layer for the $l^{th}$ bandwidth class | $W \, m^{-2}$ | computed | bandwidth specific light attenuation is computed if light_model=2 in &light | |
| $\hat{\phi}_{SW}$ | total incident shortwave radiation flux computed from the BCSM assuming clear-sky conditions | $W \, m^{-2}$ | computed | Eq. 11 and Appendix A | Bird (1984) |
| $\hat{\phi}_{DB}$ | direct beam radiation on a horizontal surface at ground level on a clear day | $W \, m^{-2}$ | computed | Eq. A19 | |
| $\hat{\phi}_{AS}$ | radiation from atmospheric scattering hitting ground level on a clear day | $W \, m^{-2}$ | computed | Eq. A20 | |
| $\hat{\phi}_{ETR}$ | extra-terrestrial radiation hitting the top of the atmosphere | $W \, m^{-2}$ | computed | Eq. A1 | |
| $\phi_{PAR}$ | downwelling photosynthetically active radiation intensity within the water column | $W \, m^{-2}$ | computed | Eq. 14 | Kirk (1994) |
| $\phi_{PAR_{BEN}}$ | light incident on the bottom of a layer, corresponding to the benthic area | $W \, m^{-2}$ | variable | | |
| $\phi_{LWin}$ | longwave radiation incident heat flux at the water surface | $W \, m^{-2}$ | variable | Eq. 20 | |
| $\phi_{LWout}$ | longwave radiation outgoing heat flux from the water surface | $W \, m^{-2}$ | variable | Eq. 19 | |
| $\phi_{LW_{net}}$ | net longwave radiation flux across the lake surface | $W \, m^{-2}$ | computed | Eq. 18 | |
| $\phi_H$ | sensible heat flux across the water surface | $W \, m^{-2}$ | computed | Eq. 22 | |
| $\phi_E$ | latent heat flux | $W \, m^{-2}$ | computed | Eq. 23 | |
| $\phi_{E_0}$ | latent heat flux under zero-wind conditions | $W \, m^{-2}$ | computed | Eq. 28a | |
| $\phi_{H_0}$ | sensible heat flux under zero-wind conditions | $W \, m^{-2}$ | computed | Eq. 28b | |
| $\phi_X$ | generic identifier for either of $\phi_E$ or $\phi_H$ | $W \, m^{-2}$ | computed | | |
| $\phi_{X_0}$ | generic identifier for either of $\phi_{E_0}$ or $\phi_{H_0}$ | $W \, m^{-2}$ | computed | | |
| $\phi_X^*$ | maximum value of either $\phi_{X_0}$ or $\phi_X$ | $W \, m^{-2}$ | selected | Eq. 27 | |
| $\phi_0$ | upward conductive heat flux through the ice and snow cover to the atmosphere | $W \, m^{-2}$ | computed | Eq. 34 | |
| $\phi_{net}$ | net incoming heat flux at the ice-atmosphere interface | $W \, m^{-2}$ | computed | Eq. 33 | Rogers et al. (1995) |
| $\phi_R$ | heat flux due to rainfall | $W \, m^{-2}$ | computed | | Rogers et al. (1995) |

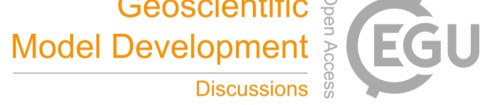

**Revision 26 Oct 2018**

| Symbol | Description | Units | Value * | Comments | Reference |
|---|---|---|---|---|---|
| $\phi_f$ | heat flux in the blue ice near the ice-water interface | W m⁻² | computed | Eq. 36 | |
| $\phi_w$ | heat flux from the water to the blue ice | W m⁻² | computed | Eq. 37 | |
| $\phi_{si}$ | heat flux per unit volume due to formation of white ice by flooding | W m⁻² | computed | | Rogers et al. (1995) |
| $\Phi_{zen}$ | solar zenith angle | radians | variable | Eq. A6 | |
| $\Phi_{day}$ | day angle | radians | computed | Eq. A2 | |
| $\Phi_{dec}$ | solar declination angle | radians | computed | Eq. A3 | |
| $\Phi_{hr}$ | hour angle | radians | computed | Eq. A5 | |
| $\Phi_{wind}$ | wind direction | degrees | time-series input | optionally provided as a boundary condition based on `fetch_mode`, set in `&fetch` | |
| $\Phi_{inf}$ | angle of the lake bed slope in the region where the inflow enters into the lake | degrees | configurable | user provided in `&inflow` | |
| $\kappa$ | von Karman's constant | - | 0.41 | | |
| $\lambda_v$ | latent heat of evaporation | J kg⁻¹ | 2.453×10⁶ | | |
| $\lambda_f$ | Latent heat of fusion | J kg⁻¹ | 3.340×10⁵ | | |
| $\lambda_{snow}$ | coefficient controlling the rate of densification of snow following rainfall | - | computed | $\lambda_{snow} = {}^*R_F\, c_{secday}/R_{snow}$ | adapted from Rogers et al. (1995) |
| $\Lambda$ | dimensionless variable associated with light penetration through ice required for heat conduction calculation | - | computed | | |
| $\theta_V$ | virtual temperature of the atmospheric boundary layer above the lake | K | computed | | |
| $\theta_a$ | temperature of the atmospheric boundary layer above the lake | K | computed | $\theta_a = f_{AT}T_x + 273.15$ | |
| $\theta_s$ | temperature of the atmosphere at the lake surface | K | variable | $\theta_s = T_s + 273.15$ | |
| $\theta_*$ | temperature scale | K | computed | | |
| $\rho_a$ | air density | kg m⁻³ | computed | computed as a function of air temperature, humidity and pressure in `atm_density` | TVA (1972) |
| $\rho_o$ | density of saturated air at the water surface temperature | kg m⁻³ | computed | | |
| $\rho_i$ | density of the $i^{th}$ layer | kg m⁻³ | variable | compute for each layer based on temperature and salinity, using TEOS-10 or UNESCO (1981); set `density_model` in `&glm_setup` | TEOS-10: teos-10.org UNESCO (1981) |
| $\rho_s$ | density of the surface water layer ($i=N_{LEV}$) | kg m⁻³ | variable | | |
| $\rho_w$ | reference water density | kg m⁻³ | 1000 | | |
| $\rho_{sml}$ | mean density of the mixed layer | kg m⁻³ | variable | | |
| $\rho_{ref}$ | average of layer densities over which reduced gravity is being computed | kg m⁻³ | computed | | |
| $\rho_{ice,snow}$ | density of the snow or ice | kg m⁻³ | selected | | |
| $\rho_{white}$ | density of white ice | kg m⁻³ | 890 | | |
| $\rho_{blue}$ | density of blue ice | kg m⁻³ | 917 | | |
| $\rho_{snow}$ | density of snow | kg m⁻³ | variable | | |

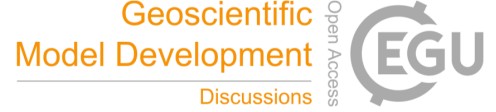

**Revision 26 Oct 2018**

| Symbol | Description | Units | Value * | Comments | Reference |
|---|---|---|---|---|---|
| $\rho_{s,min}$ | assigned minimum snow density | kg m$^{-3}$ | 50 | set in `&snowice` | |
| $\rho_{s,max}$ | assigned maximum snow density | kg m$^{-3}$ | 300 | set in `&snowice` | |
| $\rho_{snow*}$ | intermediate snow density estimate | kg m$^{-3}$ | computed | see Figure 6 | |
| $\rho_{outf}$ | density of the lake layer corresponding to the height of withdrawal, $i_{outf}$ | kg m$^{-3}$ | computed | | |
| $\rho_{i_j}$ | density of the lake layer, $i$, which is at an equivalent depth to inflow parcel $j$ | kg m$^{-3}$ | computed | | |
| $\rho_{inf}$ | density of inflowing water as it enters the lake | kg m$^{-3}$ | computed | | |
| $\rho_{ins_j}$ | density of inflowing water parcel at the $j^{th}$ increment during its transit | kg m$^{-3}$ | computed | | |
| $\rho_{ins_I}$ | density of the inflow parcel associated with inflow $I$ when it inserted | kg m$^{-3}$ | computed | | |
| $\rho_{i_{ins_I}}$ | density of the lake layer, $i$, where the inflow $I$ inserted | kg m$^{-3}$ | computed | | |
| $\sigma$ | Stefan-Boltzmann constant | W m$^{-2}$ K$^{-4}$ | 5.67×10$^{-8}$ | | |
| $\tau_i$ | total shear stress experienced at the lake bed portion of layer $i$ | N m$^{-2}$ | computed | Eq. 85 | |
| $\vartheta_s$ | dimensionless moisture content of air at water's surface | - | computed | $\vartheta_s = \kappa\, e_s / p$ | TVA (1972) |
| $\vartheta_a$ | dimensionless moisture content of the air above the lake | - | computed | $\vartheta_a = \kappa\, e_a / p$ | |
| $\nu_a$ | kinematic viscosity of air | m$^2$ s$^{-1}$ | 1.52×10$^{-5}$ | reported as 0.0548 m$^2$ hr$^{-1}$ | |
| $\nu_w$ | kinematic viscosity of water | m$^2$ s$^{-1}$ | 1.14×10$^{-6}$ | | |
| $\omega$ | ratio of molecular weight of water to molecular weight of air | - | 0.622 | | |
| $\psi_M$ | similarity function for momentum in the air above the lake | - | computed | Eq. A30 | |
| $\psi_E$ | similarity function for moisture in the air above the lake | - | computed | | |
| $\psi_H$ | similarity function for heat in the air above the lake | - | computed | | |
| $\xi$ | dimensionless parameter used for wave period calculation | - | computed | Eq. 80 | |
| $\zeta$ | dimensionless parameter used for wave period calculation | - | computed | Eq. 83 | |
| $\varsigma$ | percentage of atmospheric diffuse radiation | % | 6 | | Yajima and Yamamoto (2015) |

\* Either a numeric value for fixed constants or descriptors of the source of the value are provided. Descriptors include index, computed, configurable (default), variable, selected or time-series input, with supporting information in the comment column.

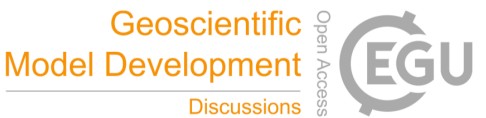

**Revision 26 Oct 2018**

**Appendix A: Bird solar radiation model**

The Bird Clear Sky Model (BCSM) was developed by (Bird, 1984) to predict clear-sky direct beam, hemispherical diffuse, and total hemispherical broadband solar radiation on a horizontal surface. Average solar radiation is computed at the model time-step (e.g., hourly) based on ten user specified input parameters (Table 1).

The solar constant in the model is taken as 1367 W m$^{-2}$, which is corrected due to the elliptical nature of the Earth's orbit and consequent change in distance to the sun. This calculation gives us the Extra-Terrestrial Radiation ($\hat{\phi}_{ETR}$), at the top of the atmosphere:

$$\hat{\phi}_{ETR} = 1367 \left(1.00011 + 0.034221\cos[\Phi_{day}] + 0.00128\sin[\Phi_{day}] + 0.000719\cos[\Phi_{day}]\right) \qquad \text{A1}$$

where the day angle, $\Phi_{day}$, is computed using, $d$, the day number:

$$\Phi_{day} = 2\pi\left(\frac{d-1}{365}\right) \qquad \text{A2}$$

The solar declination, $\Phi_{dec}$ (radians), is computed from:

$$\Phi_{dec} = \begin{array}{l} 0.006918 - 0.399912\cos[\Phi_{day}] + 0.070257\sin[\Phi_{day}] - 0.006758\cos[2\Phi_{day}] \\ + 0.000907\sin[2\Phi_{day}] - 0.002697\cos[3\Phi_{day}] + 0.00148\sin[3\Phi_{day}] \end{array} \qquad \text{A3}$$

We then solve the equation of time:

$$EQT = \begin{pmatrix} 0.0000075 + 0.001868\cos[\Phi_{day}] - 0.032077\sin[\Phi_{day}] \\ -0.014615\cos[2\Phi_{day}] - 0.040849\sin[2\Phi_{day}] \end{pmatrix} \times 229.18 \qquad \text{A4}$$

in order to compute the hour angle, $\Phi_{hr}$, is calculated with noon zero and morning positive as:

$$\Phi_{hr} = 15(hr - 12.5) + Long - 15\,TZ + \left(\frac{EQT}{4}\right) \qquad \text{A5}$$

where $TZ$ is the time-zone shift from GMT. The zenith angle, $\Phi_{zen}$ (radians), is calculated from:

$$\cos[\Phi_{zen}] = \cos[\Phi_{dec}]\cos[\Phi_{hr}]\cos[Lat] + \sin[\Phi_{dec}]\sin[Lat] \qquad \text{A6}$$

For $\Phi_{zen} < 90°$, the air mass factor is calculated as:

$$AM = \left(\cos[\Phi_{zen}] + \frac{0.15}{(93.885 - \Phi_{zen})^{1.25}}\right)^{-1} \qquad \text{A7}$$

which is corrected for atmospheric pressure, $p$ (hPa),

$$AM_p = \frac{AM\,p}{1013} \qquad \text{A8}$$

$AM_p$ is then used to calculate the Rayleigh Scattering as:
$$T_{rayleigh} = \exp\left[(-0.0903\,AM_p^{0.84}) + (1 + AM_p - AM_p^{1.01})\right] \qquad \text{A9}$$

The effect of ozone scattering is calculated by computing ozone mass, which for positive air mass is:

$$T_{ozone} = \left(1 - \left(0.1611\,(Oz\,AM)\left(1 + 139.48\,(Oz\,AM)\right)^{-0.3035}\right) - \frac{0.002715\,(Oz\,AM)}{1 + 0.044\,(Oz\,AM) + 0.0003\,(Oz\,AM)^2}\right) \qquad \text{A10}$$

The scattering due to mixed gases for positive air mass is calculated as:

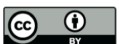

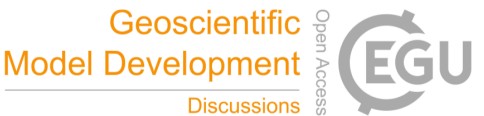

$$T_{mix} = \exp[-0.0127\ AMp^{0.26}]$$  A11

Then the water scattering is calculated by computing a water mass factor:

$$Wm = W AM_p$$  A12

where $W$ is the precipitable water vapour which defaults to 1.1. This can be approximated from dew point temperature, $T_d$, using:

$$\ln[W] = a_W T_d + b_W$$  A13

where regression coefficients of 0.09, 0.07, 0.07 and 0.08 for $a_W$, while $b_W$ is 1.88, 2.11, 2.12 and 2.01 in spring, summer, autumn and winter have been assumed (Luo et al., 2010). Then the water scattering effect is calculated as:

$$T_{watvap} = \left(1 - \frac{(2.4959\ Wm)}{1 + (79.034\ Wm)^{0.6828} + 6.385\ Wm}\right)$$  A14

The scattering due to aerosols requires the Aerosol Optical Depth at 380 nm and 500 nm:

$$TauA = 0.2758\ AOD_{380} + 0.35\ AOD_{500}$$  A15

and is calculated as:

$$T_{aerosol} = \exp[(-TauA)^{0.873}\ (1 + TauA - TauA^{0.7088})\ AM^{0.9108}]$$  A16

The absorptance of aerosols is then computed from:

$$T_{aa} = 1 - \left(0.1\ (1 - AM + AM^{1.06})\ (1 - T_{aerosol})\right)$$  A17

and we also define:

$$\epsilon = \frac{0.5(1 - T_{rayleigh}) + 0.84(1 - T_{aerosol}/T_{aa})}{1 - AM + AM^{1.02}}$$  A18

where the 0.84 value used is actually the proportion of scattered radiation reflected in the same direction as incoming radiation.

The direct beam (DB) radiation on a horizontal surface at ground level on a clear day is given by:

$$\hat{\phi}_{DB} = 0.9662\ \hat{\phi}_{ETR}\ T_{rayleigh}\ T_{ozone}\ T_{mix}\ T_{watvap}\ T_{aerosol}\ \cos[\Phi_{zen}]$$  A19

and the atmospheric scattering (AS) component is:

$$\hat{\phi}_{AS} = 0.79\ \hat{\phi}_{ETR}\ T_{ozone} T_{mix}\ T_{watvap}\ T_{aa}\ \cos[\Phi_{zen}]\ \epsilon$$  A20

The total irradiance hitting the surface (W m$^{-2}$) is therefore:

$$\hat{\phi}_{SW} = \frac{\hat{\phi}_{DB} + \hat{\phi}_{AS}}{1 - (\alpha_{SW}\ \alpha_{SKY})}$$  A21

The albedo is computed for the sky as:

$$\alpha_{SKY} = 0.068 + (1 - 0.84)\left(1 - \frac{T_{aerosol}}{T_{aa}}\right)$$  A22

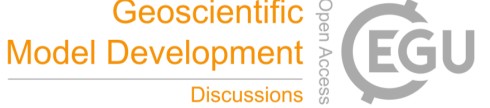

**Revision 26 Oct 2018**

**Appendix B: Non-neutral bulk transfer coefficients**

The iterative procedure used in this analysis to update correct the bulk-transfer coefficients based on atmospheric conditions is conceptually similar to the methodology discussed in detail in Launiainen and Vihma (1990). The first estimate for the neutral drag coefficient, $C_{DN}$, is specified as a function of wind speed as it is commonly observed to increase with $U_{10}$. This is
5    modelled by first estimating the value referenced to 10m height above the water from:

$$C_{DN-10}$$
$$= \begin{cases} \begin{cases} 0.001 & U_{10} \leq 5 \\ 0.001\left(1 + 0.07\left(U_{10} - 5\right)\right) & U_{10} > 5 \end{cases}, & \text{Option 1 : Francey and Garratt (1978), Hicks (1972)} \\ \\ 1.92 \times 10^{-7} U_{10}^3 + 0.00096 & , & \text{Option 2 : Babanin and Makin (2008)} \end{cases}$$

A23

and then computing the Charnock formula with the smooth flow transition (e.g., Vickers et al., 2013):

$$z_o = \frac{a u_*^2}{g} + 0.11 \frac{v_a}{u_*}$$

A24

where $a$ is the Charnock constant, and here $u_*$ is the approximated friction velocity of the atmosphere near the surface ($\sqrt{C_{DN-10} \, U_{10}^2}$), initially estimated using Eq. A23. The drag is re-computed using:

$$C_{DN-10} = \left(\frac{\kappa}{\ln\left[\frac{10}{z_o}\right]}\right)^2$$

A25

10    where $\kappa$ is the von Karman constant (Figure A1). Note the neutral humidity/temperature coefficient, $C_{HWN-10}$, is held constant at the user defined $C_H$ value and is assumed not to vary with wind speed.

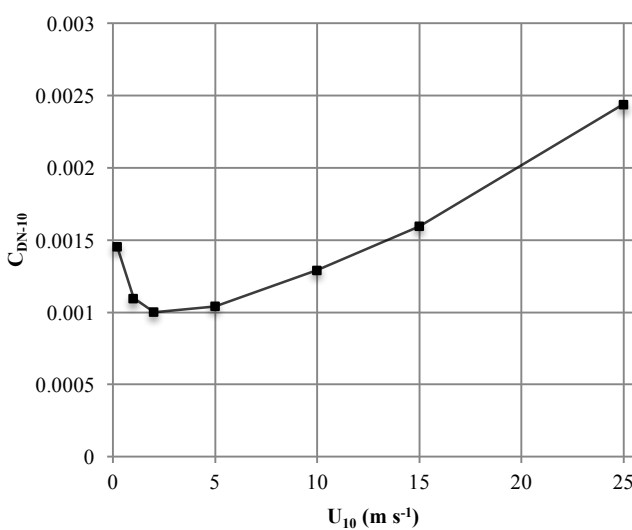

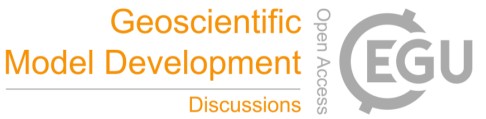

**Revision 26 Oct 2018**

**Figure A1: Scaling of the 10-m neutral drag coefficient with wind speed, $U_{10}$ (Eqns A23-25)**

Under non-neutral conditions in the atmospheric boundary layer, the transfer coefficients vary due to stratification in the air column, as was parameterised by Monin and Obukhov (1954) using the now well-known stability parameter, $z/L$, where $L$ is the Obukhov length defined as:

$$L = \frac{-\rho_a\, u_*^3\, \theta_V}{\kappa\, g \left( \frac{\phi_H}{c_a} + 0.61 \frac{\theta_a \phi_E}{\lambda_v} \right)} \qquad \text{A26}$$

where $\theta_V = \theta_a(1 + 0.61 e_a)$ is the virtual air temperature and $\phi_H$ and $\phi_E$ are the bulk fluxes. Paulson (1970) presented a solution for the vertical profiles of wind speed, temperature and moisture in the developing boundary layer as a function of the Monin-Obukhov stability parameter; the so-called flux-profile relationships:

$$U_z = \frac{u_*}{\kappa}\left( \ln\left[\frac{z}{z_o}\right] - \psi_M\left[\frac{z}{L}\right] \right) \qquad \text{A27a-c}$$

$$\theta_a - \theta_s = \frac{\theta_*}{\kappa}\left( \ln\left[\frac{z}{z_\theta}\right] - \psi_H\left[\frac{z}{L}\right] \right)$$

$$e_a - e_s = \frac{e_*}{\kappa}\left( \ln\left[\frac{z}{z_q}\right] - \psi_E\left[\frac{z}{L}\right] \right)$$

where $\psi_M$, $\psi_H$ and $\psi_E$ are the similarity functions for momentum, heat and moisture respectively, and $z_o$, $z_\theta$ and $z_q$ are their respective roughness lengths. For unstable conditions ($L<0$), the stability functions are defined as (Paulson 1970; Businger et al., 1971; Dyer, 1974):

$$\psi_M = 2\ln\left(\frac{1+x}{2}\right) + \ln\left(\frac{1+x^2}{2}\right) - 2\tan^{-1}x + \frac{\pi}{2} \qquad \text{A28a}$$

$$\psi_E = \psi_H = 2\ln\left(\frac{1+x^2}{2}\right) \qquad \text{A28b}$$

where

$$x = \left[1 - 16\left(\frac{z}{L}\right)\right]^{1/4} \qquad \text{A29}$$

During stable stratification ($L>0$) they are assumed to take a form modified from Hicks (1976):

$$\psi_M = \psi_E = \psi_H = \begin{cases} -5\left(\frac{z}{L}\right) & 0 < \frac{z}{L} < 0.5 \\ 0.5\left(\frac{z}{L}\right)^{-2} - 4.25\left(\frac{z}{L}\right)^{-1} - 7\ln\left[\frac{z}{L}\right] - 0.852 & 0.5 < \frac{z}{L} < 10 \\ \ln\left[\frac{z}{L}\right] - 0.76\left(\frac{z}{L}\right) - 12.093 & \frac{z}{L} > 10 \end{cases} \qquad \text{A30}$$

Substituting Eqns. 22-23 into A27 and ignoring the similarity functions leaves us with neutral transfer coefficients as a function of the roughness lengths:

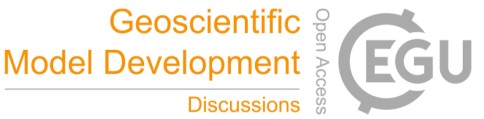

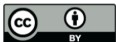

**Revision 26 Oct 2018**

$$C_{XN} = \kappa^2 \left( \ln\left[\frac{z}{z_o}\right] \right)^{-1} \left( \ln\left[\frac{z}{z_X}\right] \right)^{-1} \tag{A31}$$

where the $N$ sub-script denotes the neutral value and $X$ signifies either $D$, $H$ or $E$ for the transfer coefficient and $o$, $\theta$ or $q$ for the roughness length scale. Inclusion of the stability functions into the substitution and some manipulation (Imberger and Patterson, 1990; Launianen and Vihma, 1990) yields the transfer coefficients relative to these neutral values:

$$\frac{C_X}{C_{XN}} = \left[ 1 + \frac{C_{XN}}{\kappa^2} \left( \psi_M \psi_X - \frac{\kappa \psi_X}{\sqrt{C_{DN}}} - \frac{\kappa \psi_M \sqrt{C_{DN}}}{C_{XN}} \right) \right] \tag{A32}$$

Hicks (1975) and Launianen and Vihma (1990) suggested an iterative procedure to solve for the stability corrected transfer coefficient using Eq. A32 based on some initial estimate of the neutral values (as input by the user). The surface flux is subsequently estimated according to Eqns. 22-23 and used to provide an initial estimate for $L$ (Eq. A26). The partially corrected transfer coefficient is then recalculated and so the cycle goes. Strub and Powell (1987) and Launiainen (1995), presented an

10    alternative based on estimation of the bulk Richardson number, $Ri_B$, defined as:

$$Ri_B = \frac{gz}{\theta_V} \left( \frac{\Delta\theta + 0.61\, \theta_V \Delta e}{U_z^2} \right) \tag{A33}$$

and related as a function of the stability parameter, $z/L$, according to:

$$Ri_B = \frac{z}{L} \left( \frac{\kappa \sqrt{C_{DN}}/C_{HEN} - \psi_{H,E}}{\left[ \kappa/\sqrt{C_{DN}} - \psi_M \right]^2} \right) \tag{A34}$$

where it is specified that $C_{HN} = C_{EN} = C_{HEN}$. Figure A2 illustrates the relationship between the degree of atmospheric stratification (as described by both the bulk Richardson number and the Monin-Obukhov stability parameter) and the transfer coefficients scaled by their neutral value.

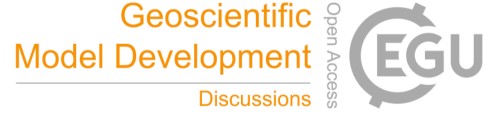

**Revision 26 Oct 2018**

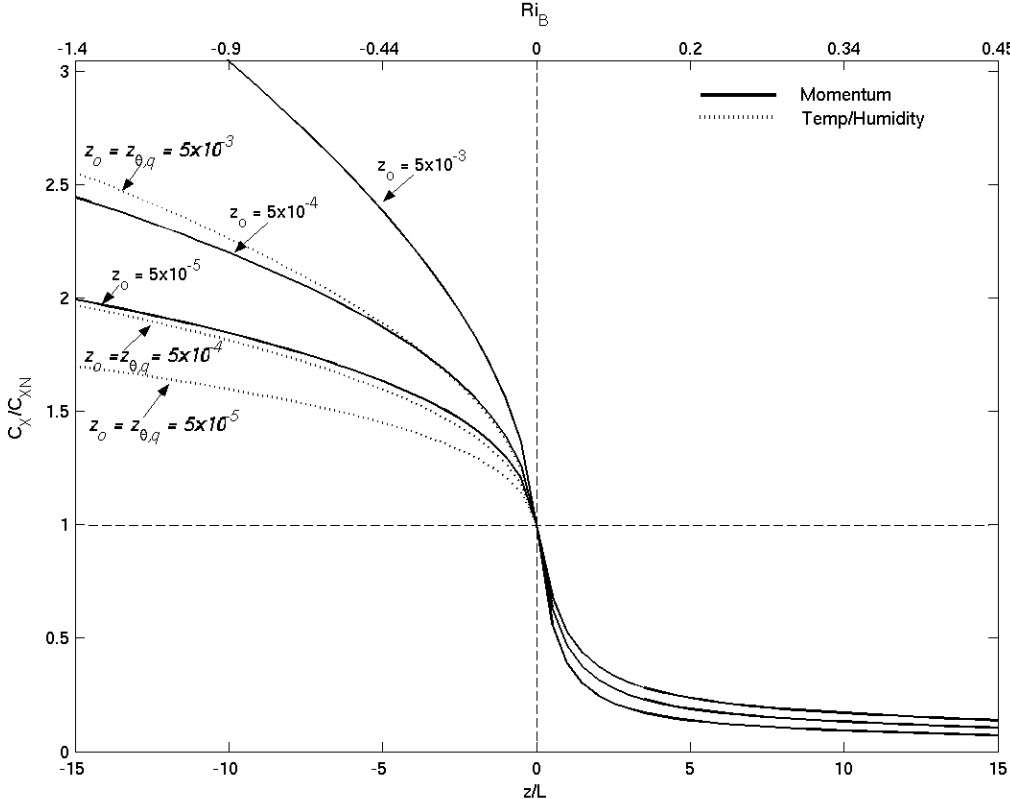

**Figure A2: Relationship between atmospheric stability (bottom axis – $z/L$, top axis – $Ri_B$) and the bulk-transfer coefficients relative to their neutral value ($C_X/C_{XN}$ where $X$ represents $D$, $H$ or $E$) for several roughness values (computed from Eq. A32). The solid line indicates the momentum coefficient variation ($C_D/C_{DN}$) and the broken line indicates humidity and temperature coefficient ($C_{HE}/C_{HEN}$) variation.**