# Peer review of "A General Lake Model (GLM 3.0) for linking with high-frequency sensor data from the Global Lake Ecological Observatory Network (GLEON)"

_Geoscientific Model Development, 2017_

## Referee Comment (RC1) · Anonymous Referee #1 · 25 Nov 2017

Review of "A General Lake Model (GLM 2.4) for linking with high-frequency sensor data from the Global Lake Ecological Observatory Network (GLEON)" by Hipsey et al.

This article describes the scientific basis of a 1-dimensional hydrodynamic lake model that can be coupled to ecosystem models. The model has already been applied to many systems in the scientific community, and I think it is useful publish the model description in a scientific paper that can be referred to for future applications of the model. That said, I stopped reviewing after equation 16, because there were simply too many errors in the equations. I therefore propose to reject the current version of the manuscript and that the authors carefully check all equations before resubmitting the manuscript to this or another journal, depending on the decision of the editors.

Errors in equations up to eq. 16:

eq. 2 and 3: I think something is wrong with the indices. $h_z$ is located between $h_{b-1}$ and $h_b$. $\alpha_b$ and $\beta_b$ describe the interpolation between $h_b$ and $h_{b+1}$. Thus, the indices in eq. 2 should be $\alpha_{b-1}$ and $\beta_{b-1}$.

eq 6: I think this equation is wrong. The right hand side is the total heat flux to the surface layer in W $m^{-2}$. This should be divided by $z_{msl}$ to get W $m^{-3}$. Then, it should be divided by the water density $\rho$ in kg $m^{-3}$ to get W/kg, and finally by $c_p$ to get °C/s for $dT_s/dt$. Therefore, the multiplication term on the left hand side should be $z_{msl}\, c_p\, \rho$, rather than $c_p/(A_S\, z_{msl})$.

eq 9b / Fig. 3: I could not reproduce the maximum of the Briegleb function at 80 degrees zenith angle. Using equation 9c yielded a monotonically increasing function between 20 and 90 degrees (with a minimum at about 20 degrees). Also the equation in the legend is wrong, it should be SZA = $\Theta_{zen}*180/2\pi$.

eq. 12: I think $\phi_{SWS}$ (i.e. the shortwave radiation absorbed in the surface layer) should be replaced by $\phi_{SW}(z=0)$ in the nominator. Otherwise, the euphotic depth increases with increasing radiation absorbed in the surface layer, which does not make sense. Same in caption to Fig 4.

eq. 16: I think in equations 16c and 16d $T_a$ should be replaced by absolute temperature (i.e., 273.15 °C should be added to $T_a$).

Also units should always be provided, especially for empirical equations (e.g. $e_a$ in eq. 16, and $U_x$, RH, and diffusive radiation in eq. 9c).

Besides that, a few other points I noticed up to page 11 (Page xx, Line yy is abbreviated as xx/yy)

In general, the paper is well written and easy to read, but there are quite a few long and complicated sentences which I think should be simplified to facilitate reading (first two examples: 2/24-29 , 3/12-17).

3/10: This list of references seems to be somewhat inconsistent. Some of the references refer to model development, some to model applications. It would be more logical to cite only model development references.

4/30: The text seems to imply that the requirement for site-specific calibration in other models is due to numerical diffusion caused by the fixed grids. If that is the intention, this should be explained. If not, the sentence should be modified.

5/7: Incomplete sentence

Figure 1: Shouldn't the local runoff, and the submerged inflows and groundwater seepage be written in blue?

eq. 1: From the text (layer volumes are determined ...), I would have expected an equation for the individual layer volumes here, but this is the integrated volume from the bottom of the lake to the top of each layer. This should be clarified in the text.

6/3: technically, it is the same, but I think it would be clearer to write $2 \leq b \leq N_{BSN}$.

6/4: how are these finer depth increments determined?

6/9: Since the Unesco (1981) equation has been replaced by TEOS-10, I think it would make sense to use the latter rather than the former in a new model. I also think it should be mentioned that the density effect of salinity in these seawater equations is quite different from that in most lakes where carbonates are usually the dominant species rather than NaCl.

6/24: heat balance **of** the surface layer

7/2-3: why is only rain but not snow multiplied with $f_R$? Also, even though this should be clear to the reader, it should probably be mentioned that S is in water equivalents.

eq 5: to be precise, this equation should be limited to a minimum of zero, as otherwise it will become negative if the rainfall is too weak.

Fig 2: Add some space between the 10 and the exponent in the y-axes labels of panels c and d. Do all these time series start on 1 January of a year?

eq. 9a: instead of subtracting $\pi/2$ within the sine functions, it would be easier to use -cos.

eq. 9b: where does the factor 1.1 in the nominator of the first term come from? Maybe I overlooked something, but I could not find it in Briegleb et al. (1989).

eq. 9c: I was not able to check this equation, as the source is in Japanese, but I did not get anything similar to what is shown in Fig. 3 trying different values for RH, U and the diffusive radiation. Please check whether the equation is correct, and specify the values used to produce Fig. 3. Furthermore, Yajima and Yamamoto is dated 2014 here but 2015 in the reference list.

Fig 7: y-axis of panel b is not depth, but elevation, y-axis of panel c is not labeled. Also it seems that $A_{BEN}$ is calculated on a different time scale than the radiation in (c). Many low radiation events are clearly visible in (b) but do not show up in (c). This probably makes sense, but the time scale should be mentioned somewhere.

11/11: It does not look like the equations were copied from Henderson-Sellers (1986), but rather from either the original sources or from Flerchinger (2009)?

---

## Editor Comment (EC1) · S. Unterstrasser (Editor) · 26 Nov 2017

Dear Authors

Referee 1 pointed out several errors, in particular in equations and interrupted his/her review. I propose the following procedure for the up-coming review process. Please revise your manuscript accordingly and upload it as soon as possible. The reviewers wait until this happens and continue their reviews based on the new version. Depending on your submission date, I would extend the discussion phase such that the reviewers have enough time and a rigorous review is ensured.

[Figure]

Best wishes, Simon
* * *

---

## Short Comment (SC1) · 13 Dec 2017

GMD is encouraging authors to provide a persistent access to the exact version of the source code used for the model version presented in the paper. As explained in https://www.geoscientific-model-development.net/about/manuscript_types.html the preferred reference to this release is through the use of a DOI which then can be cited in the paper. For projects in GitHub a DOI for a released code version can easily created using Zenodo, see https://guides.github.com/activities/citable-code/ for details. Please note that in the code accessibility section you can still point the reader to the GitHub repository for the newest version even if you use a DOI for the relevant release.

[Figure]

Lutz Gross GMD Executive Editor

---

## Referee Comment (RC2) · Anonymous Referee #2 · 3 Jan 2018

Review of "A General Lake Model (GLM2.4) for linking with high-frequency sensor data from the Global Lake Ecological Observatory Network (GLEON)" by Matthew R. Hipsey et al.

This paper presents the formulation of a one-dimensional model of thermodynamics, mixing, and evaporative and momentum fluxes from lake surfaces, which conceptually should be applicable to a wide variety of lake morphologies. However, the manuscript achieves the paradox of simultaneously containing too much information and not enough information. It goes into great detail with many equations. However, in order to present this level of detail without losing the reader requires more care and at

least some additional details.

I have a few broad suggestions that I think might help.

Before anything, you need to have a clear idea of who is in your audience. You could even have an explicit statement of this very near the beginning, and direct users with a lower level of expertise toward a simpler users' guide. This reviewer has a background explicitly in meteorology, but significant exposure to lake dynamics as well, albeit mostly regarding very large lakes. Depending on your intended audience, you might have readers who will have difficulty with terms such as aliquot and even the intended understanding of "scalar concentrations" as used on p. 27, lines 2-3.

To reduce the length of a single paper, one option is to break it into multiple papers. Another is to move more of the detail to appendices. Since it is an online journal, there is probably not a large problem with overall page length. Things as detailed as conversion of area as a function of depth to volume in a layer (eq. 1) seem like they could be skipped in the main text and relegated to an appendix.

It would be useful to have a brief introduction to each sub-section. This should start out by stating the goal of that sub-section, i.e. what will be the final equation (or set of equations) derived in the section. Then state what elements will combine to get that final equation(s). One particularly glaring issue is that sub-section 2.5.2 ends with eq. 45, defining the variable f, but nowhere does it say how f relates to any other part of the model. It seems to be something that one would multiply by the difference in temperature (or another scalar) between layers) to get the exchange of that scalar between the two layers in a single time step. Whether this is exactly correct or not, the statement is missing from the manuscript.

A simple overall schematic would be good to have early on (Fig. 15 with less detail). This would make it less abrupt when "water quality model AED2" is mentioned on p. 27 (I may have missed it, but I don't think it was mentioned before.

[Figure]

Even if only for your own reference as author, Table 1 needs to be expanded to include every variable used in every equation! And in this expanded table, include every variant of each variable based on the use of different subscripts, prime, and circumflex ("hat"). With this many equations, it is rather inevitable that you also end up using the same symbol for different things (I noticed N in particular). Then, for each variant of each variable, put additional columns for: description, units, spatial type (defined at surface only, spatially continuous, or at discrete layers), and which equations it is used in. The reader needs to have all of these carefully defined.

My high school chemistry teacher is the one who taught me how to use units in equations, and the importance of doing so, hence the need for them in the big table above. Some examples of problems with units in the equations that may indicate that the equation is simply wrong: In eq. 52, Q seems like it should have units of m cubed per s, and h units of m, so the right-hand side does not have units of velocity.

I approached this manuscript with the immediate question of what makes this model better and more useful than the many others that are available. The introduction does a pretty good job of answering this, but it may be good to give some examples of uses that are not satisfied by other models. This could be introduced with a schematic of its components, options, and functions, at a lower level of detail than in Figs. 15 and 16.

Anything that can be done to bring the reader's intuition into play when introducing equations will improve comprehension of the manuscript. Eq. 27 isn't the most problematic one, but I'll use it as an example. You might introduce it by saying something like: "Shortwave radiation is absorbed and attenuates with different e-folding depths for snow, white ice, and blue ice, and these also depend on the light's wavelength. The overall effect is..." Eq. 47 has me very mystified about how a standard definition of Richardson's number translates into this equation in terms of the angles of inflow geometry (part of the problem may be that I don't feel like I understand the meaning of the angle labeled alpha in Fig. 10; the illustration isn't helping me). It also has me asking "Richardson number at what location?" At the interface of the river water and ambient

lake water?

Several of the figures have characters that are so small as to be illegible.

What about modularity of the model? Can other schemes for pieces of the model be plugged in? The details of how may be an appendix or even another paper or guide.

Some particular examples of the general problems above are among the specific comments below.

Specific comments:

1. In nearly every review that I do, I refer to the rules of hyphens found at http://www.grammarbook.com/punctuation/hyphens.asp particularly Rule 1 on that page. Your manuscript is actually better than most, but here are the problems that I found: P. 2, line 23 should have "system scale" (no hyphen). "Time-scale" is sort of borderline; I tend to use it without a hyphen when it doesn't modify another noun. P. 4, line 15 and some figure captions: "time series". P. 13, lines 22 and 24: "Wind sheltering". P. 24, line 3: "user-specified".

2. P. 2, line 33: Change "spatial" to "vertical", to contrast with horizontal from earlier in the sentence.

3. P. 3, line 10: Stepanenko is misspelled.

4. P. 3, lines 17-18: Do you have any comment on use in small vs. large lakes? Both in terms of area and depth? There might also be considerations in terms of morphological complexity.

5. P. 3, lines 30-31 specifically mention temperature, salinity, and density. Later on, there is mention of the broader category of "scalars". Are you unnecessarily limiting yourself here? I am thinking of such things as dissolved oxygen (mentioned later) and concentrations of nutrients and contaminants.

6. P. 3, line 32: I think many readers will object to the use of "hydrodynamic" here,

since it does not explicitly represent advection of water. Continuity equations say that dynamics requires at least two dimensions.

7. P. 4, lines 31-32: "user-defined" and "set by the user" are redundant.

8. P. 5, line 8: This model doesn't include it, but in reality, vertical advection of heat can be important along with vertical mixing. Again this may depend on the size of the lake, but this factor should be acknowledged here.

9. Eq. 4 has issues with sign conventions, units, and possible missing terms depending on how one understands it. It is strange to have a positive sign on E in the right-hand side—this means that E is ordinarily a negative number. The units need to be very explicit—length per time, such as mm per day. Because of these units, E should not be called "mass flux". E multiplied by lake area should be considered a volume flux. For practical purposes these can be considered equivalent, but there would be another term for water density to convert between mass flux and volume flux. Are the evaporation, snowfall, and rainfall defined as being specifically over the area of the lake itself, or over the drainage basin? If over the lake, how is the Q term for runoff optional? I have a hard time imagining many cases of lakes in which it is not a major term in the water balance. If these variables are defined over the whole basin, is there an issue with agreement in timing between P – E and runoff?

10. P. 7, line 21: "However" is an interjection, not a conjunction. Here, it stands between two independent clauses, so preceded with either a semicolon or period. Also p. 33, line 10 and p. 35, line 11.

11. Eq. 7 is a place where I started to distinctly feel the problem of definition of variables. I had to really think through what had happened to the subscript S from eq. 6 and why the "hats" were there in eq. 7.

12. Eq. 9a has a strange step function in space. Can you justify this not being a smooth function, but rather one value for all of the northern hemisphere, another for the equator

(an infinitesimal amount of space), and another for the southern hemisphere?

13. P. 9, line 6: What makes the different formulas oceanic and lacustrine? Is there anything to help the reader's intuition for why these formulas should be different?

14. Audience issue: P. 10, line 14 departs from the main conceptual theme to give a specific file name. Detailed mathematical formulation and detailed user information don't mix very well.

15. Eq. 14: Are you using a different temperature at the skin (interfacial layer) or bulk temperature a little deeper? Specify how deep.

16. P. 11, line 10: Air temperature at how high above the surface? Standard is 2 m, but there may be adjustment needed if measurements are at a different height than that intended in the formulation.

17. Eq. 163: "Brutsaert" is misspelled.

18. P. 11, line 13: The range of octals should be 0 to 8.

19. P. 12, line 11: This should specify molecular weight of dry air, and it might be better to say mass rather than weight.

20. P. 13, lines 5 and 9: I think these should reference eqs. 17-18 rather than 16-17.

21. Eq. 23: The Latin v and Greek nu are somewhat difficult to distinguish here, and are even more difficult to distinguish in the text following.

22. P. 13, lines 16-18: Thanks for explicitly using units here. Am I correct in understanding that molecular heat conductivity is molecular heat diffusivity multiplied by heat capacity? I think it would be better not to use heat conductivity, but use diffusivity and capacity.

23. P. 13, line 22: "may be" should be two words.

24. P. 15, line 13: This use of "conductive heat flux from the ice or snow cover to the at-
mosphere" particularly triggers my thought bias as a meteorologist. That bias prompts me to assume that this means flux through the atmosphere's interfacial layer, where molecular diffusion dominates. But I also wonder whether it means heat conduction through the ice. These are not equal in general, and need to be distinguished from each other.

25. Eq. 26: Why isn't shortwave radiation here?

26. P. 16, lines 11 and 12: These are very wide ranges of albedo. Please describe the conditions under which different parts of this range manifest.

27. P. 18, line 10: The idea of energy required for mixing needs to be introduced more carefully. If you think of the water column as continuous in space, mixing is also a continuum. But the concept here is based on discrete layers, and this indicates how much energy is required to outright include a model layer in the mixed layer.

28. Eq. 31: C sub T seems completely undefined.

29. P. 20, line 14: This cannot be reproduced unless your definition of "epilimnion" and "hypolimnion" are precisely stated.

30. P. 20, line 20: Is K sub epsilon eddy diffusivity.

31. Since the vertical axis in Fig. 8 has zero at the bottom, it appears to be height above the bottom, not depth. It might be worth explaining that the varying top height of the color fill is due to varying overall depth (assuming that this is correct).

32. P. 22, line 13 and eq. 43: This seems to say that sigma should have units of inverse seconds squared, which implies that the square-bracketed part of eq. 43 has units of (m/s) squared, but exponential operations can only be performed on unitless numbers.

33. P. 24, line 5: Slope is usually defined as a ratio of rise divide by run, not an angle. I would say that the slop is the tangent of the angle.

34. The inset in Fig. 10 does not help me understand the meaning of the angle alpha,

and it could use larger lettering.

35. Equation 52 seems messed up. Units-wise, it seems to make more sense if h squared is in the denominator instead, but I also wonder why there is no dependence on width of inflow.

36. P. 26, line 16: In standard usage, dz represents an infinitesimal difference in z in continuous space, but here it is used to indicate a finite distance in discretely sectioned space.

37. P. 26, lines 16-17: It is difficult to understand the distinction between "height of withdrawal" and "edge of the withdrawal layer".

38. P. 26, line 19: Why say "fluid" rather than "water"?

39. A formatting problem put some labels that belong to Fig. 12 on p. 27, while the rest of the figure is on p. 28.

40. P. 28, lines 10-11: Removing no more than half of a layer's mass per time step seems like a reasonable way to ensure numerical stability, but it would be good to remind the reader here of the layer merging scheme that is likely to kick in. This merging and disaggregation scheme, also mentioned on p. 34, is never really described well.

41. P. 29, line 3 says "wind speed and fetch...calculated as", while eq. 61 only shows the formula for fetch.

42. To help solve the problem of small print in Fig. 16, it may be useful to transfer some of the information to a table instead.

43. P. 34, line 19 has a placeholder for a citation. This is evidence of a poor final edit on the part of the authors.

44. P. 34, line 23: If you mention calibration, it would be well to describe this process more fully, in particular which parameters you consider adjustable for purposes of calibration. Where p. 35, line 10 mentions "compare", I wonder whether this might also

imply calibration.

45. P. 48, line 12: "Anneville" misspelled?

---

## Referee Comment (RC3) · Anonymous Referee #3 · 8 Jan 2018

General Comments

This paper describes the detailed functioning of the 1D physical lake model GLM 2.4 and its application potential. The model incorporates a broad range of physical processes as surface heat exchange, snow and ice dynamics, in- and outflow, submerged inflow and groundwater seepage and can be coupled with or embedded into other models. The authors explain how GLM 2.4 has emerged as a response to the need of standardized, yet flexible and computationally effective community lake model to interpret environmental data from a broad range of lakes collected within the Global Lake Ecological Observatory Network (GLEON). The model has been formulated as a new

code in 2012, whereas layer structure, mixing algorithms and physical formulae are based on earlier peer reviewed work. The authors state that the code is computationally efficient and well suited for embedding in larger scale modelling frameworks. The authors present also an overview of pre- and post-processing utilities as well as an innovative cloud computing environment. Lastly, they elaborate on the educational use and gained experience in the classroom.

I realized that this manuscript is for a major part equivalent to an earlier manual of GLM (V2 Manual, October 2014, accessed on the 08.01.2017 from http://aed.see.uwa.edu.au/ research/models/GLM/Pages/ documentation.html). I think the authors should mention this.

The model in this paper represents with no doubt a tremendous effort in lake modelling and is of interest for modelers in various fields of environmental research. The publication of this model is a step towards better model documentation and contributes to the general scientific discussion and better lake model development. As such it falls within the scope of this journal. The paper is well written and the language is easily comprehensible. Unfortunately, this manuscript has some structural problems and there are quite a few mistakes in equations and figures. After dealing with these issues, the manuscript should be good for publishing. The main problem of this very long manuscript is that it is missing an instant overview of what is in the paper and what not. Scanning through, the reader gets lost easily in the large chapter 2 'model overview' and might miss the subsequent chapters that elaborate more on the possibilities and significance of this model for the scientific community.

I think that this problem can be fixed with some changes in the introduction: • I suggest using subtitles in the introduction. (In the introduction, the authors describe the importance of the study of lakes, the importance of GLEON, the importance of lake models, the advantages of simple models, applications and features of 1D models, the need for a flexible open-source model, how GLM 2.4 answers this need and finally an overview of the paper) • I suggest creating a new paragraph starting at p. 3 Line 19

"Nonetheless, there . . .". The need of an open source and flexible community model that can be applied to various lakes should be highlighted better. Another additional paragraph could explain how GLM 2.4 responds to this need. As I understand, GLM 2.4 is filling the gap because it provides a standard middle complexity physics 'shell' (simple yet enough complex to be applied for various lakes) that can be connected individually to or implemented into various other models (e.g. water quality or land-climate models). I think this point could be emphasized. • A figure could be helpful to draw attention to the significance of this model in the scientific community. This could also be combined with schematic overview of the model functioning (I agree with R2 that anything that gives an overview helps). • The specific limitations of GLM 2.4 (not of 1D models in general) should be mentioned in the introduction. Like this, the reader may have a quick idea whether GLM is suitable for him/her. What are the key features of this model that set it apart from other models? • On p. 4 lines 5-9, the authors explain the aims of the paper and in which of chapter 2-6 these aims are met. I think these lines are important and should be extended to a paragraph by itself to make sure the reader is fully aware what to expect from the paper. In the same paragraph, I would also expect some more information regarding what this paper is not about and mention that a companion paper by Bruce et al. (2017) is assessing the model's error structure against 31 GLEON lakes.

I think the authors did not carefully go through the complete manuscript. Many of the empirical equations are missing the definition of units for used variables. On other occasions variables where poorly described (see the examples listed below, as well as listed by R1 and R2). On several figures elevation and not the labeled depth is shown on y axis. The references are not formatted coherently. Like R2, I am of the opinion that many variable symbols are confusingly similar and that they should all be listed in a table. I also agree with R2 that all the subchapters of chapter 2 should have a small introduction paragraph. Further, I agree with the comments of R1 on the equations 1, 2, 3, 5, 9c, 12, 16 and with the comments of R2 on the equations 4, 7, 14, 26, 31, 52.

Specific comments

I think it is not very clear how the amalgamating, expanding, contracting or splitting and adding of layers works. For example, in p. 23 L 21 it is not obvious what the mentioned 'numerical criteria within the model' are. I would explain these in detail somewhere in the beginning of chapter 2.

p. 6 eq 2 and eq 3: It seems odd that the interpolation of values between levels b-1 and b are depending on b-1, b and b+1 and not only on b-1 and b.

p. 10 Line 5-8: $\varphi$SWS is defined only in text form and not as an equation, yet it is used in equation 6. There is the danger that $\varphi$SW defined in eq 10 will be confused with $\varphi$SWS. I suggest mentioning early on in this subchapter how you approach calculating $\varphi$SWS.

p. 12 eq 17: formula only for forced convection? Wind speed at what height? What are the units? I would introduce first the concepts of sensible heat (free and forced convection) and latent heat (evaporation and condensation) before showing the equations.

p. 18 L10: An intro with possible conceptual options to reproduce a surface mixed layer would be good. I would like to know how the chosen approach of a bulk mixed layer depth compares to other approaches in other models (e.g. k-epsilon turbulence closure with Fickian diffusion) and what the consequences of this approach are.

p. 22 L 15 and eq 44 and eq 45: I think an explanation of the concept behind this numerical scheme is necessary

p. 24. Figure 10: This figure is not enough self-explanatory to me.

p. 28 eq 60: Shouldn't G not just be another term in eq 4 for all cases?

p. 40 lines 21-24: move this sentence to the intro

p. 40 L 24 – 26: This needs to be better explained.

List of Corrections

p. 1 lines 31-32. Consider splitting sentence as it contains different ideas.

p. 2. Line 1: write only 'standing' as this word is comprehensible and you don't use lentic in the rest of the text.

p. 5 Line 17. Write the definite instead of the indefinite integral or otherwise phrase it in a sentence.

p.6 eq 1: could be simplified

p.6. Line 11-18: Should this go in the introduction?

p. 9 eq 9b: Contrary to R1, I managed to get the peak at 80°SZA. The equation seems to be the same as used in fig 3.

p. 9 eq 9c: Specify units, also see comments of R1.

p. 9. L 6: $U_x$ is wind speed at which height?

p.9 figure 3: Specify the values of relative humidity, wind speed and atmospheric diffusive radiation used for eq 9c. I agree with R1 that the label is wrong, but I think it should be SZA = 360° $\Phi zen/(2\pi)$ = 180° $\Phi zen/\pi$

p. 11 eq 16 a-d: Use either only °C or only K in equations, now they are mixed. I found eq 16 c in Henderson-Sellers (1986) but strangely I couldn't find this equation in Brutseart (1975).

p. 12 L 12: no units specified for latent heat of vaporization

p. 13 L 9 and L 13: I guess the authors meant eq 17 -18 and not eq 16-17?

p. 16 L 6: 'penetrating the surface', which surface?

p. 20 eq 35: What is $u_0$?

p. 22 eq 39: Please explain the variables in this equation.

Interactive
comment

p. 22 L 10. Should be eq 38, not 42

p. 22 eq 43: check index, should be hl not hi

p. 23 L 17: typo: entrain (not entrains)

p. 23 L 18: typo: the (not th)

p. 34 figure 17: increase font size and size of arrows.

p. 34 L 19: Insert the references into the place holder

p. 35 L 26: Who is testing these 'Wrappers' and examples? What is a wrapper?

p. 37 L 13: What is HTCondor?

p. 27 L 16: Start a new paragraph at 'GRAPLEr's Web service . . .' to highlight this idea.

p. 40 L 24 – 26: Explain better.

References Formatting: Some parts are underlined, remove it. Change all to coherent formatting.

P. 49 L 23, 25 same author, write same initials.

P. 51 L 13, is there a translation of this Japanese paper? Check the year (2014 in text, here 2015)

Appendix:

Table 1: If there is no default variable, can you give a range for snow density, compaction coefficient, and thermal conductivity of snow?

Table 1: Latent heat of fusion: remove the trailing zero 334

---

## Author Comment (AC1) · 8 Jan 2018

In response to the initial comments by Reviewer 1, a significantly updated version of the manuscript has been prepared. It is available for download as a supplement to this comment. Due to the volume of changes this preliminary revision does not contain track changes, but please note the significant changes that have been made, including improved consistency in notation throughout the paper, a symbol summary table, plus changes to Figures 1, 3, 6 and 10. The manuscript also includes more detail and reorientation of text to improve clarity of sub-model descriptions in Section 2.

These changes address problems that were initially identified by Reviewer 1 (which are

replied to specifically rather than in this note), but also do address several of the more recent concerns raised by Reviewer 2 and 3. A more comprehensive revision will be prepared that more fully addresses all of the reviewer comments, however, this version of the manuscript is being uploaded now to allow a more complete assessment by the reviewers during this discussion phase.

Many thanks to all reviewers for the very insightful comments, and I do hope this upgraded version will be easier to review.

Kind Regards Matt Hipsey and co-authors.

Please also note the supplement to this comment: https://www.geosci-model-dev-discuss.net/gmd-2017-257/gmd-2017-257-AC1-supplement.pdf

[Figure]

**Supplement:**

[Figure]

[Figure]

**Preliminary Revision 5 Jan 2018**

[revised manuscript text omitted]

that in a shallow-lake or wetland simulation, the layer structure may simplify to a single layer, in which case the surface and bottom layer are the same, and Eqs. 4 and 68 are effectively combined.

[Figure]

**Figure 12:** Adaptive offtake reservoir simulation; water temperatures of the adaptive offtake model assuming a constant temperature of 14 °C without (a) and with (b) mixing with bottom outlet withdrawal. The black dashed line represents the range of the variable withdrawal facility and the magenta lines the adaptive offtake and second withdrawal height.

**2.7 Wave height and bottom stress**

Wind induced resuspension of sediment from the bed of shallow lakes is sporadic and occurs as the waves created at the water surface create oscillatory currents that propagate down to the lake-bed. GLM does not predict resuspension and sediment concentration directly, but computes the bottom shear stress for later use by sediment and water quality modules. Nonetheless, even without this explicit formulation, the model can identify the areal extent and potential for bed-sediment resuspension by computing the area of the lake over which the bed shear stress exceeds some critical value required for resuspension to occur.

To compute the stress at the lake bottom the model estimates the surface wave conditions using a simple, fetch-based, steady state wave model (Laenen and LeTourneau, 1996; Ji 2008). The wave geometry (wave period, significant wave height and wave length), is predicted based on the wind speed and fetch over which the waves develop (Figure 13), calculated as:

$$F = 2\sqrt{A_s/\pi} \tag{69}$$

Using this model, the wave period, $\delta t_{wave}$, is calculated from fetch as:

$$\delta t_{wave} = 7.54\left(\frac{U_{10}}{g}\right) tanh(\xi) \, tanh\left(\frac{0.0379\left[\frac{gF}{U_{10}^2}\right]^{0.333}}{tanh(\xi)}\right) \tag{70}$$

[Figure]

**Preliminary Revision 5 Jan 2018**

where:

$$\xi = 0.833 \left[ \frac{g z_{avg}}{U_{10}^2} \right]^{0.375} \tag{71}$$

and $z_{avg}$ is the average lake depth. The typical wave length is then estimated from:

$$\delta x_{wave} = \left[ \frac{g(\delta t_{wave})^2}{2\pi} \right] tanh\left( \frac{2\pi \, z_{avg}}{\left[ \frac{g(\delta t_{wave})^2}{2\pi} \right]} \right) \tag{72}$$

and the significant wave height from:

$$\delta z_{wave} = 0.283 \left( \frac{U_{10}^2}{g} \right) tanh(\zeta) tanh\left( \frac{0.00565 \left[ \frac{gF}{U_{10}^2} \right]^{0.5}}{tanh(\zeta)} \right) \tag{73}$$

where

$$\zeta = 0.53 \left[ \frac{g z_{avg}}{U_{10}^2} \right]^{0.75} \tag{74}$$

Based on these properties the orbital wave velocity at depth (in the $i$ [th] layer) is calculated as:

$$U_{orb_i} = \frac{\pi \, \delta z_{wave}}{\delta t_{wave} \, sinh\left[ \frac{2\pi \, z_{i-1}}{\delta x_{wave}} \right]} \tag{75}$$

[Figure]

**Figure 13: Schematic of the wave estimation approach depicting the lake fetch, surface wind speed, wave height and wavelength,**
10 **and bottom stress created by the orbital velocity.**

[Figure]

[Figure]

**Preliminary Revision 5 Jan 2018**

For each layer, the total shear stress experienced at the lake bed portion of that layer (equivalent in area to $A_i - A_{i-1}$) is calculated from:

$$\tau_i = \frac{1}{2} \rho_i \left[ f_w \, U_{orb_i}{}^2 + f_c \, U_{m_i}{}^2 \right] \tag{76}$$

where $U_m$ is the mean layer velocity, which for simplicity is assumed based on the velocity estimate made during the mixing calculations (Eq. 37) in the surface mixed layer, such that:

$$U_{m_i} = \begin{cases} u_*, & i \geq k \\ 0, & i < k \end{cases} \tag{77}$$

The friction factors depend upon the characteristic particle diameter of the lake bottom sediments, $\delta_{ss}$ and the fluid velocity. For the current induced stress, we compute $f_c = 0.24/\log\left(12 z_{avg}/2.5 \delta_{ss_i}\right)$, and for waves (Kleinhans and Grasmeijer, 2006):

$$f_w = \exp\left[ -5.977 + 5.213 \left( \frac{U_{orb_i} \, \delta t_{wave}}{5\pi \, \delta_{ss_i}} \right)^{-0.194} \right] \tag{78}$$

where $\delta_{ss_i}$ is specific for each layer $i$, and may optionally be set to vary with sediment zone properties (see Sect. 4). 
[revised manuscript text omitted]

---

## Author Comment (AC2) · 8 Jan 2018

Dear Reviewer,

Many thanks for your initial thorough comments and suggestions - below are your comments in *italic* and our responses follow with the REPLY tag.

*This article describes the scientific basis of a 1-dimensional hydrodynamic lake model that can be coupled to ecosystem models. The model has already been applied to many systems in the scientific community, and I think it is useful publish the model description in a scientific paper that can be referred to for future applications of the*

[Figure]

*model. That said, I stopped reviewing after equation 16, because there were simply too many errors in the equations. I therefore propose to reject the current version of the manuscript and that the authors carefully check all equations before resubmitting the manuscript to this or another journal, depending on the decision of the editors.*

REPLY: Thank you for taking the time to review the discussion paper and identify the errors - we sincerely apologise that simple issues related to units and notation were not more thoroughly checked prior to upload. In a separate discussion comment, a revised manuscript ("Preliminary Revision 5 Jan 2008") is ready for your continued review. Please note this revision does have some substantial changes to the paper including the notation, text and (some) figures, and so would request you start reading from the beginning again (or at least from the end of the introduction). We very much look forward to your comments and suggestions on this version.

*Errors in equations up to eq. 16: eq. 2 and 3: I think something is wrong with the indices. $h_z$ is located between $h_{b-1}$ and $h_b$. $\alpha_b$ and $\beta_b$ describe the interpolation between $h_b$ and $h_{b+1}$. Thus, the indices in eq. 2 should be $\alpha_{b-1}$ and $\beta_{b-1}$.*

REPLY: You are correct; the code was looping from one step behind so we had incorrectly omitted the -1 in the equation. It is now updated

*eq 6: I think this equation is wrong. The right hand side is the total heat flux to the surface layer in W m-2. This should be divided by zmsl to get W m-3. Then, it should be divided by the water density in kg m-3 to get W/kg, and finally by cp to get °C/s for dTs/dt. Therefore, the multiplication term on the left hand side should be zmsl cp , rather than cp/(AS zmsl).*

REPLY: You are correct; it is now updated and in fact this section is now significantly revised. Table 1 now also includes all the notation and relevant units.

*eq 9b / Fig. 3: I could not reproduce the maximum of the Briegleb function at 80 degrees zenith angle. Using equation 9c yielded a monotonically increasing function*

*between 20 and 90 degrees (with a minimum at about 20 degrees). Also the equation in the legend is wrong, it should be SZA = $\Theta$Zen\*180/2$\pi$.*

REPLY: You are correct; This was an error in creating the plot rather than the model itself, and Figure 3 has been re-created and caption updated.

*eq. 12: I think $\varphi$SWS (i.e. the shortwave radiation absorbed in the surface layer) should be replaced by $\varphi$SW (z=0) in the nominator. Otherwise, the euphotic depth increases with increasing radiation absorbed in the surface layer, which does not make sense. Same in caption to Fig 4.*

REPLY: The notation has been improved to prevent confusion; $\varphi$SWS is what enters the top of the surface layer, and so the euphotic depth computation is based on the fraction of incoming light.

*eq. 16: I think in equations 16c and 16d Ta should be replaced by absolute temperature (i.e., 273.15 $^\circ$C should be added to Ta).*

REPLY: You are correct; notation updated to have K and C unit options for temperature

*Also units should always be provided, especially for empirical equations (e.g. ea in eq. 16, and Ux, RH, and diffusive radiation in eq. 9c).*

REPLY: Units have been thoroughly checked and updated in the revised upload, plus the updated Table 1 summarises units for all variables/symbols.

*Besides that, a few other points I noticed up to page 11 (Page xx, Line yy is abbreviated as xx/yy)*

*In general, the paper is well written and easy to read, but there are quite a few long and complicated sentences which I think should be simplified to facilitate reading (first two examples: 2/24-29 , 3/12- 17).*

REPLY: These examples have been re-written and the revised upload has also been check for these issues.

*3/10: This list of references seems to be somewhat inconsistent. Some of the references refer to model development, some to model applications. It would be more logical to cite only model development references.*

REPLY: Point noted, these papers were reflecting the diversity of 1D models we are aware of, but agree these should more specifically refer to development refs.

*4/30: The text seems to imply that the requirement for site-specific calibration in other models is due to numerical diffusion caused by the fixed grids. If that is the intention, this should be explained. If not, the sentence should be modified.*

REPLY: Sentence modified...

*5/7: Incomplete sentence*

REPLY: Section has been revised and reworded.

*Figure 1: Shouldn't the local runoff, and the submerged inflows and groundwater seepage be written in blue?*

REPLY: Local runoff is computed by GLM (Eq 7 in updates manuscript). Submerged inflows and seepage however has been updated to be blue in Figure 1 as they are specified.

*eq. 1: From the text (layer volumes are determined ...), I would have expected an equation for the individual layer volumes here, but this is the integrated volume from the bottom of the lake to the top of each layer. This should be clarified in the text.*

REPLY: Section has been revised and reworded to hopefully introduce the layer structure and notation more clearly.

*6/3: technically, it is the same, but I think it would be clearer to write $2 \leq b \leq NBSN$.*

REPLY: Updated.

*6/4: how are these finer depth increments determined?*

REPLY: Now summarised in symbol table (Table 1).

*6/9: Since the Unesco (1981) equation has been replaced by TEOS-10, I think it would make sense to use the latter rather than the former in a new model. I also think it should be mentioned that the density effect of salinity in these seawater equations is quite different from that in most lakes where carbonates are usually the dominant species rather than NaCl.*

REPLY: Thank-you for this suggestion. The preliminary revision upload still refers to UNESCO 1981, however, we will endeavour to implement TEOS into the code and the refer to this in the full revision, (of course, depending on the outcome of the review process)

*6/24: heat balance of the surface layer 7/2-3: why is only rain but not snow multiplied with fR? Also, even though this should be clear to the reader, it should probably be mentioned that S is in water equivalents.*

REPLY: Updated.

*eq 5: to be precise, this equation should be limited to a minimum of zero, as otherwise it will become negative if the rainfall is too weak.*

REPLY: Updated - this was the case in the code, but not properly summarised in the Eq.

*Fig 2: Add some space between the 10 and the exponent in the y-axes labels of panels c and d. Do all these time series start on 1 January of a year?*

REPLY: This panel has not yet been updated in the preliminary revision, but we will do so in the full revision. Lakes do have variable start times (for various reasons) which is why we didn't use exact date in the x-axis. The individual lake simulations are documented on our GitHub site with explanations.

*eq. 9a: instead of subtracting $\pi/2$ within the sine functions, it would be easier to use*

*-cos.*

REPLY: Apologies, there was a "-" sign wrong in this Eq so both should not have been $-\pi/2$. In the new manuscript (Eq 12a) we have the addition or subtraction of $\pi/2$ is listed properly in order to allow us to differentiate Sth vs Nth hemisphere sites.

*eq. 9b: where does the factor 1.1 in the nominator of the first term come from? Maybe I overlooked something, but I could not find it in Briegleb et al. (1989).*

REPLY: Thankyou for noticing this detail - this inclusion of 1.1 has been picked up from the implementation by Li et al (2006), who compared models and have this coefficient included.

However, for consistency, we have removed this from the Eq in the paper, and the code-base going forward. We note however, that the change made only a modest difference to the function.

Li, J., Scinocca, J., Lazare, M., McFarlane, N., Von Salzen, K. and Solheim, L., 2006. Ocean surface albedo and its impact on radiation balance in climate models. ÂăJournal of climate,Âă19(24), pp.6314-6333.

*eq. 9c: I was not able to check this equation, as the source is in Japanese, but I did not get anything similar to what is shown in Fig. 3 trying different values for RH, U and the diffusive radiation. Please check whether the equation is correct, and specify the values used to produce Fig. 3. Furthermore, Yajima and Yamamoto is dated 2014 here but 2015 in the reference list.*

REPLY: Thankyou again for checking this - the equation implemented is based on the Equation 1c of this publication, whereby y is cos(solar angle), the a and b coefficients were set by multi-variate regression, and x1 is RH(percent), x2 is wind speed (m/s) and x3 (-) is a parameter referring to the amount of atmospheric diffuse radiation.

We initially received the coefficients from the author (Yajima pers comm.) for our implementation, and following your comment, we have now updated the Figure 3, the

citation date, and the equation. The caption will also be updated to state the values of x, x2, x3 used in the graph. For your reference, R code for the algorithm is below:

$angled = 1 : 89$

$angle = angled * pi/180$

$dr = 6$

$ux = 4$

$rh = 80$

$yalbedo = max(0.02, 0.001 * rh * (1 - cos(angle))^0.33 - (0.001 * ux * (1 - cos(angle))^{(-0.57))} - (0.001 * dr * (1 - cos(angle))^{(0.829)})))$

*Fig 4: y-axis of panel b is not depth, but elevation, y-axis of panel c is not labeled.*

REPLY: You are correct; the y-axis should be height not depth. As part fo the revised manuscript we have uploaded we have undertaken a major review of notation used throughout so that depth (z), height (h) and elevation (H) are used consistently.

This figure is not yet updated in the preliminary revision but all figures will be re-created and revised in the full revision, to reflect these issues.

*Also it seems that ABEN is calculated on a different time scale than the radiation in (c). Many low radiation events are clearly visible in (b) but do not show up in (c). This probably makes sense, but the time scale should be mentioned somewhere.*

REPLY: Please note that in fact the ABEN is now being computed based on a percentage of light reduction, rather than a specific value.

The step changes in the panel (c) time-series are not due to time-scale of calculation, but rather to do with changes in the layer thickness and structure. We will review this example simulation carefully to improve this in the full revision, and anticipate it will have the same trend but smoother changes.

*11/11: It does not look like the equations were copied from Henderson-Sellers (1986), but rather from either the original sources or from Flerchinger (2009)?*

REPLY: The placing of the citation to Henderson-Sellers incorrectly gave the impression this is where the full expression was from. You are correct that that we have just chosen 4 based on original sources to implment within the model. We have now cited Henderson-Sellers and Flerchinger as sources for further description and information, rather than as a source for the algorithm set.

---

## Author Comment (AC3) · 8 Jan 2018

Thank-you for the information related to code DOI.

Due to the nature of the review comments several small changes to our code will be undertaken. Therefore we will organise a DOI for the revised version of code that will link with the revision of our manuscript.
* * *

---

## Author Comment (AC4) · 22 Jan 2018

Dear Reviewer,

Many thanks for your initial thorough comments and suggestions - below are your comments in *italic* and our responses follow with the **REPLY** tag.

*General comments*

*1. This paper presents the formulation of a one-dimensional model of thermodynamics, mixing, and evaporative and momentum fluxes from lake surfaces, which conceptually should be applicable to a wide variety of lake morphologies. However, the manuscript*

[Figure]

*achieves the paradox of simultaneously containing too much information and not enough information. It goes into great detail with many equations. However, in order to present this level of detail without losing the reader requires more care and at least some additional details.*

**REPLY**: We greatly appreciate the comments and insights. We acknowledge the paper is heavy with information, and chose the journal Geoscientific Model Development for this submission as it does support papers focusing on model description. Given the nature of the model it is our desire to have a comprehensive description of the numerous sub-models and options that is peer-reviewed and citable; this has led to a long paper compared to a traditional manuscript. As outlined in the below comments, our full revision version will endeavour to improve the narrative and flow of the paper.

Note : A "preliminary revision" is referred to below, which is on the discussion at this link: Click here

*I have a few broad suggestions that I think might help.*

*2a. Before anything, you need to have a clear idea of who is in your audience. You could even have an explicit statement of this very near the beginning, and direct users with a lower level of expertise toward a simpler users' guide.*

**REPLY**: The paper is intended as a description of the model rather than a research paper. Whilst there have been some unpublished manuals and incomplete documentation for the model during its initial development, we felt this is unsatisfactory for a model that has had increasing uptake by lake scientists. In our initial submission, we had stated the aim as:

"Given that individual applications of the model are not able to describe the full array of features and details of the model structure, the aim of this paper is to present a complete description of GLM, including the scientific background (Section 2), model code organization (Section 3), approach to coupling with biogeochemical models

(Section 4), and to overview use of the model within the context of GLEON specific requirements for model analysis, integration and education (Section 5-6)."

Our audience is therefore advanced users looking to understand the mechanics of the model, and/or end-users publishing applications that need to cite the model methods and approaches adopted in their individual simulations. We acknowledge some users will not dig into the detail and can following the "user" material we have provided on the model website. We note that we have not provided many details about how to run the model, but rather it was our intention for readers to be able to understand the science basis and model structure.

*2b.    This reviewer has a back- ground explicitly in meteorology, but significant exposure to lake dynamics as well, albeit mostly regarding very large lakes. Depending on your intended audience, you might have readers who will have difficulty with terms such as aliquot and even the intended understanding of "scalar concentrations" as used on p. 27, lines 2-3.*
**REPLY**: Thank-you for pointing these out. Already in the "preliminary revision" (upload dated 8th Jan on the discussion page) we have attempted to improve both these descriptions to improve the clarity of this terminology.

*3.    To reduce the length of a single paper, one option is to break it into multiple papers. Another is to move more of the detail to appendices. Since it is an online journal, there is probably not a large problem with overall page length. Things as detailed as conversion of area as a function of depth to volume in a layer (eq. 1) seem like they could be skipped in the main text and relegated to an appendix.*
**REPLY**: We appreciate the comment and agree with the general desire for shorter papers, and feel that in a model description paper such an argument could be made for several of the sub-model descriptions. However, a key purpose of our paper was to

describe the model from start to finish (as is done through Section 2), and the layer structure and depth-area-volume relationship is key to the model approach. Aspects such as solar radiation and atmospheric stability computation are optional modules in the code-base and not directly related to what happens within the lake model, and so are included as appendices. Nonetheless, in the preliminary revision we have made some changes to Section 2.1 (referred to specifically in this comment) and feel the new wording of this section has a stronger narrative that hopefully connects it with the subsequent sections.

*4. It would be useful to have a brief introduction to each sub-section. This should start out by stating the goal of that sub-section, i.e. what will be the final equation (or set of equations) derived in the section. Then state what elements will combine to get that final equation(s). One particularly glaring issue is that sub-section 2.5.2 ends with eq. 45, defining the variable f, but nowhere does it say how f relates to any other part of the model. It seems to be something that one would multiply by the difference in temperature (or another scalar) between layers) to get the exchange of that scalar between the two layers in a single time step. Whether this is exactly correct or not, the statement is missing from the manuscript.*
**REPLY**: The preliminary revision uploaded has attempted to address this general comment by strengthening the narrative in the sections from 2.1-2.7 (the main sub-model description sections), and we will further work on improving this for the full revision. In reference to the comment about the "$f$" function, the use of the term scalar (and symbol C) is now better introduced in the preliminary revision, placing this Eq in context. We note that the f computed in this equation is required in the equation where concentrations are updated based on the magnitude of diffusivity between layers (Eq 51 in the preliminary revision, and Eq 44 in the original version).

*5. A simple overall schematic would be good to have early on (Fig. 15 with less detail). This would make it less abrupt when "water quality model AED2" is*

*mentioned on p. 27 (I may have missed it, but I don't think it was mentioned before.*
**REPLY**: In light of other comments (8 of the general comments and 42 of the specific comments), we will prepare a revised Figure 1 and 16 that will address this.

*6. Even if only for your own reference as author, Table 1 needs to be expanded to include every variable used in every equation! And in this expanded table, include every variant of each variable based on the use of different subscripts, prime, and circumflex ("hat"). With this many equations, it is rather inevitable that you also end up using the same symbol for different things (I noticed N in particular). Then, for each variant of each variable, put additional columns for: description, units, spatial type (defined at surface only, spatially continuous, or at discrete layers), and which equations it is used in. The reader needs to have all of these carefully defined.*
**REPLY**: The preliminary revision includes a fully updated Table 1, and addresses the issue of confusion of notation by having a more consistent nomenclature scheme.

*7. My high school chemistry teacher is the one who taught me how to use units in equations, and the importance of doing so, hence the need for them in the big table above. Some examples of problems with units in the equations that may indicate that the equation is simply wrong: In eq. 52, Q seems like it should have units of m cubed per s, and h units of m, so the right-hand side does not have units of velocity.*
**REPLY**: We apologise for this mistake - this equation (now Eq 59) has been corrected (and the code was checked for consistency). The updated Table 1 now also has consistent units throughout, included for each variable.

*8. I approached this manuscript with the immediate question of what makes this model better and more useful than the many others that are available. The introduction does a pretty good job of answering this, but it may be good to give some examples of uses that are not satisfied by other models. This could be introduced with a schematic of its components, options, and functions, at a lower level of detail than in Figs. 15 and*

*16.*
**REPLY**: Thanks for this suggestion. We will add a statement to the conclusion section with this idea. We do not feel in this paper we have the scope to do a full meta-analysis of the features of all the other lake models and where they are inadequate. Given the number of sub-modules and options, and very wide array of application contexts, it becomes difficult to make judgments about what model options are required for specific lake types and we defer to users to use the model in an appropriate way. We note that Figure 1 does currently aim to be a schematic for the different model components, and functions, but this is currently not linked to applications or specific lake types. We will aim to address this in the full revision, along-with the specific comment 42 by making more clear an overview of the sub-models that may be engaged.

*9. Anything that can be done to bring the reader's intuition into play when introducing equations will improve comprehension of the manuscript. Eq. 27 isn't the most problematic one, but I'll use it as an example. You might introduce it by saying something like: "Shortwave radiation is absorbed and attenuates with different e-folding depths for snow, white ice, and blue ice, and these also depend on the light's wavelength. The overall effect is..."*
**REPLY**: Thanks for this suggestion. Some wording has been updated across sections 2.1-2.6 to try to improve the flow of the text and clarity of the descriptions. We will further update this in the full revision following compilation of the set of all reviewer comments.

*10. Eq. 47 has me very mystified about how a standard definition of Richardson's number translates into this equation in terms of the angles of inflow geometry (part of the problem may be that I don't feel like I understand the meaning of the angle labeled alpha in Fig. 10; the illustration isn't helping me). It also has me asking "Richardson number at what location?" At the interface of the river water and ambient*

*lake water?*

**REPLY**: We have cited the original source (Fischer et al 1979) of this equation in the original paper, however to improve the interpretability of this we will add some brief detail to this definition and approach to justify its basis.

*11. Several of the figures have characters that are so small as to be illegible.*
**REPLY**: We will redraft the schematic figures with larger symbol fonts.

*12. What about modularity of the model? Can other schemes for pieces of the model be plugged in? The details of how may be an appendix or even another paper or guide.*
**REPLY**: The model is not written in an object-oriented fashion, however, custom schemes can be linked to for key sub-module components (e.g do-mixing, do-deep-mixing etc) with some code-adjustments. A comment on this will be added to the code structure section (Section 3).

*Some particular examples of the general problems above are among the specific comments below.*

*Specific comments:*

*1. In nearly every review that I do, I refer to the rules of hyphens found at http://www.grammarbook.com/punctuation/hyphens.asp particularly Rule 1 on that page. Your manuscript is actually better than most, but here are the problems that I found: P. 2, line 23 should have "system scale" (no hyphen). "Time-scale" is sort of borderline; I tend to use it without a hyphen when it doesn't modify another noun. P. 4, line 15 and some figure captions: "time series". P. 13, lines 22 and 24: "Wind sheltering". P. 24, line 3: "user-specified".*
**REPLY**: Thankyou for this link and advice. These are updated now in the preliminary revision.

*2. P. 2, line 33: Change "spatial" to "vertical", to contrast with horizontal from earlier in the sentence.*
**REPLY**: This is updated in the preliminary revision (now page 2 line 32)

*3. P. 3, line 10: Stepanenko is misspelled.*
**REPLY**: This is updated in the preliminary revision (now page 2 line 7)

*4. P. 3, lines 17-18: Do you have any comment on use in small vs. large lakes? Both in terms of area and depth? There might also be considerations in terms of morphological complexity.*
**REPLY**: We will aim to refine the sentence to highlight this specific detail in the full revision.

*5. P. 3, lines 30-31 specifically mention temperature, salinity, and density. Later on, there is mention of the broader category of "scalars". Are you unnecessarily limiting yourself here? I am thinking of such things as dissolved oxygen (mentioned later) and concentrations of nutrients and contaminants.*
**REPLY**: The reference to scalars is now updated in the preliminary revision (now page 5 line 30) to be more clear, and the option of coupling to a water quality model is mentioned (page 6 line 1)

*6. P. 3, line 32: I think many readers will object to the use of "hydrodynamic" here, since it does not explicitly represent advection of water. Continuity equations say that dynamics requires at least two dimensions.*
**REPLY**: We would argue that term hydrodynamic refers to the motion of water in general, rather than specifically need to resolve advection in two dimensions. In addition to vertical mixing and transport of water and its constituents, the model does parametrise horizontal advection of inflow water (Eq 59 in the preliminary revision). Therefore, whilst the model is not resolving the Navier-Stokes equations, the model

approach is intended to capture the movement of water within the lake domain as it is filling, drying, and subject to inflows and withdrawals.

*7. P. 4, lines 31-32: "user-defined" and "set by the user" are redundant.*
**REPLY**: Section 2.1 has been significantly revised in the preliminary revision upload and this problem is removed.

*8. P. 5, line 8: This model doesn't include it, but in reality, vertical advection of heat can be important along with vertical mixing. Again this may depend on the size of the lake, but this factor should be acknowledged here.*
**REPLY**: Aside from mixing, the model captures to some degree the advection of heat and constituents in the water as brought about by inflow or outflow dynamics. For example, new inflowing water would add a layer and lift up layers above its insertion depth, which could be considered as vertical advection. Short term advection associated with internal waves, or perhaps upwelling in large lakes, however is not included. We will add clarity to this capability in this section in the full revision.

*9. Eq. 4 has issues with sign conventions, units, and possible missing terms depending on how one understands it. It is E multiplied by lake area should be considered a volume flux. For practical purposes these can be considered equivalent, but there would be another term for water density to convert between mass flux and volume flux. Are the evaporation, snowfall, and rainfall defined as being specifically over the area of the lake itself, or over the drainage basin? If over the lake, how is the Q term for runoff optional? I have a hard time imagining many cases of lakes in which it is not a major term in the water balance. If these variables are defined over the whole basin, is there an issue with agreement in timing between P – E and runoff?*
**REPLY**: Eq 4 and this section is now corrected in the preliminary revision, with units clarified in Table 1. The heat flux conversion to volume is clarified in this version, accounting for density. The $P$, $E$ and $S$ terms are computed over the active lake

surface. The inflows from the drainage basin into the lake are set by the user as an "inflow" (section 2.6). The local runoff term is not referring to runoff from the wider drainage basin where the lake sits, but the local land within the lake area ($A_{max} - A_s$) that is not inundated with water. This could be important in reservoirs where the volume is drawn down significantly from the maximum level set by the user, or for ephemeral wetlands; in both these cases this area would generally not be considered as part of the inflows from the wider drainage basin, and so Eq 7 (in the preliminary revision) allows for this "dry lake-bed area" to contribute to the water balance.

*10. P. 7, line 21: "However" is an interjection, not a conjunction. Here, it stands between two independent clauses, so preceded with either a semicolon or period. Also p. 33, line 10 and p. 35, line 11.*
**REPLY**: This is updated in the preliminary revision

*11. Eq. 7 is a place where I started to distinctly feel the problem of definition of variables. I had to really think through what had happened to the subscript S from eq. 6 and why the "hats" were there in eq. 7.*
**REPLY**: This is updated in the preliminary revision; both the updated notation in Section 2 and Table 1 should prevent this confusion.

*12. Eq. 9a has a strange step function in space. Can you justify this not being a smooth function, but rather one value for all of the northern hemisphere, another for the equator (an infinitesimal amount of space), and another for the southern hemisphere?*
**REPLY**: This function has been previously used by many authors for albedo and so is included - we acknowledge this option is simple and hence Option 2 and 3 are provided in the model (Eq 12 in the preliminary revision), with specific resolution of the solar angle.

*13. P. 9, line 6: What makes the different formulas oceanic and lacustrine? Is there anything to help the reader's intuition for why these formulas should be different?*
**REPLY**: We will add detail here in the full revision.

*14. Audience issue: P. 10, line 14 departs from the main conceptual theme to give a specific file name. Detailed mathematical formulation and detailed user information don't mix very well.*
**REPLY**: We agree with this observation and will remove the user detail in Section 2 in the full revision.

*15. Eq. 14: Are you using a different temperature at the skin (interfacial layer) or bulk temperature a little deeper? Specify how deep.*
**REPLY**: The surface temperature used in the long-wave computation is from the upper-most model layer. This is homogeneous over the depth of the surface layer (surface layer has a sub-script denoted "$s$"), however we cannot specify the exact depth, as the model computes the surface layer thickness dynamically, based on the density gradient and layer limits. A sentence will be added to clarify this for readers, mentioning the indicative depth as ranging from $0.5\Delta z_{min}$ to $0.5\Delta z_{max}$.

*16. P. 11, line 10: Air temperature at how high above the surface? Standard is 2 m, but there may be adjustment needed if measurements are at a different height than that intended in the formulation.*
**REPLY**: The air temperature used throughout the paper is now standardised to 10m and this is defined specifically in Table 1; see the preliminary revision upload. For users with data from a different height, that may use the $f_U$ parameter to scale the data appropriately.

*17. Eq. 163: "Brutsaert" is misspelled.*
**REPLY**: Updated in the preliminary revision

*18. P. 11, line 13: The range of octals should be 0 to 8.*
**REPLY**: Updated in the preliminary revision

*19. P. 12, line 11: This should specify molecular weight of dry air, and it might be better to say mass rather than weight.*
**REPLY**: Updated in the preliminary revision

*20. P. 13, lines 5 and 9: I think these should reference eqs. 17-18 rather than 16-17.*
**REPLY**: Updated in the preliminary revision

*21. Eq. 23: The Latin v and Greek nu are somewhat difficult to distinguish here, and are even more difficult to distinguish in the text following.*
**REPLY**: Updated in the preliminary revision; the notation has been standardised, thereby removing this issue.

*22. P. 13, lines 16-18: Thanks for explicitly using units here. Am I correct in understanding that molecular heat conductivity is molecular heat diffusivity multiplied by heat capacity? I think it would be better not to use heat conductivity, but use diffusivity and capacity.*
**REPLY**: Note that units in the preliminary revision are updated in the text where appropriate, and Table 1 now includes all units for all variables. You are correct that conductivity is related to diffusivity and heat capacity. The notation adopted (now Eq 27) is based on the original description in TVA (1972), however, we agree it is redundant in a way and will modify accordingly.

*23. P. 13, line 22: "may be" should be two words.*
**REPLY**: Updated in the preliminary revision

*24. P. 15, line 13: This use of "conductive heat flux from the ice or snow cover to the atmosphere" particularly triggers my thought bias as a meteorologist. That bias prompts me to assume that this means flux through the atmosphere's interfacial layer, where molecular diffusion dominates. But I also wonder whether it means heat conduction through the ice. These are not equal in general, and need to be distinguished from each other.*

**REPLY**: It does mean the upward conduction through the upper ice layer. For clarity this is depicted in the schematic in Rogers et al (1995) that we had mentioned in the text, on which the model is heavily based (Figure 5 in this source). As such, we chose not to reproduce the energy flux schematic in this manuscript, but will revise the text to prevent this confusion, and propose to add an energy flux (sub-)plot specific to the ice model in the full revision given our notation is now slightly different.

*25. Eq. 26: Why isn't shortwave radiation here?*

**REPLY**: This is the way the approach was reported in Rogers et al (1995) (Eq 7 in their paper) and we have followed that; in this case at the surface interface it is assuming the non-penetrative heat flux is in balance with the conduction of heat up to the surface. The conduction of heat up is based on the amount of light that has penetrated (as computed in Eq 31 in the revised upload). We will update the text here to prevent confusion and be more explicit that the net is referring to the non-penetrative component.

*26. P. 16, lines 11 and 12: These are very wide ranges of albedo. Please describe the conditions under which different parts of this range manifest.*

**REPLY**: The variation is discussed in the Vavrus et al (1996) paper that we cited; we will add a brief explanation about the variability to this sentence in the full revision.

*27. P. 18, line 10: The idea of energy required for mixing needs to be intro-*

*duced more carefully. If you think of the water column as continuous in space, mixing is also a continuum. But the concept here is based on discrete layers, and this indicates how much energy is required to outright include a model layer in the mixed layer.*

**REPLY**: The approach to bulk mixed layer models is well established and as such we provided suggested references for readers interested to gain a deeper understanding of the background: Kraus & Turner (1967) and Kim (1976) and Imberger & Patterson (1981). The model approach we have adopted also has been the subject of numerous prior publications that have thoroughly validated that the layer discretisation can accurately capture the mixing continuum in monomictic and polymictic lakes. To provide further clarity on this point we will revise the opening text in this section (Section 2.5), as suggested by the reviewer, in the full revision.

*28. Eq. 31: C sub T seems completely undefined.*
**REPLY**: Updated in the preliminary revision via Table 1

*29. P. 20, line 14: This cannot be reproduced unless your definition of "epilimnion" and "hypolimnion" are precisely stated.*
**REPLY**: Updated in the preliminary revision (now Eq 42.)

*30. P. 20, line 20: Is K sub epsilon eddy diffusivity.*
**REPLY**: Thankyou for noting this; the preliminary revision now includes an update to specify this.

*31. Since the vertical axis in Fig. 8 has zero at the bottom, it appears to be height above the bottom, not depth. It might be worth explaining that the varying top height of the color fill is due to varying overall depth (assuming that this is correct).*
**REPLY**: You are correct; in the preliminary revision we have been more careful in use and notation associated with depth, height and elevation, and in the full revision will

update this figures as suggested to mention height instead of depth.

*32. P. 22, line 13 and eq. 43: This seems to say that sigma should have units of inverse seconds squared, which implies that the square-bracketed part of eq. 43 has units of (m/s) squared, but exponential operations can only be performed on unitless numbers.*
**REPLY**: Thankyou again for noting this inconsistency. Both the top and bottom of the fraction in this equation (Eq 50 in the preliminary revision) are in the units of $m^2$, so the fraction for the exponential is dimensionless. The term sigma is poorly described in this sentence and we will update this in the full revision to be more explicit.

*33. P. 24, line 5: Slope is usually defined as a ratio of rise divide by run, not an angle. I would say that the slop is the tangent of the angle.*
**REPLY**: You are correct, we will update this in the full revision.

*34. The inset in Fig. 10 does not help me understand the meaning of the angle alpha, and it could use larger lettering.*
**REPLY**: Note that we have made an initial modification to this figure in the preliminary revision, however, we will refine the clarity of this schematic in the full revision to better depict the angle and slope, and readability.

*35. Equation 52 seems messed up. Units-wise, it seems to make more sense if h squared is in the denominator instead, but I also wonder why there is no dependence on width of inflow.*
**REPLY**: We apologise for this error and it has been corrected in the preliminary revision (now Eq 59). The scheme does not account for the width of the inflow, thereby assuming the inflow intrusion has a roughly uniform thickness and width as it progresses down the bathymetric depression (termed the thalweg). Whilst this may not be exactly the case, the approach implicitly assumes that any change in

width would be compensated by a change in thickness (due to conservation of mass) as the inflowing water progresses – and therefore the net area $Q$ is being divided by is therefore equivalent. This is obviously a simple approximation for an inflow parameterisation in a 1D model, and we will add clarity to the text to better explain this point in the full revision.

*36. P. 26, line 16: In standard usage, dz represents an infinitesimal difference in z in continuous space, but here it is used to indicate a finite distance in discretely sectioned space.*
**REPLY**: This expression and associated notation has been revised in the preliminary revision upload (now Eq 67). The symbol $\delta$ is now uniformly used to refer to a length-scale as is the intent here.

*37. P. 26, lines 16-17: It is difficult to understand the distinction between "height of withdrawal" and "edge of the withdrawal layer".*
**REPLY**: This description has been further developed in the revision (now Page 28 Line 10), to explicitly define how the withdrawal thickness and the relevant layer edge is determined.

*38. P. 26, line 19: Why say "fluid" rather than "water"?*
**REPLY**: You are correct that water is more specific – we will replace the use of fluid with water in the revision.

*39. A formatting problem put some labels that belong to Fig. 12 on p. 27, while the rest of the figure is on p. 28.*
**REPLY**: This has now been rectified, and will be checked in the final revision.

*40. P. 28, lines 10-11: Removing no more than half of a layer's mass per time step seems like a reasonable way to ensure numerical stability, but it would be good*

*to re- mind the reader here of the layer merging scheme that is likely to kick in. This merging and disaggregation scheme, also mentioned on p. 34, is never really described well.*

**REPLY**: Thankyou for this suggestion, we will clarify this in Section 2.6.5 and also 2.6.3 in the full revision. Note that we have revised both these sections to improve clarity of these descriptions in the preliminary revision, including making both consistent by enforcing a 90% limit. The merging and splitting scheme is introduced (page 6 line 7 of the preliminary revision), however we note the concern this is not described fully, and we will therefore address this by adding further detail in the full revision.

*41. P. 29, line 3 says "wind speed and fetch...calculated as", while eq. 61 only shows the formula for fetch.*

**REPLY**: The sentence introducing this Eq will be re-worded to remove this ambiguity in the full revision.

*42. To help solve the problem of small print in Fig. 16, it may be useful to transfer some of the information to a table instead.*

**REPLY**: Thankyou, we will address this in the full revision, along-with the general comment 8.

*43. P. 34, line 19 has a placeholder for a citation. This is evidence of a poor final edit on the part of the authors.*

**REPLY**: Updated in the preliminary revision; apologies for this oversight.

*44. P. 34, line 23: If you mention calibration, it would be well to describe this process more fully, in particular which parameters you consider adjustable for purposes of calibration. Where p. 35, line 10 mentions "compare", I wonder whether this might also imply calibration.*

**REPLY**: The initial line referred to is an introductory statement with the intent that the

text in Section 5.2 would decsribe the calibration process more fully. We will revise this section in the full revision to link to better link to our associated model validation paper (Bruce et al., 2017) and list the recommended parameters for fitting in Table 1.

*45. P. 48, line 12: "Anneville" misspelled?*
**REPLY**: Updated in the preliminary revision

References cited in this comment are as listed in the original submission.

---

## Author Comment (AC5) · 23 Jan 2018

Dear Reviewer,

Many thanks for your comments and ideas which we have found very helpful. Below are your comments in *italic* and our responses follow with the **REPLY** tag.

*General Comments*

*1. This paper describes the detailed functioning of the 1D physical lake model GLM 2.4 and its application potential. The model incorporates a broad range of*

[Figure]

*physical processes as surface heat exchange, snow and ice dynamics, in- and outflow, submerged inflow and groundwater seepage and can be coupled with or embedded into other models. The authors explain how GLM 2.4 has emerged as a response to the need of standardized, yet flexible and computationally effective community lake model to interpret environmental data from a broad range of lakes collected within the Global Lake Ecological Observatory Network (GLEON). The model has been formulated as a new code in 2012, whereas layer structure, mixing algorithms and physical formulae are based on earlier peer reviewed work. The authors state that the code is computationally efficient and well suited for embedding in larger scale modelling frameworks. The authors present also an overview of pre- and post-processing utilities as well as an innovative cloud computing environment. Lastly, they elaborate on the educational use and gained experience in the classroom.*
**REPLY**: Thankyou for this very accurate summary

*2. I realized that this manuscript is for a major part equivalent to an earlier manual of GLM (V2 Manual, October 2014, accessed on the 08.01.2017 from http://aed.see.uwa.edu.au/ research/models/GLM/Pages/ documentation.html). I think the authors should mention this.*
**REPLY**: It is our intent that this paper replaces the online manual and we have removed the available PDF cited above. The online manual was used as an interim resource to inform users until the model development efforts had stabilised. Now this is the case, the revised version of the present paper includes more detail and numerous improvements, extensions and fixes to errors and should become the key reference. Aspects of that manual associated with model-use setup that are not covered in this (science-oriented) paper have been migrated to the website pages.

*3a. The model in this paper represents with no doubt a tremendous effort in lake modelling and is of interest for modelers in various fields of environmental research. The publication of this model is a step towards better model documentation and contributes to the general scientific discussion and better lake model development. As such it falls*

*within the scope of this journal. The paper is well written and the language is easily com- prehensible. Unfortunately, this manuscript has some structural problems and there are quite a few mistakes in equations and figures. After dealing with these issues, the manuscript should be good for publishing.*

**REPLY**: Thank you for this recommendation. A "preliminary revision" was uploaded on the 8th Jan following comments by R1 and R2, and this does already address many of the points outlined below – apologies this was not made available before your review. We will endeavour to resolve the other specific issues that you and the other reviewers have identified in the full revision to be uploaded following closure of the discussion phase.

*3b. The main problem of this very long manuscript is that it is missing an instant overview of what is in the paper and what not. Scanning through, the reader gets lost easily in the large chapter 2 'model overview' and might miss the subsequent chapters that elaborate more on the possibilities and significance of this model for the scientific community.*

**REPLY**: We acknowledge this view, which is similar to a comment 2 by R2, and we will aim to provide a better "road-map" to add clarity in the introduction.

*4. I think that this problem can be fixed with some changes in the introduction: âA ÌĘc ÌĄ I suggest using subtitles in the introduction. (In the introduction, the authors describe the importance of the study of lakes, the importance of GLEON, the importance of lake models, the advantages of simple models, applications and features of 1D models, the need for a flexible open-source model, how GLM 2.4 answers this need and finally an overview of the paper) âA ÌĘ c ÌĄ I suggest creating a new paragraph starting at p. 3 Line 19 ... "Nonetheless, there . . .". The need of an open source and flexible community model that can be applied to various lakes should be highlighted better. Another additional paragraph could explain how GLM 2.4 responds to this need. As I understand, GLM 2.4 is filling the gap because it provides a standard middle complexity physics 'shell' (simple yet enough complex to be applied for various lakes) that can*

*be connected individually to or implemented into various other models (e.g. water quality or land- climate models). I think this point could be emphasized. âA ÌĘ c ÌĄ A figure could be helpful to draw attention to the significance of this model in the scientific community. This could also be combined with schematic overview of the model functioning (I agree with R2 that anything that gives an overview helps). âA ÌĘ c ÌĄ The specific limitations of GLM 2.4 (not of 1D models in general) should be mentioned in the introduction. Like this, the reader may have a quick idea whether GLM is suitable for him/her. What are the key features of this model that set it apart from other models? âA ÌĘ c ÌĄ On p. 4 lines 5-9, the authors explain the aims of the paper and in which of chapter 2-6 these aims are met. I think these lines are important and should be extended to a paragraph by itself to make sure the reader is fully aware what to expect from the paper. In the same paragraph, I would also expect some more information regarding what this paper is not about and mention that a companion paper by Bruce et al. (2017) is assessing the model's error structure against 31 GLEON lakes.*

**REPLY**: Thank you for this very useful suggestion and we will take up this idea for improving the introduction in the full revision.

*5. I think the authors did not carefully go through the complete manuscript. Many of the empirical equations are missing the definition of units for used variables. On other occasions variables where poorly described (see the examples listed below, as well as listed by R1 and R2). On several figures elevation and not the labeled depth is shown on y axis. The references are not formatted coherently. Like R2, I am of the opinion that many variable symbols are confusingly similar and that they should all be listed in a table. I also agree with R2 that all the subchapters of chapter 2 should have a small introduction paragraph. Further, I agree with the comments of R1 on the equations 1, 2, 3, 5, 9c, 12, 16 and with the comments of R2 on the equations 4, 7, 14, 26, 31, 52.*

**REPLY**: Our sincere apologies that we didn't identify these flaws in the original upload. In our responses to R1 and R2 we have detailed many fixes to these issues, including a significantly revised nomenclature and summary table with all variables and units.

The "preliminary revision" that is uploaded to the discussion area (Click here) now addresses most of the issues, and further updates are planned for the full revision, including updates to the figure scales, and improved contextual information in the Section 2 sub-sections.

*Specific comments*

*6. I think it is not very clear how the amalgamating, expanding, contracting or splitting and adding of layers works. For example, in p. 23 L 21 it is not obvious what the mentioned 'numerical criteria within the model' are. I would explain these in detail somewhere in the beginning of chapter 2.*
**REPLY**: We have revised Section 2.1 already in the preliminary revision that adds clarity to the layer scheme, and as also noted to R2 (specific comment 40), we will add some further detail in the full revision to better describe this aspect of the model.

*7. p. 6 eq 2 and eq 3: It seems odd that the interpolation of values between levels b-1 and b are depending on b-1, b and b+1 and not only on b-1 and b.*
**REPLY**: This is error is fixed in the preliminary revision

*8. p. 10 Line 5-8: $\varphi SWS$ is defined only in text form and not as an equation, yet it is used in equation 6. There is the danger that $\varphi SW$ defined in eq 10 will be confused with $\varphi SWS$. I suggest mentioning early on in this subchapter how you approach calculating $\varphi SWS$.*
**REPLY**: This notation for energy fluxes has been significantly updated in the preliminary revision; the extended Table 1 now makes the symbol definitions and units clear, and the text is updated accordingly.

*9. p. 12 eq 17: formula only for forced convection? Wind speed at what height? What are the units? I would introduce first the concepts of sensible heat (free and forced convection) and latent heat (evaporation and condensation) before showing the equations.*
**REPLY**: As per point 8 above, the notation for heat transfers has been significantly

updated in the preliminary revision. This makes explicit the reference height. As per the above suggestions for an opening paragraph in the sub-sections, we will introduce free and forced convection here in the final revision.

*10. p. 18 L10: An intro with possible conceptual options to reproduce a surface mixed layer would be good. I would like to know how the chosen approach of a bulk mixed layer depth compares to other approaches in other models (e.g. k-epsilon turbulence closure with Fickian diffusion) and what the consequences of this approach are.*
**REPLY**: A related comment by R2 (specific comment 27) was also made; as above we will improve the opening context in this sub-section in the final revision.

*11. p. 22 L 15 and eq 44 and eq 45: I think an explanation of the concept behind this numerical scheme is necessary*
**REPLY**: We will improve the description/justification for this diffusion algorithm in the final revision.

*12. p. 24. Figure 10: This figure is not enough self-explanatory to me.*
**REPLY**: A revised version of this figure has now been included in the preliminary revision, better depicting the interaction of the inflow parcels with the lake layer structure, and using the updated notation. We also propose to provide a further refinement to this figure in the final revision in response to R2 specific comment 34.

*13. p. 28 eq 60: Shouldn't G not just be another term in eq 4 for all cases?*
**REPLY**: The term $G$ is a vertical flux into the soil below the lake and is only applied to the bottom-most layer of lake ($i = 1$). This changes the thickness of the bottom layer ($\Delta z_{i=1}$), but not the thickness of the uppermost layer ($\Delta z_{i=N_{LEV}}$), or any other layer. You are correct that the change in the thickness of the bottom layer then also leads to a downward vertical shift in the elevation of all the layers above (equivalent to advection, as also discussed with R2 specific comment 8). As this step occurs separately to the surface dynamics routine we had not included a term for this in Eq 4, and instead included a sentence describing this effect (page 7 line 11 in the original upload, and
page line 7 in the preliminary revision). Given this is a potentially confusing point, we will therefore update Eq 4 in the full revision to have an additional term for height change due to inflow/outflow dynamics, and update the text to explain this.

*14a. p. 40 lines 21-24: move this sentence to the intro*
**REPLY**: In light of the proposed changes to the introduction discussed in comment 4, we will update this accordingly.

*14b. p. 40 L 24 – 26: This needs to be better explained.*
**REPLY**: The concept being referred to here is the idea that a long-term simulation from GLM can be developed for a lake, however, since the model is not as detailed as a 3D hydrodynamic model it may not fully resolve event dynamics at a sufficient level of detail (eg a flood inflow event in a reservoir or a localised algal bloom event). In this case, the GLM model can be used to provide the necessary initial conditions and boundary conditions for an event-scale simulation of a higher-resolution model. We propose to mention this in the full revision in Section 5 as part of the integration text.

*List of Corrections*

*15. p. 1 lines 31-32. Consider splitting sentence as it contains different ideas.*
**REPLY**: Thank you for the suggestion, we will revise the abstract in the full revision.

*16. p. 2. Line 1: write only 'standing' as this word is comprehensible and you don't use lentic in the rest of the text.*
**REPLY**: Thank you for the suggestion, we will adopt this change.

*17. p. 5 Line 17. Write the definite instead of the indefinite integral or otherwise phrase it in a sentence.*
**REPLY**: We will update the integral to be between $H_0$ and $H_{max}$.

*18. p.6 eq 1: could be simplified*

**REPLY**: Thank you for the suggestion, we will adopt this change.

*19. p.6. Line 11-18: Should this go in the introduction?*
**REPLY**: This section has been improved in the preliminary revision, and we will reconsider the placement of this statement when revising the introduction in the full revision.

*20. p. 9 eq 9b: Contrary to R1, I managed to get the peak at 80ÅŮęSZA. The equation seems to be the same as used in fig 3.*
**REPLY**:

*21. p. 9 eq 9c: Specify units, also see comments of R1.*
**REPLY**: This is updated in the preliminary revision and also see the reply to R1.

*22. p. 9. L 6: Ux is wind speed at which height?*
**REPLY**: Within the preliminary revision, all meteorological variables are now referenced to 10m, $U_{10}$. This is computed from the user input data ($U_x$) as $U_{10} = f_U U_x$. The updated equation in the preliminary revision (now Eq 12c) however still needs correcting from $U_x$ to $U_{10}$.

*23. p.9 figure 3: Specify the values of relative humidity, wind speed and atmospheric diffusive radiation used for eq 9c. I agree with R1 that the label is wrong, but I think it should be SZA = 360ÅŮę $\Phi zen/(2\pi)$ = 180ÅŮę $\Phi zen/\pi$*
**REPLY**: This figure is updated in the preliminary revision, and, we will extend the caption to specify the values.

*24. p. 11 eq 16 a-d: Use either only ÅŮęC or only K in equations, now they are mixed. I found eq 16 c in Henderson-Sellers (1986) but strangely I couldn't find this equation in Brutseart (1975).*
**REPLY**: The notation is now updated so as to distinguish between them ($T$ and $\theta$, respectively).

*25. p. 12 L 12: no units specified for latent heat of vaporization*
**REPLY**: Table 1 in the preliminary revision has been updated to have unit descriptions of all variables.

*26. p. 13 L 9 and L 13: I guess the authors meant eq 17 -18 and not eq 16-17?*
**REPLY**: Yes, corrected in the preliminary revision.

*27. p. 16 L 6: 'penetrating the surface', which surface?*
**REPLY**: This is referring to the ice/snow surface, now corrected in the preliminary revision.

*28. p. 20 eq 35: What is u0?*
**REPLY**: Following our updates to the notation in the preliminary revision it is now $u_{b_{old}}$; it is the layer velocity of the previous time-step. We will explain this more fully in the full revision (and add to Table 1 as it has been overlooked).

*29. p. 22 eq 39: Please explain the variables in this equation.*
**REPLY**: Apologies for this oversight and confusing notation. The preliminary revision includes the description of $V$ further down (and in Table 1), however we will move this first description up to this point. The symbol for the surface layer thickness is also now matching the earlier use. However, we noted a further error since the k in this expression should be squared, and will be updated as $k_{TKE}^2 = \frac{c_{wn}A_S}{V z_{sml}}$.

*30. p. 22 eq 43: check index, should be hl not hi*
**REPLY**: This is updated in the preliminary revision (now Eq 50).

*31. p. 23 L 17: typo: entrain (not entrains)*
**REPLY**: This is updated in the preliminary revision (now Page 24 Line 17).

*32. p. 23 L 18: typo: the (not th)*
**REPLY**: This is updated in the preliminary revision.

*33. p. 34 figure 17: increase font size and size of arrows.*
**REPLY**: We will update the resolution and fonts in this figure for the full revision.

*34. p. 34 L 19: Insert the references into the place holder*
**REPLY**: This is updated in the preliminary revision.

*35. p. 35 L 26: Who is testing these 'Wrappers' and examples? What is a wrapper?*
**REPLY**: We will update this sentence in the full revision.

*36. p. 37 L 13: What is HTCondor?*
**REPLY**: We will update this sentence in the full revision.

*37. p. 27 L 16: Start a new paragraph at 'GRAPLEr's Web service . . .' to highlight this idea.*
**REPLY**: Agreed - we will update this in the full revision.

*38. p. 40 L 24 – 26: Explain better.*
**REPLY**: See the reply to comment 14b above

*39. References Formatting: Some parts are underlined, remove it. Change all to coherent formatting.*
**REPLY**: The reference formatting will be refined in the full revision.

*40. P. 49 L 23, 25 same author, write same initials.*
**REPLY**: Agreed - we will update this in the full revision.

*41. P. 51 L 13, is there a translation of this Japanese paper? Check the year (2014 in text, here 2015)*
**REPLY**: Unfortunately there is no English translation of this article, however, Yajima pers comm provided an English summary of the algorithm performance and coefficients. Year has been corrected in the preliminary revision.

*42. Table 1: If there is no default variable, can you give a range for snow density, compaction coefficient, and thermal conductivity of snow?*
**REPLY**: Table 1 has been updated to specify "computed" and in the comments the relevant equation is now referred to. We will aim to add the ranges to the table where possible in the full revision.

*43. Table 1: Latent heat of fusion: remove the trailing zero 334*
**REPLY**: Updated in the preliminary revision.

---

## Author Comment (AC6) · 25 Jan 2018

Dear Editor & Reviewers,

We thank you for the opportunity for our paper to be considered for publication in GMD, and are very grateful for the comments received during the discussion phase of the paper. We have found the comments very detailed and insightful and they will guide us to significantly improve the next version of the manuscript which we will upload in due course.

[Figure]

We have already provided specific replies back to each of the reviewer comments in the discussion forum in order to describe how we will revise the manuscript. Whist there are numerous changes required, we list a summary of the main issues identified as:

**1.** *Numerous issues associated with mistakes in the notation and units of several equations, plus the ambiguous use of some symbols;*
**2.** *Lack of context/justification for the adoption of some of approaches/equations in the model description sub-sections (eg. surface mixing and inflows);*
**3.** *A long and potentially hard to navigate structure;*
**4.** *Issues with figure readability and axis labels;*
**5.** *Numerous typographical and minor editorial issues, plus some errors with reference citations and the formatting consistency in the reference list;*
**6.** *Requirement for a DOI to be included for the code-base.*

The obvious issues with mistakes in the notation and units initially noted by R1 led to the development of a "*preliminary revision*" that was uploaded to the discussion forum (dated 8th Jan). This revision includes significant changes to Section 2 of the paper and primarily addresses Item 1 above, but also many of the issues relevant to Items 2-5 are already resolved in this upload.

Following the discussion comments, a fully revised paper that builds on changes in the preliminary revision is under development to address the outstanding issues. The proposed major change relates to a re-worked introduction to better introduce the need for the model and explain the structure of the paper. A few remaining notation/equation issues remain (following further reflection on R2 and R3 comments), and some improved descriptions of selected algorithms is required. Additional improvements to several of the conceptual figures and simulation results have also been suggested. Please refer to the individual responses for specific details.

The comments have also led to slight code adjustments, and so the model version will be updated accordingly on GitHub to match the updated paper version, in order to ensure the paper and code DOI are consistent.

We thank you again for the significant time and effort that have gone into the discussion.

Kind Regards
*Matthew Hipsey*, on behalf of all co-authors.

---

## Author Response (AR1)

*8th Mar 2018*

Dear GMD Editor & Reviewers,

**Re. Revision of gmd-2017-257: A General Lake Model (GLM)**

We thank you for the opportunity for our paper to be considered for publication in GMD, and are very grateful for the comments received during the discussion phase of the paper. We have found the comments very detailed and insightful and they have guided us to significantly improve this version of the manuscript.

We have already provided specific replies back to each of the reviewer comments in the discussion forum explaining our approach to the revision. A summary of the main issues identified were:

1. Numerous issues associated with mistakes in the notation and units of several equations, plus the ambiguous use of some symbols;
2. Lack of context/justification for the adoption of some of approaches/equations in the model description sub-sections (eg. surface mixing and inflows);
3. A long and potentially hard to navigate structure;
4. Issues with figure readability and axis labels;
5. Numerous typographical and minor editorial issues, plus some errors with reference citations and the formatting consistency in the reference list;
6. Requirement for a DOI to be included for the code-base.

The obvious issues with mistakes in the notation and units initially noted by R1 led to the development of a "preliminary revision" that was uploaded to the discussion forum (dated 8[th] Jan). This revision included significant changes to Section 2 of the paper and primarily addresses Item 1 above, and many of the issues relevant to Items 2-5 were resolved in this upload.

Here we upload a fully revised paper that further builds on these changes to address all the comments, where possible.  The major change relates to a re-worked introduction to better introduce the need for the model and explain the structure of the paper. Further remaining notation/equation issues have also been resolved and improved descriptions of selected algorithms has been added. Additional improvements to several of the conceptual figures and simulation results have also been undertaken. Please refer to the individual responses for specific details. A tracked-changes version is at the back of this document.

The comments have also led to significant code adjustments, and so the model version associated with this version of the manuscript is updated on GitHub and is now v3.0. A DOI for the code is to be added if the paper is accepted for publication.

We thank you again for the significant time and effort that have gone into the discussion and look forward to your decision.

Kind Regards

Matthew Hipsey, on behalf of all co-authors.

**Anonymous Referee #1**

General comments

This article describes the scientific basis of a 1-dimensional hydrodynamic lake model that can be coupled to ecosystem models. The model has already been applied to many systems in the scientific community, and I think it is useful publish the model description in a scientific paper that can be referred to for future applications of the model. That said, I stopped reviewing after equation 16, because there were simply too many errors in the equations. I therefore propose to reject the current version of the manuscript and that the authors carefully check all equations before resubmitting the manuscript to this or another journal, depending on the decision of the editors.

REPLY: Thank you for taking the time to review the discussion paper and identify the errors - we sincerely apologise that simple issues related to units and notation were not more thoroughly checked prior to upload. Please note this revision does have some substantial changes to the paper including the notation, text and (some) figures. We very much look forward to your comments and suggestions on this version.

Errors in equations up to eq. 16:

eq. 2 and 3: I think something is wrong with the indices. $h_z$ is located between $h_{b-1}$ and $h_b$. $\alpha_b$ and $\beta_b$ describe the interpolation between $h_b$ and $h_{b+1}$. Thus, the indices in eq. 2 should be $\alpha_{b-1}$ and $\beta_{b-1}$.

REPLY: You are correct; the code was looping from one step behind so we had incorrectly omitted the -1 in the equation. It is now updated

eq 6: I think this equation is wrong. The right hand side is the total heat flux to the surface layer in W m$^{-2}$. This should be divided by $z_{msl}$ to get W m$^{-3}$. Then, it should be divided by the water density $\rho$ in kg m$^{-3}$ to get W/kg, and finally by $c_p$ to get °C/s for $dT_s/dt$. Therefore, the multiplication term on the left hand side should be $z_{msl} c_p \rho$, rather than $c_p/(A_S z_{msl})$.

REPLY: You are correct; it is now updated and in fact this section is now significantly revised. Table 1 now also includes all the notation and relevant units.

eq 9b / Fig. 3: I could not reproduce the maximum of the Briegleb function at 80 degrees zenith angle. Using equation 9c yielded a monotonically increasing function between 20 and 90 degrees (with a minimum at about 20 degrees). Also the equation in the legend is wrong, it should be SZA = $\Theta_{zen}*180/2\pi$.

REPLY: You are correct; This was an error in creating the plot rather than the model itself, and Figure 3 has been re-created and caption updated.

eq. 12: I think $\phi_{SWS}$ (i.e. the shortwave radiation absorbed in the surface layer) should be replaced by $\phi_{SW}$ (z=0) in the nominator. Otherwise, the euphotic depth increases with increasing radiation absorbed in the surface layer, which does not make sense. Same in caption to Fig 4.

REPLY: The notation has been improved to prevent confusion; $\phi_{SWS}$ is what is absorbed and $\phi_{SW0}$ is what arrives at the top of the surface layer; The euphotic depth computation is now based on the fraction of incoming light.

eq. 16: I think in equations 16c and 16d $T_a$ should be replaced by absolute temperature (i.e., 273.15 °C should be added to $T_a$).

REPLY: You are correct; notation updated to have K and C unit options for temperature

Also units should always be provided, especially for empirical equations (e.g. $e_a$ in eq. 16, and $U_x$, RH, and diffusive radiation in eq. 9c).

REPLY: Units have been thoroughly checked and updated in the revised upload, plus the updated Table 1 summarises units for all variables/symbols to avoid confusion.

Besides that, a few other points I noticed up to page 11 (Page xx, Line yy is abbreviated as xx/yy)

In general, the paper is well written and easy to read, but there are quite a few long and complicated sentences which I think should be simplified to facilitate reading (first two examples: 2/24-29 , 3/12- 17).

REPLY: These examples have been re-written and the revised version has also been check for these issues as best as we could identify.

3/10: This list of references seems to be somewhat inconsistent. Some of the references refer to model development, some to model applications. It would be more logical to cite only model development references.

REPLY: Point noted, these papers were reflecting the diversity of 1D models we are aware of, but agree these should more specifically refer to development refs; introduction is updated.

4/30: The text seems to imply that the requirement for site-specific calibration in other models is due to numerical diffusion caused by the fixed grids. If that is the intention, this should be explained. If not, the sentence should be modified.

REPLY: Sentence modified.

5/7: Incomplete sentence

REPLY: Section has been revised and reworded.

Figure 1: Shouldn't the local runoff, and the submerged inflows and groundwater seepage be written in blue?

REPLY: Local runoff is computed by GLM (Eq 7 in updates manuscript). Submerged inflows and seepage however has been updated to be blue in Figure 1 as they are specified.

eq. 1: From the text (layer volumes are determined ...), I would have expected an equation for the individual layer volumes here, but this is the integrated volume from the bottom of the lake to the top of each layer. This should be clarified in the text.

REPLY: Section has been revised and reworded to hopefully introduce the layer structure and notation more clearly. The layer volume balance is not presented here due to the complexity of options and it is not appropriate at this early stage of the paper; however, edits later on have aimed to make this more clear. Notation now is V for cumulative volume to a point, and delV for layer volume.

6/3: technically, it is the same, but I think it would be clearer to write $2 \leq b \leq N_{BSN}$.

REPLY: Updated.

6/4: how are these finer depth increments determined?

REPLY: Now summarised in symbol table (Table 1).

6/9: Since the Unesco (1981) equation has been replaced by TEOS-10, I think it would make sense to use the latter rather than the former in a new model. I also think it should be mentioned that the density effect

of salinity in these seawater equations is quite different from that in most lakes where carbonates are usually the dominant species rather than NaCl.

REPLY: Thank-you for this suggestion. The revised code now has the TEOS option included, accessible via the setting density_model =1. The UNESCO option is also included, and facility for users to add a custom option.

6/24: heat balance **of** the surface layer 7/2-3: why is only rain but not snow multiplied with $f_R$? Also, even though this should be clear to the reader, it should probably be mentioned that S is in water equivalents.

REPLY: Updated.

eq 5: to be precise, this equation should be limited to a minimum of zero, as otherwise it will become negative if the rainfall is too weak.

REPLY: Updated - this was the case in the code, but not properly summarised in the Eq.

Fig 2: Add some space between the 10 and the exponent in the y-axes labels of panels c and d. Do all these time series start on 1 January of a year?

REPLY: Lakes do have variable start times (for various reasons) which is why we didn't use exact date in the x-axis. The individual lake simulations are documented on our GitHub site with explanations and we didn't want to make it too confusing here with too many details.

eq. 9a: instead of subtracting $\pi/2$ within the sine functions, it would be easier to use -cos.

REPLY: Apologies, there was a "-" sign wrong in this Eq so both should not have been - $\pi/2$. In the new manuscript (Eq 12a) we have the addition or subtraction of $\pi/2$ is listed properly in order to allow us to differentiate Sth vs Nth hemisphere sites.

eq. 9b: where does the factor 1.1 in the nominator of the first term come from? Maybe I overlooked something, but I could not find it in Briegleb et al. (1989).

REPLY: Thankyou for noticing this detail - this inclusion of 1.1 has been picked up from the implementation by Li et al (2006), who compared models and have this coefficient included.

However, for consistency, we have removed this from the Eq in the paper, and the code-base going forward. We note however, that the change made only a modest difference to the function.

Li, J., Scinocca, J., Lazare, M., McFarlane, N., Von Salzen, K. and Solheim, L., 2006. Ocean surface albedo and its impact on radiation balance in climate models. *Journal of climate*, *19*(24), pp.6314-6333.

eq. 9c: I was not able to check this equation, as the source is in Japanese, but I did not get anything similar to what is shown in Fig. 3 trying different values for RH, U and the diffusive radiation. Please check whether the equation is correct, and specify the values used to produce Fig. 3. Furthermore, Yajima and Yamamoto is dated 2014 here but 2015 in the reference list.

REPLY: Thankyou again for checking this - the equation implemented is based on the Equation 1c of this publication, whereby y is cos(solar angle), the a and b coefficients were set by multi-variate regression, and x1 is RH(%), x2 is wind speed (m/s) and x3 (-) is a parameter referring to the amount of atmospheric diffuse radiation.

We initially received the coefficients from the author (Yajima pers comm.) for our implementation, and following your comment we have now updated the Figure 3, the citation date, and the equation. The caption also now states the values of x, x2, x3 used in the graph. For your reference, R code for the algorithm is below:

angled = 1:89
angle = angled * pi/180
dr = 6
ux = 4
rh = 80
yalbedo = max( 0.02, 0.001 * rh * (1-cos(angle))^0.33 - (0.001 * ux * (1-cos(angle))^(-0.57)) - (0.001 * dr * (1-cos(angle))^(0.829)) )

Fig 4: y-axis of panel b is not depth, but elevation, y-axis of panel c is not labeled.

REPLY: You are correct; the y-axis should be height not depth. As part of the revised manuscript we have uploaded we have undertaken a major review of notation used throughout so that depth (z), height (h) and elevation (H) are used consistently.

All figures have been re-created and revised in the revision, to resolve these issues.

Also it seems that $A_{BEN}$ is calculated on a different time scale than the radiation in (c). Many low radiation events are clearly visible in (b) but do not show up in (c). This probably makes sense, but the time scale should be mentioned somewhere.

REPLY: Please note that in fact the ABEN is now being computed based on a percentage of light reduction (I/I0), rather than a specific light intensity.

The step changes in the panel (c) time-series are not due to time-scale of calculation, but rather to do with changes in the layer thickness and structure.

11/11: It does not look like the equations were copied from Henderson-Sellers (1986), but rather from either the original sources or from Flerchinger (2009)?

REPLY: The placing of the citation to Henderson-Sellers incorrectly gave the impression this is where the full expression was from. You are correct that that we have just chosen 4 based on original sources to implement within the model. We have now cited Henderson-Sellers and Flerchinger as sources for further description and information, rather than as a source for the algorithm set.

**Anonymous Referee #2**

General comments

1. This paper presents the formulation of a one-dimensional model of thermodynamics, mixing, and evaporative and momentum fluxes from lake surfaces, which conceptually should be applicable to a wide variety of lake morphologies. However, the manuscript achieves the paradox of simultaneously containing too much information and not enough information. It goes into great detail with many equations. However, in order to present this level of detail without losing the reader requires more care and at least some additional details.

REPLY: We greatly appreciate the authors comments and insights. We acknowledge the paper is heavy with information, and chose the journal Geoscientific Model Development for this submission as it does support papers focusing on model description. Given the nature of the model it is our desire to have a comprehensive description of the numerous sub-models and options that is peer-reviewed and citable, and, as a result, it has led to a long paper compared to a traditional manuscript. As outlined in the below comments, in this revised version we have endeavoured to improve the narrative and flow of the paper.

I have a few broad suggestions that I think might help.

2a. Before anything, you need to have a clear idea of who is in your audience. You could even have an explicit statement of this very near the beginning, and direct users with a lower level of expertise toward a simpler users' guide.

REPLY: The paper is intended as a description of the model rather than a research paper. Whilst there have been some unpublished manuals and incomplete documentation for the model during its initial development, we felt this is unsatisfactory for a model that has had increasing uptake by lake scientists. In our initial submission, we had stated the aim as:

"Given that individual applications of the model are not able to describe the full array of features and details of the model structure, the aim of this paper is to present a complete description of GLM, including the scientific background (Section 2), model code organization (Section 3), approach to coupling with biogeochemical models (Section 4), and to overview use of the model within the context of GLEON specific requirements for model analysis, integration and education (Section 5-6)."

Our audience is therefore to advanced users looking to understand the mechanics of the model, and/or users publishing applications that need to cite the model methods and approaches adopted in their individual simulations. We acknowledge some users will not dig into the detail and can following the "user" material we have provided on the model website. We note that we have not provided many details about how to run the model, but rather it was our intention for readers to be able to understand the science basis and model structure.

The updated introduction, will hopefully make this intent more clear.

2b. This reviewer has a back- ground explicitly in meteorology, but significant exposure to lake dynamics as well, albeit mostly regarding very large lakes. Depending on your intended audience, you might have readers who will have difficulty with terms such as aliquot and even the intended understanding of "scalar concentrations" as used on p. 27, lines 2-3.

REPLY: Thank-you for pointing these out. We have attempted to improve both these descriptions to improve the clarity of this terminology.

3. To reduce the length of a single paper, one option is to break it into multiple papers. Another is to move more of the detail to appendices. Since it is an online journal, there is probably not a large problem with

overall page length. Things as detailed as conversion of area as a function of depth to volume in a layer (eq. 1) seem like they could be skipped in the main text and relegated to an appendix.

REPLY: We appreciate the comment and agree with the general desire for shorter papers, and feel that in a model description paper such an argument could be made for several of the sub-model descriptions. However, a key purpose of our paper was to describe the model from start to finish (as is done through Section 2), and the layer structure and depth-area-volume relationship is key to the model approach. Aspects such as solar radiation and atmospheric stability computation are optional modules in the code-base and not directly related to what happens within the lake model, and so are included as appendices. Nonetheless, in the preliminary revision we have made some changes to Section 2.1 (referred to specifically in this comment) and feel the new wording of this section has a stronger narrative that hopefully connects it with the subsequent sections.

4. It would be useful to have a brief introduction to each sub-section. This should start out by stating the goal of that sub-section, i.e. what will be the final equation (or set of equations) derived in the section. Then state what elements will combine to get that final equation(s). One particularly glaring issue is that sub-section 2.5.2 ends with eq. 45, defining the variable f, but nowhere does it say how f relates to any other part of the model. It seems to be something that one would multiply by the difference in temperature (or another scalar) between layers) to get the exchange of that scalar between the two layers in a single time step. Whether this is exactly correct or not, the statement is missing from the manuscript.

REPLY: The revised version has attempted to address this general comment by strengthening the narrative in the sections from 2.1-2.7 (the main sub-model description sections). In reference to the comment about the "f" function, the use of the term scalar (and symbol C) is now better introduced, placing this Eq in context. We note that the f computed in this equation is required in the equation where concentrations are updated based on the magnitude of diffusivity between layers (Eq 54 in the revision, and Eq 44 in the original version).

5. A simple overall schematic would be good to have early on (Fig. 15 with less detail). This would make it less abrupt when "water quality model AED2" is mentioned on p. 27 (I may have missed it, but I don't think it was mentioned before.

REPLY: In light of other comments (#8 of the general comments and #42 of the specific comments), we re-drafted Figure 16 and improved clarity in the Fig 1 schematic, but in the ned chose not to make another schematic figure early in the paper. Hopefully the improved text flow in Section 2 will make this more clear.

6. Even if only for your own reference as author, Table 1 needs to be expanded to include every variable used in every equation! And in this expanded table, include every variant of each variable based on the use of different subscripts, prime, and circumflex ("hat"). With this many equations, it is rather inevitable that you also end up using the same symbol for different things (I noticed N in particular). Then, for each variant of each variable, put additional columns for: description, units, spatial type (defined at surface only, spatially continuous, or at discrete layers), and which equations it is used in. The reader needs to have all of these carefully defined.

REPLY: The revision includes a fully updated Table 1, and addresses the issue of confusion of notation by having a more consistent nomenclature scheme.

7. My high school chemistry teacher is the one who taught me how to use units in equations, and the importance of doing so, hence the need for them in the big table above. Some examples of problems with units in the equations that may indicate that the equation is simply wrong: In eq. 52, Q seems like it should have units of m cubed per s, and h units of m, so the right-hand side does not have units of velocity.

REPLY: We apologise for this mistake - this equation (now Eq 62) has been corrected (and the code was checked for consistency). The updated Table 1 now also has consistent units throughout, included for each variable.

8. I approached this manuscript with the immediate question of what makes this model better and more useful than the many others that are available. The introduction does a pretty good job of answering this, but it may be good to give some examples of uses that are not satisfied by other models. This could be introduced with a schematic of its components, options, and functions, at a lower level of detail than in Figs. 15 and 16.

REPLY: Thanks for this suggestion. We do not feel in this paper we have the scope to do a full meta-analysis of the features of all the other lake models and where they are inadequate relative to this effort. Given the number of sub-modules and options, and very wide array of application contexts, it becomes difficult to make judgments about what model options are required for specific lake types and we defer to users to use the model in an appropriate way. We note that Figure 1 does currently aim to be a schematic for the different model components, and functions, but agree this is currently not linked to applications or specific lake types. However, we were worried about adding another Figure and hope the improved introduction may help resolve this issue.

9. Anything that can be done to bring the reader's intuition into play when introducing equations will improve comprehension of the manuscript. Eq. 27 isn't the most problematic one, but I'll use it as an example. You might introduce it by saying something like: "Shortwave radiation is absorbed and attenuates with different e-folding depths for snow, white ice, and blue ice, and these also depend on the light's wavelength. The overall effect is..."

REPLY: Thanks for this suggestion. Wording has been extensively updated across sections 2.1-2.6 to try to improve the flow of the text and clarity of the descriptions.

10. Eq. 47 has me very mystified about how a standard definition of Richardson's number translates into this equation in terms of the angles of inflow geometry (part of the problem may be that I don't feel like I understand the meaning of the angle labeled alpha in Fig. 10; the illustration isn't helping me). It also has me asking "Richardson number at what location?" At the interface of the river water and ambient lake water?

REPLY: We have cited the original source (Fischer et al 1979) of this equation and approach in the original paper, however to improve the interpretability of this we have extended the sentence linking it to the figure.

11. Several of the figures have characters that are so small as to be illegible.

REPLY: We have redrafted the schematic figures with larger symbol fonts.

12. What about modularity of the model? Can other schemes for pieces of the model be plugged in? The details of how may be an appendix or even another paper or guide.

REPLY: The model is not written in an object-oriented fashion, however, custom schemes can be linked to for key sub-module components (e.g do_mixing, do_deep_mixing etc) with some code-adjustments. A comment on this will be added to the code structure section (Section 3).

Some particular examples of the general problems above are among the specific comments below.

Specific comments:

1. In nearly every review that I do, I refer to the rules of hyphens found at http://www.grammarbook.com/punctuation/hyphens.asp particularly Rule 1 on that page. Your manuscript is actually better than most, but here are the problems that I found: P. 2, line 23 should have "system scale" (no hyphen). "Time-scale" is sort of borderline; I tend to use it without a hyphen when it doesn't modify another noun. P. 4, line 15 and some figure captions: "time series". P. 13, lines 22 and 24: "Wind sheltering". P. 24, line 3: "user-specified".

REPLY: Thankyou for this link and advice. These are updated now in the revision.

2. P. 2, line 33: Change "spatial" to "vertical", to contrast with horizontal from earlier in the sentence.

REPLY: This is updated in the revision

3. P. 3, line 10: Stepanenko is misspelled.

REPLY: This is updated in the revision

4. P. 3, lines 17-18: Do you have any comment on use in small vs. large lakes? Both in terms of area and depth? There might also be considerations in terms of morphological complexity.

REPLY: We have now mentioned size when describing 1D models.

5. P. 3, lines 30-31 specifically mention temperature, salinity, and density. Later on, there is mention of the broader category of "scalars". Are you unnecessarily limiting yourself here? I am thinking of such things as dissolved oxygen (mentioned later) and concentrations of nutrients and contaminants.

REPLY: The reference to scalars is now updated in the preliminary revision (now page 5 line 30) to be more clear, and the option of coupling to a water quality model is mentioned (page 6 line 1)

6. P. 3, line 32: I think many readers will object to the use of "hydrodynamic" here, since it does not explicitly represent advection of water. Continuity equations say that dynamics requires at least two dimensions.

REPLY: We would argue that term hydrodynamic refers to the motion of water in general, rather than specifically need to resolve advection in two dimensions. In addition to vertical mixing and transport of water and its constituents, the model does parametrise horizontal advection of inflow water (Eq 62 in the revision). Therefore, whilst the model is not resolving the Navier-Stokes equations, the model approach is intended to capture the movement of water within the lake domain as it is filling, drying, and subject to inflows and withdrawals.

7. P. 4, lines 31-32: "user-defined" and "set by the user" are redundant.

REPLY: Section 2.1 has been significantly revised in the revision upload and this problem is removed.

8. P. 5, line 8: This model doesn't include it, but in reality, vertical advection of heat can be important along with vertical mixing. Again this may depend on the size of the lake, but this factor should be acknowledged here.

REPLY: Aside from mixing, the model captures to some degree the advection of heat and constituents in the water as brought about by inflow or outflow dynamics. For example, new inflowing water would add a layer and lift up layers above its insertion depth, which could be considered as vertical advection. Short term advection associated with internal waves, or perhaps upwelling in large lakes, however is not included.

9. Eq. 4 has issues with sign conventions, units, and possible missing terms depending on how one understands it. It is E multiplied by lake area should be considered a volume flux. For practical purposes these can be considered equivalent, but there would be another term for water density to convert between mass flux and volume flux. Are the evaporation, snowfall, and rainfall defined as being specifically over the area of the lake itself, or over the drainage basin? If over the lake, how is the Q term for runoff optional? I have a hard time imagining many cases of lakes in which it is not a major term in the water balance. If these variables are defined over the whole basin, is there an issue with agreement in timing between P – E and runoff?

REPLY: Eq 4 and this section is now corrected in the revision, with units clarified in Table 1. The heat flux conversion to volume is clarified in this version, accounting for density. The P, E and S terms are computed over the active lake surface. The inflows from the drainage basin into the lake are set by the user as an "inflow" (section 2.7). The local runoff term is not referring to runoff from the wider drainage basin where the lake sits, but the local land within the lake area ($A_{max}$-$A_s$) that is not inundated with water. This could be important in reservoirs where the volume is drawn down significantly from the maximum level set by the user, or for ephemeral wetlands; in both these cases this area would generally not be considered as part of the inflows from the wider drainage basin, and so Eq 7 (in the preliminary revision) allows for this "dry lake-bed area" to contribute to the water balance.

10. P. 7, line 21: "However" is an interjection, not a conjunction. Here, it stands between two independent clauses, so preceded with either a semicolon or period. Also p. 33, line 10 and p. 35, line 11.

REPLY: This is updated in the revision

11. Eq. 7 is a place where I started to distinctly feel the problem of definition of variables. I had to really think through what had happened to the subscript S from eq. 6 and why the "hats" were there in eq. 7.

REPLY: This is updated in the revision; both the updated notation in Section 2 and Table 1 should prevent this confusion.

12. Eq. 9a has a strange step function in space. Can you justify this not being a smooth function, but rather one value for all of the northern hemisphere, another for the equator (an infinitesimal amount of space), and another for the southern hemisphere?

REPLY: This function has been previously used by many authors for albedo and so is included - we acknowledge this option is simple and hence Option 2 and 3 are provided in the model (Eq 12 in the revision), with specific resolution of the solar angle.

13. P. 9, line 6: What makes the different formulas oceanic and lacustrine? Is there anything to help the reader's intuition for why these formulas should be different?

REPLY: We looked into this, and I think they are just different empirical approaches based on the indivdual researchers choice.

14. Audience issue: P. 10, line 14 departs from the main conceptual theme to give a specific file name. Detailed mathematical formulation and detailed user information don't mix very well.

REPLY: We agree with this observation and have remove the user detail in Section 2.

15. Eq. 14: Are you using a different temperature at the skin (interfacial layer) or bulk temperature a little deeper? Specify how deep.

REPLY: The surface temperature used in the long-wave computation is from the upper-most model layer. This is homogeneous over the depth of the surface layer (surface layer has a sub-script denoted "s"),

however we cannot specify the exact depth, as the model computes the surface layer thickness dynamically, based on the density gradient, mixing intensity and layer limits.

16. P. 11, line 10: Air temperature at how high above the surface? Standard is 2 m, but there may be adjustment needed if measurements are at a different height than that intended in the formulation.

REPLY: The air temperature used throughout the paper is now standardised to 10m and this is defined specifically in Table 1. For users with data from a different height, that may use the f_U parameter to scale the data appropriately.

17. Eq. 163: "Brutsaert" is misspelled.

REPLY: Updated in the revision

18. P. 11, line 13: The range of octals should be 0 to 8.

REPLY: Updated in the revision

19. P. 12, line 11: This should specify molecular weight of dry air, and it might be better to say mass rather than weight.

REPLY: Updated in the revision

20. P. 13, lines 5 and 9: I think these should reference eqs. 17-18 rather than 16-17.

REPLY: Updated in the revision

21. Eq. 23: The Latin v and Greek nu are somewhat difficult to distinguish here, and are even more difficult to distinguish in the text following.

REPLY: Updated in the revision; the notation has been standardised, thereby removing this issue.

22. P. 13, lines 16-18: Thanks for explicitly using units here. Am I correct in understanding that molecular heat conductivity is molecular heat diffusivity multiplied by heat capacity? I think it would be better not to use heat conductivity, but use diffusivity and capacity.

REPLY: Note that units in the revision are updated in the text where appropriate, and Table 1 now includes all units for all variables. You are correct that conductivity is related to diffusivity and heat capacity. The notation adopted (now Eq 28) is based on the original description in TVA (1972)

23. P. 13, line 22: "may be" should be two words.

REPLY: Updated in the revision

24. P. 15, line 13: This use of "conductive heat flux from the ice or snow cover to the atmosphere" particularly triggers my thought bias as a meteorologist. That bias prompts me to assume that this means flux through the atmosphere's interfacial layer, where molecular diffusion dominates. But I also wonder whether it means heat conduction through the ice. These are not equal in general, and need to be distinguished from each other.

REPLY: It does mean the upward conduction through the upper ice layer. For clarity this is depicted in the schematic in Rogers et al (1995) that we had mentioned in the text, on which the model is heavily based (Figure 5 in this source). As such, we chose not to reproduce the energy flux schematic in this manuscript, and have updated the text, but can consider a sub-plot if deemed necessary.

25. Eq. 26: Why isn't shortwave radiation here?

REPLY: This is the way the approach was reported in Rogers et al (1995) (Eq 7 in their paper) and we have followed that; in this case at the surface interface it is assuming the non-pentrative heat flux is in balance with the conduction of heat up to the surface. The conduction of heat up is based on the amount of light that has penetrated (as computed in Eq 31 in the revised upload). We updated the text here to prevent confusion and be more explicit that the net is referring to the non-penetrative component.

26. P. 16, lines 11 and 12: These are very wide ranges of albedo. Please describe the conditions under which different parts of this range manifest.

REPLY: The variation is discussed in the Vavrus et al (1996) paper that we cited; we have added a brief explanation about the variability to this sentence in the revision.

27. P. 18, line 10: The idea of energy required for mixing needs to be introduced more carefully. If you think of the water column as continuous in space, mixing is also a continuum. But the concept here is based on discrete layers, and this indicates how much energy is required to outright include a model layer in the mixed layer.

REPLY: The approach to bulk mixed layer models is well established and as such we provided suggested references for readers interested to gain a deeper understanding of the background: Kraus & Turner (1967) and Kim (1976) and Imberger & Patterson (1981). The model approach we have adopted also has been the subject of numerous prior publications that have thoroughly validated that the layer discretisation can accurately capture the mixing continuum in monomictic and polymictic lakes.

28. Eq. 31: $C_T$ seems completely undefined.

REPLY: Updated in the revision via Table 1

29. P. 20, line 14: This cannot be reproduced unless your definition of "epilimnion" and "hypolimnion" are precisely stated.

REPLY: Updated in the revision (now Eq 42.)

30. P. 20, line 20: Is $K_\epsilon$ eddy diffusivity.

REPLY: Thankyou for noting this; the revision now includes an update to specify this.

31. Since the vertical axis in Fig. 8 has zero at the bottom, it appears to be height above the bottom, not depth. It might be worth explaining that the varying top height of the color fill is due to varying overall depth (assuming that this is correct).

REPLY: You are correct; in the revision we have been more careful in use and notation associated with depth, height and elevation, and have updated the figures as suggested to use height instead of depth.

32. P. 22, line 13 and eq. 43: This seems to say that sigma should have units of inverse seconds squared, which implies that the square-bracketed part of eq. 43 has units of (m/s) squared, but exponential operations can only be performed on unitless numbers.

REPLY: Thankyou again for noting this inconsistency. Both the top and bottom of the fraction in this equation (Eq 53 in the revision) are in the units of m^2, so the fraction for the exponential is dimensionless. Text has been updated.

33. P. 24, line 5: Slope is usually defined as a ratio of rise divide by run, not an angle. I would say that the slop is the tangent of the angle.

REPLY: You are correct, this sentence is updated.

34. The inset in Fig. 10 does not help me understand the meaning of the angle alpha, and it could use larger lettering.

REPLY: We have made a modification to this figure to improve readability and detail.

35. Equation 52 seems messed up. Units-wise, it seems to make more sense if h squared is in the denominator instead, but I also wonder why there is no dependence on width of inflow.

REPLY: We apologise for this error and it has been corrected in the revision (now Eq 62). The scheme does not account for the width of the inflow explicitly, but it is approximated based on the water height and channel angle (as depicted in the inset of Fig 10).

36. P. 26, line 16: In standard usage, dz represents an infinitesimal difference in z in continuous space, but here it is used to indicate a finite distance in discretely sectioned space.

REPLY: This expression and associated notation has been revised (now Eq 69). The symbol delta is now uniformly used to refer to a length-scale, as is the intent here.

37. P. 26, lines 16-17: It is difficult to understand the distinction between "height of withdrawal" and "edge of the withdrawal layer".

REPLY: This description has been further developed in the revision (now Page 30), to explicitly define how the withdrawal thickness and the relevant layer edge is determined.

38. P. 26, line 19: Why say "fluid" rather than "water"?

REPLY: You are correct that water is more specific – we have replaced the use of fluid with water in the revision.

39. A formatting problem put some labels that belong to Fig. 12 on p. 27, while the rest of the figure is on p. 28.

REPLY: This has now been rectified

40. P. 28, lines 10-11: Removing no more than half of a layer's mass per time step seems like a reasonable way to ensure numerical stability, but it would be good to re- mind the reader here of the layer merging scheme that is likely to kick in. This merging and disaggregation scheme, also mentioned on p. 34, is never really described well.

REPLY: Thankyou for this suggestion. We have revised both these sections to improve clarity of these descriptions in the preliminary revision, including making both consistent by enforcing a 90% limit. The merging and splitting scheme is introduced (page 6 line 7 of the revision), however we note the concern this is not described fully, and have referred to it where relevant now, but probably should consider a new sub-section.

41. P. 29, line 3 says "wind speed and fetch. . .calculated as", while eq. 61 only shows the formula for fetch.

REPLY: The sentence introducing this Eq has been be re-worded to remove this ambiguity.

42. To help solve the problem of small print in Fig. 16, it may be useful to transfer some of the information to a table instead.

REPLY: Thankyou, we have re-drafted this figure, to prevent the small fonts.

43. P. 34, line 19 has a placeholder for a citation. This is evidence of a poor final edit on the part of the authors.

REPLY: Updated in the revision; apologies for this oversight.

44. P. 34, line 23: If you mention calibration, it would be well to describe this process more fully, in particular which parameters you consider adjustable for purposes of calibration. Where p. 35, line 10 mentions "compare", I wonder whether this might also imply calibration.

REPLY: The initial line referred to is an introductory statement with the intent that the text in Section 5.2 would describe the calibration process more fully. We have mentioned in this section the link to our associated model validation paper (Bruce et al., 2018).

45. P. 48, line 12: "Anneville" misspelled?

REPLY: Updated in the revision

**Anonymous Referee #3**

General Comments

   1. This paper describes the detailed functioning of the 1D physical lake model GLM 2.4 and its application potential. The model incorporates a broad range of physical processes as surface heat exchange, snow and ice dynamics, in- and outflow, submerged inflow and groundwater seepage and can be coupled with or embedded into other models. The authors explain how GLM 2.4 has emerged as a response to the need of standardized, yet flexible and computationally effective community lake model to interpret environmental data from a broad range of lakes collected within the Global Lake Ecological Observatory Network (GLEON). The model has been formulated as a new code in 2012, whereas layer structure, mixing algorithms and physical formulae are based on earlier peer reviewed work. The authors state that the code is computationally efficient and well suited for embedding in larger scale modelling frameworks. The authors present also an overview of pre- and post-processing utilities as well as an innovative cloud computing environment. Lastly, they elaborate on the educational use and gained experience in the classroom.

REPLY: Thankyou for this very accurate summary

   2. I realized that this manuscript is for a major part equivalent to an earlier manual of GLM (V2 Manual, October 2014, accessed on the 08.01.2017 from http://aed.see.uwa.edu.au/ research/models/GLM/Pages/ documentation.html). I think the authors should mention this.

REPLY: It is our intent that this paper replaces the online manual and we have removed the available PDF cited above. The online manual was used as an interim resource to inform users until the model development efforts had stabilised. Now this is the case, the revised version of the present paper includes more detail and numerous improvements, extensions and fixes to errors and should become the key reference. Aspects of that manual associated with model-use & setup that are not covered in this (science-oriented) paper have been migrated to the website pages.

   3a. The model in this paper represents with no doubt a tremendous effort in lake modelling and is of interest for modelers in various fields of environmental research. The publication of this model is a step towards better model documentation and contributes to the general scientific discussion and better lake model development. As such it falls within the scope of this journal. The paper is well written and the language is easily com- prehensible. Unfortunately, this manuscript has some structural problems and there are quite a few mistakes in equations and figures. After dealing with these issues, the manuscript should be good for publishing.
 REPLY: Thank you for this recommendation. We have endeavoured to resolve the specific issues that you and the other reviewers have identified in this revision.

   3b. The main problem of this very long manuscript is that it is missing an instant overview of what is in the paper and what not. Scanning through, the reader gets lost easily in the large chapter 2 'model overview' and might miss the subsequent chapters that elaborate more on the possibilities and significance of this model for the scientific community.
 REPLY: We acknowledge this view, which is similar to a comment #2 by R2, and have updated the introduction significantly with the aim of providing a better "road-map" for the paper structure.

   4. I think that this problem can be fixed with some changes in the introduction:  ć I suggest using subtitles in the introduction. (In the introduction, the authors describe the importance of the study of lakes, the importance of GLEON, the importance of lake models, the advantages of simple models, applications and features of 1D models, the need for a flexible open-source model, how GLM 2.4 answers this need and finally an overview of the paper)   ć I suggest creating a new paragraph starting at p. 3 Line 19 ... "Nonetheless, there . . .". The need of an open source and flexible community model that can be applied to various lakes should be highlighted better. Another additional paragraph could explain how GLM 2.4 responds to this need. As I understand, GLM 2.4 is filling the gap because it provides a standard middle

complexity physics 'shell' (simple yet enough complex to be applied for various lakes) that can be connected individually to or implemented into various other models (e.g. water quality or land- climate models). I think this point could be emphasized. âAˇ ć    A figure could be helpful to draw attention to the significance of this model in the scientific community. This could also be combined with schematic overview of the model functioning (I agree with R2 that anything that gives an overview helps). âAˇ ć    The specific limitations of GLM 2.4 (not of 1D models in general) should be mentioned in the introduction. Like this, the reader may have a quick idea whether GLM is suitable for him/her. What are the key features of this model that set it apart from other models? âAˇ ć    On p. 4 lines 5-9, the authors explain the aims of the paper and in which of chapter 2-6 these aims are met. I think these lines are important and should be extended to a paragraph by itself to make sure the reader is fully aware what to expect from the paper. In the same paragraph, I would also expect some more information regarding what this paper is not about and mention that a companion paper by Bruce et al. (2017) is assessing the model's error structure against 31 GLEON lakes.

REPLY: Thank you for this very useful suggestion and we have heavily modified the introduction in the revision, thought stopped short at adding another figure as our attempt at such a figure didn't seem worthy.

5. I think the authors did not carefully go through the complete manuscript. Many of the empirical equations are missing the definition of units for used variables. On other occasions variables where poorly described (see the examples listed below, as well as listed by R1 and R2). On several figures elevation and not the labeled depth is shown on y axis. The references are not formatted coherently. Like R2, I am of the opinion that many variable symbols are confusingly similar and that they should all be listed in a table. I also agree with R2 that all the subchapters of chapter 2 should have a small introduction paragraph. Further, I agree with the comments of R1 on the equations 1, 2, 3, 5, 9c, 12, 16 and with the comments of R2 on the equations 4, 7, 14, 26, 31, 52.

REPLY: Our sincere apologies that we didn't identify these flaws in the original upload. In our responses to R1 and R2 we have detailed many fixes to these issues, including a significantly revised nomenclature and summary table with all variables and units. The revision addresses most of the issues, including updates to the nomenclature and figures, and improved contextual information in the Section 2 sub-sections.

Specific comments
6. I think it is not very clear how the amalgamating, expanding, contracting or splitting and adding of layers works. For example, in p. 23 L 21 it is not obvious what the mentioned 'numerical criteria within the model' are. I would explain these in detail somewhere in the beginning of chapter 2.

REPLY: We have revised Section 2.1 to add clarity to the layer scheme, and as also noted to R2 (specific comment #40), and also better descriptions in other module sections.

7. p. 6 eq 2 and eq 3: It seems odd that the interpolation of values between levels b-1 and b are depending on b-1, b and b+1 and not only on b-1 and b.

REPLY: This error is fixed in the revision

8. p. 10 Line 5-8: $\phi_{SWS}$ is defined only in text form and not as an equation, yet it is used in equation 6. There is the danger that $\phi_{SW}$ defined in eq 10 will be confused with $\phi_{SWS}$. I suggest mentioning early on in this subchapter how you approach calculating $\phi_{SWS}$.

REPLY: This notation for energy fluxes has been significantly improved and checked in the new revision; the extended Table 1 now makes the symbol definitions and units clear, and the text is updated accordingly.

9. p. 12 eq 17: formula only for forced convection? Wind speed at what height? What are the units? I would introduce first the concepts of sensible heat (free and forced con- vection) and latent heat (evaporation and condensation) before showing the equations.

REPLY: As per point #8 above, the notation for heat transfers has been significantly updated in the revised version. This makes explicit the reference height. As per the above suggestions for an opening paragraph in

the sub-sections, we have re-worded the opening and other parts of this sub-section to better introduce free and forced convection.

10. p. 18 L10: An intro with possible conceptual options to reproduce a surface mixed layer would be good. I would like to know how the chosen approach of a bulk mixed layer depth compares to other approaches in other models (e.g. k-epsilon turbulence closure with Fickian diffusion) and what the consequences of this approach are.
REPLY: We have not delved into a critical assessment of the model approach in this sub-section. We acknowledge your interest in the approach relative to alternatives but at 24000 words already we decided a review and critical assessment was not appropriate here, and the paper should focus on description. We have published a companion paper that compares the model performance in capturing stratification (Bruce et al, 2018) and others have done 1D model inter-comparisons.

11. p. 22 L 15 and eq 44 and eq 45: I think an explanation of the concept behind this numerical scheme is necessary
REPLY: We have provided some additional explanation to this description for this diffusion algorithm in the revised version.

12. p. 24. Figure 10: This figure is not enough self-explanatory to me.
REPLY: A revised version of this figure has now been included, better depicting the interaction of the inflow "parcels" with the lake layer structure, and using the updated notation (consistent with Table 1).

13. p. 28 eq 60: Shouldn't G not just be another term in eq 4 for all cases?
REPLY:
You are correct that the change in the thickness of the bottom layer then also leads to a downward vertical shift in the elevation of all the layers above (equivalent to advection, as also discussed with R2 specific comment #8). The wording of the seepage section is now updated. As this step occurs separately to the surface dynamics routine we had not included a term for this in Eq 4, and instead included a sentence describing this effect :

" However, in addition to the terms in Eq. 4, $h_S$ is modified due to volume changes associated with river inflows, withdrawals, seepage or overflows, which are described in subsequent sections."

14a. p. 40 lines 21-24: move this sentence to the intro
REPLY: In light of the proposed changes to the introduction discussed in comment #4, we have updated accordingly.

14b. p. 40 L 24 – 26: This needs to be better explained.
REPLY: We have removed this

List of Corrections
15. p. 1 lines 31-32. Consider splitting sentence as it contains different ideas.
REPLY:  Thank you for the suggestion, we have revise the abstract.
16. p. 2. Line 1: write only 'standing' as this word is comprehensible and you don't use lentic in the rest of the text.
REPLY:  Thank you for the suggestion, we have made this change.
17. p. 5 Line 17. Write the definite instead of the indefinite integral or otherwise phrase it in a sentence.
REPLY: We have updated the integral to be between $H_0$ and $H_{max}$.
18. p.6 eq 1: could be simplified
REPLY:  Thank you for the suggestion, we have made this change.
19. p.6. Line 11-18: Should this go in the introduction?
REPLY: This section has been updated
20. p. 9 eq 9b: Contrary to R1, I managed to get the peak at 80◦SZA. The equation seems to be the same as used in fig 3.

REPLY: Fig 3 is updated and checked - problem with the bracket location!

21. p. 9 eq 9c: Specify units, also see comments of R1.
REPLY: This is updated in the revised version and also see the reply to R1.

22. p. 9. L 6: Ux is wind speed at which height?
REPLY: Within the revised vresion, all meteorological variables are now referenced to 10m, $U\_10$. This is computed from the user input data ($U\_x$) as $U\_10 = f\_UU\_x$. The updated equation in the preliminary revision (now Eq 12c) however still needs correcting from $U\_x$ to $U\_10$.

23. p.9 figure 3: Specify the values of relative humidity, wind speed and atmospheric diffusive radiation used for eq 9c. I agree with R1 that the label is wrong, but I think it should be SZA = 360° Φzen/(2π) = 180° Φzen/π
REPLY: This figure is updated in the revision, and we have extended the caption to specify the values.

24. p. 11 eq 16 a-d: Use either only °C or only K in equations, now they are mixed. I found eq 16 c in Henderson-Sellers (1986) but strangely I couldn't find this equation in Brutseart (1975).
REPLY: The notation is now updated so as to distinguish between them ($T$ and $\theta$, respectively).

25. p. 12 L 12: no units specified for latent heat of vaporization
REPLY: Table 1 in the revision has been updated to have unit descriptions of ALL variables.

26. p. 13 L 9 and L 13: I guess the authors meant eq 17 -18 and not eq 16-17?
REPLY: Yes, corrected in the revision.

27. p. 16 L 6: 'penetrating the surface', which surface?
REPLY: This is referring to the ice/snow surface, now corrected in the revision.

28. p. 20 eq 35: What is u0?
REPLY: Following our updates to the notation in this revision, it is now $u\_b\_old$; it is the layer velocity at the previous time-step. We have added to Table 1 as it has been overlooked.

29. p. 22 eq 39: Please explain the variables in this equation.
REPLY: Apologies for this oversight and confusing notation. The revision includes the description of $\tildaV$ further down (and in Table 1), however we have moved this first description up to this point. The symbol for the surface layer thickness is also now matching the earlier use. However, we noted a further error since the k in this expression should be squared and it is now updated.

30. p. 22 eq 43: check index, should be hl not hi
REPLY: This is updated in the revision (now Eq 53).

31. p. 23 L 17: typo: entrain (not entrains)
REPLY: This is updated in the revision.

32. p. 23 L 18: typo: the (not th)
REPLY: This is updated in the revision.

33. p. 34 figure 17: increase font size and size of arrows.
REPLY: We have updated the resolution and fonts in this figure in the revision.

34. p. 34 L 19: Insert the references into the place holder
REPLY: This is updated in the revision.

35. p. 35 L 26: Who is testing these 'Wrappers' and examples? What is a wrapper?
REPLY: Updated to remove wrappers.

36. p. 37 L 13: What is HTCondor?
REPLY: Updated.

37. p. 27 L 16: Start a new paragraph at 'GRAPLEr's Web service . . .' to highlight this idea.
REPLY: Agreed

38. p. 40 L 24 – 26: Explain better.
REPLY: See the reply to comment #14b above

39. References Formatting: Some parts are underlined, remove it. Change all to coherent formatting.
REPLY: The reference formatting has been refined in the revision.

40. P. 49 L 23, 25 same author, write same initials.
REPLY: Agreed

41. P. 51 L 13, is there a translation of this Japanese paper? Check the year (2014 in text, here 2015)
REPLY: Unfortunately there is no English translation of this article, however, Yajima pers comm provided an English summary of the algorithm performance and coefficients. Year has been corrected in the revision.

42. Table 1: If there is no default variable, can you give a range for snow density, compaction coefficient, and thermal conductivity of snow?
REPLY: Table 1 has been updated to specify "computed" and in the comments the relevant equation is now referred to.

43. Table 1: Latent heat of fusion: remove the trailing zero 334
REPLY: Updated in the revision.

[Figure]

[Figure]

[revised manuscript text omitted]
} \cancel{(1 + 0.275\,C)(1 - 0.261\exp[-0.000777\,T_a{}^2])} & \cancel{\text{Option 1: Idso and Jackson (1969)}} \\ \cancel{(1 + 0.17\,C^2)\,(9.365{\times}10^{-6}[T_a + 273.15]^2)} & \cancel{\text{Option 2: Swinbank (1963)}} \\ \cancel{(1 + 0.275\,C)\,0.642\left(\frac{e_a}{T_a}\right)^{\frac{1}{7}}} & \cancel{\text{Option 3: Brutseart (1975)}} \\ \cancel{\left[(1 - C^{2.796})\,1.24\left(\frac{e_a}{T_a}\right)^{\frac{1}{7}} + 0.955\,C^{2.796}\right]} & \cancel{\text{Option 4: Yajima and Yamamoto (2014)}} \end{cases} \cancel{\varepsilon_a^*} \tag{$\cancel{16a}$20a -d}$$

$$= \begin{cases} (1 + 0.275\,C_x)(1 - 0.261\exp[-0.000777\,T_a{}^2]), & \text{Option 1: Idso and Jackson (1969)} \\ (1 + 0.17\,C_x^2)\,(9.365{\times}10^{-6}(\theta_a)^2), & \text{Option 2: Swinbank (1963)} \\ (1 + 0.275\,C_x)\,0.642\,(e_a/\theta_a)^{1/7}, & \text{Option 3: Brutsaert (1975)} \\ (1 - C_x^{2.796})\,1.24\,(e_a/\theta_a)^{1/7} + 0.955\,C_x^{2.796}, & \text{Option 4: Yajima and Yamamoto (2015)} \end{cases}$$

where, $\cancel{C}C_x$ is the cloud cover fraction (0-1), $e_a$ the air vapour pressure calculated from relative humidity, and options 1-4 are chosen via the cloud_mode variable. Note that cloud cover is typically reported in octals (0-8) with each value depicting a fraction of 8  thus a value of 1 would correspond to a fraction of 0.125. Some data may also include cloud type

and their respective heights. If this is the case, good results have been reported by averaging the octal values for all cloud types to get an average  cloud cover.

If longwave radiation data does not exist and cloud data is also not available, but solar irradiance is measured, then  GLM `rad mode` setting 3 will instruct the model to compare the measured and theoretical  clear-sky solar irradiance (estimated by the BCSM; Eq. 10) to approximate the cloud cover fraction.  by assuming that $ \phi_{SW_x}/\hat{\phi}_{SW} = [C_x]$. Note that if neither shortwave or longwave radiation is provided, then the model will use the BCSM to compute incoming solar irradiance, and cloud cover will be assumed to be 0 (noting that this is likely to be an overestimate of downwelling shortwave radiation).

**2.3.3 Sensible and latent heat transfer**

The model accounts for the surface fluxes of sensible heat and latent heat using commonly adopted bulk aerodynamic formulae. For sensible heat:

$$\phi_H = -\rho_a c_a C_H  U_{10} (T_s - T_a)$$

$$(21)$$

where $c_a$ is the specific heat capacity of air  is the bulk aerodynamic coefficient for sensible heat transfer ~~(~ 1.3x10$^{-3}$), $T_a$,(°C)T_s(°C).is in (kg m$^{-3}$ and)~~ is computed from $\rho_a = 0.348\,(1 + r)/(1 + 1.61r)\,p/T_a$, where $p$ is air pressure (hPa) and $r$ is the mixing ratio, which is used to compute the gas constant.

For latent heat:

$$\phi_E = -\rho_a C_E  \lambda_v\, U_{10}\, \frac{\omega}{p}\, (e_s[T_s] - e_a[T_a])$$

$$(22)$$

where $C_E$ is the bulk aerodynamic coefficient for latent heat transfer, $e_a$ the air vapour pressure, $e_s$ the saturation vapour pressure (hPa) at the surface layer temperature (°C), $~~\kappa~isweightweight\lambda~is~~\lambda_v$ the latent heat of vaporisation. The vapour pressure can be calculated by the following formulae:

$$ \qquad \text{Option 1 : TVA (1972) - Magnus-Tetens}$$

$$(23a)$$

$$ \qquad \text{Option 2 : August-Roche-Magnus}$$

$$(23b)$$

$$ \qquad \text{Option 3 : Tabata (1973) - Linear}$$

$$(23c)$$

$$e_s[T_s] = \begin{cases} exp\left[2.3026\left(7.5\frac{T_s}{T_s+237.3}\right)+0.7858\right], & \text{Option 1 : TVA (1972) - Magnus-Tetens} \\ exp\left[6.1094\left(\frac{17.625\,T_s}{T_s+243.04}\right)\right], & \text{Option 2 : August-Roche-Magnus} \\ 10^{\left(9.28603523\frac{2322.37885\,T_s}{T_s+273.15}\right)}, & \text{Option 3 : Tabata (1973) - Linear} \end{cases}$$

and

$$e_a[T_a] = \frac{\cancel{RH}}{\cancel{100}}\cancel{e_s}f_{RH}RH_x\,e_s[T_a] \tag{$\cancel{20}$ (24)}$$

Correction for non-neutral atmospheric stability : For long-time (e.g., seasonal) integrations , the bulk-transfer coefficients for momentum, $\cancel{C_D}C_{D}$, sensible heat, $\cancel{C_H}C_{H}$ and latent heat, $\cancel{C_E}C_{E}$ can be assumed approximately constant because of the negative feedback between surface forcing and the temperature response of the water body (e.g, Strub and Powell, 1987). At finer timescales (hours to weeks), the thermal inertia of the water body is too great and so the transfer coefficients must be specified as a function of the degree of atmospheric stratification experienced in the internal boundary layer that develops over the water (Woolway et al. 2017). Monin and Obukhov (1954) parameterised the stratification in the air column using the now well-known stability parameter, $\cancel{z/L}z/L$, which is used to define corrections to the bulk aerodynamic coefficients $\cancel{C_H}C_H$ and $\cancel{C_E}C_{E}$ using the numerical scheme presented in Appendix B. The corrections may be optionally applied within a simulation, and if enabled, the transfer coefficients used above are automatically updated. This option requires  the measurement of wind speed, air temperature and relative humidity within the internal boundary layer over the lake surface  supplied at an hourly resolution.

[revised manuscript text omitted]

$$\phi_{E_0} = \rho_s \, \lambda \lambda_v \, \alpha_e \, (C_\theta - C_a)(\vartheta_s - \vartheta_a) \tag{2227a-}$$

$$\phi_{H_0} = \alpha_h (T_s - T_a) \tag{b)}$$

$$\alpha_e = 2.283 \times 10^{-3} \, \xi \, \frac{v}{c_p \rho_s} \left[ g \, \frac{|\rho_a - \rho_o|}{\rho_a \, v \, a} \right]^{1/3} 0.137 \, f_0 \, \frac{K_{air}}{c_a \rho_s} \left( g \, \frac{|\rho_a - \rho_o|}{\rho_a \, v_a \, D_a} \right)^{1/3} \tag{2328a-}$$

$$\alpha_h = 2.283 \times 10^{-3} \, \xi \, v \left[ g \, \frac{|\rho_a - \rho_o|}{\rho_a \, v \, a} \right]^{1/3} 0.137 \, f_0 \, K_{air} \left( g \, \frac{|\rho_a - \rho_o|}{\rho_a \, v_a \, D_a} \right)^{1/3} \tag{b)}$$

where $C\vartheta = \kappa \, e/p$, with the appropriate vapour pressure values, $e$, for both surface and ambient atmospheric values. Here, $v K_{air}$ is the molecular heat conductivity of air ($0.1 \, kJ$ m$^{-1}$ $h$s$^{-1}$ C$^{-1}$ K$^{-1}$), $v$), $v_a$ is the kinematic viscosity of the air ($0.0548$ m$^2$ $h$s$^{-1}$). $\rho_o$ is the density of the saturated air at the water surface temperature, $\rho_s$ is the density of the surface water, $\xi f_0$ is a dimensionless roughness correction coefficient for the lake surface $(0.5)$ and $a D_a$ is the molecular heat diffusivity of air ($0.077$

5  m$^2$ $h$s$^{-1}$). Note that the impact of low wind speeds on the drag coefficient is captured by the modified Charnock relation (Eq. A24), which includes an additional term for the smooth flow transition (see also Figure A1).

10

[Figure]

[Figure]

**Figure 5: A two-year times-series of the simulated daily heat fluxes for** five example lakes, a-e, that were depicted in Figure 2. The heat balance components summarised are depicted schematically in the inset, as described in Sect. 2.3.

GLM, users may specify the degree of sheltering or fetch limitation by optionally supplying the model with the wind direction, and a table linking direction and a wind scaling factor. Alternatively, if the direction-specific data is not available, an effective wind-sheltering coefficient has been implemented that reduces the effective surface area for heat and momentum fluxes:

$$A_E = \begin{cases} A_S \tanh\left(\dfrac{A_S}{A_C}\right) & \text{Yeates \& Imberger (2003)} & (24a) \\ \dfrac{D^2}{2}\cos^{-1}\left(\dfrac{x_F}{D}\right) - \dfrac{x_F}{D}\sqrt{D^2 - x_F^2} & \text{Markfort et al. (2009)} & (24b) \end{cases}$$

where $A_C$ is the critical area. In GLM, the ratio of the effective area to the total area of the lake $A_E/A_S$ is then used to scale $U_x$

10 as a means of capturing the average wind speed over the entire lake surface.

[Figure]

**2.4 Snow and ice dynamics**

The algorithms for GLM ice and snow dynamics are based on previous ice modelling studies (Patterson and Hamblin, 1988; Gu and Stefan, 1993; Rogers et al., 1995; Vavrus et al., 1996; Launiainen and Cheng, 1998; Magee et al., 2016). To solve the heat transfer equation, the ice model uses a quasi-steady assumption that the time scale for heat conduction through the ice is

5 short relative to the time scale of changes in meteorological forcing (Patterson and Hamblin, 1988; Rogers et al., 1995). The steady-state conduction equations,  are used with a three-layer ice model that includes blue ice (or black ice), white ice (or snow ice) and snow (see Eq. 1 and Fig. 5 of Rogers et al., 1995), and forced at the surface based on shortwave radiation which is partitioned into two components, a visible ($f_{VIS}$) and an infra-red ($1 - f_{VIS}$) spectral band. Blue ice is formed

10 through direct freezing of lake water into ice whereas white ice is generated in response to flooding, when the mass of snow that can be supported by the buoyancy of the ice cover is exceeded ( Rogers et al., 1995). By assigning appropriate boundary conditions to the interfaces and solving the quasi-steady state equation for heat transfer numerically, the model computes the upward conductive heat flux through the ice and snow cover to the atmosphere, termed $\phi_0$. The estimation of $\phi_0$ applies an empirical equation  to estimate snow conductivity , $K_{snow}$, from

15 its density (Ashton, 1986; Figure 6).

At the solid surface (ice or snow), a heat flux balance is employed to provide the condition for surface melting:

$$\phi_0 \cancel{(T_0)}[T_0] + \phi_{net} \cancel{(T_0)}[T_0] = 0 \qquad\qquad T_0 < T_m$$

$$\cancel{---} \quad \phi_{net} \cancel{(T_0)}[T_0] \cancel{= \rho L \frac{dh_i}{dt}} = -\rho_{ice,snow}\, \lambda_f \frac{d\Delta z_{ice,snow}}{dt} \qquad\qquad T_0 = T_m \qquad \cancel{(25}$$

$$(29)$$

$$(30)$$

where $\cancel{L}\lambda_f$ is the latent heat of fusion , $\cancel{h_i}\Delta z_{ice,snow}$ is the height of the upper snow or ice layer,  $\cancel{\rho}\rho_{ice,snow}$ is the density of either the snow or ice, determined from the surface medium properties, $\cancel{T_0}T_0$ is the

20 temperature at the solid surface, $\cancel{T_m}T_m$ is the melt-water temperature ($\cancel{0\,°C}0\ ^oC$) and $\cancel{\phi_{net}(T_0)}\phi_{net}[T_0]$ is the net incoming heat flux, for non-pentrative radiation at the solid surface:

$$\phi_{net} \cancel{(T_0)}[T_0] = \phi_{LWin} - \phi_{LWout} \cancel{(T_0)}[T_0] + \phi_H \cancel{(T_0)}[T_0] + \phi_E \cancel{(T_0)}[T_0] + \phi_R \cancel{(T_0)}[T_0] \qquad (\cancel{26}31)$$

[revised manuscript text omitted]

**Revision 8 Mar 2018**

[revised manuscript text omitted]

$$T_{aerosol} = e^{(-TauA)^{0.873} \left(1 + TauA - TauA^{0.7088}\right) AM^{0.9108}}$$ (A16)

We also define:

$$T_{aa} = 1 - \left[0.1 \left(1 - AM + AM^{1.06}\right)\left(1 - T_{aerosol}\right)\right]$$ (A17)

and:

$$\frac{0.5\left(1 - T_{rayleigh}\right) + 0.84(1 - T_{as})}{1 - AM + AM^{1.02}}$$ (A18)

5 where the 0.84 value used is actually the proportion of scattered radiation reflected in the same direction as incoming radiation.

The direct beam radiation on a horizontal surface at ground level on a clear day is given by,

$$\hat{\phi}_{DB} = 0.9662 \, \hat{\phi}_{ETR} \, T_{rayleigh} \, T_{ozone} \, T_{mix} \, T_{watvap} \, T_{aerosol} \, cos(\Phi_{zen})$$ (A19)

$$\hat{\phi}_{AS} = 0.79 \, \hat{\phi}_{ETR} \, T_{ozone} T_{mix} \, T_{watvap} \, T_{aa} \, cos(\Phi_{zen})$$ (A20)

The total irradiance hitting the surface is therefore (W m$^{-2}$):

$$\hat{\phi}_{SW} = \frac{\hat{\phi}_{DB} + \hat{\phi}_{AS}}{1 - (\alpha_{SW} \, \alpha_{SKY})}$$ (A21)

10 The albedo is computed for the sky as:

$$\alpha_{SKY} = 0.068 + (1 - 0.84)\left(1 - \frac{T_{aerosol}}{T_{aa}}\right)$$ (A22)

[Figure]

**Appendix B: Non-neutral bulk transfer coefficients**

The iterative procedure used in this analysis  correct the bulk-transfer coefficients based on atmospheric conditions is conceptually similar to the methodology discussed in detail in Launiainen and Vihma (1990). The first estimate for the neutral drag coefficient, $C_{DN}$, is specified as a function of wind speed as it is  commonly observed

5    to increase with $U_{10}$. This is modelled by first  estimating the value referenced to 10m height above the water from:

$$C_{DN-10} = \begin{cases} 0.001 & U_{10} \leq 5 \\ 0.001\,(1 + 0.07[U_{10} - 5]) & U_{10} > 5 \end{cases} \qquad \text{Option 1 : Francey and Garratt (1978), Hicks (1972)} \qquad \text{(A23)}$$

$$C_{DN-10} = 1.92{\times}10^{-7}U_{10}^3 + 0.00096 \qquad \text{Option 2 : Babanin and Makin (2008)}$$

and then computing the Charnock formula with the smooth flow transition (e.g., Vickers et al., 2013):

$$z_o = \frac{\cancel{\alpha u_*^2}}{\cancel{g}}\frac{a u_*^2}{g} + 0.11\frac{\cancel{\nu}\, \nu_a}{\cancel{u_*}\, u_*} \qquad \text{(A24)}$$

where $a$ is the Charnock constant (0.012), $u_*$ is the approximated friction velocity ($\sqrt{C_{DN-10}\, U_{10}^2}$) using Eq A23. The drag is re-computed using:

$$C_{DN-10} = \left[\frac{\cancel{k}}{\ln\left(\frac{10}{\cancel{z_o}}\right)}\right]^{\cancel{2}} \left[\frac{\kappa}{\ln\left(\frac{10}{z_o}\right)}\right]^2 \qquad \text{(A25)}$$

10    where $\kappa$ is the von Karman constant (Figure A1). Note the neutral humidity/temperature coefficient, $C_{HWN-10}$, is held constant at the user defined $C_H$ value and is assumed not to vary with wind speed.

[Figure]

[Figure]

Revision 8 Mar 2018

[Figure]

**Figure A1: Scaling of the 10-m neutral drag coefficient with wind speed, $U_{10}$ (Eqns A23-25)**

Under non-neutral conditions in the atmospheric boundary layer, the transfer coefficients vary due to stratification  in the

5    air column, as was parameterised by Monin and Obukhov (1954) using the now well-known stability parameter, $z/L$, where $L$

is the Obukhov length defined as:

$$L = \frac{-\rho_a u_*^3 \theta_V}{\kappa \, g \left( \frac{\phi_H}{c_a} + 0.61 \frac{\theta_a \phi_E}{\lambda_v} \right)} \tag{A26}$$

where $\theta_V = \theta_a (1 + 0.61 e_a)$ is the virtual air temperature and $\phi_H$ and $\phi_E$ are the bulk fluxes. Paulson (1970)

presented a solution for the vertical profiles of wind speed, temperature and moisture in the developing boundary layer as a

function of the Monin-Obukhov stability parameter; the so-called flux-profile relationships:

$$U_z = \frac{u_*}{\kappa} \left[ \ln \left( \frac{z}{z_o} \right) - \psi_M \left( \frac{z}{L} \right) \right] \tag{A27a-c}$$

$$\theta_a - \theta_s = \frac{\theta_*}{\kappa} \left[ \ln \left( \frac{z}{z_\theta} \right) - \psi_H \left( \frac{z}{L} \right) \right] \tag{A27b}$$

$$e_a - e_s = \frac{e_*}{\kappa} \left[ \ln \left( \frac{z}{z_e} \right) - \psi_E \left( \frac{z}{L} \right) \right] \tag{A27c}$$

10    where $\psi_M$, $\psi_H$ and $\psi_E$ are the similarity functions for momentum, heat and moisture respectively, and $z_o$, $z_\theta$ and $z_e$ are their

respective roughness lengths. For unstable conditions ($L<0$), the stability functions are defined as (Paulson 1970; Businger et

al., 1971; Dyer, 1974):

$$\psi_M = 2 \ln \left( \frac{1 + x}{2} \right) + \ln \left( \frac{1 + x^2}{2} \right) - 2 \tan^{-1} x + \frac{\pi}{2} \tag{A28a}$$

$$\psi_E = \psi_H = 2\ln\left(\frac{1+x^2}{2}\right) \tag{A28b}$$

where

$$x = \left[1 - 16\left(\frac{z}{L}\right)\right]^{1/4} \tag{A29}$$

During stable stratification (*L>0*) they take the form:

$$\psi_M = \psi_E = \psi_H = \begin{cases} -5\left(\frac{z}{L}\right) & 0 < \frac{z}{L} < 0.5 \\ 0.5\left(\frac{z}{L}\right)^{-2} - 4.25\left(\frac{z}{L}\right)^{-1} - 7\left(\frac{z}{L}\right) - 0.852 & 0.5 < \frac{z}{L} < 10 \\ \ln\left(\frac{z}{L}\right) - 0.76\left(\frac{z}{L}\right) - 12.093 & \frac{z}{L} > 10 \end{cases} \tag{A30}$$

5    Substituting Eqns. 20-21 into (A27) and ignoring the similarity functions leaves us with neutral transfer
coefficients as a function of the roughness lengths:

$$C_{XN} = \cancel{k^2}\kappa^2 \left[\ln\left(\frac{z}{z_o}\right)\right]^{-1}\left[\ln\left(\frac{z}{z_X}\right)\right]^{-1} \tag{A31}$$

where the N sub-script denotes the neutral value and *X* signifies either *D*, *H* or *E* for the transfer coefficient and *o*, *θ* or e for
the roughness length scale. Inclusion of the stability functions into the substitution and some manipulation (Imberger and
Patterson, 1990; Launianen and Vihma, 1990) yields the transfer coefficients relative to these neutral values:

$$\frac{C_X}{C_{XN}} = \left[1 + \frac{\cancel{C_{XN}}}{\cancel{k^2}}\left(\cancel{\psi_M\psi_X} - \frac{\cancel{k\psi_X}}{\sqrt{\cancel{C_{DN}}}} - \frac{\cancel{k\psi_M\sqrt{C_{DN}}}}{\cancel{C_{XN}}}\right)\right]\left[1 + \frac{C_{XN}}{\kappa^2}\left(\psi_M\psi_X - \frac{\kappa\psi_X}{\sqrt{C_{DN}}} - \frac{\kappa\psi_M\sqrt{C_{DN}}}{C_{XN}}\right)\right] \tag{A32}$$

Hicks (1975) and Launianen and Vihma (1990) suggested an iterative procedure to solve for the stability corrected transfer
coefficient using (A32) based on some initial estimate of the neutral values (as input by the user). The surface flux is
subsequently estimated according to Eqns. 20-21 and used to provide an initial estimate for *L* (Eq. A26). The
15   partially corrected transfer coefficient is then recalculated and so the cycle goes. Strub and Powell (1987) and Launiainen
(1995), presented an alternative based on estimation of the bulk Richardson number, *Ri_B*, defined as:

$$Ri_B = \frac{gz}{\theta_V}\left(\frac{\cancel{\Delta\theta + 0.61\,\theta_V\Delta q}}{\cancel{U_z^2}}\right)\left(\frac{\Delta\theta + 0.61\,\theta_V\Delta e}{U_z^2}\right) \tag{A33}$$

and related as a function of the stability parameter, *z/L*, according to:

$$\cancel{Ri_B = \frac{z}{L}\left(\frac{k\sqrt{C_{DN}}/C_{HWN} - \psi_{HW}}{\left[k/\sqrt{C_{DN}} - \psi_M\right]^2}\right)}Ri_B = \frac{z}{L}\left(\frac{\kappa\sqrt{C_{DN}}/C_{HWN} - \psi_{HW}}{\left[\kappa/\sqrt{C_{DN}} - \psi_M\right]^2}\right) \tag{A34}$$

where it is specified that $C_{HN} = C_{WN} = C_{HWN}$. Figure A2 illustrates the relationship between the degree of atmospheric
stratification (as described by both the bulk Richardson number and the Monin-Obukhov stability parameter) and the transfer
20   coefficients scaled by their neutral value.

[Figure]

[Figure]

[Figure]

**Figure A2: Relationship between atmospheric stability (bottom axis – *z/L*, top axis – *Ri_B*) and the bulk-transfer coefficients relative to their neutral value (*C_X/C_XN* where *X* represents *D*, *H* or *W*) for several roughness values (computed from Eq. A32). The solid line indicates the momentum coefficient variation (*C_D/C_DN*) and the broken line indicates humidity and temperature coefficient (*C_HW/C_HWN*) variation.**

---

## Referee Report (RR1)

**Review of "A General Lake Model (GLM 3.0) for linking with high-frequency sensor data from the Global Lake Ecological Observatory Network (GLEON)" by Hipsey et al.**

**Overview**

This paper describes a one-dimensional hydrodynamic lake model GLM, which has already been used for a number of applications in the scientific literature. The paper is generally well written and structured. It gives a comprehensive overview of the model structure and equations, and shortly describes available tools for pre- and post-processing and for linking the model to other models.

I have rejected the previous version of this manuscript because there were far too many errors already in the first few equations. This has certainly been improved in the current version. However, I feel there are still too many errors and inconsistencies. These are problematic for two reasons. First, given the large number of errors reported in the first and this second review, I have to assume that there are still quite a few remaining errors that were not spotted by any of the reviewers. This reduces the trust both in the model description and in the model itself, as the reader never knows whether the equations are only wrongly written or also wrongly implemented. Second, the inconsistencies in the notation make it sometimes hard for the model user to understand the details. I am aware that writing a flawless model description is a very tedious work, but this can't be avoided.

**General comments to model description**

The total lake water balance is missing in the model description. It is shortly mentioned in the caption of Figure 2, but it should be described in detail in a separate section.

Section 2.3.3: I don't understand why three different parameterizations for saturation vapor pressure are included, which should give the same results within 1-2% at most, but only one parameterization for the latent heat flux (eq. 22), where different parameterizations can yield very different evaporation rates (see, e.g., Rosenberry et al., 2007, doi:10.1016/j.jhydrol.2007.03.018).

Section 2.6.1: It is somewhat confusing that first a total energy balance is introduced as being relevant for mixing, but subsequently only parts of that energy balance are used in each step. I think it would be clearer for the reader to first describe the individual energy components, and then the different steps in the mixing calculation, and remove the total energy balance, which is not explicitly used in the model.

Section 2.6.2: It is not clear to which part of the water column this mixing regime applies (the wording includes "below the epilimnion", "below the thermocline" and "in the hypolimnion". I assume it applies to all layers that are not within the surface mixed layer, according to Eq. (53), but this should also be made clear in the text.

Section 2.8. The wave properties are calculated based on the average lake depth, but the wave velocity in the $i^{th}$ layer is then calculated from the local depth of this layer. I don't really know, but wouldn't it be more consistent to estimate also the wave properties from the local depth of the layer? It seems to me you could potentially underestimate local wave heights and thus resuspension in shallow waters with the given approach.

Code availability: I was not able to clearly identify the source code for GLM version 2.4. or 3.0 (probably not yet available?) or previously published versions on the GitHub repository. Also, the information on the GLM website

(http://aed.see.uwa.edu.au/research/models/GLM/Pages/documentation.html) is not updated and does not link to the GitHub repository. All release versions, also past versions to allow reproducing previous calculations, should be published on GitHub and linked from the GLM website.

**Comments to Figures**

Figure 2: Fonts are rather small. The water balance should be described in the text rather than in the figure caption. I also think the notation used here is not entirely consistent with the model description.

Figure 3: increase fonts, albedo_mode is not introduced in the text.

Figure 4: what is the grey area in Fig 4a? The area where $f_{BEN} < 0.2$? if so, the color scale should be adapted accordingly. Also, the grey area is at least 80% of $A_S$, but in Fig 4c, the fraction is always >30%. Is this for the same lake?

Figure 5: How is it possible that the net heat balance is always positive for Ellen Brook Nature Reserve? This should lead to a massive heating of the lake throughout the year. Also net LW radiation is not shown in this figure. Is it missing from the budget? Please check. In the legend, $\phi_{LW}$ should be replaced by $\phi_{LW\,Net}$

Figure 6: use larger font for conditions, these are hardly readably even zoomed to > 100%. Shouldn't $S_F$ be multiplied with $\Delta t$ in the top boxes (and $R_F$ in box 4 from the top)? Are the equations in boxes 6 and 7 from the top correct? $R_F$ has a unit, so it should not be in the exponent. And typical values of $R_F$ (unit m/s) are on the order of $10^{-6}$, so $\exp(-R_F)$ is virtually nothing. In the third box from the bottom, the first equation must be wrong (check units). It is not clear what happens in the lowermost case (transition from snow to white ice?).

Figure 9: What drives vertical transport in case a) where vertical diffusivity in the hypolimnion should be zero? And what causes the local minima in the vertical profiles in cases a) and c)? If a tracer is released in the bottom cell and moves upward by diffusion, it's concentration should always be monotonically decreasing upward. If other processes (such as river intrusions) are responsible for these minima, why don't they exist in case b)?

Figure 10: $\alpha_{inf}$ is not very clearly drawn, also what exactly is $\delta z_{infj}$? In general, it would be useful if the discretisation in Eqs. (60) ff could be more clearly linked to the Figure, using exactly the same notation ($\Delta z_j$ and $\delta x_{inf}$ in the figure, but $\Delta z_{jinf}$ and $\Delta x_{inf}$ in the equations).

Figure 12c: why is there bottom water withdrawal around day 20?

Figure 14: increase fonts, the subscripts are impossible to read at 100% size. $H_S$, L and T are not used in the figure or text, remove from caption. Is the lower limit of the height axes at 1 m rather than 0 m on purpose?

Figure 15: This is a useful overview of the code structure. However, I think two things are nowhere explained in the text: what is the difference between do_model and do_model_nonavg? And what is do_bubbler?

Figure 16: Fonts are very small at 100% size.

Figure 17: "exposed sediment" is impossible to read at 100% size. Figure 17b can be removed.

Figure 18: As I understand, the grey shaded areas show the 80% uncertainty range of model projections based on the posterior uncertainty of the model parameters. However, it seems strange to me that this projection can deviate so much from the observed hypolimnion temperatures. These observations have a very small uncertainty (maybe 0.1 °C?), and such large deviations should lead to very low values for the likelihood in the Bayesian analysis.

**Detailed comments to equations**

Eq. (1): replace $A_{b-1} + 0.5(A_b-A_{b-1})$ with $0.5(A_b + A_{b-1})$.

Eqs. (2) and (3) are now correct, but might be easier to read if $V_b$ was defined as the level below $H_{mi}$ rather than $V_{b-1}$.

Eq. (4): According to Eq. (8), $\phi_E$ is positive if water is evaporated, thus E is negative (line 26) and should be added rather than subtracted in Eq (4), or maybe better, the minus sign should be removed in line 26?

Explain in a few words the intention of Eqs. (5) and (6), where, depending on meteorological conditions, precipitation will either be added to the water volume, or to the snow cover, referring to section 2.4. Also, I think the notation is inconsistent between here and section 2.4. Here, $S_F$, and $R_F$ are the fractions of rain and snow that are added to the fluid water volume. In section 2.4, they are used also for rain and snow that are added to the snow cover.

Eq. (8). In order to optimize the calculation speed, consider replacing the calculated density of the surface layer by an average water density. This is an option for all equations where a calculated density is used as a multiplying (or dividing) factor. Density variations in lakes are in almost all cases < 1%, which is negligible compared to other uncertainties in model parameters. These are generally only relevant if density differences are calculated.

Eq. (9): Why is $f_{sw}$ not included in the third option? I think it would make sense to have the option available to scale SW radiation also in this case.

Eq. (10) replace C with $C_x$

Eq. (11) I did not see the base for this equation in Luo et al (2010).

Eq. (12a) $\sin(x-\pi/2)$ and $\sin(x + \pi/2)$ could be simplified to $-\cos(x)$ and $\cos(x)$, respectively.

Eq. (12c) I think $RH_x$ should be multiplied rather than divided by 100.

I have the impression that Eqs (20c) and (20d) should be valid for different units of $e_a$ as they are written here. Please check.

Eq. (23) Something must be wrong with all three equations. They all yield results that are far away from the correct saturation vapor pressure. Saturation vapor pressure is 6.1 hPa for 0 °C. But for $T_S$ = 0 °C, Eq. (23a) results in exp(0.7858) = 2.2, Eq. (23b) in exp (0) = 1 and Eq. (23c) in 10^0 = 1. Also the exponent in c is far too high for any temperature other than 0°C.

Eq. (25c) The nominator should be 2, rather than $L_D$ in front of the square root according to Markfort et al. (also because otherwise the units are not correct). Furthermore, $\cos^{-1}(x)$ is ambiguous. It is sometimes used for arccos (x), and sometimes for 1/cos(x).

Eq. (32) Shouldn't it be $\phi_{SW0}$ also in the second line of the equation?

Eq. (39): Should it be $\rho_{SML}$ instead of $\rho_i$ in the last term? If not, the term in parentheses could be simplified to sum of $\Delta z_i \rho_i (h_l-h_{sml})$

Eq. (41) g needs to be added to the two equations for reduced gravity.

Eq. (42) I think it should be $\Delta t$ rather than $t$ in the upper equation?

Eq. (49): I think this equation should have a minus sign. $N^2$ is usually defined to be positive if the stratification is stable, i.e. if the density of the upper layer is smaller than the density of the lower layer. Here it is the opposite.

Eq. (52): I don't understand the last term of this equation. $h_s - h_{iinsl}$ seems to me to be almost the same as $z_{infinsl}$ (depending on how exactly the latter is defined) so the term in parentheses should be almost zero?

Eq. (53): the meaning of $\sigma$ is unclear. The unit of $N^2$ is $s^{-2}$, that of its variance would be $s^{-4}$, but this obviously can't be what is meant here.

Eq. (54): should be $f_{dif}$ in the exponents.

Eqs (56) ff. Maybe I looked at the wrong place, but I was not able to find all the corresponding equations in Fischer et al. (1979) and Antenucci et al. (2005). Since Fischer et al. is rather voluminous, it might be useful to mention the respective equation number from the original publication. I did not see Eqs. (56) and (57) anywhere in Fischer, and the corresponding Eq. (2) looks rather different in Antenucci. Eq. (58) corresponds to eq (4) in Antenucci, but $\tan \alpha$ is replaced by $\tan \phi_{inf}$, and the tan is in the denominator here and in the numerator in Antenucci (not sure whether that is on purpose there, though).

Eq. (61) and following line: I don't completely this. The second equation in line 9 can be simplified, using eq. 61, to $z_{infj} = z_{infj-1} + \delta z_{infj-1}$. But according to the first equation in line 9 after reducing the index j by 1, also $z_{infj-1} + \delta z_{infj-1} = h_s-h_{ij-2}$. So that means $z_{infj} = h_s-h_{ij-2}$?

Eq. (62) For a given $\Delta z_{infj}$, and a given discharge, the flow velocity should be smaller if the channel is wider, i.e. if the angle $\alpha_{inf}$ is larger. But this equation implies the opposite. Please check.

Eq. (71): the equation calculating the seepage as a function of lake head is somewhat inconsistent as it applies the entire lake head to the entire area, and all water is removed from the bottom layer, i.e. it in fact treats the lake as a rectangular box.

**Details in the text (pxx/lyy means page xx line yy)**

p5/l15: $\Delta H_{mi}$ is 0.1 m here and 0.01 m in Table 1

p6/l4: Section 2.6 only describes layer merging by mixing from above, i.e. in the surface mixed layer. Can layer merging also occur in deeper waters, and if yes, under what conditions?

p8/l16: remove "either"?

p 9/10: $\varsigma$ is not atmospheric diffusive radiation, but "a constant related to atmospheric diffuse radiation" according to Table 1. It remains unclear, what this means.

p11/l8: $\phi_{SWS}$ is defined here as the fraction heating up the surface layer but in Table 1 as the radiation flux crossing the water surface (should be $\phi_{SW0}$ in the table?).

p11/l23: $A_{Ben}/A_s$ is a fraction, not a percentage.

p11/l15: unit for vapor pressure needs to be given here or in Table 1.

p15/l20ff: I am not sure I understand the procedure for calculating ice melting here. I understand that first $\phi_{SW0}$ is calculated from eq. 31, then $T_0$ is calculated such that $\phi_0 = \phi_{net}$ (how is this done?), , and if $T_0$ is then equal or larger than the melting temperature $T_m$, melting is determined from eq 30. Is that correct? Consider revising the text in this section to clarify.

p18/l5: replace possible with available?

p19/l9: the notation $z_{msl}$ is inconsistent, as all other layer thicknesses are named $\Delta z$.

p20/l2: difference to what?

p21/l16-18: this is not clear.

p23/l6: if $\alpha_{TKE}$ is interpreted as diffusivity, why then $Kz = C_{HYP}$ and not $Kz = \alpha_{TKE}$?

p23/l10: "contains 85% of N2" sounds strange, as $N^2$ is not a property that can be reasonably summed up across layers. Table 1 defines the same variable as the "fraction that contains 85% of the $N^2$ *variance*". This is even less clear. Is it calculated by summing up all $N^2$ values and then taking the volume for which the sum is 85% of the sum for the entire lake? If so, this is virtually the same as the volume that contains 85% of the density difference between the top and the bottom of the lake.

p23/l15: which density difference?

p25/l9: delete "the tangent of"

p26/l9: really daily time step? The time scale for river intrusions is usually rather minutes to hours than days. So does this ever take more than one step?

p26/l14: should be $\Delta z_{infj}$ instead of $z_{infj}$ in the equation.

p27/l12: should refer to Eq. (52)?

p31/l5: verb is missing.

p31/l8: If no weir is present I assume $Q_{Ovfl}$ is the same as Eq. (73), but with $Q_{weir} = 0$?

p31/l9: Is $\Delta h_s$ the result of eq. (4)?

p35/l5: iti -> it

p37/l2: I can't remember having read anything about solar shading in the model description.

p37/l16: In Figure 16, glmtools is called GLMr.

References: I did not check the list, but noticed that Spigel and Imberger (1980) is missing.

**Comments to Table 1**

This table is very useful, but it needs to be thoroughly checked and corrected. I checked only a small part of the variables, the following list of inconsistencies and missing variables is therefore certainly incomplete:

- I think the following variables used in the paper are missing in Table 1 (incomplete list): $H_0$, $N_{SW}$, $e_a$, $\vartheta_S$, $\vartheta_a$, $\delta_{zsoil}$, $K_{soil}$, $c_{wn}$

- The variables $\alpha_b$ and $\beta_b$ should go to Lake domain.
- Use per mil rather than ppt for parts per thousand, as the latter is generally used for parts per trillion.
- Check definition of $\theta_S$
- Use Kelvin without degree sign.
- $A_c$ is $A_{ws}$ in Eq. 25b?
- most of the h's should be height above datum rather than height above bottom.
- $f_w$ is calculated in Eq. (83), not Eq. (78).

Furthermore, the variables are mostly ordered alphabetically, but not completely, and the assignment of variables to the different classes of variables can be ambiguous. It is therefore often rather difficult to find a variable in the Table.

Finally, is unclear which parameter values are hard-coded and which can be modified by the user. Some parameters are defined as configurable, but for some others, which can also be defined by the user according to the text, just one value is given.

---

## Author Response (AR2)

*25th June 2018*

Dear GMD Editor & Reviewers,

**Re. Revision #2 of gmd-2017-257:  A General Lake Model (GLM)**

We thank you again for the opportunity for our paper to be considered for publication in GMD, and are very grateful for the additional comments received on the first revision of the paper. We have found the comments very insightful and they have guided us to significantly improve this version of the manuscript.

The revised paper includes the suggested corrections to the equations figures and text, and updated detail in some of the figures and sub-sections as suggested by the reviewers. The detailed replies are provided below.

Based on this revision, we are now confident to release the code-base associated with this paper and results presented herein. The code is therefore stamped on GitHub as v3.0. A DOI for the code has been added to the paper, and a separate DOI for the example files presented herein has also been generated.

We thank you again for the significant time and effort that have gone into the discussion and look forward to your decision.

Kind Regards

Matthew Hipsey, on behalf of all co-authors.

**Overview**

This paper describes a one-dimensional hydrodynamic lake model GLM, which has already been used for a number of applications in the scientific literature. The paper is generally well written and structured. It gives a comprehensive overview of the model structure and equations, and shortly describes available tools for pre- and post-processing and for linking the model to other models.

I have rejected the previous version of this manuscript because there were far too many errors already in the first few equations. This has certainly been improved in the current version. However, I feel there are still too many errors and inconsistencies. These are problematic for two reasons. First, given the large number of errors reported in the first and this second review, I have to assume that there are still quite a few remaining errors that were not spotted by any of the reviewers. This reduces the trust both in the model description and in the model itself, as the reader never knows whether the equations are only wrongly written or also wrongly implemented. Second, the inconsistencies in the notation make it sometimes hard for the model user to understand the details. I am aware that writing a flawless model description is a very tedious work, but this can't be avoided

REPLY: We very much appreciate the substantial time invested by the reviewer in combing through the details of this paper, and apologise we have rushed previous submissions without an adequate level of checking. The review process has not only led to improved manuscript but in going through the review the model code has advanced in several areas. In particular, the consistency of the notation has matured over the course of the reviews, and we feel the current revision has now stabilised in terms of notation and has improved clarity. We would be happy to acknowledge the reviewer specifically in the paper given the significant improvements they have made, should they wish to forego anonymity.

**General comments to model description**

The total lake water balance is missing in the model description. It is shortly mentioned in the caption of Figure 2, but it should be described in detail in a separate section.

REPLY: Agreed, we have updated Section 2.2 to include this.

Section 2.3.3: I don't understand why three different parameterizations for saturation vapor pressure are included, which should give the same results within 1-2% at most, but only one parameterization for the latent heat flux (eq. 22), where different parameterizations can yield very different evaporation rates (see, e.g., Rosenberry et al., 2007, doi:10.1016/j.jhydrol.2007.03.018).

REPLY: We understand clearly the point. The reason is due to the organic nature of the model development where different users and developers have sought to use specific algorithms for particular applications. It is our hope that over time we can enrich the options for other algorithms (such as evaporation, as suggested) to ensure the model's wide applicability to diverse lake types.

Section 2.6.1: It is somewhat confusing that first a total energy balance is introduced as being relevant for mixing, but subsequently only parts of that energy balance are used in each step. I think it would be clearer for the reader to first describe the individual energy components, and then the different steps in the mixing calculation, and remove the total energy balance, which is not explicitly used in the model.

REPLY: We considered this suggestion and would prefer to provide the overall energy requirement and amount available. We note that most aspects of the model are also similar, for example the water balance or surface heat balance, in that they have a sequential solution within the model. In our response to Rev 2 we aimed to improve the description in this section and feel the overall summary of the energy terms in

this form helps the reader see the overall picture before the specific steps are explained.

Section 2.6.2: It is not clear to which part of the water column this mixing regime applies (the wording includes "below the epilimnion", "below the thermocline" and "in the hypolimnion". I assume it applies to all layers that are not within the surface mixed layer, according to Eq. (53), but this should also be made clear in the text.

REPLY: It is right that the mixing algorithm is applied to all layers that are not in the surface mixed layer. We modified the text throughout this section to improve clarity and consistency.

Section 2.8. The wave properties are calculated based on the average lake depth, but the wave velocity in the ith layer is then calculated from the local depth of this layer. I don't really know, but wouldn't it be more consistent to estimate also the wave properties from the local depth of the layer? It seems to me you could potentially underestimate local wave heights and thus resuspension in shallow waters with the given approach.

REPLY: The horizontally averaged nature of the model and this parameterisation means that we are only able to compute a single set of wave characteristics across the entire lake surface (ie. wave length and wave height are assumed constant) - and for this computation the depth to the lake bottom is required, for which we assume an average lake depth to get the mean wave conditions. Once computed, we then translate the orbital velocity based on these wave characteristics down the water column to get as a function of depth from the water surface, and this is where the layer specific depths are used. This is now clarified in the opening text of the section.

Code availability: I was not able to clearly identify the source code for GLM version 2.4. or 3.0 (probably not yet available?) or previously published versions on the GitHub repository. Also, the information on the GLM website (http://aed.see.uwa.edu.au/research/models/GLM/Pages/documentation.html) is not updated and does not link to the GitHub repository. All release versions, also past versions to allow reproducing previous calculations, should be published on GitHub and linked from the GLM website.

REPLY: It appears you accessed an archived version of the GLM web page dating back some time. The main download page that has version 2.4 and 3.0 bundles is here, including links to source code in GitHub:

http://aed.see.uwa.edu.au/research/models/GLM/latest_release.html

**Comments to Figures**

Figure 2: Fonts are rather small. The water balance should be described in the text rather than in the figure caption. I also think the notation used here is not entirely consistent with the model description.

REPLY: Figure updated, improved fonts and updated caption. New opening to Section 2.2 now provides overview of lake scale water balance.

Figure 3: increase fonts, albedo_mode is not introduced in the text.

REPLY: albedo_mode is removed from the caption text to be consistent with other sections (input configuration file details will be reported under the user instructions on the model website rather than in this paper). Fonts will be updated at proofing

Figure 4: what is the grey area in Fig 4a? The area where fBEN < 0.2? if so, the color scale should be adapted accordingly. Also, the grey area is at least 80% of AS, but in Fig 4c, the fraction is always >30%. Is this for the same lake?

REPLY: Figure updated.

Figure 5: How is it possible that the net heat balance is always positive for Ellen Brook Nature Reserve? This should lead to a massive heating of the lake throughout the year. Also net LW radiation is not shown in this figure. Is it missing from the budget? Please check. In the legend, ϕLW should be replaced by ϕLW Net

REPLY: Apologies, there was a problem with that plot for that version of the model that has been rectified.

Figure 6: use larger font for conditions, these are hardly readably even zoomed to > 100%. Shouldn't SF be multiplied with Δt in the top boxes (and RF in box 4 from the top)? Are the equations in boxes 6 and 7 from the top correct? RF has a unit, so it should not be in the exponent. And typical values of RF (unit m/s) are on the order of 10-6, so exp(-RF) is virtually nothing. In the third box from the bottom, the first equation must be wrong (check units). It is not clear what happens in the lowermost case (transition from snow to white ice?).

REPLY: Thanks for highlighting the problems with the figure; the figure has been re-drafted. The readability is improved plus we now distinguish more clearly between the rainfall and snowfall components that land on the water vs on the ice/snow, and have corrected the other issues following review of the corresponding code.

Figure 9: What drives vertical transport in case a) where vertical diffusivity in the hypolimnion should be zero? And what causes the local minima in the vertical profiles in cases a) and c)? If a tracer is released in the bottom cell and moves upward by diffusion, it's concentration should always be monotonically decreasing upward. If other processes (such as river intrusions) are responsible for these minima, why don't they exist in case b)?

REPLY: To clarify, the passive tracer diffusing out of the sediment is not just entering into the bottom layer, but also into all layers simultaneously. For the bottom layer the area of the sediment in contact with the water layer is 100%, but for the remaining layers the area of sediment in contact with the water depends on the layer height range relative to the hypsographic shape. For layers near the surface the sediment area relative to layer area is very small, so the input is less noticeable. It may be the case that layers in the hypolimnion (or elsewhere) have a different layer area to sediment area ratio, and this leads to a relatively different areal loading rate. As it is a 1D model, the input is instantly homogenised across the layer leading to the vertical variation in concentration noticeable in the layers of the hypolminion (Figure 9a). For the case of constant diffusivity, the diffusion rate is high and it becomes smoothed and monotonic (Figure 9b), but for the depth dependent diffusivity model the diffusion rate is less with distance from the thermocline and so the smoothing is more modest (Figure 9c). Figure caption updated to avoid confusion.

Figure 10: αinf is not very clearly drawn, also what exactly is δzinfj? In general, it would be useful if the discretisation in Eqs. (60) ff could be more clearly linked to the Figure, using exactly the same notation (Δzj and δxinf in the figure, but Δzjinf and Δxinf in the equations).

REPLY: Figure updated, including consistent symbol notation, and improved depiction of the vertical excursion length scale.

Figure 12c: why is there bottom water withdrawal around day 20?

REPLY: Thanks for pointing this out. In the test case with the activated bottom outlet, the bottom withdrawal was only opened during flooding conditions. We have added this information to the Figure caption

Figure 14: increase fonts, the subscripts are impossible to read at 100% size. HS, L and T are not used in the figure or text, remove from caption. Is the lower limit of the height axes at 1 m rather than 0 m on purpose?

REPLY: Figure updated.

Figure 15: This is a useful overview of the code structure. However, I think two things are nowhere explained in the text: what is the difference between do_model and do_model_nonavg? And what is do_bubbler?

REPLY: Figure updated, and in addition the (now three) entry points to the model are explained.

Figure 16: Fonts are very small at 100% size.

REPLY: Updated

Figure 17: "exposed sediment" is impossible to read at 100% size. Figure 17b can be removed.

REPLY: Updated, without the (b) panel, and updated detail.

Figure 18: As I understand, the grey shaded areas show the 80% uncertainty range of model projections based on the posterior uncertainty of the model parameters. However, it seems strange to me that this projection can deviate so much from the observed hypolimnion temperatures. These observations have a very small uncertainty (maybe 0.1 °C?), and such large deviations should lead to very low values for the likelihood in the Bayesian analysis.

REPLY: The figure does have three shades, and it is the winder band (80th) that is deviating notably in the hypolimnion, as you have noted. The optimisation for this simulation used surface and bottom and thermocline depth to optimise the objective function. In this case the model has some sensitive parameters related to mixing that lead to full mixing vs stratification. This parameter combinations were less likely, but did contribute to a wide tail in the posterior probability. It is possible the algorithm could have been conditioned better, and this was intended to highlight the potential for this approach with GLM.

**Detailed comments to equations**

Eq. (1): replace Ab-1 + 0.5(Ab-Ab-1) with 0.5(Ab + Ab-1).

REPLY: Updated

Eqs. (2) and (3) are now correct, but might be easier to read if Vb was defined as the level below Hmi rather than Vb-1.

REPLY: Yes indeed! Updated.

Eq. (4): According to Eq. (8), $\phi E$ is positive if water is evaporated, thus E is negative (line 26) and should be added rather than subtracted in Eq (4), or maybe better, the minus sign should be removed in line 26?

REPLY: Agreed, updated

Explain in a few words the intention of Eqs. (5) and (6), where, depending on meteorological conditions, precipitation will either be added to the water volume, or to the snow cover, referring to section 2.4. Also, I think the notation is inconsistent between here and section 2.4. Here, SF, and RF are the fractions of rain and snow that are added to the fluid water volume. In section 2.4, they are used also for rain and snow that are added to the snow cover.

REPLY: Thank you for this suggestion, and noting the slightly different intention of these variables between section 2.2 and 2.4. We have updated accordingly (including updating Figure 6).

Eq. (8). In order to optimize the calculation speed, consider replacing the calculated density of the surface layer by an average water density. This is an option for all equations where a calculated density is used as a

multiplying (or dividing) factor. Density variations in lakes are in almost all cases < 1%, which is negligible compared to other uncertainties in model parameters. These are generally only relevant if density differences are calculated.

REPLY: Thank you for this suggestion, we will review the code with this in mind. For now, we compute the density for all layers anyway for mixing, and utilise that for this equation.

Eq. (9): Why is fsw not included in the third option? I think it would make sense to have the option available to scale SW radiation also in this case.

REPLY: Agreed - we added this to the code and updated Eq.

Eq. (10) replace C with Cx

REPLY: Updated

Eq. (11) I did not see the base for this equation in Luo et al (2010).

REPLY: The equation came from our own data; Luo was cited as an alternative source for a similar function. Wording slightly modified.

Eq. (12a) $\sin(x-\pi/2)$ and $\sin(x + \pi/2)$ could be simplified to $-\cos(x)$ and $\cos(x)$, respectively.

REPLY: we left as is

Eq. (12c) I think RHx should be multiplied rather than divided by 100.

REPLY: We have settled on RH as %, therefore, this division by 100 is removed.

I have the impression that Eqs (20c) and (20d) should be valid for different units of ea as they are written here. Please check.

REPLY: To the best of our knowledge they are in hPa, consistent with our units for ea.

Eq. (23) Something must be wrong with all three equations. They all yield results that are far away from the correct saturation vapor pressure. Saturation vapor pressure is 6.1 hPa for 0 °C. But for TS = 0 °C, Eq. (23a) results in $\exp(0.7858) = 2.2$, Eq. (23b) in $\exp(0) = 1$ and Eq. (23c) in $10^0 = 1$. Also the exponent in c is far too high for any temperature other than 0°C.

REPLY: Apologies for this poor checking. The Tabata (1973) is the main one being used within the code and this was implemented properly, but the equation as written in the paper with a spurious Ts; we have now removed the other two options (which weren't being used) and updated accordingly.

Eq. (25c) The nominator should be 2, rather than LD in front of the square root according to Markfort et al. (also because otherwise the units are not correct). Furthermore, cos-1 (x) is ambiguous. It is sometimes used for arccos (x), and sometimes for 1/cos(x).

REPLY: Updated

Eq. (32) Shouldn't it be φSW0 also in the second line of the equation?

REPLY: Yes, updated.

Eq. (39): Should it be ρSML instead of ρi in the last term? If not, the term in parentheses could be simplified to sum of $\Delta z_i \rho_i (h_l-h_{sml})$

REPLY: Simplified as suggested

Eq. (41) g needs to be added to the two equations for reduced gravity.

REPLY: Updated

Eq. (42) I think it should be $\Delta t$ rather than t in the upper equation?

REPLY: Checked and updated

Eq. (49): I think this equation should have a minus sign. N2 is usually defined to be positive if the stratification is stable, i.e. if the density of the upper layer is smaller than the density of the lower layer. Here it is the opposite.

REPLY: Updated - thank you for spotting this. Our function was correct in the code, and took in rho1 (upper) and rho2 (lower), and then computed rho2-rho1, which was reported backwards in the paper.

Eq. (52): I don't understand the last term of this equation. hs – hiinsl seems to me to be almost the same as zinfinsl (depending on how exactly the latter is defined) so the term in parentheses should be almost zero?

REPLY: This term is the depth over which the inflow plunges in one day (rather than the amount from the beginning of its transit, which you may be expecting?)

Eq. (53): the meaning of σ is unclear. The unit of N2 is s-2, that of its variance would be s-4, but this obviously can't be what is meant here.

REPLY: Apologies for the confusion here again, this was checked during the revision for unit consistency but somehow the notation update didn't occur. The intention for σ was the length scale of the water column corresponding to the N2 distribution variance. The variance of the buoyancy distribution is computed by first finding the 1st moment of the N2 distribution (ie. centre of buoyancy in metres above the bottom). This symbol should have been changed to $dz_\sigma^2$. In this case the units are $m^2$.

Eq. (54): should be fdif in the exponents.

REPLY: Updated

Eqs (56) ff. Maybe I looked at the wrong place, but I was not able to find all the corresponding equations in Fischer et al. (1979) and Antenucci et al. (2005). Since Fischer et al. is rather voluminous, it might be useful to mention the respective equation number from the original publication. I did not see Eqs. (56) and (57) anywhere in Fischer, and the corresponding Eq. (2) looks rather different in Antenucci. Eq. (58) corresponds to eq (4) in Antenucci, but tan α is replaced by tan ϕinf, and the tan is in the denominator here and in the numerator in Antenucci (not sure whether that is on purpose there, though).

REPLY: Thank you again for your keen eye here. We have cleaned up this section with careful cross checking that the angles are reported right, and a review of the original references on which these were based. Since the Fischer book is harder to access we make a general reference to it, and now rely on the Imberger and Patterson eq (which uses a slightly different form), the Antenucci paper, and we now also cite a nice analysis using a similar approach by Ayala et al 2014.

Eq. (61) and following line: I don't completely this. The second equation in line 9 can be simplified, using eq. 61, to zinfj = zinfj-1 + δzinfj-1. But according to the first equation in line 9 after reducing the index j by 1, also zinfj-1 + δzinfj-1 = hs-hij-2.  So that means zinfj = hs-hij-2?

REPLY: This section has been reviewed and updated to be more clear, and these sub-scripts updated to prevent confusion here.

Eq. (62) For a given Δzinfj, and a given discharge, the flow velocity should be smaller if the channel is wider, i.e. if the angle αinf is larger. But this equation implies the opposite. Please check.

REPLY: Thank you for noticing this mistake; The velocity is Q divided by channel area, which is computed based on the water depth (Δz). In the code it is Q * cos($\alpha_{inf}$) / (ΔzΔz sin($\alpha_{inf}$)), which we mistakenly wrote in Eq 62 with tan($\alpha_{inf}$) as a numerator.

Eq. (71): the equation calculating the seepage as a function of lake head is somewhat inconsistent as it applies the entire lake head to the entire area, and all water is removed from the bottom layer, i.e. it in fact treats the lake as a rectangular box.

REPLY: Yes, this is the case. Mostly this option is used in shallow systems, but in a future iteration we plan to take this suggestion and partition the seepage from the overlying layers based on their respective extent of exposure to sediment area.

**Details in the text**

p5/l15: ΔHmi is 0.1 m here and 0.01 m in Table 1

REPLY: Updated in Table 1

p6/l4: Section 2.6 only describes layer merging by mixing from above, i.e. in the surface mixed layer. Can layer merging also occur in deeper waters, and if yes, under what conditions?

REPLY: Layer merging and splitting below the thermocline occurs if they contract or expand due to inflow/outflow processes affecting these layers, beyond the prescribed thickness limits. The sentence below this in this paragraph is now:

"*Layer volumes change due to depth-specific changes in mixing, inflows and outflows, and thickness limits, $\Delta z_{min}$ and $\Delta z_{max}$, are enforced to adequately resolve the vertical density gradient, generally with fine resolution occurring in the metalimnion and thicker cells where gradients are weak.*"

p8/l16: remove "either"?

REPLY: Updated

p 9/10: ς is not atmospheric diffusive radiation, but "a constant related to atmospheric diffuse radiation" according to Table 1. It remains unclear, what this means.

REPLY: Updated; This is as reported in the supporting reference.

p11/l8: φSWS is defined here as the fraction heating up the surface layer but in Table 1 as the radiation flux crossing the water surface (should be φSW0 in the table?).

REPLY: These two symbols are different (the former is the amount remained in the SML, whereas the latter is the total incoming amount. The table is updated to have both.

p11/l23: ABen/As is a fraction, not a percentage.

REPLY: Updated

p11/l15: unit for vapor pressure needs to be given here or in Table 1.

REPLY: This was in Table 1, listed as hPa

p15/l20ff: I am not sure I understand the procedure for calculating ice melting here. I understand that first

$\phi$SW0 is calculated from eq. 31, then T0 is calculated such that $\phi$0 = $\phi$net (how is this done?), , and if T0 is then equal or larger than the melting temperature Tm, melting is determined from eq 30. Is that correct? Consider revising the text in this section to clarify.

REPLY: This subsection has been re-written to more clearly explain the heat conduction approach and associated boundary conditions that control ice layer changes.

p18/l5: replace possible with available?

REPLY: 18 doesnt seem relevant to this comment, so I am assuming this refers to P19 Line 5. Updated as suggested.

p19/l9: the notation zmsl is inconsistent, as all other layer thicknesses are named $\Delta z$.

REPLY: The notation is that $\Delta z$ is referring to a layer thickness whereas z is referring to depth from the surface. In this case they are essentially equivalent, but since the SML may have more than one layer, we prefer to keep the dzsml/dt notation here. The text is however updated however to refer to depth rather than thickness.

p20/l2: difference to what?

REPLY: The sentence is updated: " ... *the potential energy that would be released by mixed layer deepening is computed as the difference in the moments of layer masses in the epilimnion (surface mixed layer) about the lake bottom, relative to the well-mixed condition.*"

p21/l16-18: this is not clear.

REPLY: P21 is a figure so I am assuming this refers to P20 Line 16-18, where the shear velocity is computed. This text is updated.

p23/l6: if $\alpha$TKE is interpreted as diffusivity, why then Kz = CHYP and not Kz = $\alpha$TKE?

REPLY: Thanks for noticing the discrepancy - when we revised the paper we replaced $\alpha$TKE with CHYP, but missed this reference in the text.

p23/l10: "contains 85% of N2" sounds strange, as N2 is not a property that can be reasonably summed up across layers. Table 1 defines the same variable as the "fraction that contains 85% of the N2 variance". This is even less clear. Is it calculated by summing up all N2 values and then taking the volume for which the sum is 85% of the sum for the entire lake? If so, this is virtually the same as the volume that contains 85% of the density difference between the top and the bottom of the lake.

REPLY: Thanks for pointing out that this remains confusing, we have updated to make this more clear. To clarify, N2 varies following an approximate normal distribution with height, and the centre of the distribution is the height where the centre of buoyancy is located, computed as the 1st moment of the vertical N2 distribution. The vertical length scale which captures one standard deviation of this distribution is then computed (see also comment above; $dz_\sigma^2$). The volume being referred to here, $\tilde{V}_{N^2}$ is the volume of the lake above the height where this standard deviation is positioned. For convenience, this is called the 85% volume since a symmetric normal distribution is assumed and the volume does not account for the volume below the 1st standard deviation (~15% of N2 variance).

p23/l15: which density difference?

REPLY: This is referring to the rate of working doen by the inflow as it plunges. Updated to say " ... *the difference in density between the inflow water and layer into which it inserts*".

p25/l9: delete "the tangent of"

REPLY: Updated

p26/l9: really daily time step? The time scale for river intrusions is usually rather minutes to hours than days. So does this ever take more than one step?

REPLY: For many applications of modest to large sized lakes this can take many days; for small lakes and reservoirs though it can complete the insertion in just one step.

p26/l14: should be $\Delta z_{infj}$ instead of $z_{infj}$ in the equation.

REPLY: I think it is correct as is, the equation is computing the vertical change in elevation of the parcel from the previous time; the term in the brackets (h-z) is the height of the top of the inflow after the days travel where as h-$\Delta$z is the thickness of water above the inflow.

p27/l12: should refer to Eq. (52)?

REPLY: Updated

p31/l5: verb is missing.

REPLY: description updated.

p31/l8: If no weir is present I assume $Q_{Ovfl}$ is the same as Eq. (73), but with $Q_{weir}$ = 0?

REPLY: Yes, description updated.

p31/l9: Is $\Delta h_s$ the result of eq. (4)?

REPLY: Yes, description updated.

p35/l5: iti -> it

REPLY: Updated

p37/l2: I can't remember having read anything about solar shading in the model description.

REPLY: Updated - shading now mentioned when introducing $f_{SW}$, and cross referenced to this point also.

p37/l16: In Figure 16, glmtools is called GLMr.

REPLY: Figure updated

References: I did not check the list, but noticed that Spigel and Imberger (1980) is missing

REPLY: Updated to include this reference. Reference list also checked for formatting throughout.

**Comments to Table 1**

This table is very useful, but it needs to be thoroughly checked and corrected. I checked only a small part of the variables, the following list of inconsistencies and missing variables is therefore certainly incomplete:

I think the following variables used in the paper are missing in Table 1 (incomplete list):

$C_{wn}$, $H_0$, $NSW$, $K_{soil}$ $e_a$, $\theta_S$, $\theta_a$, $\delta_{zsoil}$

- The variables αb and βb should go to Lake domain.

- Use per mil rather than ppt for parts per thousand, as the latter is generally used for parts   per trillion.

- Check definition of θS

- Use Kelvin without degree sign.

- AC is AWS in Eq. 25b?

- most of the h's should be height above datum rather than height above bottom.

- fw is calculated in Eq. (83), not Eq. (78).

REPLY: The variables ea, θS, θa, δzsoil were already present, but we have added H0, NSW, Ksoil  so thank you for identifying these .The other changes have been updated, except the lower case h's are all above the lake bottom by definition (lower case h is all form the lake bottom, and upper case H is from the datum).

Furthermore, the variables are mostly ordered alphabetically, but not completely, and the assignment of variables to the different classes of variables can be ambiguous. It is therefore often rather difficult to find a variable in the Table.

REPLY: The table has been refined to include all variables under sequential headings:

Indices

Time variables

Lake domain (volumes, areas, heights and depths)

Other simulation variables and general parameters (listed alphabetically)

Finally, is unclear which parameter values are hard-coded and which can be modified by the user. Some parameters are defined as configurable, but for some others, which can also be defined by the user according to the text, just one value is given

REPLY: In the case where the numbers in the "value" column are for hard-coded parameters the otherwise they are the default suggestions. We have now referred to in the comments column the section of the configuration file where the parameters can be set. A new footnote is added to clarify the entries in this column.

**Anonymous Referee #2**

**Overview**

I appreciate the work that has been done on this revision. The layer splitting and merging scheme is now much better described and makes sense (end of p. 4 to middle of p. 6). The big Table 1, even though it might feel excessive, makes sure that everything has its proper definition and units. I still have some comments, mostly minor, and will recommend publication contingent on minor revision, even though I feel like I still haven't sufficiently combed the equations for inconsistencies.

REPLY: Many thanks for the time you have spent on the revised manuscript and we are grateful for the suggestions for corrections and improvements. Specific responses are provided below.

**Some things that I still consider somewhat major:**

1. P. 5, 1st paragraph: This starts to explain the split/merge scheme, and refers to Fig. 1, which doesn't show layers at all. You might want to add a separate figure that schematically shows a split and a merge. Fig. 1 can also benefit from having sensible heat flux included. So you'll want to reference the current Fig. 1 for the processes that lead to splitting and merging, and another figure (or panel) for the split/merge scheme itself.

REPLY: We note that Figure 1 does depict layers, albeit relatively faint. You are right they do not depict the merging/splitting scheme. We have adopted the suggestion for a new figure in the mixing section (now Figure 8) that depicts the mixing changes on the layer structure. It is somewhat idealised but can help readers interpret the content in the section.

2. The paragraph on p. 8, lines 9-13 is a rather good example of a brief description of the general concept that the following sub-sub-sections will fill in. Other sections should have more like this.

REPLY: Thanks for noting this and the suggestion. Sub-Sections that previously include a brief general concept description include 2.8, 2.7 and 2.3; we have therefore made modifications to the openings of sub-sections 2.2, 2.4, 2.5 and 2.6, plus we revisited 2.8.

3. Eqs. 37 and 38 are some that I decided to take a fairly close look at. In eq. 37, it seems to have the correct units, but the only thing in the equation that is suggestive of a wave number is the K-H billow length, which only has a single value, and other terms don't even have that. The "per wave number" part of the definition suggests that you might be able to do an integral across wave numbers to get a value per mass, but I'm not seeing how this works. In eq. 38, the units don't work out right. The part inside the brackets seems to have the right units (meters squared per second squared), but then is multiplied by a length.

REPLY: Many thanks for identifying this. We introduced some confusion in the units table definitions for these two terms. Both terms are based on an approximation of TKE made available per mass (m2/s2), that is then integrated over the depth of the layer being raised. As you correctly identified, this was accounting for the depth integration in the case of $E_{TKE}$, and not for $E_{PE}$. Whilst the units matched the definition per wavenumber, that was not our intent as this is a bulk estimate across the spectrum, and the unit description is corrected.

4. Table 1 should have an explanation in the header that says that a hyphen in the units column means no units. This made me ask myself what a hyphen in the value column means, and I don't think it is all that consistent. Some are distinctly independent variables (t and z). Some indicate the space-time location of a feature of the system (such as t sub b and H). To what extent might these be characterized as calculated? [t] seems to be more of a device used in coding the algorithm. E sub TKE and E sub PE seem more clearly to be calculated. U sub 10 seems to be time-series input (observed). e sub * is altogether undefined in the table. E sub PE seems like it should be defined as potential energy within the stratified water column *per mass*, but see the previous comment about eq. 38.

REPLY: Table 1 has been updated, including fixes to the above mistakes, and consistency with the value descriptions (and associated footnote).

5. Fig. 10: The inset should have a better 3-D effect. Consult an expert to add color and sheen so that it is unambiguous which direction each surface points. I don't understand why alpha sub inf is called a "half angle". It's the angle between the two sloping sides of the idealized river channel, right?

REPLY: The figure has been updated for this revision; whilst we didnt get the 3D effect it is hopefully less ambiguous now what the angles are referring to.

**Minor comments:**

1. P. 3, lines 12-14: I hesitate to call FLake a 1-D model. Maybe more like 0.5-D, because of its predefined shapes of temperature profile curves.

REPLY: Yes, this is a bulk model; the citation to Kirillin et al., (2011) is removed (but remains referred to elsewhere so is left in the list).

2. P. 3, line 26: It would be more convincing that GLEON is a large group if you explicitly state a number of participants.

REPLY: Thank you for the suggestion; it now reads:

*"... availability to a broad community (e.g., GLEON has >500 members from around 50 countries)."*

3. P. 6, line 3: This sentence seems very confusing. When you say energy, to you mean heat energy, TKE, or something else? Density instability promotes mixing, so it doesn't make sense to overcome instability to trigger mixing. And "accounting for" can be taken as either a cause or an effect; I'd rather see it expressed more explicitly as an effect of mixing.

REPLY: Thank you for observing this inaccurate wording. The sentence is redrafted as :

"*When density instabilities occur between adjacent layers, or when sufficient turbulent kinetic energy becomes available to overcome stable density gradients, then layers merge, thereby accounting for the process of mixing.*"

4. Hyphen police: See my previous comments. P. 6, line 19: "Well mixed" is an adverb + adjective, so should not have a hyphen. Captions to Figs. 2 and 5: "Time series" should have no hyphen. P. 14, line 2: "Length scale" without a hyphen.

REPLY: Updated; we should have them all now, and will also consider as the paper goes through to type-setting.

5. P. 6, line 27: The word "latent" should be before "heat flux".

REPLY: Updated

6. Eqs. 5 and 6: It seems that you are defining the water level as the level of liquid water if no ice were displacing it, while ice thickness is simply ice volume/area. In other words, when freezing occurs, water is withdrawn from the top layer of liquid water and transferred to the ice layer. This needs to be very explicit in order to understand these equations.

REPLY: Section 2.2 introduction has been modified to include the volume balance, and this is now made explicit.

7. P 8, line 10: "Uppermost" should be one word.

REPLY: Updated

8. P. 8, line 13: I don't think you've defined "RHS". Some might know this, but perhaps not all.

REPLY: Updated

9. P. 10, eq. 13: Is zenith-angle dependence already built into the definition of incoming light? Otherwise direct sunlight should have this formula with z divided by the cosine of the zenith angle, and diffuse light should use some effective average zenith angle.

REPLY: Depending on the solar radiation option (Eq 10 in the new revision), users can either input measured incoming solar data or predict it based on the bird model. The albedo for all options (Eq 13) accounts for the zenith angle, if the albedo options 2 or 3 are adopted.

10. P. 10, line 11: "Adsorption" refers to material being incorporated; for radiation, it should be "absorption".

REPLY: Updated

11. P. 12, line 9: Add "water vapour" before "mixing ratio".

REPLY: Updated

12. The journal's spelling standard apparently is British English, and this manuscript mostly follows that, but I noticed at least one occurrence of "vapor" on p. 16, line 2.

REPLY: Updated

13. P. 12, eq. 23b: Is there a citation for August-Roche-Magnus?

REPLY: Note this Eq has been updated, and this options removed.

14. P. 13, line 4: I'm not sure whether "within the internal boundary layer" implies that you need to define an internal boundary-layer height and measure at multiple heights within that range, or whether it only needs to be measured near the surface.

REPLY: To improve clarity we have reworded this sentence:

*"To ensure data provided is from within the internal boundary layer over the lake surface, this option requires the provision of wind speed, air temperature and relative humidity data from near the lake surface (e.g., 2-10 m, depending on lake size), supplied at approximately hourly resolution."*

15. P. 15, last line: "Penetrative" misspelled.

REPLY: Updated

16. P. 20, line 12: The formula at the beginning of this line should be multiplied by g.

REPLY: Updated

17. P. 23, line 11: There should be a semi-colon before "however".

REPLY: Updated

18. P. 25, line 9: Phi sub inf should be the angle of slope. I think this is better than just saying "slope" which is often expressed as a ratio, and I think "tangent" here is unintentional.

REPLY: Updated

19. The citation and the reference give different dates for Makler-Pick et al.

REPLY: Updated

20. Mueller et al. (2016) is out of alphabetical order.

REPLY: Updated

21. Snortheim et al. is missing a date.

REPLY: Updated

22. Tennessee Valley Authority (TVA) needs to be alphabetized according to the way that it is stated in the citations. It is not alphabetized correctly for either "TVA" or "Tennessee Valley Authority". I suggest leaving the citations as they are and alphabetizing it as "TVA (Tennessee Valley Authority)".

REPLY: Updated as suggested

[Figure]

[Figure]

**Revision 25 Jun 2018**

[revised manuscript text omitted]

---

## Author Response (AR3)

*21st September 2018*

Dear Dr. Unterstrasser,

**Re. Revision #3 of gmd-2017-257:  General Lake Model (GLM)**

Please find attached an updated revision of our paper including the two corrections as noted by Reviewer 1, and responses to suggestions from Reviewer 4. A detailed response to the reviewer's comments is included over page.

In addition, to address the concerns raised by Reviewer 1, the authorship team have made a significant effort to proof-read the entire manuscript and cross-check with the code to identify any final editorial mistakes. The track-changes version of the manuscript is included after the review comments; note the several edits in the Appendices were to bring the style in line with the main body text (particularly around the use of brackets and capitalisation) and do not relate to changes in functionality. We have also carefully checked and made some improvements to the large parameter table.

We thank you again for the significant time and effort that have gone into the discussion and look forward to your decision.

Kind Regards

Matthew Hipsey, on behalf of all co-authors.

**Anonymous Referee #1**

I decided not to further invest any time in reviewing this manuscript after noticing that the following two equations, which I had mentioned to be wrong in my previous review, are still wrong in the revised version:

Equation 21c: Brutsaert's equation is 1.24*(ea/Ta)^1/7 with ea in hPa or mbar (see eq. 11 in his paper). If the leading factor is 0.642 as here in equation 21c, then the unit of ea should be Pa (as 100^(1/7) = 1.93 = 1.24/0.642) and not hPa as stated by the authors in their reply.

Sorry, we misunderstood this and are thankful for the clarity. We have updated the coefficient from 0.642 to 1.24 (Page 12)

Equation 24: This equation still yields values on the order of 10^100, which is obviously not correct.

We did identify the missing "-" sign following the comment on the previous draft, but must have lost the change during the many revisions, for which we are very embarrassed. The -ve sign in Eq 24 is now added (Page 13).

I am sorry to make this decision, as I really would like this to be published. But it is very important to be accurate in model descriptions, as wrong equations will confuse the users of the model, and they might also cause errors in follow-up studies. Maybe the authors are unlucky and these are the only two errors remaining in the equations. But given the history of revisions, with numerous errors in both previous versions of the manuscript, in my opinion it is not acceptable that the third version still contains such basic errors.

We apologise the above issues caused frustration and are very grateful the previous errors were identified in the earlier drafts - a lot of effort has gone into evolving the model capability and the manuscript quality during the revision process and the sheer volume of changes has meant a few issues remained.

We do hope not to abandon it here when it is so close to being final. For this revision, we have spent further time combing the equations and cross-checking the code; there are a few additional editorial mistakes we note (shown in track changes), but we do not feel that they warrant rejection.

In the context of model reproducibility, a large effort has been put into corrections and improvements of the drafts, and we note that this round of corrections relate to minor editorial issues bearing in mind the overall scope of the model. We would also like to highlight that a) the code is fully open-source and has been version controlled prior to and during the revision process (i.e. all changes are traceable), and b) all the simulations reported in the paper using the code are available in full, with instructions for operation. While we acknowledge and apologise for the errors in the manuscript noted by Reviewer 1 above, our efforts in rectifying these errors and improving accessibility to the code and examples ensure that the scientific community now have the ability to re-create the model, follow the specific algorithms in the code, and run a diverse set of example simulations that test different aspects of model functionality.

To further aid readers link the code operation and the manuscript notations, we have included a table on the model website: http://aquatic.science.uwa.edu.au/research/models/GLM/getting_started.html. This will help assist reproducibility and prevent confusion about the notation changes we have made.

Anonymous Referee #4

**Overview**

The manuscript describes the one-dimensional hydrodynamic lake model GLM, which has been widely adopted by the scientific community. The model represents a tremendous effort in lake modelling which needs to be well documented. As such, it falls within the scope of this journal. The paper is well written and structured. It gives a comprehensive overview of the model structure and equations, while available tools for pre and post-processing and for linking the model to other models are described.

The current version is the third version submitted by the authors and corrects most of the errors and structural deficiencies and inconsistencies found in the previous versions. In that sense, the authors have done a remarkable work. I'm also glad that, although it is besides the scope of the manuscript, the authors have improved the GLM web page. Since great part of the deficiencies of the previous versions have already been corrected and the review process can't be eternal, I only have some specific comments to do.

Thank you for the positive comments about the manuscript.

**Specific comments**

P10L12: Kw is set as a fix value (variable if the model is couple with a WQ model, ie: AED2). This value have a no-negligible effect into the stratification and temperature evolution of the lakes. Most users have series of Kw (or Secchi disk), so it would be interesting to implement Kw as an input time series in the future versions of the model. Just a suggestion.

Thank you for the suggestion, we agree some users wish for Kw to vary over time without having to run AED2. In fact a forked version of the code already had this implemented, which we have merged into the latest code-base. A brief update to the text is made to mention this option.

Figure4a: The 2D map is not a direct result of the model. For that figure, the authors assume horizontal homogeneity of Kw. Not sure if it is a good idea to interpreter in a 2D map the results of a 1D model. In any case, what is the lake's size?

The model is 1D with a uniform Kw, but we wished to map the results back onto a real domain to help users visualise how they may use the 1D model to understand benthic habitat conditions. We have updated the caption with a disclaimer. The lake size and all other details are included in the online simulation setups, linked to via the DOI.

P13L16: Does it implies that the origin of the meteorological date must be within the lake to apply the corrections for non-neutral atmospheric stability? Shore? If so, please specify.

Yes this is our recommendation; wording updated to clarify (Page 14)

P24L7 & Fig.8. It would be interesting to include a (g) panel to include the effect of K-H mixing by splitting the layers from the (f) panel to have a more visual effect.

This is a useful suggestion, though we have not made the change at this stage as it requires the figure to be re-drafted. We will consider and update is possible if the paper is accepted.

P28L14: How is calculated the temperature of the local runoff?

It is assumed to be the air-temperature, with zero salinity. Text updated (Page 28)

*"Inflows have a prescribed composition (temperature, salinity, and scalars), except local runoff which is assumed to be at air temperature, with zero salinity"*

P33L14: What about an adaptive height water withdrawal? That would be useful for some reservoir management.

This is possible in the model, covered in Section 2.7.4.

End of review

[Figure]

[Figure]

**Revision 21 Sep 2018**

[revised manuscript text omitted]

---

## Author Response (AR4)

*26th October 2018*

Dear Dr. Unterstrasser,

**Re. Revision #3 of gmd-2017-257:  General Lake Model (GLM)**

With this upload please find our finalised version of the manuscript.

The manuscript is as per the previous upload but now including the update to Figure 8 to include an additional panel for depicting K-H mixing (as suggested in the previous revision), and some associated text.

There was also a small change to Eq 13, adding a new albedo option. Due to these small changes, no track changes version is included below.

We thank you and the reviewers again for the significant time and effort that have gone into the discussion and are very pleased it can now be accepted for publication.

Kind Regards

Matthew Hipsey, on behalf of all co-authors.